# `crowd-hpo`: Realistic Hyperparameter Optimization and Benchmarking for Learning from Crowds with Noisy Labels

**Marek Herde**                                    *marek.herde@uni-kassel.de*
*Intelligent Embedded Systems*
*University of Kassel*
*Kassel, Hesse, Germany*

**Lukas Lührs**                                    *lukas.luehrs@uni-kassel.de*
*Intelligent Embedded Systems*
*University of Kassel*
*Kassel, Hesse, Germany*

**Denis Huseljic**                                 *dhuseljic@uni-kassel.de*
*Intelligent Embedded Systems*
*University of Kassel*
*Kassel, Hesse, Germany*

**Bernhard Sick**                                  *bsick@uni-kassel.de*
*Intelligent Embedded Systems*
*University of Kassel*
*Kassel, Hesse, Germany*

**Reviewed on OpenReview:** *https://openreview.net/forum?id=SaKfhylVLK*

## Abstract

Crowdworking is a cost-efficient solution for acquiring class labels. Since these labels are subject to noise, various approaches to learning from crowds have been proposed. Typically, these approaches are evaluated using default hyperparameter configurations, which often result in unfair and suboptimal performance, or using hyperparameter configurations tuned via a validation set with ground truth class labels, which represents an often unrealistic scenario. Moreover, both setups can yield different approach rankings, complicating study comparisons. Therefore, we introduce `crowd-hpo` as a framework for evaluating approaches to learning from crowds, together with criteria for selecting well-performing hyperparameter configurations using only noisy crowd-labeled validation data. Extensive experiments with neural networks demonstrate that these criteria select hyperparameter configurations that improve the learning from crowds approaches' generalization performances, measured on separate test sets with ground truth labels. Hence, incorporating such criteria into experimental studies is essential for enabling fairer and more realistic benchmarking.

## 1 Introduction

Crowdworking represents a popular and cost-efficient solution to label data instances for classification tasks (Vaughan, 2018). However, the corresponding crowdworkers are error-prone for various reasons, e.g., missing domain knowledge, lack of concentration, or even adversarial behavior (Herde et al., 2021). Training deep neural networks with noisy crowd-labeled data decreases generalization performance because these networks tend to memorize the false class labels (Zhang et al., 2017). Hence, many approaches intend to improve the robustness against noisy labels. Together, they form the research area of *learning from noisy labels* (LNL) with the core topics of regularization, sample selection, robust loss functions, or dedicated neural

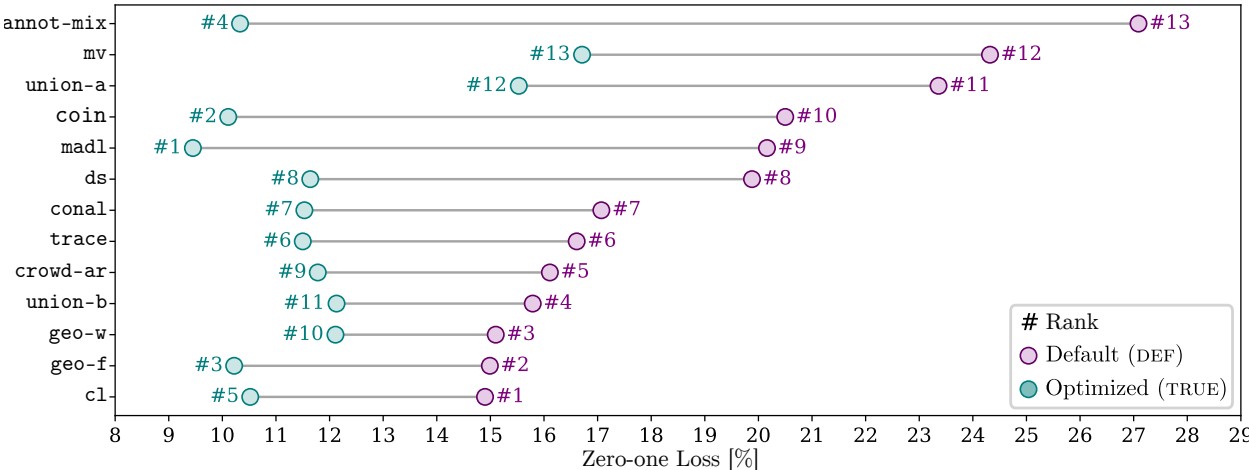

Figure 1: **Default** (DEF) **versus optimized** (TRUE) **HPCs for LFC approaches**. The $y$-axis lists the LFC approaches, and the $x$-axis the zero-one loss evaluated on a test set with true class labels of the `reuters-full` dataset (Rodrigues et al., 2017), whose training set contains noisy class labels from crowdworkers. Default HPCs result in substantially worse performance than HPCs optimized via validation data with true class labels. Further, HPO alters the approaches' ranking. For example, `cl` (Rodrigues & Pereira, 2018) performs best under default and only fifth-best under optimized HPCs, whereas `madl` (Herde et al., 2023) moves from the ninth place with the default HPCs to the first place after optimization.

network architectures (Song et al., 2022). Within this area, *learning from crowds*[1] (LFC, Raykar et al., 2010) approaches explicitly handle crowd-labeled data, where each instance receives a (potentially varying) number of noisy class labels and where we know which label originates from which crowdworker. Accordingly, these approaches estimate the crowdworkers' performances (e.g., labeling accuracies) to infer the instances' true (i.e., ground truth) class labels. Many experimental evaluation studies have demonstrated the performance gains of such approaches (Rodrigues & Pereira, 2018; Chu et al., 2021; Nguyen et al., 2024).

Translating these gains into practice demands effective *hyperparameter optimization* (HPO) to find well-performing *hyperparameter configurations* (HPCs). While approaches for training standard neural networks are tuned against a validation set with true class labels, LFC approaches have no access to such a set if all class labels originate from crowdworkers. As a result, HPO becomes a more difficult challenge. Potential workarounds are using data-agnostic default HPCs (Chen et al., 2021) or explicitly requiring access to a validation set with true class labels (Herde et al., 2023). We refer to both procedures as *hyperparameter selection* (HPS) criteria because each chooses HPCs for the LFC approaches. Figure 1 exemplifies that both HPS criteria lead to different test losses per LFC approach and even rankings between LFC approaches. Specifying default HPCs is realistic because it requires no true class labels. Nevertheless, such an HPS criterion often yields suboptimal performance results that are heavily influenced by nonobjective choices such as the experimenters' or software frameworks' presets, undermining fairness (Bagnall & Cawley, 2017). By contrast, HPO on a validation set with true labels can produce superior and fairer results when every LFC approach receives the same search budget. Nonetheless, this HPS criterion is unrealistic in an LFC setting where only noisy crowdworkers provide labels. Existing literature lacks HPS criteria to perform experiments for a fair (involving HPO) and realistic (with access only to crowd-labeled validation data) comparison of LFC approaches. Motivated by these observations and related ones in areas such as partial label learning (Wang et al., 2025), we analyze the following *research questions* (RQs):

---

[1]We use the term learning from crowds, whereas other publications in the same research area refer to multiple annotators (Li et al., 2022) or labelers (Rodrigues et al., 2013) instead of crowdworkers.

---

**`crowd-hpo`: Research Questions and Contributions**

$RQ_1$: *Given access only to crowd-labeled validation data, which evaluated hyperparameter selection criterion yields the highest performances for LFC approaches?*

$RQ_2$: *Given the best-evaluated hyperparameter selection criterion for crowd-labeled validation data, how do learning from crowds approaches compare in performance?*

Based on these research questions, we propose `crowd-hpo` for learning from crowds approaches with crowd-labeled validation data contributing:

- a framework of hyperparameter selection criteria based on empirical risk measures to be combined in a robust ensemble,

- an extensive experimental study benchmarking 13 learning from crowds approaches across 5 real-world datasets, each with 7 variants of noisy labels from humans,

- a guideline for realistic and fair experimentation to compare learning from crowds approaches' performances in combination with hyperparameter optimization,

- and a comprehensive codebase[a] to reproduce and perform experimental studies for learning from crowds approaches in combination with hyperparameter optimization.

---

[a]`https://github.com/ies-research/multi-annotator-machine-learning/tree/crowd-hpo`

## 2 Related Work

This section presents a discussion on foundational and related works on LFC (Raykar et al., 2010) approaches for classification tasks, their experimental studies, and validation with noisy class labels in the broader area of LNL (Song et al., 2022). In short, this discussion confirms that most experimental studies on LFC follow different experimentation protocols, none of which consider HPO with noisy crowd-labeled validation data. Although other works on LNL address validation issues in the presence of noisy labels across various contexts, they do not explicitly validate with crowdworkers of varying performances.

Table 1: **Overview of experimental studies of LFC approaches training neural networks for classification tasks.** Each row represents one study sorted by publication years, while the columns refer to the characteristics of such a study. We denote counts by the # symbol. We account for multiple simulation methods for the same single-labeled and variants for the same crowd-labeled dataset by ($\times \ldots$). The symbols $\checkmark_{\text{TL}}$ (**True Labels**) and $\checkmark_{\text{NL}}$ (**Noisy Labels**) denote the validation label type, whereas $\times$ indicates that the respective aspect has been ignored. If no information is available, we denote **?** as a symbol.

| Study | Venue | Approaches [#] | | Datasets [#] | | Hyperparameter Optimization | | Early Stopping |
|---|---|---|---|---|---|---|---|---|
| | | Two-stage | One-stage | Simulated | Real | Per Dataset | Per Approach | |
| Rodrigues & Pereira | AAAI | 3 | 4 | 1 | 1 | $\checkmark_{\text{TL}}$ | $\times$ | $\times$ |
| Cao et al. | ICLR | 1 | 4 | 3 ($\times$ 6) | 1 | $\times$ | $\times$ | $\times$ |
| Tanno et al. | CVPR | 1 | 5 | 2 ($\times$ 2) | 0 | $\times$ | $\times$ | $\checkmark_{\text{TL}}$ |
| Li et al. | TMM | 4 | 6 | 4 | 2 | $\times$ | $\times$ | $\times$ |
| Wei et al. | TNNLS | 1 | 6 | 4 ($\times$ 4) | 2 | $\times$ | $\times$ | $\times$ |
| Li et al. | MLJ | 2 | 5 | 4 ($\times$ 2) | 2 | $\times$ | $\checkmark_{\text{TL}}$ | $\checkmark_{\text{NL}}$ |
| Herde et al. | TMLR | 1 | 6 | 4 ($\times$ 4) | 2 | $\checkmark_{\text{TL}}$ | $\checkmark_{\text{TL}}$ | $\checkmark_{\text{TL}}$ |
| Ibrahim et al. | ICLR | 2 | 8 | 2 ($\times$ 2) | 2 | $\checkmark_{\text{TL}}$ | $\checkmark_{\text{TL}}$ | $\checkmark_{\text{TL}}$ |
| Cao et al. | SIGIR | 5 | 5 | 0 | 3 | ? | ? | ? |
| Herde et al. | ECAI | 2 | 9 | 6 | 5 | $\checkmark_{\text{TL}}$ | $\times$ | $\checkmark_{\text{TL}}$ |
| Zhang et al. | AAAI | 1 | 6 | 2 ($\times$ 4) | 3 | $\times$ | $\times$ | $\checkmark_{\text{TL}}$ |
| Li et al. | TPAMI | 6 | 7 | 4 ($\times$ 5) | 3 | $\times$ | $\times$ | $\times$ |
| Nguyen et al. | NeurIPS | 1 | 5 | 2 ($\times$ 3) | 3 | $\times$ | $\times$ | $\times$ |
| Han et al. | NeurIPS | 3 | 9 | 13 ($\times$ 2) | 2 | $\times$ | $\times$ | $\times$ |
| Guo et al. | NeurIPS | 2 | 7 | 2 ($\times$ 3) | 4 | $\times$ | $\times$ | $\checkmark_{\text{NL}}$ |
| Herde et al. | NeurIPS | 2 | 10 | 0 | 1 ($\times$ 7) | $\checkmark_{\text{TL}}$ | $\times$ | $\times$ |
| `crowd-hpo` | TMLR | 2 | 11 | 0 | 5 ($\times$ 7) | $\checkmark_{\text{NL}}$ | $\checkmark_{\text{NL}}$ | $\times$ |

**Learning from Crowds Approaches** Literature differs between two-stage and one-stage LFC approaches (Li et al., 2022). Two-stage approaches aggregate the noisy crowd-labeled class labels per instance in the first stage and use these aggregated labels as true class label estimates for training neural networks in the second stage. The most common aggregation algorithm is majority voting (`mv`), which implicitly assumes equal performances across the crowdworkers (Chen et al., 2022; Jiang et al., 2022). In contrast, the Dawid-Skene algorithm (`ds`, Dawid & Skene, 1979) leverages the *expectation-maximization* (EM) algorithm, where the true label probabilities are estimated in the E-step to update the crowdworkers' confusion matrices in the M-step. Typically, such label aggregation approaches operate with the given labels as only inputs (Zhang et al., 2016) and expect more than one class label per instance (Khetan et al., 2018). One-stage approaches aim to overcome these limitations by jointly training a neural network for estimating the true labels and a model for evaluating the crowdworkers' performances (Herde et al., 2023). The latter model is often implemented as weights of noise adaptation layers (Rodrigues & Pereira, 2018; Chu et al., 2021) or probabilistic confusion matrices (Tanno et al., 2019; Chu et al., 2021; Ibrahim et al., 2023) to model crowdworkers' class-dependent performances. More complex models, designed as (deep) neural networks, estimate performances as a function of instances and crowdworkers (Zhang et al., 2020; Li et al., 2022; Cao et al., 2023; Herde et al., 2024b).

**Experimental Studies for Learning from Crowds** For a better understanding of experimenting with LFC approaches, Table 1 overviews and characterizes recent experimental studies of LFC approaches. Most studies focus on presenting a new LFC approach compared to state-of-the-art competitors. We report the number of evaluated two-stage and one-stage LFC approaches for each study. Here, we count individual approaches if they incorporate distinct methodological ideas. In addition, we report the number of datasets used in each study. We distinguish between simulated and real crowd-labeled datasets. Simulated datasets are built on top of standard single-labeled datasets, such as `cifar10` (Krizhevsky, 2009), by simulating the labeling process of the crowdworkers. For the simulated data, most experimental studies consider multiple single-labeled datasets and multiple simulation methods for the noisy class labels. Analogously, multiple variants of crowd-labeled datasets can be constructed by subsampling the crowdworkers' labels, e.g., by retaining only a certain number of class labels per instance (Wei et al., 2022; Herde et al., 2024a). We take both procedures into account by denoting the product term (# datasets × # variants). Central to our analysis is the handling of the *hyperparameters* (HPs) for the LFC approaches. Here, we note the distinction between HPO, which involves systematically searching for the best HPC, and early stopping, a regularization technique that halts training once validation performance deteriorates, thereby preventing overfitting. If the HPO is only performed per dataset, e.g., to select the basic architecture and optimizer parameters, we set a check mark at "per dataset". If the HPO is only performed to determine values of specific HPs for an individual approach over multiple datasets, e.g., the best value for a regularization term, we set a check mark at "per approach". If HPO is performed for each dataset and approach, we set a check mark at "per dataset" and "per approach". If no HPO is performed, we set a cross for both columns. We also mark if noisy validation labels from crowdworkers are used for the HPO or if access to a validation set with true labels is assumed. For studies without any HPO, some experimentation relies on standard architectures with default HPCs across their study (Tanno et al., 2019; Zhang et al., 2024; Li et al., 2024; Nguyen et al., 2024; Han et al., 2024; Guo et al., 2024). In contrast, others specify the HPC for each dataset and approach without further explanation (Cao et al., 2019; Li et al., 2021; Wei et al., 2023). Several studies (Tanno et al., 2019; Herde et al., 2023; 2024b; Zhang et al., 2024; Nguyen et al., 2024; Guo et al., 2024) provide an extra ablation study for the HPs of their own LFC approaches.

**Validation with Noisy Class Labels** A few works exist on different aspects of validation with noisy class labels in the broader area of LNL. Chen et al. (2021) theoretically prove that for diagonally-dominant confusion matrices, the validation accuracy remains a reliable indicator of true performance. However, in practice, complex types of noise can still pose challenges, especially when the noise is systematic or when not enough data are available to average it out. For example, the empirical findings of Kuo et al. (2023) indicate that even small amounts of (not necessarily label) noise in the validation signal can significantly degrade HPO outcomes. The observations of Inouye et al. (2017) also confirm that standard validation can be misleading for localized, systematic label noise. Their proposed solution injects synthetic label noise into the training data (based on an estimated noise model) while keeping validation labels unchanged. This penalizes models that overfit spurious patterns and improves over standard cross-validation. Guo et al. (2024) evaluate

LFC approaches with early stopping using noisy validation data. However, no analysis regarding the effects of such an early stopping is reported. Yuan et al. (2024) also recognize the issues of training and validating with noisy class labels in the context of early stopping. Therefore, they propose a solution for early stopping that does not rely on a separate validation set. However, they do not perform any HPO but instead focus on demonstrating the robustness of their solution across different HPCs. In contrast, Wang et al. (2025) tackle the issue of HPO by proposing HPS criteria when learning from partial labels.

## 3 Hyperparameter Optimization with Noisy Labels from Crowds

This section first formalizes the problem setting and approaches to LFC, then outlines the basics of HPO, and finally introduces corresponding HPS criteria for handling noisy crowd-labeled validation data.

### 3.1 Problem Setting

Here, we describe the data generation process to define the objective of LFC approaches.

**Data Generation Process** Figure 2 depicts the probabilistic graphical model of the commonly assumed data generation process in LFC settings (Li et al., 2022; Herde et al., 2024b). Let the multiset[2] $\mathcal{X} := \{\boldsymbol{x}_n\}_{n=1}^N \subset \Omega_X, N \in \mathbb{N}_{\geq 1}$ denote the observed instances, which are independently drawn from $\Pr(\boldsymbol{x})$. Then, their one-hot encoded true class labels, denoted as the multiset $\mathcal{Y} := \{\boldsymbol{y}_n\}_{n=1}^N \subseteq \Omega_Y := \{\boldsymbol{e}_c\}_{c=1}^C$ with $C \in \mathbb{N}_{\geq 2}$ as the number of classes, are distributed according to $\Pr(\boldsymbol{y}|\boldsymbol{x}_n)$ and latent. Only the multiset $\mathcal{Z} := \{\boldsymbol{z}_{nm}\}_{n=1,m=1}^{N,M} \subseteq \Omega_Z := \Omega_Y \cup \{\boldsymbol{0}\}$ of one-hot encoded conditionally independent noisy class labels provided by $M \in \mathbb{N}_{\geq 2}$ crowdworkers is observable. Since not every crowdworker is requested to label each instance, some class labels from the crowdworkers are unobserved, denoted as an all-zero vector $\boldsymbol{0}$. An observed class label $\boldsymbol{z}_{nm}$ with $m \in \mathcal{M}_n := \{m|\boldsymbol{z}_{nm} \neq \boldsymbol{0}\}_{m=1}^M$ is assumed to be drawn from the instance-, class-, and crowdworker-specific distribution $\Pr(\boldsymbol{z}|\boldsymbol{x}_n, \boldsymbol{y}_n, m)$.

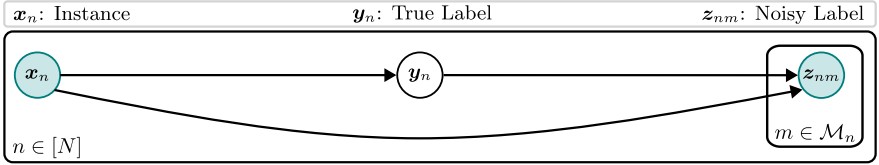

Figure 2: **Probabilistic graphical model of LFC.** Arrows show dependencies between random variables, while shaded circles indicate observed variables and unshaded latent ones. The set $\mathcal{M}_n \subseteq [M] := \{1, \dots, M\}$ indicates that an instance $\boldsymbol{x}_n$ is not necessarily labeled by all $M$ crowdworkers.

**Objective** LFC approaches aim to optimize the parameters $\boldsymbol{\theta} \in \Omega_\Theta$ of the data classification model $\boldsymbol{f}_{\boldsymbol{\theta}} : \Omega_X \to \Delta_C$ by minimizing its expected risk:

$$\boldsymbol{\theta}^\star := \underset{\boldsymbol{\theta} \in \Omega_\Theta}{\arg\min} \left( \mathbb{E}_{\Pr(\boldsymbol{x}, \boldsymbol{y})} \left[ L\left( \boldsymbol{y}, \boldsymbol{f}_{\boldsymbol{\theta}}(\boldsymbol{x}) \right) \right] \right), \tag{1}$$

where $\Delta_C$ is a probability simplex and $L : \Delta_C \times \Delta_C \to \mathbb{R}$ denotes an appropriate loss function. Throughout this article, we employ the zero-one loss (Vapnik, 1995) to assess the data classification model's predictions:[3]

$$L_{0/1}(\boldsymbol{y}, \hat{\boldsymbol{y}}) := 1 - \left( \underset{\boldsymbol{e}_c \in \Omega_Y}{\arg\max} \left( \boldsymbol{e}_c^{\mathrm{T}} \boldsymbol{y} \right) \right)^{\mathrm{T}} \left( \underset{\boldsymbol{e}_c \in \Omega_Y}{\arg\max} \left( \boldsymbol{e}_c^{\mathrm{T}} \hat{\boldsymbol{y}} \right) \right). \tag{2}$$

### 3.2 Approaches to Learning from Crowds

Given the objective in Eq. (1), LFC approaches do not directly optimize the outputs of the data classification model $\boldsymbol{f}_{\boldsymbol{\theta}}$ due to the lack of true labels $\mathcal{Y}$. Instead, the noisy class labels $\mathcal{Z}$ are used to train a crowdworker

---

[2]A multiset is a set that can contain duplicates.

[3]The dot product of two one-hot encoded label vectors is one if and only if they represent the same class.

classification model $\boldsymbol{g_\phi} : \Omega_X \times [M] \to \Delta_C$ with parameters $\boldsymbol{\phi} \in \Omega_\Phi$. This model predicts the probability distribution over all class labels for each instance-crowdworker pair, where each label's value indicates the probability that the given crowdworker will assign that label to the given instance. The estimates of both classification models are typically linked through transformations based on confusion matrices (Tanno et al., 2019) or noise adaptation layers (Rodrigues & Pereira, 2018), which try to separate the crowdworkers' noise from the true class label distribution. Furthermore, such a noise separation allows defining a crowdworker performance model $h_{\boldsymbol{\psi}} : \Omega_X \times [M] \to [0,1]$ with parameters $\boldsymbol{\psi} \in \Omega_\Psi$ quantifying crowdworkers' labeling accuracies. These three different models' predictions have the following probabilistic interpretations:

$$[\boldsymbol{f_\theta}(\boldsymbol{x}_n)]_c := \Pr(\boldsymbol{y}_n = \boldsymbol{e}_c | \boldsymbol{x}_n, \boldsymbol{\theta}), \tag{3}$$

$$[\boldsymbol{g_\phi}(\boldsymbol{x}_n, m)]_c := \Pr(\boldsymbol{z}_{nm} = \boldsymbol{e}_c | \boldsymbol{x}_n, m, \boldsymbol{\phi}), \tag{4}$$

$$h_{\boldsymbol{\psi}}(\boldsymbol{x}_n, m) := \Pr(\boldsymbol{z}_{nm}^{\mathrm{T}} \boldsymbol{y}_n = 1 | \boldsymbol{x}_n, m, \boldsymbol{\psi}), \tag{5}$$

where $[\cdot]_c$ denotes the $c$-th element of a vector. Throughout the main text, we regard the three models as black-box functions, potentially with shared parameters. Appendix A summarizes concrete LFC implementations and how they estimate the probabilities in Eqs. (3)-(5).

## 3.3 Hyperparameter Optimization

Let us define the dataset $\mathcal{D} := \{(\boldsymbol{x}_n, \mathcal{Z}_n)\}_{n=1}^N$ with $\mathcal{Z}_n = \{\boldsymbol{z}_{nm}\}_{m \in \mathcal{M}_n}$ to encompass only instances with their observed class labels. Then, a learning algorithm $\boldsymbol{A_\lambda}$ (corresponding to an LFC approach) with the HPC $\boldsymbol{\lambda} \in \Omega_\Lambda$ outputs all required parameters $\boldsymbol{A_\lambda}(\mathcal{D}) \in \Omega_\Pi$, of which model-specific parameters are (possibly overlapping) projections with $\boldsymbol{\pi_f}(\boldsymbol{A_\lambda}(\mathcal{D})) \in \Omega_\Theta$, $\boldsymbol{\pi_g}(\boldsymbol{A_\lambda}(\mathcal{D})) \in \Omega_\Phi$, and $\boldsymbol{\pi_h}(\boldsymbol{A_\lambda}(\mathcal{D})) \in \Omega_\Psi$. Each dimension in the HP search space $\Omega_\Lambda$ corresponds to a single HP, e.g., the number of epochs (integer), the learning rate (continuous), or the type of the optimizer (categorical). Ideally, we find the optimal HPC $\boldsymbol{\lambda^\star} \in \Omega_\Lambda$ such that our learning algorithm outputs the optimal classification model parameters (see Eq. (1)):

$$\boldsymbol{\pi_f}(\boldsymbol{A_{\lambda^\star}}(\mathcal{D})) = \boldsymbol{\theta^\star}. \tag{6}$$

In practice, finding the optimal solution is difficult due to many challenges, of which two critical ones are:

① Evaluating each HPC $\boldsymbol{\lambda} \in \Omega_\Lambda$ is computationally infeasible for a large HP search space $\Omega_\Lambda$.

② We can only estimate the expected risk (see Eq. (1)) because $\Pr(\boldsymbol{x}, \boldsymbol{y})$ is unknown.

In this article, we focus exclusively on challenge ② because the risk estimation is difficult and underexplored with access to only crowd-labeled validation data (see Section 2). Challenge ① is not part of our contributions. Instead, we briefly review established solutions as context for the HPO loop shown in Figure 3.

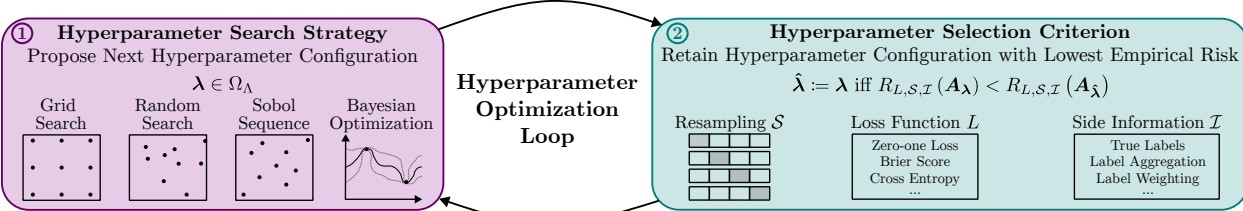

Figure 3: **HPO loop.** In an iterative process, HPO techniques explore the HP search space by retraining and evaluating the learning algorithm with different HPCs.

① **Hyperparameter Search Strategy**  Given true labels, research on HPO focuses primarily on improving the search strategy for (iteratively) proposing a set of candidate HPCs $\Lambda \subset \Omega_\Lambda$ by balancing the exploration-exploitation trade-off within the HP search space $\Omega_\Lambda$ given a budget of $|\Lambda| \in \mathbb{N}_{\geq 1}$ evaluated HPCs. For this purpose, random search is a popular choice that samples HP values randomly from predefined ranges, often

outperforming exhaustive grid search in high-dimensional spaces (Bergstra & Bengio, 2012). Meanwhile, Sobol sequences (Sobol, 1998) and Bayesian optimization (Wang et al., 2023) guide the search of candidate HPCs even more efficiently. Bayesian optimization usually excels because it adaptively selects each new HPC. However, we employ Sobol sequences so that every HPS criterion evaluates the same set $\Lambda$ of candidate HPCs. Hence, any performance differences between the HPS criteria come solely from choosing different HPCs.

② **Hyperparameter Selection Criterion**   Ideally, an HP search strategy has access to a reliable empirical risk estimation (Vapnik, 1995), which assigns a learning algorithm a scalar value $R_{L,\mathcal{S},\mathcal{I}}(\boldsymbol{A_\lambda}) \in \mathbb{R}$. Thereby, $L$ denotes the loss function (see Eq. (2)), $S$ a resampling technique, and $\mathcal{I}$ additional side information. Formally, we represent a resampling technique, e.g., hold-out, cross-validation, or bootstrapping, through a set of $K \in \mathbb{N}_{\geq 1}$ disjoint training ($\mathcal{T}$) and validation ($\mathcal{V}$) splits of the full training set $\mathcal{D}$:

$$\mathcal{S} \coloneqq \{(\mathcal{T}_k, \mathcal{V}_k) | \mathcal{T}_k \cup \mathcal{V}_k = \mathcal{D} \wedge \mathcal{T}_k \cap \mathcal{V}_k = \emptyset\}_{k=1}^K. \tag{7}$$

Side information $\mathcal{I}$ encompasses all required inputs beyond the loss function $L$ and resampling technique $\mathcal{S}$ for computing the empirical risk. Based on such empirical risk estimates, we define an HPS criterion as a rule picking the evaluated HPC with the lowest empirical risk:

$$\hat{\boldsymbol{\lambda}} \coloneqq \arg\min_{\boldsymbol{\lambda} \in \Lambda} \left(R_{L,\mathcal{S},\mathcal{I}}(\boldsymbol{A_\lambda})\right). \tag{8}$$

We treat the empirical risk measure $R_{L,\mathcal{S},\mathcal{I}}$ as a placeholder, whose explicit definition depends on the respective criterion. For example, suppose the true class labels are obtained from an expert as side information such that $\mathcal{I} \coloneqq \mathcal{Y}$. Then, the true empirical risk of the learning algorithm $\boldsymbol{A_\lambda}$ is computed as:

$$R_{L,\mathcal{S},\mathcal{Y}}(\boldsymbol{A_\lambda}) \coloneqq \sum_{(\mathcal{T}_k, \mathcal{V}_k) \in \mathcal{S}} \sum_{(\boldsymbol{x}_n, \mathcal{Z}_n) \in \mathcal{V}_k} \frac{1}{K \cdot |\mathcal{V}_k|} L\left(\boldsymbol{y}_n, \boldsymbol{f}_{\boldsymbol{\pi_f}(\boldsymbol{A_\lambda}(\mathcal{T}_k))}(\boldsymbol{x}_n)\right). \tag{9}$$

Since we have only access to noisy crowd-labeled validation data in an LFC setting, the HPS criterion based on the true empirical risk $R_{L,\mathcal{S},\mathcal{Y}}$ represents our upper baseline criterion for HPO, denoted as TRUE (plug Eq. (9) into Eq. (8)). In contrast, our lower baseline criterion DEF (plug Eq. (10) into Eq. (8)) constantly outputs a default HPC $\boldsymbol{\lambda}_{\text{DEF}} \in \Omega_\Lambda$, which corresponds to a naive risk estimation using the default HPC as side information, i.e., $\mathcal{I} \coloneqq \boldsymbol{\lambda}_{\text{DEF}}$, such that:

$$R_{L,\mathcal{S},\boldsymbol{\lambda}_{\text{DEF}}}(\boldsymbol{A_\lambda}) \coloneqq \delta(\boldsymbol{\lambda}_{\text{DEF}} \neq \boldsymbol{\lambda}), \tag{10}$$

where $\delta : \{\text{false}, \text{true}\} \to \{0, 1\}$ denotes an indicator function.

### 3.4 Hyperparameter Selection Criteria for Crowd-labeled Validation Data

If the validation set is labeled only by the crowd, the HPS criterion TRUE cannot be used. We therefore introduce proxy[4] HPS criteria. Each proxy plugs a specific empirical risk measure $R_{L,\mathcal{S},\mathcal{I}}$ into the minimum-risk selection rule (see Eq. (8)). Again, the side information $\mathcal{I}$ is only a placeholder that is instantiated in accordance with each risk definition. Rather than listing formulas, Figure 4 highlights commonalities and differences between the HPS criteria by presenting them as leaves of a tree. The path from root to leaf records the design choices and assumptions about the empirical risk template, how crowdworker performances are modeled, and how labels are weighted. Which combination works best is unknown and depends on the data and the LFC approach. Hence, we also introduce an ensemble of HPS criteria that combines their risk estimates, aiming for a more robust selection than any single criterion. In the following, we proceed through the tree's levels beyond its root. Finally, we note that this tree-structured overview is illustrative, not exhaustive, and future HPS criteria may introduce additional design choices.

---

[4]We use the term proxy to avoid confusion with classical surrogate losses that have proven consistency guarantees. For further guidance regarding our empirical viewpoint and its limitations, we refer to Section 4.3 and Section 5.

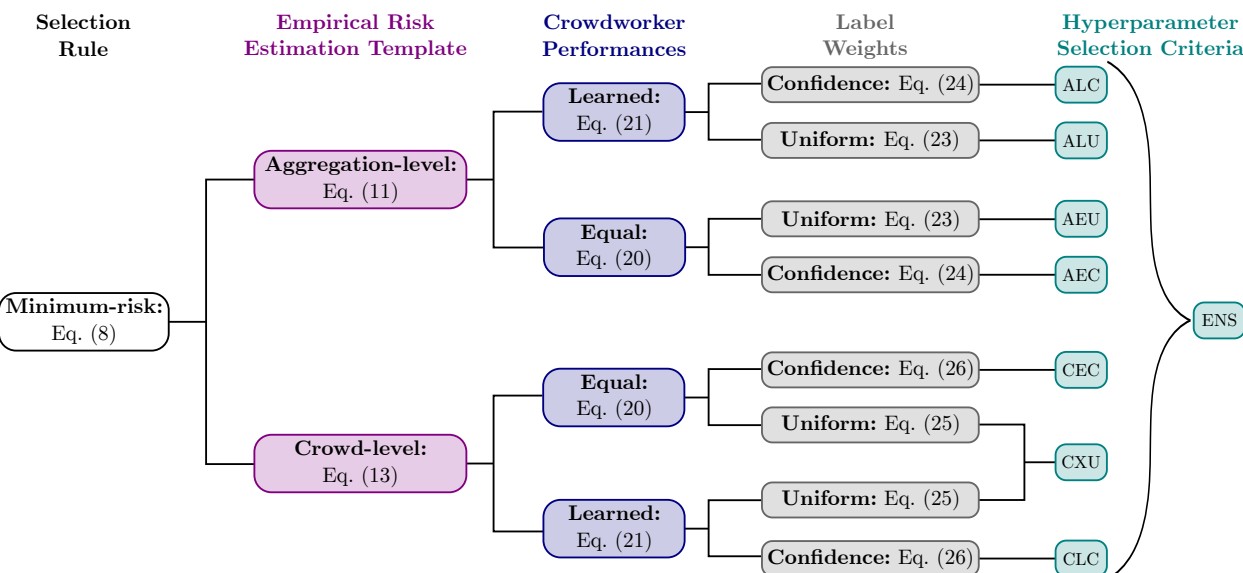

Figure 4: **HPS criteria for noisy crowd-labeled validation data.** Each HPS criterion is defined by composing the fixed root equation with the equations at branching nodes along its path. Fixed equations elsewhere are implicit. The criterion's name is formed from the internal nodes' initials on its path. The abbreviation CXU indicates that the paths with the abbreviations CEU and CLU lead to identical HPC selections. The ensemble-based criterion ENS combines all criteria for improved robustness.

**Empirical Risk Estimation Template**  LFC approaches typically involve the joint training of the data classification model $\boldsymbol{f_\theta}$ and the crowdworker classification model $\boldsymbol{g_\phi}$ (see Section 3.2), whose predictive performances capture distinct facets of the training outcome. We therefore introduce two empirical risk templates that become concrete risk measures when their components are instantiated at the tree's subsequent levels. On the one hand, we assess the data classification model $\boldsymbol{f_\theta}$ by computing the **aggregation-level** empirical risk through:

$$R_{L,\mathcal{S},\{\overline{\boldsymbol{z}},w\}}(\boldsymbol{A_\lambda}) \coloneqq \sum_{(\mathcal{T}_k,\mathcal{V}_k)\in\mathcal{S}} \sum_{(\boldsymbol{x}_n,\mathcal{Z}_n)\in\mathcal{V}_k} \frac{w(\boldsymbol{x}_n,\mathcal{Z}_n)}{K\cdot W_k} L\left(\overline{\boldsymbol{z}}(\boldsymbol{x}_n,\mathcal{Z}_n), \boldsymbol{f}_{\boldsymbol{\pi_f}(\boldsymbol{A_\lambda}(\mathcal{T}_k))}(\boldsymbol{x}_n)\right) \text{ with} \tag{11}$$

$$W_k \coloneqq \sum_{(\boldsymbol{x}_n,\mathcal{Z}_n)\in\mathcal{V}_k} w(\boldsymbol{x}_n,\mathcal{Z}_n) \in \mathbb{R}_{>0}, \tag{12}$$

where the two functions in $\mathcal{I} \coloneqq \{\overline{\boldsymbol{z}}, w\}$ denote the side information. The label aggregation function $\overline{\boldsymbol{z}} : \Omega_X \times \mathcal{B}(\Omega_Y) \to \Delta_C$ aims to infer the latent true class labels with $\mathcal{B}(\Omega_Y)$ referring to the set of all finite multisets over the class labels in $\Omega_Y$. The label weighting function $w : \Omega_X \times \mathcal{B}(\Omega_Y) \to \mathbb{R}_{\geq 0}$ weights individual loss contributions of the aggregated class labels. On the other hand, we compute the **crowd-level** empirical risk to assess the crowdworker classification model $\boldsymbol{g_\phi}$ through:

$$R_{L,\mathcal{S},v}(\boldsymbol{A_\lambda}) \coloneqq \sum_{(\mathcal{T}_k,\mathcal{V}_k)\in\mathcal{S}} \sum_{(\boldsymbol{x}_n,\mathcal{Z}_n)\in\mathcal{V}_k} \sum_{z_{nm}\in\mathcal{Z}_n} \frac{v(\boldsymbol{x}_n,m,\mathcal{Z}_n)}{K\cdot V_k} L\left(\boldsymbol{z}_{nm}, \boldsymbol{g}_{\boldsymbol{\pi_g}(\boldsymbol{A_\lambda}(\mathcal{T}_k))}(\boldsymbol{x}_n,m)\right) \text{ with} \tag{13}$$

$$V_k \coloneqq \sum_{(\boldsymbol{x}_n,\mathcal{Z}_n)\in\mathcal{V}_k} \sum_{z_{nm}\in\mathcal{Z}_n} v(\boldsymbol{x}_n,m,\mathcal{Z}_n) \in \mathbb{R}_{>0}, \tag{14}$$

where the label weighting function $v : \Omega_X \times [M] \times \mathcal{B}(\Omega_Y) \to \mathbb{R}_{\geq 0}$ as side information $\mathcal{I} \coloneqq v$ weights the individual loss contributions of the crowdworkers' labels. Intuitively, both label weighting functions are to downweight the influence of labels that are likely false, allowing the more reliable ones to dominate the loss and thereby enhancing robustness to label noise. Both risk templates come with potential downfalls. While the aggregation-level risk requires us to estimate the true class labels, and therefore evaluates the model on

data we do not have, the crowd-level risk evaluates the model's capacity in predicting the labels provided by the individual crowdworkers, and therefore evaluates the model on a different objective. Both can serve as proxies under a reasonable modeling of crowdworkers' labeling behavior.

**Crowdworker Performances** Starting from different assumptions about crowdworker performances, we estimate true class label probabilities. This creates a direct connection between crowdworker performance and risk estimation, because these posteriors induce the label aggregation function $\overline{z}$ and the label weighting functions $w, v$ as parts of the risk estimation template. For this purpose, let us assume that we have the estimated confusion probabilities $\widehat{\Pr}(z_{nm}|x_n, y_n, m)$ for each crowdworker $m \in \mathcal{M}_n$ and instance $x_n \in \mathcal{X}$. Then, we estimate the posterior true class label probabilities, i.e., after observing the crowdworkers' labels $\mathcal{Z}_n$, through:

$$\widehat{\Pr}(y_n = e_c|x_n, \mathcal{Z}_n) \stackrel{(\star)}{\propto} \widehat{\Pr}(y_n = e_c|x_n)\widehat{\Pr}(\mathcal{Z}_n|x_n, y_n = e_c) \tag{15}$$

$$\stackrel{(\dagger)}{=} \widehat{\Pr}(y_n = e_c|x_n) \prod_{m \in \mathcal{M}_n} \widehat{\Pr}(z_{nm}|x_n, y_n = e_c, m), \tag{16}$$

where the transformation $(\star)$ corresponds to Bayes' theorem and $(\dagger)$ to the assumed crowdworkers' conditional independence. Even with strictly disjoint training and validation instances, if we employed the data classification model $f_\theta$ to predict $\widehat{\Pr}(y_n|x_n)$, we would bias the posterior true class label probability estimation $\widehat{\Pr}(y_n = e_c|x_n, \mathcal{Z}_n)$ toward the data classification model's own predictions (see Proposition 1 in Appendix B.1), yielding a model-dependent (circular) validation target and over-optimistic risk estimates. Instead, we assume uniform prior class probabilities, i.e., before observing any crowdworkers' labels $\mathcal{Z}_n$:

$$\widehat{\Pr}(y_n = e_c|x_n) \coloneqq \frac{1}{C}. \tag{17}$$

If we employed the full crowdworkers' confusion probability estimates $\widehat{\Pr}(z_{nm}|x_n, y_n, m)$, the posterior computation would be vulnerable to class-specific biases (see Proposition 2 in Appendix B.2). Hence, we model the confusion probabilities (only during validation) via the crowdworker's instance-wise performance estimate $\widehat{\Pr}(z_{nm}^T y_n = 1|x_n, m)$ by defining:

$$\widehat{\Pr}(z_{nm} = e_k|x_n, y_n = e_c, m) \coloneqq \widehat{\Pr}(z_{nm}^T y_n = 1|x_n, m)^{e_k^T e_c} \left( \frac{\widehat{\Pr}(z_{nm}^T y_n = 0|x_n, m)}{C - 1} \right)^{1 - e_k^T e_c}. \tag{18}$$

By summarizing each crowdworker's behavior on a given instance with one scalar performance value, we apply the same performance value to every class. All classes are, therefore, treated identically, and no systematic bias toward any particular class arises. As a result of Eq. (17) and Eq. (18), the posterior estimation from Eq. (16) reduces to:

$$\widehat{\Pr}(y_n = e_c|x_n, \mathcal{Z}_n) \propto \prod_{m \in \mathcal{M}_n} \widehat{\Pr}(z_{nm}^T y_n = 1|x_n, m)^{z_{nm}^T e_c} \left( \frac{\widehat{\Pr}(z_{nm}^T y_n = 0|x_n, m)}{C - 1} \right)^{1 - z_{nm}^T e_c}, \tag{19}$$

where we distinguish between:

$$\widehat{\Pr}(z_{nm}^T y_n = 1|x_n, m) \coloneqq p \in (1/C, 1] \qquad \text{as \textbf{equal} crowdworker performances,} \tag{20}$$

$$\widehat{\Pr}(z_{nm}^T y_n = 1|x_n, m) \coloneqq h_{\pi_h(A_\lambda(\mathcal{T}_k))}(x_n, m) \qquad \text{as \textbf{learned} crowdworker performances.} \tag{21}$$

In the first case, the exact value of the labeling accuracy $p$ is irrelevant. Instead, $p$ only encodes the assumption that all crowdworkers have the same performance across all instances, which is better than randomly guessing. In the second case, the crowdworker- and potentially instance-wise performances are estimated by the crowdworker performance model $h_{\pi_h(A_\lambda(\mathcal{T}_k))}$ obtained after training with the respective LFC approach $A_\lambda$ on the $k$-th training fold using the candidate HPC $\lambda$. In both cases, the label aggregation function $\overline{z}$ outputs the *maximum a posteriori* (MAP) estimate of the true class label:

$$\overline{z}(x_n, \mathcal{Z}_n) \coloneqq \underset{e_c \in \Omega_Y}{\arg\max} \left( \widehat{\Pr}(y_n = e_c|x_n, \mathcal{Z}_n) \right). \tag{22}$$

When all crowdworkers are assumed to perform equally, the MAP estimate reduces to simple majority voting. In contrast, it naturally becomes weighted majority voting once their performances are learned (see Proposition 3 in Appendix B.3).

**Label Weights**    The label weighting functions control a class label's impact on the risk estimate. A uniform weighting gives every label the same weight, whereas confidence weighting scales a label's weight in proportion to its estimated correctness probability. Accordingly, the label weighting functions for aggregation-level risk estimation take the forms:

$$w(\boldsymbol{x}_n, \mathcal{Z}_n) \coloneqq 1 \qquad\qquad\qquad \text{as } \textbf{uniform} \text{ weighting,} \qquad (23)$$

$$w(\boldsymbol{x}_n, \mathcal{Z}_n) \coloneqq \max_{\boldsymbol{e}_c \in \Omega_Y} \left( \widehat{\Pr}\left( \boldsymbol{y}_n = \boldsymbol{e}_c | \boldsymbol{x}_n, \mathcal{Z}_n \right) \right) \qquad \text{as } \textbf{confidence} \text{ weighting,} \qquad (24)$$

whereas the label weighting functions for the crowd-level risk take the forms:

$$v(\boldsymbol{x}_n, m, \mathcal{Z}_n) \coloneqq 1 \qquad\qquad\qquad \text{as } \textbf{uniform} \text{ weighting,} \qquad (25)$$

$$v(\boldsymbol{x}_n, m, \mathcal{Z}_n) \coloneqq \widehat{\Pr}\left( \boldsymbol{y}_n = \boldsymbol{z}_{nm} | \boldsymbol{x}_n, \mathcal{Z}_n \right) \qquad \text{as } \textbf{confidence} \text{ weighting.} \qquad (26)$$

We evaluate only these two label weighting schemes because they serve as simple, assumption-light baselines. Nevertheless, other weighting schemes are possible in principle, e.g., discarding highly uncertain labels or tempering confidence scores.

**Hyperparameter Selection Criteria**    The different combinations of risk estimation template, crowdworker performance modeling, and label weighting correspond to $J = 7$ distinct HPS criteria with their associated risk measures $\mathcal{R} \coloneqq \{R_{L,\mathcal{S},\mathcal{I}_j}\}_{j=1}^J$. As an example, let us look at the path of the criterion AEU in Figure 4. This criterion instantiates the aggregation-level risk template from Eq. (11) with the aggregation function from Eq. (22), which evaluates the class posteriors from Eq. (19) with the equal crowdworker performance model from Eq. (20), and uniform label weights from Eq. (23). The resulting risk measure reduces to computing the average zero-one loss between the data classification model's predictions with plain majority vote labels as targets for the validation instances. Each empirical risk measure in $\mathcal{R}$ is only a proxy the true empirical risk (see Eq. (9)):

$$R_{L,\mathcal{S},\mathcal{I}_j}(\boldsymbol{A}_{\boldsymbol{\lambda}}) = R_{L,\mathcal{S},\mathcal{Y}}(\boldsymbol{A}_{\boldsymbol{\lambda}}) + \epsilon_{L,\mathcal{S},\mathcal{Y},\mathcal{I}_j}(\boldsymbol{A}_{\boldsymbol{\lambda}}) \ \text{ with } \epsilon_{L,\mathcal{S},\mathcal{Y},\mathcal{I}_j}(\boldsymbol{A}_{\boldsymbol{\lambda}}) \in \mathbb{R}. \qquad (27)$$

The difference arises through multiple sources, e.g., imperfect label aggregation, imprecise crowdworker performance estimates, or even different target models. For example, the crowd-level risk estimation measures the risk of the crowdworker classification model, which is different from estimating the risk of the data classification model as in Eq. (9). Because of these issues, we propose combining our empirical risk measures into an ensemble-based selection criterion ENS. For this purpose, we adopt the classic Borda count (de Borda, 1781), a rank-aggregation rule used in robust meta-evaluation (Abdulrahman et al., 2018). Let $\mathcal{O} \coloneqq \{o_j : \Lambda \to \{1, \ldots, |\Lambda|\}\}_{j=1}^J$ denote the set of ranking functions such that:

$$o_j(\boldsymbol{\lambda}) \coloneqq 1 + \sum_{\boldsymbol{\lambda}' \in \Lambda} \delta(R_{L,\mathcal{S},\mathcal{I}_j}(\boldsymbol{A}_{\boldsymbol{\lambda}}) > R_{L,\mathcal{S},\mathcal{I}_j}(\boldsymbol{A}_{\boldsymbol{\lambda}'})). \qquad (28)$$

Accordingly, $o_j(\boldsymbol{\lambda}) = 1$ corresponds to the HPC with the lowest empirical risk. The empirical risk estimate (proxy) based on the Borda count is then defined as:

$$R_{L,\mathcal{S},\mathcal{O}}(\boldsymbol{A}_{\boldsymbol{\lambda}}) \coloneqq \sum_{o_j \in \mathcal{O}} o_j(\boldsymbol{\lambda}), \qquad (29)$$

where the ranking functions serve as our side information such that $\mathcal{I} \coloneqq \mathcal{O}$. As a result, the criterion ENS outputs the HPC with the minimum rank sum (plug Eq. 29 into Eq. (8)). Intuitively, summing rankings is expected to stabilize decisions by balancing individual biases of each noisy risk measure in $\mathcal{R}$, increasing the likelihood of choosing a robust HPC.

## 4 Experimental Study

This section starts with a comprehensive description of our experimental setup. Subsequently, we analyze our experimental results to answer $RQ_1$ and $RQ_2$ as our two central research questions. The corresponding answers serve as a basis for formulating a guideline to design realistic and fair experiments when benchmarking LFC approaches in the future.

### 4.1 Experimental Setup

Our setup covers datasets, neural network architectures, LFC approaches, HP search, and HPS criteria. We describe our design choices for each of these aspects in the following.

**Datasets**   Realistic datasets are a requirement for a meaningful evaluation of LFC approaches. Therefore, we rely only on real-world datasets annotated by error-prone humans, mostly actual crowdworkers. Table 2 overviews these datasets by detailing their key attributes. The dataset `mgc` (Tzanetakis & Cook, 2002) originally contains $30s$ audio files of songs to be classified according to their music genres. A subset of the well-known image benchmark dataset `label-me` (Russell et al., 2008) concerns the classification of scenes, while the dataset `dopanim` (Herde et al., 2024a) targets the classification of doppelganger (groups of highly similar) animals. There are two text datasets, which are a subset of the dataset `reuters` (Lewis, 1987) for news document classification and a subset of the dataset `spc` (Pang & Lee, 2005) for sentiment polarity classification of movie review sentences. Based on large sets of noisy labels resulting from the datasets' labeling campaigns, we follow the ideas of Wei et al. (2022) and Herde et al. (2024a) by introducing variants of these noisy label sets. These variants simulate different levels of crowdworker performance and varying amounts of label redundancy. For each instance, we either retain only the labels produced by the `worst` crowdworkers, i.e., we keep false labels if any are available, or we select labels uniformly at `random`. The suffixes `-1`, `-2`, and `-v` then specify how many labels to keep per instance: exactly one, exactly two, or a variable number, respectively. Because we preserve only a subset of all submitted labels, the total number of crowdworkers contributing labels can change across variants. The variant `full` refers to the originally published set of class labels from crowdworkers. Together, these dataset variants cover a wide range of different LFC settings. Specifically, the number of crowdworkers ranges from a small group of $M = 20$ people to a large group of $M = 203$ people. Label noise, defined as the fraction of erroneous labels from crowdworkers, ranges from approximately 20 % to 87 %. When these labels are aggregated via majority voting, the resulting aggregation noise, i.e., the fraction of aggregated labels that are incorrect, ranges from circa 11 % to 87 %. Finally, the dataset variants encompass scenarios ranging from no label redundancy, i.e., only one class label per instance, to those exhibiting substantial label redundancy, i.e., over five class labels per instance.

**Neural Network Architectures**   The original audio files are unavailable for the crowd-labeled dataset `mgc`. Instead, Rodrigues et al. (2013) published features extracted via a music information retrieval tool. Similarly, only term counts published by Rodrigues et al. (2017) are available for the crowd-labeled dataset `reuters` for which we apply a *term frequency-inverse document frequency* (TF-IDF) transformation. As a result, the instances for these two datasets correspond to plain feature vectors. Thus, we employ *multi-layer perceptrons* (MLPs) as the data classification model $f_\theta$. Apart from the input dimension, which depends on the respective dataset, the MLPs share two hidden layers (256 and 128 neurons) enhanced by batch normalization (Ioffe & Szegedy, 2015) and *rectified linear unit* (ReLU, Glorot et al., 2011) activation functions. For all image datasets, where the actual images with their associated noisy class labels from the crowdworkers are published, we employ a DINOv2 vision transformer (vit-s/14, Oquab et al. (2023)) as a backbone model. Analogously, we use an MPNet sentence transformer (all-mpnet-base-v2, Song et al., 2020; Reimers & Gurevych, 2019) as the backbone for the sentences of the dataset `spc`. Both backbones' pre-trained weights remain fixed to preserve the robust feature representations as inputs to an MLP head as the data classification model $f_\theta$ with the same architecture (apart from the input dimensions) as for the other datasets.

**Learning from Crowds Approaches**   Table 1 lists the 13 LFC approaches evaluated in our study. We focus on one-stage LFC approaches, whose end-to-end training has yielded state-of-the-art performances (Nguyen et al., 2024; Herde et al., 2024b; Ibrahim et al., 2023). Of this type, we include approaches that model

Table 2: **Dataset overview.** The first column indicates the names of the datasets, while the remaining columns refer to the datasets' attributes. We denote counts by the # symbol, fractions by the % symbol, and means are supplemented by standard deviations.

| Dataset | Variant | Labeling Campaign | Training Instances [#] | Test Instances [#] | Classes [#] | Workers [#] | Labels per Instance [#] | Label Noise [%] | Aggregation Noise [%] |
|---|---|---|---|---|---|---|---|---|---|
| | | | | Audio Data | | | | | |
| | worst-1 | | | | | 32 | $1.0_{\pm 0.0}$ | 87.4 | 87.4 |
| | worst-2 | | | | | 37 | $1.9_{\pm 0.3}$ | 72.5 | 69.4 |
| | worst-v | | | | | 42 | $2.5_{\pm 1.6}$ | 59.2 | 58.6 |
| mgc | rand-1 | Rodrigues et al. | 700 | 300 | 10 | 37 | $1.0_{\pm 0.0}$ | 47.1 | 47.1 |
| | rand-2 | | | | | 43 | $1.9_{\pm 0.3}$ | 45.7 | 43.9 |
| | rand-v | | | | | 43 | $2.6_{\pm 1.6}$ | 44.6 | 38.3 |
| | full | | | | | 44 | $4.2_{\pm 2.0}$ | 44.0 | 30.3 |
| | | | | Image Data | | | | | |
| | worst-1 | | | | | 57 | $2.5_{\pm 0.6}$ | 41.1 | 41.1 |
| | worst-2 | | | | | 59 | $2.0_{\pm 0.2}$ | 30.8 | 30.1 |
| | worst-v | | | | | 59 | $1.8_{\pm 0.8}$ | 31.6 | 32.5 |
| label-me | rand-1 | Rodrigues et al. | 1,000 | 1,188 | 8 | 57 | $1.0_{\pm 0.0}$ | 23.9 | 23.9 |
| | rand-2 | | | | | 59 | $2.0_{\pm 0.2}$ | 25.5 | 25.7 |
| | rand-v | | | | | 59 | $1.8_{\pm 0.8}$ | 25.5 | 25.0 |
| | full | | | | | 59 | $2.5_{\pm 0.6}$ | 26.0 | 23.7 |
| | worst-1 | | | | | | $1.0_{\pm 0.0}$ | 77.6 | 77.6 |
| | worst-2 | | | | | | $2.0_{\pm 0.0}$ | 62.7 | 62.2 |
| | worst-v | | | | | | $3.0_{\pm 1.4}$ | 45.2 | 46.9 |
| dopanim | rand-1 | Herde et al. | 10,484 | 4,500 | 15 | 20 | $1.0_{\pm 0.0}$ | 32.5 | 32.5 |
| | rand-2 | | | | | | $2.0_{\pm 0.0}$ | 32.8 | 33.2 |
| | rand-v | | | | | | $3.0_{\pm 1.4}$ | 32.7 | 26.3 |
| | full | | | | | | $5.0_{\pm 0.2}$ | 32.7 | 19.3 |
| | | | | Text Data | | | | | |
| | worst-1 | | | | | | $1.0_{\pm 0.0}$ | 69.2 | 69.2 |
| | worst-2 | | | | | | $2.0_{\pm 0.2}$ | 54.0 | 54.0 |
| | worst-v | | | | | | $2.0_{\pm 1.0}$ | 50.8 | 51.8 |
| reuters | rand-1 | Rodrigues et al. | 1,786 | 4,217 | 8 | 38 | $1.0_{\pm 0.0}$ | 38.5 | 38.5 |
| | rand-2 | | | | | | $2.0_{\pm 0.2}$ | 39.9 | 40.9 |
| | rand-v | | | | | | $2.0_{\pm 1.0}$ | 40.8 | 38.4 |
| | full | | | | | | $3.0_{\pm 1.0}$ | 40.4 | 35.5 |
| | worst-1 | | | | | 185 | $1.0_{\pm 0.0}$ | 63.4 | 63.4 |
| | worst-2 | | | | | 199 | $2.0_{\pm 0.0}$ | 47.1 | 47.0 |
| | worst-v | | | | | 202 | $3.2_{\pm 1.6}$ | 31.6 | 32.5 |
| spc | rand-1 | Rodrigues et al. | 3,000 | 1,999 | 2 | 184 | $1.0_{\pm 1.0}$ | 21.2 | 21.2 |
| | rand-2 | | | | | 200 | $2.0_{\pm 0.0}$ | 20.8 | 20.6 |
| | rand-v | | | | | 202 | $3.3_{\pm 1.6}$ | 21.1 | 14.9 |
| | full | | | | | 203 | $5.5_{\pm 0.7}$ | 20.9 | 11.0 |

class-dependent and instance-dependent crowdworker performances. Further, we consider two two-stage approaches, of which `mv` serves as a lower baseline because it trains the data classification model $f_\theta$ on majority vote labels. Implementing the crowdworker classification model $g_\phi$ and the crowdworker performance model $h_\psi$ depends on the respective LFC approach, of which Appendix A provides more detailed descriptions.

**Hyperparameter Search** Table 1 lists general (approach-agnostic) HPs and approach-specific HPs. All approaches share the general HPs. Specifically, we fix RAdam (Liu et al., 2019) as the optimizer, combined with a cosine annealing learning rate scheduler (Loshchilov & Hutter, 2017) without restarts to gradually reduce the learning rate throughout the training process, thereby promoting stable convergence. For the remaining general HPs, we define suitable search spaces derived from related literature and default values of PyTorch (Paszke et al., 2019) optimizers (e.g., no weight decay). The distributions for sampling HP values are denoted as `uniform` and `log-uniform`. For approach-specific HPs, we adopt default values from the publications or codebases. If available, the search spaces are also extracted from these two sources. Otherwise, they are defined based on reasonable value ranges. The defined HP search spaces are sampled using Sobol sequences (Sobol, 1998) as HP search strategy (see Section 3.3 for our rationale). A total of 50 distinct HPCs are generated per combination of LFC approach and dataset variant. Together with the default HPC, this yields a set of $|\Lambda| = 51$ HPCs, from which the HPS criteria tries to choose the best performer.

**Hyperparameter Selection Criteria** Each HPC is evaluated via a $K = 5$-fold cross-validation given the crowd-labeled training set to obtain risk estimates for the respective HPS criterion. The HPC chosen by

Table 3: **Overview of LFC approaches' general and individual HP search spaces.** For each HP, we define a default value and a search space as the basis for the HPO. The notation *not applicable* (N/A) indicates that an LFC approach does not introduce additional HPs or that an HP is not optimized. The expressions `uniform` and `log-uniform` define the search spaces as distributions used for generating HPCs.

| Approach | Reference | Hyperparameter | Default Value | Search Space |
|---|---|---|---|---|
| General | N/A | optimizer | RAdam | N/A |
| | | learning rate scheduler | cosine annealing | N/A |
| | | number of epochs | 30 | $\texttt{uniform}(\{5, 30, 50\})$ |
| | | batch size | 32 | $\texttt{uniform}(\{16, 32, 64\})$ |
| | | initial learning rate | $10^{-3}$ | $\texttt{loguniform}([10^{-4}, 10^{-1}])$ |
| | | weight decay | 0 | $\texttt{loguniform}([10^{-6}, 10^{-3}])$ |
| | | dropout rate | 0.0 | $\texttt{uniform}([0.0, 0.5])$ |
| *Two-stage Approach without Crowdworker Performance Modeling* | | | | |
| `mv` | N/A | N/A | N/A | N/A |
| *Two-stage Approach with Class-dependent Crowdworker Performance Modeling* | | | | |
| `ds` | Dawid & Skene | N/A | N/A | N/A |
| *One-stage Approaches with Class-dependent Crowdworker Performance Modeling* | | | | |
| `cl` | Rodrigues & Pereira | N/A | N/A | N/A |
| `trace` | Tanno et al. | confusion matrix regularization ($\lambda$) | $10^{-2}$ | $\texttt{loguniform}([10^{-3}, 10^{-1}])$ |
| `conal` | Chu et al. | confusion matrix regularization ($\lambda$) | $10^{-5}$ | $\texttt{loguniform}([10^{-6}, 10^{-3}])$ |
| | | embedding dimension | 20 | $\texttt{uniform}(\{20, 40, 60, 80\})$ |
| `union-a` `union-b` | Wei et al. | confusion matrix initialization ($\epsilon$) | $10^{-5}$ | $\texttt{loguniform}([10^{-6}, 10^{-4}])$ |
| `geo-f` `geo-w` | Ibrahim et al. | confusion matrix regularization ($\lambda$) | $10^{-3}$ | $\texttt{loguniform}([10^{-4}, 10^{-2}])$ |
| *One-stage Approaches with Instance-dependent Crowdworker Performance Modeling* | | | | |
| `madl` | Herde et al. | confusion matrix initialization ($\eta$) | 0.8 | $\texttt{uniform}([0.75, 0.95])$ |
| | | gamma distribution parameter ($\alpha$) | 1.25 | $\texttt{uniform}([1.0, 1.5])$ |
| | | gamma distribution parameter ($\beta$) | 0.25 | $\texttt{uniform}([0.25, 0.5])$ |
| | | embedding dimension ($Q$) | 16 | $\texttt{uniform}(\{8, 16, 32\})$ |
| `crowd-ar` | Cao et al. | loss balancing | 0.9 | $\texttt{uniform}([0.5, 1.0])$ |
| `annot-mix` | Herde et al. | confusion matrix initialization ($\eta$) | 0.9 | $\texttt{uniform}([0.75, 0.95])$ |
| | | mixup ($\alpha$) | 1.0 | $\texttt{uniform}([0.0, 2.0])$ |
| `coin` | Nguyen et al. | outlier regularization ($\mu_1$) | $10^{-2}$ | $\texttt{loguniform}([10^{-3}, 10^{-1}])$ |
| | | volume regularization ($\mu_2$) | $10^{-2}$ | $\texttt{loguniform}([10^{-3}, 10^{-1}])$ |

the respective criterion (see Eq. (8)) is then tested on the hold-out test set with true labels for five different initializations of the neural networks' weights, ensuring a reliable assessment of the data classification model's generalization performance. We include two variants of default HPCs. On the one hand, we use the default values specified in Table 3 across all datasets. This criterion, to which we refer as DEF (see Eq. (10)), is the most naive one since there is no consideration of the datasets' individual requirements. A more advanced and commonly used alternative in LFC evaluation studies is to fix one default HPC per dataset, denoted DEF-DATA. For each dataset variant, we first perform a conventional HPO: a vanilla classification model, ignoring the LFC setting, is trained and validated on the true class labels, tuning only the approach-agnostic HPs (Herde et al., 2024a). The best HPC obtained from this search is then frozen and transferred to every LFC approach. Consequently, every approach shares the same general HPC for that dataset variant while its approach-specific HPs stay at their default values. For well-studied benchmarks, one could alternatively adopt HPCs reported in the literature (Tanno et al., 2019), yet we rely on the HPO variant as DEF-DATA criterion to guarantee consistency across datasets. This DEF-DATA criterion differs from our upper baseline criterion TRUE, where all HPs of an LFC approach are optimized by training on the noisy crowd-labeled data and validating on the true labels (see Eq. (9)). Despite this difference, DEF-DATA and TRUE are the only HPS criteria requiring access to the true validation labels.

## 4.2 Experimental Results

Our setup encompasses 5 datasets, each with 7 variants, 13 LFC approaches, and 11 HPS criteria. Due to the numerous combinations, we present only the main results addressing our two research questions here and refer to Appendix C.2 for the complete list of results. These supplementary results also contain more in-depth analyses regarding the impact of different noise levels, numbers of candidate HPCs, and an architecture search for the data classification model.

**RQ₁**: *Given access only to crowd-labeled validation data, which evaluated HPS criterion yields the highest performance for LFC approaches?*

We compare the HPS criteria with each other across multiple datasets and LFC approaches. Therefore, we report performance results in the form of rankings and zero-one losses in Table 4. In addition, this table lists for each criterion the number of significant wins, significant losses, and non-significant comparisons (ties) against all other criteria. We treated each pair of dataset variant and LFC approach as a block and applied a Friedman test (Friedman, 1937) at the significance level $\alpha := 0.05$, which was significant. Thus, we performed all pairwise Wilcoxon signed-rank tests with Holm correction (Holm, 1979). For each pair of criteria, the comparison was counted as a win (or loss) for a criterion if the adjusted $p$-value was below $\alpha$ (with the sign of the performance difference determining win versus loss). Otherwise, the comparison was counted as a tie. The complete table of pairwise comparisons is given in Appendix C.2. One central finding is that the criterion DEF significantly ranks poorly, supporting our motivation that using default HPCs leads to underestimating the approaches' potential performances. Moreover, the poor rank of the DEF-DATA criterion indicates that using well-tuned HPCs from standard classification tasks (without any noisy labels) does not account for the unique requirements of each LFC approach. As anticipated, the TRUE criterion, which utilizes the true class labels for selecting the best HPC per LFC approach and per dataset, performs superiorly to

Table 4: **HPS criteria's results.** One column per criterion reports the rank, number of significant wins, significant losses, non-significant ties compared to the other criteria, and the zero-one loss reductions (absolute as percentage points [%$_p$] and relative as percentages [%]) compared to DEF as criterion. Means and standard errors are computed over all combinations of dataset variants and LFC approaches (excluding the approach `mv` that is not compatible with each criterion). The arrows show whether a smaller (↓) or higher (↑) value is better. The **best** and second best values are marked per result type. A ⋆ marks criteria that had access to the true validation labels.

| | Baseline | | | Aggregation-level | | | | Crowd-level | | | Ensemble |
|---|---|---|---|---|---|---|---|---|---|---|---|
| TRUE⋆ | DEF-DATA⋆ | DEF | AEU | AEC | ALU | ALC | CXU | CEC | CLC | ENS |
| | | | | | Ranks (↓) | | | | | | |
| **4.64** | 7.84 | 9.76 | 5.66 | 5.57 | 5.29 | 5.60 | 5.46 | 5.36 | 5.91 | 4.89 |
| ± 0.15 | ± 0.17 | ± 0.11 | ± 0.11 | ± 0.11 | ± 0.11 | ± 0.13 | ± 0.11 | ± 0.11 | ± 0.13 | ± 0.09 |
| | | | Significant Wins [#] (↑) / Non-significant Ties [#] / Significant Losses [#] (↓) | | | | | | | | |
| **9 / 1 / 0** | 1/0/9 | 0/0/10 | 2/6/2 | 2/6/2 | 3/5/2 | 2/6/2 | 2/6/2 | 2/6/2 | 2/5/3 | **9 / 1 / 0** |
| | | | Absolute Zero-one Loss Reductions Compared to DEF ($\Delta_{\text{DEF}} L_{0/1}$ [%$_p$] ↑) | | | | | | | | |
| + **5.36** | + 0.76 | + 0.00 | + 4.55 | + 4.59 | + 4.73 | + 4.43 | + 4.52 | + 4.57 | + 4.07 | + 5.15 |
| ± 0.24 | ± 0.31 | ± 0.00 | ± 0.26 | ± 0.26 | ± 0.26 | ± 0.27 | ± 0.29 | ± 0.28 | ± 0.31 | ± 0.26 |
| | | | Relative Zero-one Loss Reductions Compared to DEF ($\Delta_{\text{DEF}} L_{0/1}$ [%] ↑) | | | | | | | | |
| +**18.20** | + 3.65 | + 0.00 | +16.18 | +16.27 | +16.48 | +14.66 | +15.48 | +15.93 | +13.30 | +17.60 |
| ± 0.73 | ± 1.12 | ± 0.00 | ± 0.76 | ± 0.76 | ± 0.80 | ± 0.81 | ± 0.87 | ± 0.84 | ± 0.91 | ± 0.77 |

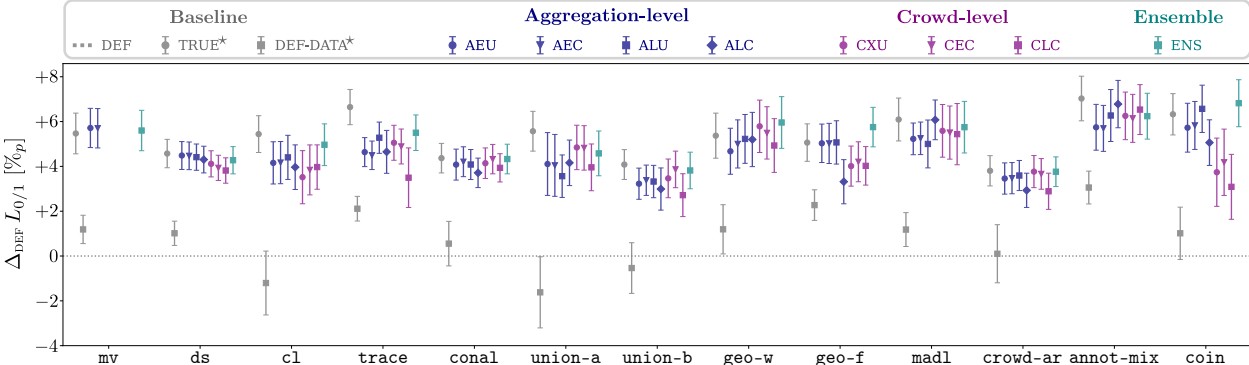

Figure 5: **Absolute zero-one loss reductions of HPS criteria.** For each LFC approach ($x$-axis), the scatter plot displays the mean and standard error of a criterion's reduction in zero-one loss ($y$-axis) as percentage points [%$_p$] compared to the criterion DEF (see Appendix C.2 for relative reductions as percentages [%]). Higher reductions correspond to greater improvements. A ⋆ marks criteria that had access to the true validation labels.

Table 5: **LFC approaches' results with** ENS **as HPS criterion.** One column per approach reports the rank, number of significant wins, significant losses, non-significant ties compared to the other approaches and the zero-one loss reductions (absolute as percentage points [$\%_p$] and relative as percentages [%]) compared to the approach `mv` trained with its default (DEF) HPC. Means and standard errors are computed over all dataset variants. The arrows indicate whether a smaller (↓) or higher (↑) value is better. The **best** and second best values are marked per result type.

| Baseline | | Class-dependent | | | | | | | Instance-dependent | | | |
|---|---|---|---|---|---|---|---|---|---|---|---|---|
| mv | ds | cl | trace | conal | union-a | union-b | geo-w | geo-f | madl | crowd-ar | annot-mix | coin |
| Ranks (↓) | | | | | | | | | | | | |
| 10.57 | 9.09 | 7.90 | 7.31 | 7.37 | 7.60 | 7.61 | 5.81 | 4.10 | 5.36 | 8.97 | 5.34 | **3.96** |
| ± 0.38 | ± 0.53 | ± 0.64 | ± 0.53 | ± 0.53 | ± 0.67 | ± 0.55 | ± 0.59 | ± 0.45 | ± 0.63 | ± 0.43 | ± 0.65 | ± 0.50 |
| Significant Wins [#] (↑) / Non-significant Ties [#] / Significant Losses [#] (↓) | | | | | | | | | | | | |
| 0/3/9 | 0/7/5 | 0/11/1 | 1/9/2 | 1/9/2 | 0/10/2 | 1/8/3 | 4/8/0 | **8 / 4 / 0** | 3/9/0 | 1/6/5 | 3/9/0 | 7/5/0 |
| Absolute Zero-one Loss Reductions Compared to mv with DEF ($\Delta_{\text{mv[DEF]}} L_{0/1}$ [$\%_p$] ↑) | | | | | | | | | | | | |
| + 5.60 | + 6.05 | + 8.04 | + 7.81 | + 7.18 | + 7.05 | + 7.42 | + 9.70 | **+10.05** | + 8.52 | + 6.87 | + 9.21 | + 9.88 |
| ± 0.90 | ± 1.17 | ± 1.19 | ± 1.09 | ± 0.92 | ± 0.97 | ± 1.05 | ± 1.28 | ± 1.08 | ± 1.33 | ± 0.97 | ± 0.93 | ± 0.99 |
| Relative Zero-one Loss Reductions Compared to mv with DEF ($\Delta_{\text{mv[DEF]}} L_{0/1}$ [%] ↑) | | | | | | | | | | | | |
| +19.31 | +21.04 | +23.95 | +24.99 | +24.20 | +21.67 | +23.82 | +28.75 | +30.79 | +27.66 | +22.68 | +28.22 | **+30.86** |
| ± 2.75 | ± 3.14 | ± 2.95 | ± 3.06 | ± 2.83 | ± 2.92 | ± 2.86 | ± 2.99 | ± 2.86 | ± 3.45 | ± 2.87 | ± 2.60 | ± 2.84 |

every other criterion. Concretely, it ranks best and provides the highest loss reductions compared to the DEF criterion across all dataset variants. Among the criteria relying solely on crowd-labeled validation data, the ensemble-based ENS criterion is the clear runner-up: it ranks better and provides higher loss reductions than all other competing criteria apart from the upper baseline criterion TRUE. Notably, ENS is the only criterion without a significant loss compared to TRUE. Reducing the size of the ensemble decreases its performance (see ablation study in Appendix C.2). At the same time, the extra computation demanded by ENS remains negligible (see time complexity analysis in Appendix B.4): the expensive steps are training and inferring the predictions of the models, and once those predictions are available, they can be reused to compute every risk estimate. Finally, although the AEU criterion, which aggregates the noisy validation labels per instance via majority voting, performs worse than the criteria TRUE and ENS, it still yields substantial improvements over default HPCs. Together, these results highlight the benefits of HPO in LFC settings with crowd-labeled validation data. Besides these results averaged over the LFC approaches, we also examine how each criterion performs in combination with each approach individually. For this purpose, Figure 5 breaks down the absolute zero-one loss reductions according to the LFC approaches. This way, we measure how much a specific criterion improves an approach relative to using that approach's default HPC. It does not indicate which LFC approach is superior to others. We observe that for a few LFC approaches, e.g., `coin` and `geo-f`, the criterion TRUE does not achieve the highest loss reduction on average. An explanation is that cross-validation uses only subsets of the data for training, so the HPC that minimizes the validation zero-one loss on the subsets may not be optimal when training is performed on the full dataset. This observation also suggests an interdependence between the criterion and the approach. Another example of this interdependence can be found when comparing the results for the `union-a` approach, where criteria that rely on crowd-level risk estimates outperform those that use aggregation-level risks, to the results for the `coin` approach, where aggregation-level criteria take the lead. Consequently, when it is unclear which criterion will pair best with a given approach, the ensemble-based criterion ENS offers a robust compromise.

> **RQ$_1$: Takeaway**
>
> Aggregation-level and crowd-level HPS criteria, estimating risks only from noisy crowdworker labels, enable effective HPO in LFC. Combining them via the ensemble-based criterion ENS yields the highest performance across dataset variants and LFC approaches.

**RQ$_2$**: *Given the best-evaluated HPS criterion for crowd-labeled validation data, how do LFC approaches compare in performance?*

We compare the LFC approaches whose HPCs have been selected via the best-evaluated criterion ENS across all dataset variants and report the performance results again as rankings and zero-one loss reductions in

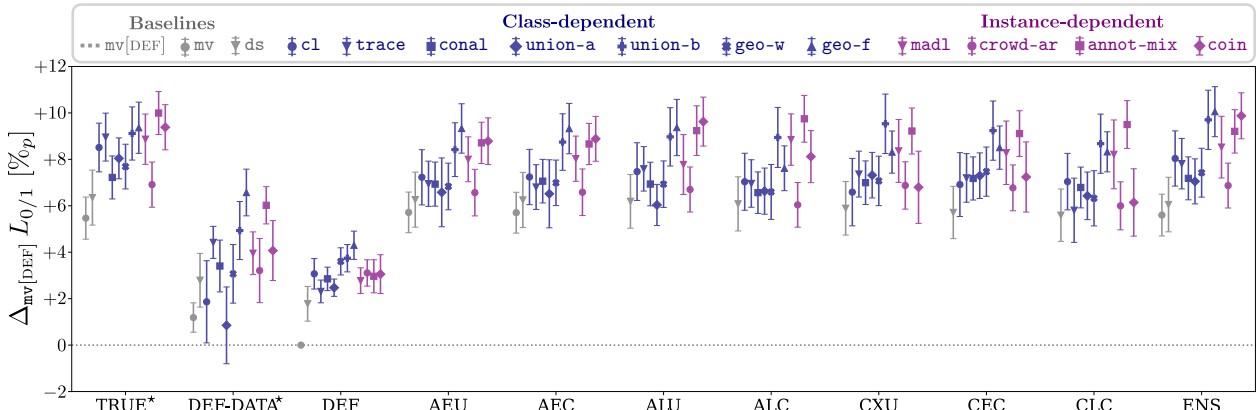

Figure 6: **Absolute zero-one loss reductions of LFC approaches.** For each HPS criterion ($x$-axis), the scatter plot displays the mean and standard error of an LFC approach's reduction in zero-one loss ($y$-axis) as percentage points [%$_p$] compared to majority voting (mv) trained with its default (DEF) HPC (see Appendix C.2 for relative reductions as percentages [%]). Higher reductions correspond to greater improvements. A ⋆ marks criteria that had access to the true validation labels.

Table 5. In addition, this table lists for each approach the number of significant wins, significant losses, and non-significant ties against all other approaches. These numbers are obtained via the same test procedure as in $RQ_1$ with each dataset variant as a block. The results underscore the benefits of one-stage LFC approaches that estimate class- or instance-dependent crowdworker performances, as the two-stage baseline approaches mv and ds (Dawid & Skene, 1979) attain the worst ranks and have the most significant losses. Moreover, they provide the lowest zero-one loss reductions compared to training mv with its default (DEF) HPC. The two one-stage approaches geo-f (Ibrahim et al., 2023) and coin (Nguyen et al., 2024) provide significant performance gains to most of their competitors with an average zero-one loss reduction of around 10%$_p$ corresponding to a relative improvement of over 30% compared to training mv with DEF as a criterion. Both approaches' idea is to identify the crowdworkers' confusion matrices via special regularization terms, where geo-f estimates a single confusion matrix per worker, while coin relaxes this assumption by modeling instance-dependent outlier terms for the crowdworkers' confusion matrices. Figure 6 breaks down the approaches' absolute zero-one loss reductions compared to training mv with the default (DEF) HPC according to the HPS criteria. We observe that similar absolute reductions of the zero-one loss around 10%$_p$ are also achievable when relying on other criteria, such as ALC paired with the annot-mix approach (Herde et al., 2024b). Moreover, the results indicate that the comparison between the approaches' performances is affected by the choice of the criterion. For example, the loss reduction of the approach coin is much inferior to the reduction of the approach annot-mix for the baseline criterion DEF-DATA, but is superior when relying on ENS as a criterion.

To systematically analyze the impact of a criterion on comparing the approaches, we compute the Kendall rank correlation coefficient (Kendall, 1945) to determine whether the various criteria disagree on the ranking of the approaches. Figure 7 shows violin plots of these coefficients across all dataset variants for 6 example criteria pairs. Each coefficient is a kind of distance measurement between two rankings of the approaches obtained after using two different criteria for the same dataset variant. The coefficients are in the interval $[-1, 1]$, where $-1$ indicates a perfect negative ordering, 0 no ordering, and 1 a perfect positive ordering. In the absence of ties, a Kendall $\tau$-$b$ coefficient of 0.50 between two rankings of the approaches means that if you randomly sample a pair of approaches, the probability that both criteria rank the same approach higher than the other exceeds the probability that they disagree by 50%$_p$ (75% vs 25%). The criteria DEF and DEF-DATA paired with ENS or TRUE have an average Kendall $\tau$-$b$ coefficient of around 0.25, showing only modest agreement in the ranking of LFC approaches. Criterion ENS versus criterion TRUE has an average Kendall $\tau$-$b$ coefficient of about 0.40, reflecting a moderate positive overlap of rankings. The violin plots, however, make clear that these average Kendall $\tau$-$b$ coefficients mask considerable dispersion: dataset-wise coefficients spread widely around each average, and some even take negative values, showing that two criteria more often disagree than agree on the ordering of approaches.

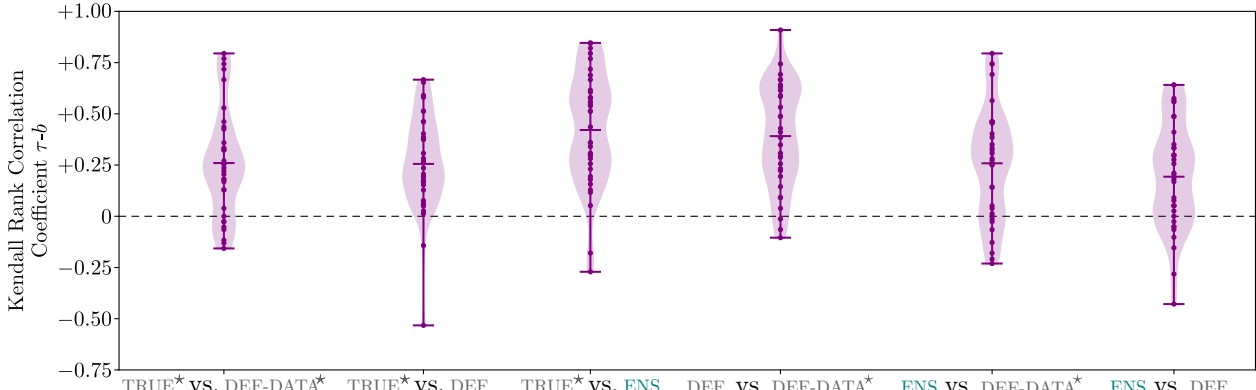

Figure 7: **Rank correlation between HPS criteria.** Each violin plot shows the mean and distribution of pairwise Kendall $\tau$-$b$ coefficients, shown as violet dots and obtained from 35 dataset variants when comparing the ranking of LFC approaches with their HPCs selected via baseline and ensemble-based criteria. Higher coefficients indicate stronger correlation. A $\star$ marks criteria that had access to the true validation labels.

---

**RQ$_2$: Takeaway**

With the HPS criterion ENS, all one-stage LFC approaches, particularly `geo-f` and `coin`, outperform the two-stage baselines. The approaches' gains and rankings vary with the criterion, underscoring the importance of a criterion for a fair and realistic evaluation.

---

### 4.3 Hyperparameter Optimization in Learning from Crowds: A Guideline

Based on `crowd-hpo` with its experimental study and related experimental studies (see Section 2), we propose a **guideline** to perform HPO given a (new) crowd-labeled dataset. Figure 8 summarizes this guideline as a flowchart that can be used both by researchers, who design and evaluate LFC approaches on benchmark datasets, and by practitioners, who apply an LFC approach to a crowd-labeled target dataset. The flowchart is to be applied to each pair of a target dataset and an LFC approach to obtain an optimized HPC. Any subsequent steps conducted depend on the specific objectives of the application or research context.

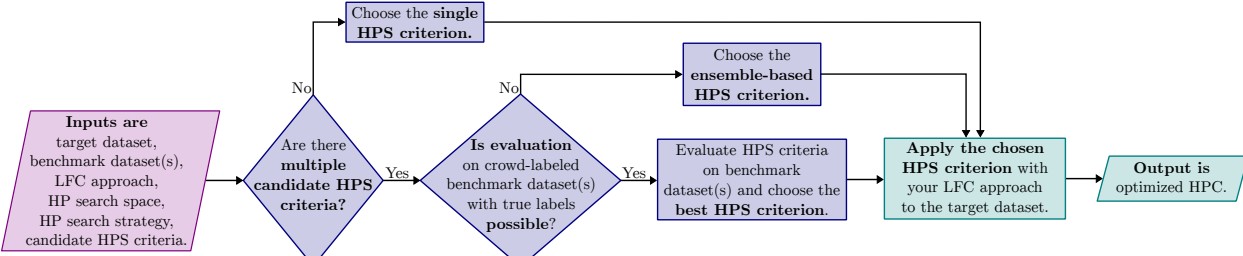

Figure 8: **Guideline to perform HPO for a (new) crowd-labeled dataset.** If only one candidate HPS criterion is available, it is applied directly. If multiple criteria exist, they are either evaluated on benchmark datasets (when available) to choose the best HPS criterion (which may itself be an ensemble) or, in the absence of suitable benchmarks, combined into an ensemble. The chosen HPS criterion is then used to optimize the HPC on the target dataset.

For each run of the flowchart, we need to fix the following **inputs**:

- *Target dataset:* For practitioners, the crowd-labeled target dataset is typically fixed by the application, and the flowchart is run once for a chosen LFC approach to obtain an optimized HPC. Researchers may consider multiple benchmark datasets and various LFC approaches, and therefore run the flowchart

repeatedly, treating each benchmark in turn as the target dataset, while the remaining benchmarks serve as auxiliary datasets.

- *Benchmark datasets:* Both practitioners and researchers may use crowd-labeled benchmark datasets with additional true labels to compare candidate HPS criteria. When curating benchmarks, we recommend datasets with noisy class labels from human crowdworkers (see Table 2) rather than purely simulated crowdworkers. Where possible, multiple noisy label variants can be derived from these datasets (Wei et al., 2022; Herde et al., 2024a) to study different noise levels and label redundancies.

- *Learning from crowds approach:* From a practitioner's perspective, an LFC approach is ideally state-of-the-art and empirically robust (see results in Table 5 for orientation). In contrast, a researcher repeatedly applies the flowchart to cover multiple LFC approaches that differ in their training principles and crowdworker performance modeling, enabling systematic comparisons. For example, including both two-stage and one-stage training approaches (Li et al., 2022) allows us to contrast label aggregation as a separate step with end-to-end training. Likewise, comparing methods that model class-dependent and instance-dependent crowdworker performance (Herde et al., 2023) reveals when the additional modeling complexity is beneficial.

- *Hyperparameter search space and strategy:* The efficiency and effectiveness of HPO depend on the HP search space and strategy (Bergstra & Bengio, 2012). For example, overly broad spaces waste budget and obscure findings. Practitioners commonly favor tight, well-motivated spaces around settings known to work robustly, combined with simple strategies such as random search or Sobol sequences. Researchers may explore richer spaces, but should still focus on HPs that are critical for the behavior of the LFC approach (see Table 3 for orientation). We therefore advocate that new LFC approaches provide concrete recommendations for their approach-specific search spaces. When such guidance is unavailable, reasonable ranges should be derived from theoretical considerations and the functional role of each HP rather than being chosen arbitrarily.

- *Candidate hyperparameter selection criteria:* All HPS criteria that are compatible with the chosen LFC approach may serve as candidates. Relying on default (DEF) HPCs typically results in suboptimal performance of an LFC approach. For a target dataset, where assuming the availability of a separate validation set with true labels is reasonable, validating with those labels (corresponding to the criterion TRUE) is fine. Otherwise, criteria designed for crowd-labeled validation data, such as those provided by `crowd-hpo`, are to be used. In practice, the optimal HPS criterion depends on the LFC approach and target dataset, so we recommend including at least one simple crowd-based criterion (e.g., AEU) and a robust ensemble-based criterion (e.g., ENS) among the candidates. In the future, new LFC approaches are ideally introduced in conjunction with HPS criteria tailored to their specific characteristics, reflecting that HPO under noisy validation labels is a central challenge in related settings, such as partial label learning (Wang et al., 2025).

After fixing the inputs, we must **choose an HPS criterion**. There is only a single candidate HPS criterion when, for example, prior evidence already identifies a suitable choice for the given LFC approach, or when TRUE is applicable as an HPS criterion because true validation labels are available for the target dataset. If multiple candidate HPS criteria are available, an extra choice is required. When suitable crowd-labeled benchmark datasets with true test labels or sufficient computational resources are unavailable, we recommend using the ensemble-based criterion ENS as the default. Otherwise, one may instead compare the candidate criteria on these benchmarks, as in our study: run HPO for each criterion on the benchmark datasets and select the one that, on average, yields the best test performance, e.g., the lowest mean zero-one loss[5].

Finally, we obtain an **optimized HPC** by running HPO on the target dataset, applying the chosen HPS criterion. Specifically, for each candidate HPC, the LFC approach is trained on crowd-labeled training data and evaluated on (mostly crowd-labeled) validation data using the chosen HPS criterion. Practitioners typically perform this procedure once on their application dataset for deployment, whereas researchers repeat it across benchmark datasets and LFC approaches as the basis for comparative analyses.

---

[5]This procedure does not necessarily require separate HPO runs with model retrainings: when using static HP search strategies, e.g., Sobol sequences, grid search, or random search, multiple HPS criteria can be evaluated on the same set of tried HPCs, so only the decision which HPC is best under the chosen HPS criterion needs to be recomputed.

## 5   Limitations

Our three core contributions, i.e., the proposed HPS criteria (see Section 3.4), the experimental study (see Section 4.2), and the guideline for HPO in LFC settings (see Section 4.3), are subject to limitations.

These proxy HPS criteria, whose underlying empirical risk estimates access only crowd-labeled validation data, offer no theoretical consistency guarantees, i.e., that they will asymptotically select the HPC minimizing the expected risk of an LFC approach. In general, the true validation labels are not identifiable from crowd labels alone: different combinations of latent true labels and crowdworker performance can induce the same distribution over observed noisy labels (Tanno et al., 2019). Any general guarantee, therefore, requires explicit structural assumptions on the noise in the labels, e.g., crowdworkers being better than random or identifiable noise models. As we do not impose such assumptions, we can not claim universal guarantees. Instead, we provide a comprehensive empirical evaluation of the proposed HPS criteria as practical proxies for validation performance with true labels on real crowd-labeled datasets.

Our **experimental study** does not investigate aspects, such as the HPO search strategy, the number of folds $K$ for cross-validation, and fine-tuning pre-trained backbones. These aspects impact the computational complexity of HPO. Furthermore, our analysis based on computing means across all datasets does not account for the influence of specific dataset attributes, including the number of crowdworkers and label redundancy. While our observations refer to the zero-one loss function, alternative loss functions such as the Brier score (Brier, 1950) may also be relevant when assessing probabilistic estimates. An initial analysis (see Appendix C.2) confirms the benefit of employing the proposed HPS criteria for the Brier score as the target loss function, while the choice of the best HPS criterion changes. Finally, we refer to the number of candidate HPCs $|\Lambda|$ as the computational budget, which does not account for different training and inference complexities between the LFC approaches.

Our **guideline** for HPO in LFC settings is limited to empirically motivated advice on how to choose an HPS criterion for a given LFC approach and (new) target dataset. In particular, selecting an HPS criterion based on experiments with related benchmark datasets does not guarantee that this criterion will also be optimal on the target dataset. A possible direction for future work is to exploit latent structure in dataset characteristics to inform the choice of HPS criterion, or to adaptively tune the weights in the ensemble-based criterion ENS using meta-learning techniques (Brazdil et al., 2022). Finally, while the guideline outputs an optimized HPC, it does not address the practical requirement for a reliable, independent test set to confirm the classification model's performance before deployment.

## 6   Conclusion

We introduced `crowd-hpo` as a framework to enable more realistic and fairer benchmarking of LFC approaches by leveraging HPO with only access to crowd-labeled validation data. We started with exemplary results demonstrating notable zero-one loss reductions and changes in the rankings of LFC approaches when performing HPO with true class labels in the validation set compared to default HPCs. Subsequently, we identified a lack of research regarding HPO with crowd-labeled validation data. Therefore, we proposed and evaluated HPS criteria accounting for the potential noise in class labels from crowdworkers. Across extensive experiments, our proposed HPS criteria strongly reduced the losses of the LFC approaches relative to their default HPCs. This applies particularly to the ensemble-based criterion ENS, which is also easily extensible by including future empirical risk measures for crowd-labeled validation data. However, the ranking of LFC approaches shifted with the criterion applied. These findings grounded our guideline for future experimentation and benchmarking in LFC settings. To further improve the HPS, future work should rigorously explore advanced HP search strategies, particularly Bayesian optimization (Wang et al., 2023), and examine how they interact with criteria accounting for crowd-labeled validation data. Progress can also be achieved through HPS criteria with alternative loss functions, label weighting, label aggregation, or resampling schemes, e.g., by allowing per-class crowdworker biases during validation. Finally, `crowd-hpo` serves as a starting point for developing empirical risk measures that are not only suited for improving HPS given crowd-labeled validation data but ultimately should provide reliable estimates of the approaches' generalization performances given only crowd-labeled test data.

**Broader Impact Statement**

In a broader context, we identify an impact on real-world applications and crowdworkers as two branches of potential societal consequences of `crowd-hpo`. (1) On the one hand, the validation with noisy class labels from the crowd makes the LFC approaches' employment more practical in real-world applications because potential users do not need to rely on default or manually picked HPCs when training the LFC approaches. However, selecting the best HPC is different from accurate risk estimates. In the first case, we only need to rank the HPCs, while in the latter case, we require a precise estimate of the risk associated with the resulting data classification model. In other words, despite being able to optimize the selection of the HPC, we do not know the actual generalization performance of the data classification model after training. Correspondingly, there is still no solution to how practitioners can obtain such an estimate without access to a separate test set with true labels. Therefore, practitioners must only employ LFC approaches in safety-critical applications after thorough evaluation based on a separate test set with true labels. (2) On the other hand, wider adoption of LFC approaches can raise demand for crowdworkers. However, crowdworkers often endure difficult working conditions (Bhatti et al., 2020), e.g., insufficient payments and job insecurity. Therefore, collecting crowd-labeled data for training LFC approaches should always be coupled with explicit provisions for fair working conditions.

**Acknowledgements**

This work was funded by the ALDeep (P/681) and CIL (P/710, P/1082) projects through the University of Kassel.

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

## A  Inference Mechanisms of Learning from Crowds Approaches

This appendix overviews the common inference mechanisms of LFC approaches to better understand the connections between the data classification model $f_{\boldsymbol{\theta}}$, the crowdworker classification model $g_{\boldsymbol{\phi}}$, and the crowdworker performance model $h_{\boldsymbol{\psi}}$. Moreover, the probabilistic estimates of Eqs. (3)-(5) are required to evaluate our presented HPS criteria. To explain the inference, we distinguish between two architecture types employed by LFCs approaches: the ones with probabilistic confusion matrices and those with non-probabilistic noise adaptation layers.

## A.1 Probabilistic Confusion Matrices

Many LFC approaches (Dawid & Skene, 1979; Tanno et al., 2019; Ibrahim et al., 2023; Cao et al., 2023; Herde et al., 2023; 2024b; Nguyen et al., 2024) estimate crowdworker performances through confusion matrices, which we formalize as a function $\boldsymbol{Q_\beta} : \Omega_X \times [M] \to \Delta_C^C$ with parameters $\boldsymbol{\beta} \in \Omega_B$. Thereby, a confusion matrix entry has the following probabilistic interpretation:

$$[\boldsymbol{Q_\beta}(\boldsymbol{x}_n, m)]_{c,k} \coloneqq \Pr\left(\boldsymbol{z}_{nm} = \boldsymbol{e}_k | \boldsymbol{x}_n, \boldsymbol{y}_n = \boldsymbol{e}_c, m, \boldsymbol{\beta}\right). \tag{30}$$

This confusion matrix entry in row $c \in [C]$ and column $k \in [C]$ is the probability that crowdworker $m$ assigns the class label $\boldsymbol{e}_k$ to instance $\boldsymbol{x}_n$ with $\boldsymbol{e}_c$ as its true class label. Depending on the assumptions of the LFC approach, there are confusion matrices differing in their degrees of freedom $\nu \in \mathbb{N}_{>0}$ (Herde et al., 2023). Here, we distinguish between class-independent ($\nu = 1$) and class-dependent ($\nu = C(C-1)$) confusion matrices. Moreover, the confusion matrices can be modeled as instance-independent:

$$\forall \boldsymbol{x}_n, \boldsymbol{x}_l \in \Omega_X : \boldsymbol{Q_\beta}(\boldsymbol{x}_n, m) = \boldsymbol{Q_\beta}(\boldsymbol{x}_l, m), \tag{31}$$

or as an instance-dependent function. Despite different assumptions about confusion matrices, the LFC approaches share the inference schemes for their crowdworker classification model with parameters $\boldsymbol{\phi} \coloneqq (\boldsymbol{\theta}, \boldsymbol{\beta})$:

$$\boldsymbol{g_\phi}(\boldsymbol{x}_n, m) \coloneqq \boldsymbol{Q}_{\boldsymbol{\beta}}^{\mathrm{T}}(\boldsymbol{x}_n, m) \boldsymbol{f_\theta}(\boldsymbol{x}_n) \tag{32}$$

and for their crowdworker performance model with parameters $\boldsymbol{\psi} \coloneqq (\boldsymbol{\theta}, \boldsymbol{\beta})$:

$$h_{\boldsymbol{\psi}}(\boldsymbol{x}_n, m) \coloneqq \sum_{c \in [C]} [\boldsymbol{f_\theta}(\boldsymbol{x}_n)]_c \cdot [\boldsymbol{Q_\beta}(\boldsymbol{x}_n, m)]_{c,c}. \tag{33}$$

## A.2 Non-probabilistic Noise Adaptation Layers

In contrast to probabilistic confusion matrices, we refer to noise adaptation layers in LFC approaches (Rodrigues & Pereira, 2018; Chu et al., 2021; Wei et al., 2023) as unconstrained transformations of the estimated true class probabilities. For this purpose, the approach `cl` (Rodrigues & Pereira, 2018) introduces a noise adaptation layer $\boldsymbol{W}_m \in \mathbb{R}^{C \times C}$ for each crowdworker $m \in [M]$. Then, the crowdworker classification model with parameters $\boldsymbol{\phi} \coloneqq (\boldsymbol{\theta}, \boldsymbol{W}_1, \dots, \boldsymbol{W}_M)$ performs inference via:

$$\boldsymbol{g_\phi}(\boldsymbol{x}_n, m) \coloneqq \mathrm{softmax}\left(\boldsymbol{W}_m^{\mathrm{T}} \boldsymbol{f_\theta}(\boldsymbol{x}_n)\right). \tag{34}$$

The approach `conal` (Chu et al., 2021) extends the crowdworker-specific noise adaptation layers by another layer $\overline{\boldsymbol{W}} \in \mathbb{R}^{C \times C}$ modeling common confusions across crowdworkers, which leads to:

$$\boldsymbol{g_\phi}(\boldsymbol{x}_n, m) \coloneqq (1 - s_{\boldsymbol{\gamma}}(\boldsymbol{x}_n, m)) \cdot \mathrm{softmax}\left(\boldsymbol{W}_m^{\mathrm{T}} \boldsymbol{f_\theta}(\boldsymbol{x}_n)\right) + s_{\boldsymbol{\gamma}}(\boldsymbol{x}_n, m) \cdot \mathrm{softmax}\left(\overline{\boldsymbol{W}}^{\mathrm{T}} \boldsymbol{f_\theta}(\boldsymbol{x}_n)\right), \tag{35}$$

where $s_{\boldsymbol{\gamma}} : \Omega_X \times [M] \to [0, 1]$ is an auxiliary network with parameters $\boldsymbol{\gamma} \in \Omega_\Gamma$. It outputs an instance- and crowdworker-dependent scalar as the estimated degree to which a crowdworker's class label distribution follows common confusions across crowdworkers. Accordingly, the crowdworker classification model in Eq. (35) has the parameters $\boldsymbol{\phi} \coloneqq (\boldsymbol{\theta}, \boldsymbol{W}_1, \dots, \boldsymbol{W}_M, \overline{\boldsymbol{W}}, \boldsymbol{\gamma})$. Another variant of a noise adaptation layer is implemented by the LFC approaches `union-a` and `union-b` (Wei et al., 2023). Instead of treating the crowdworkers independently, the two approaches model the crowdworkers as a union via a single noise adaptation layer $\widetilde{\boldsymbol{W}} \in \mathbb{R}^{C \times (C \cdot M)}$. Therefore, they do not directly implement a crowdworker classification model but a classification model $\widetilde{\boldsymbol{g}}_{\boldsymbol{\phi}} : \Omega_X \to \Delta_{C \cdot M}$ parameterized by $\boldsymbol{\phi} \coloneqq (\boldsymbol{\theta}, \widetilde{\boldsymbol{W}})$ and treating the crowdworkers' class labels as a union:

$$\widetilde{\boldsymbol{g}}_{\boldsymbol{\phi}}(\boldsymbol{x}_n) \coloneqq \mathrm{softmax}\left(\widetilde{\boldsymbol{W}}^{\mathrm{T}} \boldsymbol{f_\theta}(\boldsymbol{x}_n)\right) \qquad (\texttt{union-a}), \tag{36}$$

$$\widetilde{\boldsymbol{g}}_{\boldsymbol{\phi}}(\boldsymbol{x}_n) \coloneqq \mathrm{softmax}\left(\widetilde{\boldsymbol{W}}\right)^{\mathrm{T}} \boldsymbol{f_\theta}(\boldsymbol{x}_n) \qquad (\texttt{union-b}), \tag{37}$$

where the softmax is applied row-wise in the case of `union-b`. As a workaround for approximating the crowdworker classification model, we normalize the outputs associated with each crowdworker, which corresponds to:

$$\boldsymbol{g}_{\boldsymbol{\phi}}(\boldsymbol{x}_n, m) := \text{normalize}\left([\widetilde{\boldsymbol{g}}_{\boldsymbol{\phi}}(\boldsymbol{x}_n)]_{(m-1)\cdot C+1:m\cdot C}\right), \tag{38}$$

where $[\cdot]_{i:j}$ denotes the entries from index $i$ to index $j$ in a vector. For all these LFC approaches, which do not explicitly implement a probabilistic confusion matrix per crowdworker, we resort to using marginal alignment accuracy, which is computed as the agreement between the predicted crowdworker distribution and the predicted true label distribution as an instance-level proxy measure for crowdworker accuracy. Consequently, the crowdworker performance model with parameters $\boldsymbol{\psi} := (\boldsymbol{\theta}, \boldsymbol{\phi})$ performs inference via:

$$h_{\boldsymbol{\psi}}(\boldsymbol{x}_n, m) := \boldsymbol{f}_{\boldsymbol{\theta}}^{\mathrm{T}}(\boldsymbol{x}_n)\boldsymbol{g}_{\boldsymbol{\phi}}(\boldsymbol{x}_n, m). \tag{39}$$

## B   Theoretical Analysis of Hyperparameter Selection Criteria

This appendix expands on our design decisions underlying the HPS criteria framework introduced in Section 3. Using simple propositions, we show that richer modeling assumptions can inject class-specific bias into the posterior computation of Eq. (16). We then prove that the computation we adopt in Eq. (17) is immune to this issue and remains class-agnostic. Finally, we overview the time complexities of the HPS criteria.

### B.1   Prior Class Probabilities

Proposition 1 motivates our design choice not to use non-uniform prior class probabilities when computing the posterior class probabilities in Eq. (16). Intuitively, if the prior class probabilities are sufficiently biased toward a particular class, this bias will dominate in the posterior regardless of the observed class labels from the crowdworkers. Therefore, relying on the class probabilities estimated by the data classification model $\boldsymbol{f}_{\boldsymbol{\theta}}$ as prior can bias the aggregated class labels towards the data classification model's predictions.

**Proposition 1** *For an instance $\boldsymbol{x}_n \in \mathcal{X}$, let us assume strictly positive likelihoods for the class labels such that $\forall \boldsymbol{e}_c \in \Omega_Y$ and $\forall m \in \mathcal{M}_n$:*

$$\widehat{\Pr}(\boldsymbol{z}_{nm}|\boldsymbol{x}_n, \boldsymbol{y}_n = \boldsymbol{e}_c, m) > 0. \tag{40}$$

*Then, for any non-empty allocation of observed class labels $\mathcal{Z}_n \in \mathcal{B}(\Omega_Y) \setminus \{\emptyset\}$ and for any class label $\boldsymbol{e}_k \in \Omega_Y$, there exists a constant $\varepsilon \in (0, 1)$ such that from $\widehat{\Pr}(\boldsymbol{y}_n = \boldsymbol{e}_k|\boldsymbol{x}_n) > \varepsilon$ follows:*

$$\overline{\boldsymbol{z}}(\boldsymbol{x}_n, \mathcal{Z}_n) = \boldsymbol{e}_k. \tag{41}$$

**Proof**  For a fixed instance $\boldsymbol{x}_n \in \mathcal{X}$ and class label $\boldsymbol{e}_k \in \Omega_Y$, we first define

$$l_c := \prod_{m \in \mathcal{M}_n} \widehat{\Pr}(\boldsymbol{z}_{nm}|\boldsymbol{x}_n, \boldsymbol{y}_n = \boldsymbol{e}_c, m), \tag{42}$$

$$l_{\max} := \max_{c \in [C] \setminus \{k\}} (l_c), \tag{43}$$

$$\varepsilon := \frac{l_{\max}}{l_{\max} + l_k}. \tag{44}$$

Now, let $\widehat{\Pr}(\boldsymbol{y}_n = \boldsymbol{e}_k|\boldsymbol{x}_n) > \varepsilon$. Our goal is to show that for any $\boldsymbol{e}_c \in \Omega_Y \setminus \{\boldsymbol{e}_k\}$ we have $\widehat{\Pr}(\boldsymbol{y}_n = \boldsymbol{e}_k|\boldsymbol{x}_n, \mathcal{Z}_n) > \widehat{\Pr}(\boldsymbol{y}_n = \boldsymbol{e}_c|\boldsymbol{x}_n, \mathcal{Z}_n)$, since then we get $\overline{\boldsymbol{z}}(\boldsymbol{x}_n, \mathcal{Z}_n) = \boldsymbol{e}_k$. Starting from Bayes' theorem in Eq. (15) and the

conditional independence assumption in Eq. (16), we have

$$\widehat{\Pr}(\boldsymbol{y}_n = \boldsymbol{e}_k | \boldsymbol{x}_n, \mathcal{Z}_n) = \frac{1}{\widehat{\Pr}(\mathcal{Z}_n | \boldsymbol{x}_n)} \underbrace{\widehat{\Pr}(\boldsymbol{y}_n = \boldsymbol{e}_k | \boldsymbol{x}_n)}_{> \frac{l_{\max}}{l_{\max} + l_k}} \underbrace{\prod_{m \in \mathcal{M}_n} \widehat{\Pr}(\boldsymbol{z}_{nm} | \boldsymbol{x}_n, \boldsymbol{y}_n = \boldsymbol{e}_k, m)}_{= l_k} \tag{45}$$

$$> \frac{1}{\widehat{\Pr}(\mathcal{Z}_n | \boldsymbol{x}_n)} \frac{l_{\max}}{l_{\max} + l_k} l_k \tag{46}$$

$$= \frac{1}{\widehat{\Pr}(\mathcal{Z}_n | \boldsymbol{x}_n)} \frac{l_k}{l_{\max} + l_k} l_{\max}. \tag{47}$$

While the first factor in Eq. (47) with $\widehat{\Pr}(\mathcal{Z}_n | \boldsymbol{x}_n) > 0$ is for normalization, we get for the second factor

$$\frac{l_k}{l_{\max} + l_k} = \frac{l_{\max} + l_k - l_{\max}}{l_{\max} + l_k} \tag{48}$$

$$= 1 - \frac{l_{\max}}{l_{\max} + l_k} \tag{49}$$

$$> 1 - \widehat{\Pr}(\boldsymbol{y}_n = \boldsymbol{e}_k | \boldsymbol{x}_n) \tag{50}$$

$$= \widehat{\Pr}(\boldsymbol{y}_n \neq \boldsymbol{e}_k | \boldsymbol{x}_n) \tag{51}$$

$$\geq \widehat{\Pr}(\boldsymbol{y}_n = \boldsymbol{e}_c | \boldsymbol{x}_n), \tag{52}$$

where Eq. (50) follows by definition of $\varepsilon$ as $-\widehat{\Pr}(\boldsymbol{y}_n = \boldsymbol{e}_k | \boldsymbol{x}_n) < -\varepsilon = -\frac{l_{\max}}{l_{\max} + l_k}$ and Eq. (52) follows by monotonicity as $\boldsymbol{y}_n = \boldsymbol{e}_c \implies \boldsymbol{y}_n \neq \boldsymbol{e}_k$. For the third factor, we have

$$l_{\max} \geq l_c = \prod_{m \in \mathcal{M}_n} \widehat{\Pr}(\boldsymbol{z}_{nm} | \boldsymbol{x}_n, \boldsymbol{y}_n = \boldsymbol{e}_c, m) \tag{53}$$

by definition of $l_{\max}$. Incorporating both inequalities into Eq. (47), we get

$$\frac{1}{\widehat{\Pr}(\mathcal{Z}_n | \boldsymbol{x}_n)} \frac{l_k}{l_{\max} + l_k} l_{\max} > \frac{1}{\widehat{\Pr}(\mathcal{Z}_n | \boldsymbol{x}_n)} \widehat{\Pr}(\boldsymbol{y}_n = \boldsymbol{e}_c | \boldsymbol{x}_n) \prod_{m \in \mathcal{M}_n} \widehat{\Pr}(\boldsymbol{z}_{nm} | \boldsymbol{x}_n, \boldsymbol{y}_n = \boldsymbol{e}_c, m) \tag{54}$$

$$= \widehat{\Pr}(\boldsymbol{y}_n = \boldsymbol{e}_c | \boldsymbol{x}_n, \mathcal{Z}_n) \tag{55}$$

and therefore $\widehat{\Pr}(\boldsymbol{y}_n = \boldsymbol{e}_k | \boldsymbol{x}_n, \mathcal{Z}_n) > \widehat{\Pr}(\boldsymbol{y}_n = \boldsymbol{e}_c | \boldsymbol{x}_n, \mathcal{Z}_n)$ as desired. ■

### B.2 Crowdworker Confusion Probabilities

Proposition 2 motivates our design choice not to use full confusion probability estimates for computing the posterior class probabilities in Eq. (16). Intuitively, suppose the full confusion probabilities for the crowdworkers are sufficiently biased toward a particular class (see sufficient condition in Eq. (58)). In that case, this bias will dominate in the posterior regardless of uniform prior class probabilities and observed class labels from the crowdworkers. Therefore, relying on full confusion matrices, as estimated by many LFC approaches in the form of $\boldsymbol{Q}_\beta$ in Appendix A, can bias the aggregated class labels towards their own predictions.

**Proposition 2** *For an instance $\boldsymbol{x}_n \in \mathcal{X}$, suppose uniform prior class probabilities according to Eq. (17). Then, for any non-empty allocation of observed class labels $\mathcal{Z}_n \in \mathcal{B}(\Omega_Y) \setminus \{\emptyset\}$ and for any class label $\boldsymbol{e}_k \in \Omega_Y$, there exist confusion probability estimates $\widehat{\Pr}(\boldsymbol{z}_{nm} | \boldsymbol{x}_n, \boldsymbol{y}_n, m)$ such that:*

$$\overline{\boldsymbol{z}}(\boldsymbol{x}_n, \mathcal{Z}_n) = \boldsymbol{e}_k. \tag{56}$$

**Proof** With uniform prior class probabilities and $\widehat{\Pr}(\mathcal{Z}_n|\boldsymbol{x}_n) > 0$ for normalization, the posterior probability for class label $\boldsymbol{e}_c \in \Omega_Y$ from Eq. (15) and Eq. (16) is given by:

$$\widehat{\Pr}(\boldsymbol{y}_n = \boldsymbol{e}_c|\boldsymbol{x}_n, \mathcal{Z}_n) = \frac{1}{\widehat{\Pr}(\mathcal{Z}_n|\boldsymbol{x}_n)} \frac{1}{C} \prod_{m \in \mathcal{M}_n} \widehat{\Pr}(\boldsymbol{z}_{nm}|\boldsymbol{x}_n, \boldsymbol{y}_n = \boldsymbol{e}_c, m). \tag{57}$$

Therefore, by requiring that $\forall \boldsymbol{e}_c \in \Omega_Y \setminus \{\boldsymbol{e}_k\}$ and $\forall m \in \mathcal{M}_n$:

$$\widehat{\Pr}(\boldsymbol{z}_{nm}|\boldsymbol{x}_n, \boldsymbol{y}_n = \boldsymbol{e}_k, m) > \widehat{\Pr}(\boldsymbol{z}_{nm}|\boldsymbol{x}_n, \boldsymbol{y}_n = \boldsymbol{e}_c, m), \tag{58}$$

we obtain $\forall \boldsymbol{e}_c \in \Omega_Y \setminus \{\boldsymbol{e}_k\}$:

$$\widehat{\Pr}(\boldsymbol{y}_n = \boldsymbol{e}_k|\boldsymbol{x}_n, \mathcal{Z}_n) = \frac{1}{\widehat{\Pr}(\mathcal{Z}_n|\boldsymbol{x}_n)} \frac{1}{C} \prod_{m \in \mathcal{M}_n} \widehat{\Pr}(\boldsymbol{z}_{nm}|\boldsymbol{x}_n, \boldsymbol{y}_n = \boldsymbol{e}_k, m) \tag{59}$$

$$> \frac{1}{\widehat{\Pr}(\mathcal{Z}_n|\boldsymbol{x}_n)} \frac{1}{C} \prod_{m \in \mathcal{M}_n} \widehat{\Pr}(\boldsymbol{z}_{nm}|\boldsymbol{x}_n, \boldsymbol{y}_n = \boldsymbol{e}_c, m) \tag{60}$$

$$= \widehat{\Pr}(\boldsymbol{y}_n = \boldsymbol{e}_c|\boldsymbol{x}_n, \mathcal{Z}_n), \tag{61}$$

which implies that $\boldsymbol{e}_k \in \Omega_Y$ is our MAP estimate and, thus, our aggregated class label. ∎

### B.3 Maximum-a-posterior Estimate as Weighted Majority Vote

Proposition 3 motivates our design choice to use uniform prior class probabilities as in Eq. (17) and a scalar performance-based model for the instance-wise crowdworkers' confusion probabilities as in Eq. (18). Intuitively, the posterior class probabilities correspond to a soft weighted majority vote, so the MAP estimate is the label with the most "soft votes". This weighted majority voting treats all classes symmetrically, and any difference in posterior probabilities or the MAP estimate arises only from the class labels provided by the crowdworkers, never from a built-in preference for a particular class.

**Proposition 3** *Given the posterior class probability computation from Eq. (19) for an instance $\boldsymbol{x}_n \in \mathcal{X}$ with any non-empty allocation of observed class labels $\mathcal{Z}_n \in \mathcal{B}(\Omega_Y) \setminus \{\emptyset\}$ and assuming $\widehat{\Pr}(\boldsymbol{z}_{nm}^{\mathrm{T}}\boldsymbol{y}_n = 1|\boldsymbol{x}_n, m) \in (0, 1)$ for all $m \in \mathcal{M}_n$, the aggregated label in Eq. (22) equals a weighted majority vote such that:*

$$\overline{\boldsymbol{z}}(\boldsymbol{x}_n, \mathcal{Z}_n) = \arg\max_{\boldsymbol{e}_c \in \Omega_Y} \left( \sum_{m \in \mathcal{M}_n} \ln\left( \frac{\widehat{\Pr}(\boldsymbol{z}_{nm}^{\mathrm{T}}\boldsymbol{y}_n = 1|\boldsymbol{x}_n, m)(C-1)}{\widehat{\Pr}(\boldsymbol{z}_{nm}^{\mathrm{T}}\boldsymbol{y}_n = 0|\boldsymbol{x}_n, m)} \right) \cdot (\boldsymbol{z}_{nm}^{\mathrm{T}}\boldsymbol{e}_c) \right). \tag{62}$$

**Proof** For ease of notation, let us define:

$$\alpha_{nm} := \widehat{\Pr}(\boldsymbol{z}_{nm}^{\mathrm{T}}\boldsymbol{y}_n = 1|\boldsymbol{x}_n, m) \in (0, 1), \tag{63}$$

as the estimated (e.g., via the crowdworker performance model) labeling accuracy of the crowdworker $m$ given instance $\boldsymbol{x}_n$. Then, we can rewrite (with the help of basic logarithmic laws) the un-normalized posterior

class probability from Eq. (19) according to:

$$
\begin{aligned}
\widehat{\Pr}(\boldsymbol{y}_n = \boldsymbol{e}_c | \boldsymbol{x}_n, \mathcal{Z}_n) &\propto \exp\left( \ln\left( \prod_{m \in \mathcal{M}_n} (\alpha_{nm})^{\boldsymbol{z}_{nm}^{\mathrm{T}} \boldsymbol{e}_c} \left( \frac{1 - \alpha_{nm}}{C - 1} \right)^{1 - \boldsymbol{z}_{nm}^{\mathrm{T}} \boldsymbol{e}_c} \right) \right) \\
&= \exp\left( \sum_{m \in \mathcal{M}_n} \ln\left( (\alpha_{nm})^{\boldsymbol{z}_{nm}^{\mathrm{T}} \boldsymbol{e}_c} \left( \frac{1 - \alpha_{nm}}{C - 1} \right)^{1 - \boldsymbol{z}_{nm}^{\mathrm{T}} \boldsymbol{e}_c} \right) \right) \\
&= \exp\left( \sum_{m \in \mathcal{M}_n} \ln\left( (\alpha_{nm})^{\boldsymbol{z}_{nm}^{\mathrm{T}} \boldsymbol{e}_c} \right) + \ln\left( \left( \frac{1 - \alpha_{nm}}{C - 1} \right)^{1 - \boldsymbol{z}_{nm}^{\mathrm{T}} \boldsymbol{e}_c} \right) \right) \\
&= \exp\left( \sum_{m \in \mathcal{M}_n} (\boldsymbol{z}_{nm}^{\mathrm{T}} \boldsymbol{e}_c) \ln(\alpha_{nm}) + \left(1 - \boldsymbol{z}_{nm}^{\mathrm{T}} \boldsymbol{e}_c\right) \ln\left( \frac{1 - \alpha_{nm}}{C - 1} \right) \right) \\
&= \exp\left( \sum_{m \in \mathcal{M}_n} (\boldsymbol{z}_{nm}^{\mathrm{T}} \boldsymbol{e}_c) \left( \ln(\alpha_{nm}) - \ln\left( \frac{1 - \alpha_{nm}}{C - 1} \right) \right) + \ln\left( \frac{1 - \alpha_{nm}}{C - 1} \right) \right) \\
&\propto \exp\left( \sum_{m \in \mathcal{M}_n} (\boldsymbol{z}_{nm}^{\mathrm{T}} \boldsymbol{e}_c) \left( \ln(\alpha_{nm}) - \ln\left( \frac{1 - \alpha_{nm}}{C - 1} \right) \right) \right) \\
&= \exp\left( \sum_{m \in \mathcal{M}_n} \ln\left( \frac{\alpha_{nm}(C - 1)}{1 - \alpha_{nm}} \right) \cdot (\boldsymbol{z}_{nm}^{\mathrm{T}} \boldsymbol{e}_c) \right).
\end{aligned}
\tag{64}
$$

Replacing the labeling accuracy $\alpha_{nm}$ with its probabilistic definition from Eq. (63), we get the input of the $\arg\max$ function in Eq. (62). ∎

## B.4 Worst-case Time Complexity Analysis

We analyze the time complexity when evaluating an HPS criterion as one part of the HPO pipeline. More specifically, Table 6 presents the worst-case time complexity of individual steps involved by the non-default HPS criteria in O-notation. Typically, the training time complexity is the most expensive part of HPO. Figure 9 provides an example of this. Here, the training times are almost 100 times higher than computing the empirical risks (including previous steps such as prediction computation) for evaluating the HPS criterion ENS. Even when using a GPU, the difference can be higher for more complex neural network architectures, such as when fully fine-tuning a backbone. Accordingly, differences in time complexity when evaluating the various HPS criteria are of minor importance. Since the training time complexity is identical for each HPS criterion, we do not further analyze this complexity and instead focus on the steps related to the actual evaluation of an HPS criterion in the following.

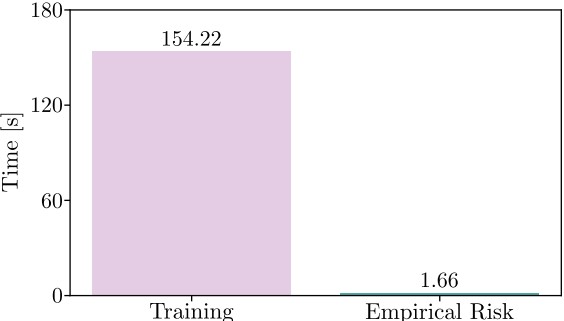

Figure 9: **Computation times.** Given an AMD Ryzen 9 7950X as CPU, the sum of training times for the LFC approach `coin` (Nguyen et al., 2024) with its default HPC and the times for computing all empirical risks in $\mathcal{R}$ (see Section 3.4) are reported in seconds across a $K = 5$-fold cross-validation for the dataset variant `dopanim-full`.

**Prediction Computation**  Computing model predictions represents the first step of an HPO criterion's evaluation. This step's complexity is affected by which of the three models, i.e., data classification model $\boldsymbol{f_\theta}$, crowdworker classification model $\boldsymbol{g_\phi}$, or crowdworker performance model $h_{\boldsymbol{\psi}}$, is required to make predictions. If only the data classification model is involved, we need class probability predictions for each validation instance. When performing a $K$-fold cross-validation, each instance is used for validation once. Accordingly, this step scales linearly with $N$ as the number of observed instances and $|\Lambda|$ as the number of candidate HPCs.

Table 6: **Worst-case time complexity analysis.** Each row corresponds to one HPS criterion, with its name indicated by the first column. The other column headings refer to the steps of evaluating an HPS criterion. The colors distinguish between baseline, aggregation-level, crowd-level, and ensemble-based criteria. There are $N$ observed instances, $M$ crowdworkers, $|\Lambda|$ candidate HPCs, and $C$ classes. The variable $T$ denotes the worst-case time complexity of performing a single forward pass for any model trained by an LFC approach. A $\star$ marks criteria that had access to the true validation labels.

| Hyperparameter Selection Criterion | Prediction Computation | Posterior True Label Probability Estimation | Empirical Risk Computation | Winner Selection |
|---|---|---|---|---|
| TRUE$^\star$ | $O(|\Lambda| \cdot N \cdot T)$ | N/A | $O(|\Lambda| \cdot N \cdot C)$ | $O(|\Lambda|)$ |
| AEU | $O(|\Lambda| \cdot N \cdot T)$ | $O(|\Lambda| \cdot N \cdot M \cdot C)$ | $O(|\Lambda| \cdot N \cdot C)$ | $O(|\Lambda|)$ |
| AEC | $O(|\Lambda| \cdot N \cdot T)$ | $O(|\Lambda| \cdot N \cdot M \cdot C)$ | $O(|\Lambda| \cdot N \cdot C)$ | $O(|\Lambda|)$ |
| ALU | $O(|\Lambda| \cdot N \cdot M \cdot T)$ | $O(|\Lambda| \cdot N \cdot M \cdot C)$ | $O(|\Lambda| \cdot N \cdot C)$ | $O(|\Lambda|)$ |
| ALC | $O(|\Lambda| \cdot N \cdot M \cdot T)$ | $O(|\Lambda| \cdot N \cdot M \cdot C)$ | $O(|\Lambda| \cdot N \cdot C)$ | $O(|\Lambda|)$ |
| CXU | $O(|\Lambda| \cdot N \cdot M \cdot T)$ | N/A | $O(|\Lambda| \cdot N \cdot M \cdot C)$ | $O(|\Lambda|)$ |
| CEC | $O(|\Lambda| \cdot N \cdot M \cdot T)$ | $O(|\Lambda| \cdot N \cdot M \cdot C)$ | $O(|\Lambda| \cdot N \cdot M \cdot C)$ | $O(|\Lambda|)$ |
| CLC | $O(|\Lambda| \cdot N \cdot M \cdot T)$ | $O(|\Lambda| \cdot N \cdot M \cdot C)$ | $O(|\Lambda| \cdot N \cdot M \cdot C)$ | $O(|\Lambda|)$ |
| ENS | $O(|\Lambda| \cdot N \cdot M \cdot T)$ | $O(J \cdot |\Lambda| \cdot N \cdot M \cdot C)$ | $O(J \cdot |\Lambda| \cdot N \cdot M \cdot C)$ | $O(J \cdot |\Lambda| \cdot \ln |\Lambda|)$ |

The variable $T$ denotes the time complexity for obtaining a prediction from one of the three models. The step's complexity increases if the crowdworker classification or crowdworker performance model is involved, because then we need to compute for each of the $N \cdot M$ instance-crowdworker pairs a prediction in the worst case, where each crowdworker has labeled every observed instance.

**Posterior True Label Probability Estimation** The HPS criteria TRUE and CXU do not require any posterior probabilities. For the other criteria, the posterior probabilities are obtained according to Eq. (19) by iterating over all of the $M$ (worst case) observed class labels per instance for each of the $C$ classes. Accordingly, this computation has a complexity of $O(M \cdot C)$ and must be repeated for each of the $N$ observed instances and for each of the $|\Lambda|$ candidate HPCs. For the ensemble-based criterion ENS, this process is additionally repeated for each of its $J$ members.

**Empirical Risk Computation** Given the probabilistic model predictions and the targets, this step refers to computing the zero-one losses, which are then averaged to obtain the empirical risk measurement. For the HPS criterion TRUE and the aggregation level HPS criteria, we need to find one of the $C$ class labels with the maximum probability predicted by the classification model for each of the $N$ observed instances. In the case of the crowd-level criteria, this step extends to each of the $N \cdot M$ instance-crowdworker pairs. We need to repeat this step for each of the $|\Lambda|$ candidate HPCs. For the HPS criterion ENS, this step must also cover each of its $J$ members.

**Winner Selection** For nearly all HPS criteria, we find the winner HPC by selecting the one with the lowest empirical risk from the $|\Lambda|$ candidate HPCs. Only for the ensemble-based HPS criterion, we need to compute the Borda count. In this case, the complexity of sorting the $|\Lambda|$ candidate HPCs according to each of their $J$ empirical risk measurements is dominating the subsequent summation and finding of the minimum. In other words, the reported worst-case complexity corresponds to sorting $J$ lists of $|\Lambda|$ elements with the merge sort algorithm.

## C   Experimental Evaluation

This appendix describes our computational resources for experimentation and provides additional results to the experimental evaluation presented in Section 4.

### C.1   Computational Resources

Table 14 lists the results for all 35 dataset variants, 13 LFC approaches, and 11 HPS criteria. Moreover, we report the results for training with the ground truth (`gt`) class labels as the upper baseline approach. Each test zero-one loss value is the result of determining the selected HPC from a candidate set of $|\Lambda| = 51$ HPCs

via a $K = 5$-fold cross-validation, of which 50 HPCs have been generated via Sobol sequences (Sobol, 1998) and one corresponds to the default (DEF) HPC. Subsequently, each selected HPC is tested with 5 different initializations of the respective neural network architecture. In total, this corresponds to almost

$$\underbrace{5}_{\text{datasets}} \cdot \underbrace{7}_{\text{variants}} \cdot \underbrace{14}_{\text{approaches}} \left( \underbrace{51}_{\text{HPCs}} \cdot \underbrace{5}_{\text{training \& validation}} + \underbrace{11}_{\text{criteria}} \cdot \underbrace{5}_{\text{training \& testing}} \right) = 151{,}900 \qquad (65)$$

training and evaluation runs. We executed all runs on a compute cluster equipped with NVIDIA A100 and V100 GPU servers, which we used to pre-compute the image and text embeddings. Almost all other computations were executed with AMD EPYC 7742 CPU servers. The time measurements from Appendix B.4 are the only exception because they refer to an AMD Ryzen 9 7950X as the CPU of a local workstation.

## C.2 Supplementary Results

We present supplementary results regarding the HPS criteria' and LFC approaches' performances. Further, we ablate the members' impact on the ensemble-based criterion ENS, analyze the impact of the noise level, the number of candidate HPCs, an architecture search for the data classification model, and the loss function.

**Hyperparameter Selection Criteria** Figure 10 (analogous to Figure 5) depicts the criteria's relative zero-one loss reductions compared to the lower baseline criterion DEF. The results confirm our observations for the absolute zero-one loss reductions, namely, that a criterion's benefit varies across the LFC approaches, while ENS is the most robust one for noisy crowd-labeled validation data.

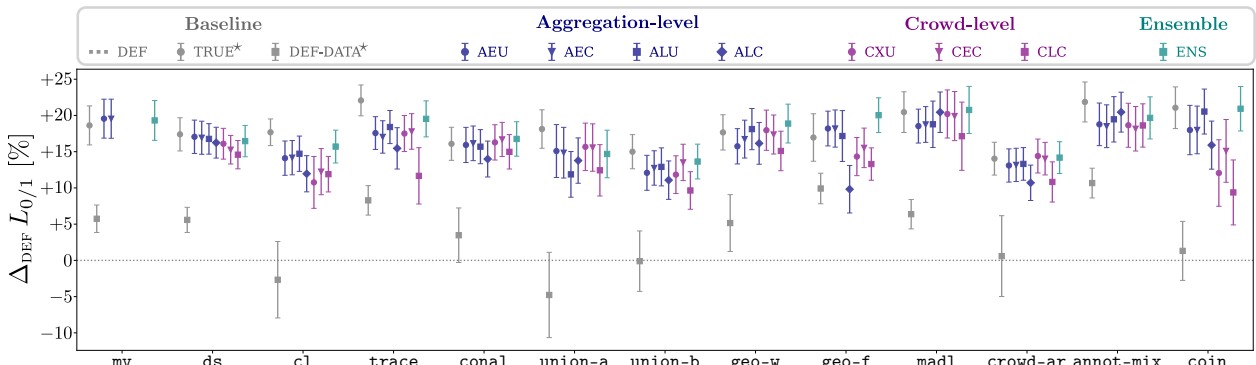

Figure 10: **Relative zero-one loss reductions of HPS criteria.** For each LFC approach ($x$-axis), the scatter plot displays the mean and standard error of a criterion's reduction in zero-one loss ($y$-axis) as a percentage [%] relative to training with default (DEF) HPC. Higher reductions correspond to greater improvements. A $\star$ marks criteria that had access to the true validation labels.

Beyond reporting average reductions in zero-one loss relative to the default HPC, we also examine how performance is distributed across pairs of HPS criteria. Figure 11, therefore, presents a pairwise win-rate matrix, including the results from the pairwise significance tests, where each cell shows the proportion of experiments in which the row criterion beats the column criterion. The matrix shows that the TRUE criterion dominates all alternatives: its win-rate is consistently higher than its loss-rate. Several cell pairs do not sum to one, indicating ties in which the two criteria selected HPCs with identical performance. For the default baselines, DEF loses to every other criterion, confirming that a single global default HPC per approach is inadequate. DEF-DATA, which transfers HPCs optimized on classification tasks without any noisy labels, performs somewhat better yet still lags behind. Accordingly, optimizing the HPC of a standard classification model per dataset with access to true validation labels does not satisfy the LFC approaches' individual requirements. Among criteria that explicitly account for noisy crowd-labeled validation data, the ensemble-based criterion performs best: against every rival except TRUE, it wins more often than it loses. Together with TRUE, it is the only criterion whose loss-rate versus DEF stays below 10%.

| | TRUE* | DEF-DATA* | DEF | AEU | AEC | ALU | ALC | CXU | CEC | CLC | ENS |
|---|---|---|---|---|---|---|---|---|---|---|---|
| **Row criterion wins more often than column criterion.** | | | | | **Row criterion loses more often than column criterion.** | | | | | |
| ● Comparison is significant. | | | | | | | | | | | |
| TRUE* | N/A | 78.46 | 93.85 | 41.76 | 41.10 | 44.05 | 47.62 | 48.33 | 46.43 | 53.10 | 39.12 |
| DEF-DATA* | 19.78 | N/A | 72.97 | 25.93 | 26.37 | 25.00 | 29.76 | 26.67 | 24.76 | 30.95 | 22.86 |
| DEF | 5.93 | 27.03 | N/A | 10.77 | 10.11 | 8.81 | 11.43 | 12.86 | 10.71 | 13.81 | 7.03 |
| AEU | 28.13 | 72.75 | 88.57 | N/A | 5.27 | 22.62 | 34.76 | 28.57 | 25.95 | 37.62 | 15.60 |
| AEC | 28.57 | 71.87 | 89.23 | 7.47 | N/A | 22.86 | 34.76 | 28.57 | 25.00 | 37.62 | 15.38 |
| ALU | 31.43 | 73.57 | 89.05 | 28.10 | 26.90 | N/A | 29.29 | 29.52 | 31.43 | 40.00 | 18.57 |
| ALC | 28.81 | 68.10 | 86.67 | 37.38 | 35.95 | 24.76 | N/A | 30.48 | 31.90 | 25.71 | 22.38 |
| CXU | 32.38 | 72.14 | 85.00 | 34.76 | 32.14 | 24.76 | 33.33 | N/A | 12.14 | 30.95 | 16.90 |
| CEC | 32.38 | 73.57 | 87.14 | 34.52 | 30.95 | 28.33 | 35.24 | 12.62 | N/A | 30.71 | 15.95 |
| CLC | 26.90 | 67.62 | 84.05 | 34.76 | 33.33 | 28.33 | 19.52 | 22.86 | 20.95 | N/A | 17.38 |
| ENS | 34.73 | 75.16 | 91.65 | 30.55 | 27.69 | 26.67 | 36.67 | 26.67 | 25.00 | 37.86 | N/A |

Figure 11: **Pairwise win-rate matrix for HPS criteria.** Across all datasets and LFC approaches, a cell reports the percentage [%] on which the row criterion selects an HPC outperforming the HPC selected by the column criterion. A cell's color decodes whether the row criterion has more wins or loses than the column criterion, whereas a circle ● indicates whether this comparison is significant. The diagonal shows *not applicable* (N/A) because a criterion is not compared with itself. A ⋆ marks criteria that had access to the true validation labels.

**Learning from Crowds Approaches** Figure 12 (analogous to Figure 6) depicts the approaches' relative zero-one loss reductions compared to the lower baseline approach mv[DEF], which corresponds to training with the majority vote labels and the default HPC. The results confirm our observations for the absolute zero-one loss reductions, namely, that one-stage approaches can achieve high performance gains, while the choice of the criterion affects the comparison of the approaches with each other.

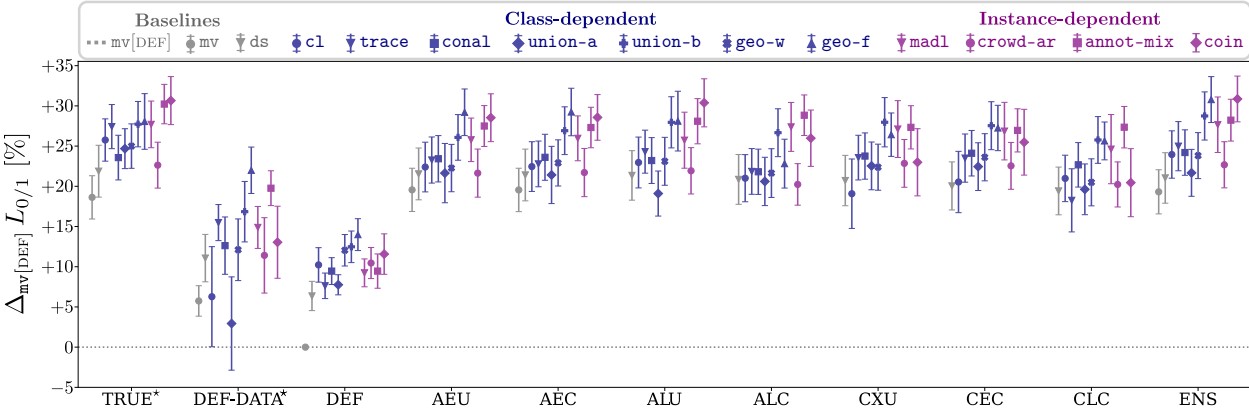

Figure 12: **Relative zero-one loss reductions of LFC approaches.** For each HPS criterion ($x$-axis), the scatter plot displays the mean and standard error of an LFC approach's reduction in zero-one loss ($y$-axis) as percentage [%] compared to majority voting (mv) trained with the default (DEF) HPC. Higher reductions correspond to greater improvements. A ⋆ marks criteria that had access to the true validation labels.

Beyond reporting average reductions in zero-one loss relative to the lower baseline approach mv[DEF], we also examine how performance is distributed across pairs of approaches. Figure 13, therefore, presents a pairwise win-rate matrix, including the results from the pairwise significance tests, with ENS as the HPS criterion, where each cell shows the proportion of dataset variants on which the row approach beats the column approach. Several cell pairs do not sum to one, indicating ties in which the two approaches reached identical performance. The two-stage baseline mv loses to every other approach, confirming that estimating

■ Row approach wins more often than column approach.  ■ Row approach loses more often than column approach.
● Comparison is significant.

| | mv | ds | cl | trace | conal | union-a | union-b | geo-w | geo-f | madl | crowd-ar | annot-mix | coin |
|---|---|---|---|---|---|---|---|---|---|---|---|---|---|
| mv | N/A | 22.86 | 34.29 | 20.00● | 11.43● | 31.43● | 17.14● | 14.29● | 5.71● | 20.00● | 25.71● | 22.86● | 2.86● |
| ds | 48.57 | N/A | 45.71 | 28.57 | 28.57 | 45.71 | 25.71 | 25.71● | 11.43● | 28.57● | 45.71 | 25.71● | 17.14● |
| cl | 65.71 | 54.29 | N/A | 51.43 | 37.14 | 48.57 | 45.71 | 25.71 | 20.00● | 37.14 | 57.14 | 37.14 | 28.57 |
| trace | 80.00● | 71.43 | 48.57 | N/A | 42.86 | 48.57 | 54.29 | 51.43 | 25.71● | 25.71 | 62.86 | 31.43 | 25.71● |
| conal | 88.57● | 71.43 | 62.86 | 57.14 | N/A | 42.86 | 51.43 | 37.14 | 20.00● | 22.86 | 62.86 | 28.57 | 17.14● |
| union-a | 68.57 | 54.29 | 51.43 | 51.43 | 57.14 | N/A | 48.57 | 31.43 | 22.86● | 34.29 | 68.57 | 34.29 | 14.29● |
| union-b | 82.86● | 74.29 | 54.29 | 45.71 | 48.57 | 51.43 | N/A | 22.86● | 11.43● | 37.14 | 60.00 | 25.71 | 22.86● |
| geo-w | 85.71● | 74.29● | 71.43 | 48.57 | 62.86 | 68.57 | 77.14● | N/A | 31.43 | 42.86 | 71.43● | 45.71 | 37.14 |
| geo-f | 94.29● | 88.57● | 80.00● | 74.29● | 80.00● | 74.29● | 88.57● | 68.57 | N/A | 51.43 | 88.57● | 51.43 | 48.57 |
| madl | 80.00● | 71.43● | 62.86 | 74.29 | 77.14 | 65.71 | 60.00 | 57.14 | 48.57 | N/A | 82.86● | 48.57 | 34.29 |
| crowd-ar | 74.29● | 54.29 | 42.86 | 37.14 | 37.14 | 31.43 | 40.00 | 28.57 | 11.43● | 17.14● | N/A | 20.00 | 8.57● |
| annot-mix | 77.14● | 74.29● | 62.86 | 68.57 | 71.43 | 65.71 | 74.29 | 54.29 | 48.57 | 51.43 | 80.00● | N/A | 37.14 |
| coin | 97.14● | 82.86● | 71.43 | 74.29● | 82.86● | 82.86● | 77.14● | 62.86 | 51.43 | 65.71 | 91.43● | 62.86 | N/A |

Figure 13: **Pairwise win-rate matrix for LFC approaches with** ENS **as HPS criterion.** Across all dataset variants, a cell reports the percentage [%] on which the row approach outperforms the column approach. A cell's color decodes whether the row approach has more wins or loses than the column approach, whereas a circle ● indicates whether this comparison is significant. The diagonal shows *not applicable* (N/A) because an approach is not compared with itself.

crowdworkers' performances is beneficial. The other two-stage approach `ds` (Dawid & Skene, 1979), estimating crowdworker performances for label aggregation in the first stage, performs somewhat better yet still lags behind. Accordingly, one-stage LFC approaches combining the crowdworker performance and true label estimation in one joint training lead to superior performances. Among these one-stage approaches, the `coin` approach (Nguyen et al., 2024) dominates all alternatives: its win-rate is consistently higher than its loss-rate. Together with `geo-f` (Ibrahim et al., 2023), it is the only approach whose loss-rate versus `mv` stays below 10%.

**Ablation Study**   In Table 7, we ablate the individual members' impact on the ensemble-based HPS criterion ENS. Therefore, we demonstrate the effect of removing risk measures from $\mathcal{R}$ (see Section 3.4) in the ranking order of their respective criterion from Table 4, starting with the worst one. We observe that removing members mostly leads to a lower absolute and relative reduction of the zero-one loss compared to the DEF criterion. This observation confirms the importance of individual members. Nevertheless, there might be unexplored combinations of multiple members leading to higher reductions than the full ensemble.

Table 7: **Ablation study for** ENS**.** In comparison to DEF as HPS criterion, each column reports the zero-one loss reductions (absolute as percentage points [$\%_p$] and relative as percentages [%]) for one subset of risk measures in $\mathcal{R}$ when employing the HPS criterion ENS. The full set is given in the leftmost column, while each succeeding column shows the results after removing one criterion with its associated risk measure. The rightmost column refers to the case when only one member is remaining, corresponding to the criterion ALU. The colors distinguish between baseline, aggregation-level, crowd-level, and ensemble-based criteria. Means and standard errors are computed over all combinations of dataset variants and LFC approaches (excluding the approach `mv` that is not compatible with each criterion's empirical risk measure). The arrows indicate that higher (↑) values are better. The **best** and second best values are marked per result type.

| ENS | −CLC | −AEU | −ALC | −AEC | −CXU | −CEC = ALU |
|---|---|---|---|---|---|---|
| | | Absolute Zero-one Loss Reductions Compared to DEF ($\Delta_{\mathrm{DEF}} L_{0/1}$ [$\%_p$] ↑) | | | | |
| + **5.15** | + 5.03 | + 5.09 | + 4.96 | + 4.84 | + 4.96 | + 4.73 |
| ± 0.26 | ± 0.26 | ± 0.27 | ± 0.26 | ± 0.27 | ± 0.27 | ± 0.26 |
| | | Relative Zero-one Loss Reductions Compared to DEF ($\Delta_{\mathrm{DEF}} L_{0/1}$ [%] ↑) | | | | |
| +**17.60** | +17.49 | +17.43 | +17.17 | +16.56 | +17.14 | +16.48 |
| ± 0.77 | ± 0.76 | ± 0.78 | ± 0.77 | ± 0.78 | ± 0.78 | ± 0.80 |

**Analysis Stratified by Noise** To assess how label noise impacts the performances of HPS criteria and LFC approaches, we group the dataset variants into low and high noise via a class-count–invariant correction of the aggregation noise. Let $\chi \in [0, 1]$ be the aggregation noise rate, i.e., the fraction of instances whose majority vote label disagrees with the true label, from Table 2. We define the class-corrected aggregation noise:

$$\widetilde{\chi} := \frac{\chi}{1 - 1/C}. \tag{66}$$

This normalization removes the chance-level dependence of $1/C$, making noise levels more comparable across datasets with different numbers of classes. We use a median split across all dataset variants (median $\widetilde{\chi}_{0.5} := 0.439$) such that 18 dataset variants with $\widetilde{\chi} \leq \widetilde{\chi}_{0.5}$ are assigned to the *low-noise* group, and 17 dataset variants with $\widetilde{\chi} > \widetilde{\chi}_{0.5}$ to the *high-noise* group. Note that this grouping does not purely reflect the noise level, as confounding factors such as the number of labels per instance or the inherent difficulty of a dataset may also play a role. Table 8 breaks down the results from Table 4 according to these two groups of noise levels. We observe that the benefit of HPO is higher in the case of high-noise dataset variants, especially in combination with the HPS criteria TRUE and ENS. For both groups of noise levels, the results confirm that ENS is the best performing HPS criterion with only access to crowd-labeled validation data. For Table 5 with the results of the LFC approaches, an analogous breakdown is given by Table 9. Here, the performance gains of one-stage over two-stage LFC approaches are higher for the high- in comparison to the low-noise group of dataset variants. Accordingly, the ranking of the LFC approaches is also affected by the noise level. For example, `cl` (Rodrigues & Pereira, 2018), `union-a` (Wei et al., 2023), and `annot-mix` (Herde et al., 2024b) achieve much better ranks for the high-noise level. Figure 14 extends our stratified noise level analysis to the violin plots from Figure 7. Here, Kendall $\tau$-*b* coefficients between the two groups are differently distributed. In particular, there is a higher spread of the Kendall $\tau$-*b* coefficients for the group of high-noise dataset variants when comparing rankings for the default (DEF, DEF-DATA) HPCs with the rankings after HPO (via TRUE or ENS), although their means remain similar.

Table 8: **HPS criteria's results stratified by noise levels.** One column per criterion reports the rank compared to the other criteria and the zero-one loss reductions (absolute as percentage points [$\%_p$] and relative as percentages [%]) compared to DEF as criterion. Means and standard errors are computed over all combinations of dataset variants in a noise level and LFC approaches (excluding the approach `mv` that is not compatible with each criterion). The arrows show whether a smaller ($\downarrow$) or higher ($\uparrow$) value is better. The **best** and second best values are marked per result type and noise level. A $\star$ marks criteria that had access to the true validation labels.

| Noise | Baseline | | | Aggregation-level | | | | Crowd-level | | | Ensemble |
|---|---|---|---|---|---|---|---|---|---|---|---|
| **Level** | TRUE[$\star$] | DEF-DATA[$\star$] | DEF | AEU | AEC | ALU | ALC | CXU | CEC | CLC | ENS |
| | | | | | | Ranks ($\downarrow$) | | | | | |
| High | **4.60** | 8.62 | 9.56 | 5.71 | 5.55 | 5.22 | 5.58 | 5.35 | 5.40 | 5.70 | 4.70 |
| | ± 0.21 | ± 0.20 | ± 0.16 | ± 0.17 | ± 0.17 | ± 0.16 | ± 0.19 | ± 0.15 | ± 0.15 | ± 0.18 | ± 0.13 |
| Low | **4.69** | 7.10 | 9.94 | 5.62 | 5.60 | 5.36 | 5.62 | 5.57 | 5.33 | 6.11 | 5.07 |
| | ± 0.21 | ± 0.26 | ± 0.14 | ± 0.15 | ± 0.14 | ± 0.17 | ± 0.18 | ± 0.16 | ± 0.16 | ± 0.18 | ± 0.13 |
| | | | | Absolute Zero-one Loss Reductions Compared to DEF ($\Delta_{\text{DEF}} L_{0/1}$ [$\%_p$] $\uparrow$) | | | | | | | |
| High | **+ 7.24** | + 1.01 | + 0.00 | + 5.99 | + 6.08 | + 6.43 | + 5.97 | + 6.23 | + 6.15 | + 5.64 | + 7.06 |
| | ± 0.41 | ± 0.42 | ± 0.00 | ± 0.47 | ± 0.48 | ± 0.47 | ± 0.50 | ± 0.52 | ± 0.52 | ± 0.55 | ± 0.47 |
| Low | **+ 3.59** | + 0.53 | 0.00 | + 3.18 | + 3.18 | + 3.13 | + 2.97 | + 2.91 | + 3.09 | + 2.58 | + 3.34 |
| | ± 0.19 | ± 0.45 | ± 0.00 | ± 0.18 | ± 0.18 | ± 0.20 | ± 0.21 | ± 0.23 | ± 0.21 | ± 0.25 | ± 0.19 |
| | | | | Relative Zero-one Loss Reductions Compared to DEF ($\Delta_{\text{DEF}} L_{0/1}$ [%] $\uparrow$) | | | | | | | |
| High | **+19.84** | + 4.29 | + 0.00 | +17.63 | +17.81 | +18.58 | +15.85 | +17.64 | +17.51 | +15.11 | +19.76 |
| | ± 1.17 | ± 1.08 | ± 0.00 | ± 1.32 | ± 1.33 | ± 1.32 | ± 1.34 | ± 1.43 | ± 1.43 | ± 1.45 | ± 1.31 |
| Low | **+16.66** | + 3.05 | + 0.00 | +14.81 | +14.82 | +14.49 | +13.54 | +13.44 | +14.44 | +11.59 | +15.56 |
| | ± 0.88 | ± 1.92 | ± 0.00 | ± 0.78 | ± 0.76 | ± 0.92 | ± 0.93 | ± 0.99 | ± 0.89 | ± 1.13 | ± 0.81 |

**Analysis Stratified by Hyperparameter Budget** To assess how the size of the candidate set $|\Lambda|$ (our proxy for the HP budget) affects the performances of non-default HPS criteria and LFC approaches, we complement the main setting $|\Lambda| = 51$ in Section 4.2 with evaluations at $|\Lambda| \in \{11, 31\}$. Table 10 breaks down the results from Table 4 according to these two additional HP budgets. As one might expect, the benefit of the non-default HPS criteria is lowered by reducing the HP budget, while still achieving notable improvements over the default HPS criteria. This is likely because the selection is restricted due to the lower

Table 9: **LFC approaches' results stratified by noise levels with** ENS **as HPS criterion.** One column per approach reports the rank compared to the other approaches and the zero-one loss reductions (absolute as percentage points [%$_p$] and relative as percentages [%]) compared to the approach mv trained with its default (DEF) HPC. Means and standard errors are computed over all dataset variants of the respective label noise level. The arrows indicate whether a smaller (↓) or higher (↑) value is better. The **best** and second best values are marked per result type and noise level.

| Noise | Baseline | | Class-dependent | | | | | | | Instance-dependent | | | |
|---|---|---|---|---|---|---|---|---|---|---|---|---|---|
| Level | mv | ds | cl | trace | conal | union-a | union-b | geo-w | geo-f | madl | crowd-ar | annot-mix | coin |
| | | | | | | Ranks (↓) | | | | | | | |
| High | 11.32 | 10.03 | 6.68 | 7.35 | 8.59 | 6.26 | 7.88 | 5.15 | 4.35 | 5.59 | 9.06 | **4.35** | 4.38 |
| | ± 0.25 | ± 0.71 | ± 0.98 | ± 0.83 | ± 0.63 | ± 0.91 | ± 0.73 | ± 0.91 | ± 0.51 | ± 0.86 | ± 0.66 | ± 0.76 | ± 0.80 |
| Low | 9.86 | 8.19 | 9.06 | 7.28 | 6.22 | 8.86 | 7.36 | 6.44 | 3.86 | 5.14 | 8.89 | 6.28 | **3.56** |
| | ± 0.68 | ± 0.73 | ± 0.76 | ± 0.71 | ± 0.76 | ± 0.89 | ± 0.82 | ± 0.74 | ± 0.75 | ± 0.95 | ± 0.59 | ± 1.01 | ± 0.63 |
| | | | Absolute Zero-one Loss Reductions Compared to mv with DEF ($\Delta_{\mathtt{mv[DEF]}}\, L_{0/1}$ [%$_p$] ↑) | | | | | | | | | | |
| High | + 6.23 | + 6.83 | +11.53 | + 9.81 | + 8.34 | + 9.30 | + 9.41 | **+13.44** | +13.15 | +10.45 | + 8.18 | +12.16 | +12.64 |
| | ± 1.71 | ± 2.32 | ± 2.04 | ± 1.96 | ± 1.73 | ± 1.64 | ± 1.95 | ± 2.16 | ± 1.74 | ± 2.54 | ± 1.79 | ± 1.41 | ± 1.60 |
| Low | + 5.00 | + 5.32 | + 4.74 | + 5.93 | + 6.09 | + 4.93 | + 5.55 | + 6.17 | + 7.13 | + 6.71 | + 5.63 | + 6.42 | **+ 7.27** |
| | ± 0.71 | ± 0.71 | ± 0.68 | ± 0.87 | ± 0.73 | ± 0.86 | ± 0.70 | ± 0.82 | ± 0.89 | ± 0.86 | ± 0.77 | ± 0.82 | ± 0.84 |
| | | | Relative Zero-one Loss Reductions Compared to mv with DEF ($\Delta_{\mathtt{mv[DEF]}}\, L_{0/1}$ [%] ↑) | | | | | | | | | | |
| High | +16.86 | +18.81 | +27.71 | +25.16 | +22.25 | +22.97 | +24.03 | **+31.90** | +31.71 | +26.92 | +21.65 | +30.05 | +31.30 |
| | ± 4.92 | ± 5.85 | ± 5.27 | ± 5.46 | ± 5.12 | ± 4.94 | ± 5.28 | ± 5.27 | ± 4.91 | ± 6.38 | ± 5.13 | ± 4.47 | ± 5.00 |
| Low | +21.63 | +23.15 | +20.40 | +24.83 | +26.04 | +20.44 | +23.61 | +25.77 | +29.92 | +28.35 | +23.65 | +26.50 | **+30.45** |
| | ± 2.67 | ± 2.71 | ± 2.74 | ± 3.14 | ± 2.71 | ± 3.36 | ± 2.65 | ± 3.02 | ± 3.19 | ± 3.19 | ± 2.91 | ± 2.87 | ± 3.04 |

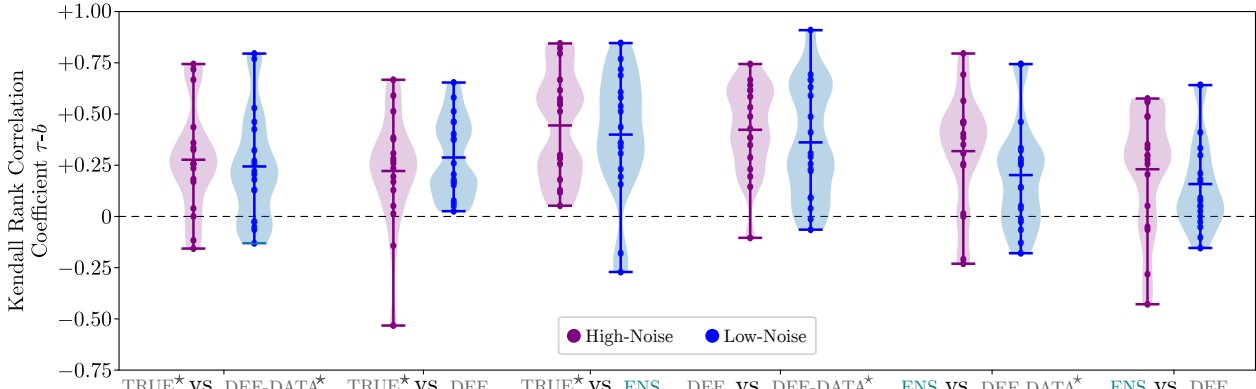

Figure 14: **Rank correlation between HPS criteria stratified by noise levels.** Each violin plot shows the mean and distribution of pairwise Kendall $\tau$-$b$ coefficients, visualized as dots and obtained from the high- and low-noise groups of dataset variants when comparing the ranking of LFC approaches with their HPCs selected via baseline and ensemble-based criteria. Higher coefficients indicate stronger correlation. A ⋆ marks criteria that had access to the true validation labels.

number of candidate HPCs. The results also confirm that ENS is the best performing HPS criterion with only access to crowd-labeled validation data across all tested budgets. For Table 5 containing the main results of the LFC approaches, an analogous breakdown is given by Table 11. Here, the performances of the LFC approaches also decrease with a decreasing HP budget, while the one-stage approaches still take the lead over the two-stage approaches. The ranking of the approaches is also affected by the HP budget. For example, the approaches conal (Chu et al., 2021) and madl (Herde et al., 2023) achieve worse ranks for lower HP budgets. Figure 15 extends our stratified HP budget analysis to the violin plots from Figure 7. The rankings obtained via the criterion TRUE and ENS get more similar for a lower HP budget, as indicated by higher Kendall $\tau$-$b$ coefficients. This can be explained by a lower selection variability for a smaller set of candidate HPCs.

**Hyperparameter Optimization with Data Classification Model Architecture Search** In the main experiments of Section 4, the architecture of the data classification model $f_{\boldsymbol{\theta}}$ is fixed, tuning only the optimizer and LFC approach-specific HPs. This design keeps the HP space $\Omega_\Lambda$ tractable for our budget of $|\Lambda| = 51$ candidate HPCs. To demonstrate feasibility rather than to redefine the benchmark, we additionally

Table 10: **HPS criteria's results stratified by HP budgets.** One column per criterion reports the rank compared to the other criteria and the zero-one loss reductions (absolute as percentage points $[\%_p]$ and relative as percentages $[\%]$) compared to DEF as criterion. For the given HP budget $|\Lambda|$, means and standard errors are computed over all combinations of dataset variants and LFC approaches (excluding the approach mv that is not compatible with each criterion). The arrows show whether a smaller ($\downarrow$) or higher ($\uparrow$) value is better. The **best** and second best values are marked per result type and HP budget. A $\star$ marks criteria that had access to the true validation labels.

| HP Budget | Baseline | | | Aggregation-level | | | | Crowd-level | | | Ensemble |
|---|---|---|---|---|---|---|---|---|---|---|---|
| | TRUE$^\star$ | DEF-DATA$^\star$ | DEF | AEU | AEC | ALU | ALC | CXU | CEC | CLC | ENS |
| | Ranks ($\downarrow$) | | | | | | | | | | |
| $|\Lambda| = 31$ | **4.59** ± 0.14 | 7.78 ± 0.17 | 9.77 ± 0.11 | 5.62 ± 0.11 | 5.58 ± 0.11 | 5.41 ± 0.11 | 5.57 ± 0.13 | 5.57 ± 0.12 | 5.36 ± 0.11 | 5.90 ± 0.12 | 4.85 ± 0.09 |
| $|\Lambda| = 11$ | **4.75** ± 0.12 | 7.24 ± 0.19 | 9.36 ± 0.13 | 5.71 ± 0.10 | 5.62 ± 0.10 | 5.41 ± 0.10 | 5.84 ± 0.12 | 5.43 ± 0.10 | 5.42 ± 0.10 | 5.99 ± 0.12 | 5.22 ± 0.07 |
| | Absolute Zero-one Loss Reductions Compared to DEF ($\Delta_{\text{DEF}} L_{0/1}$ $[\%_p]$ $\uparrow$) | | | | | | | | | | |
| $|\Lambda| = 31$ | + **5.18** ± 0.24 | + 0.76 ± 0.31 | + 0.00 ± 0.00 | + 4.41 ± 0.25 | + 4.40 ± 0.25 | + 4.56 ± 0.26 | + 4.19 ± 0.30 | + 4.38 ± 0.29 | + 4.51 ± 0.28 | + 3.98 ± 0.31 | + 5.00 ± 0.26 |
| $|\Lambda| = 11$ | + **4.50** ± 0.23 | + 0.76 ± 0.31 | + 0.00 ± 0.00 | + 3.70 ± 0.25 | + 3.74 ± 0.25 | + 3.94 ± 0.27 | + 3.36 ± 0.32 | + 3.94 ± 0.27 | + 3.96 ± 0.27 | + 3.14 ± 0.33 | + 4.33 ± 0.25 |
| | Relative Zero-one Loss Reductions Compared to DEF ($\Delta_{\text{DEF}} L_{0/1}$ $[\%]$ $\uparrow$) | | | | | | | | | | |
| $|\Lambda| = 31$ | +**17.70** ± 0.72 | + 3.65 ± 1.12 | + 0.00 ± 0.00 | +15.76 ± 0.76 | +15.73 ± 0.76 | +16.04 ± 0.80 | +14.09 ± 0.95 | +14.87 ± 0.87 | +15.65 ± 0.82 | +13.31 ± 0.91 | +17.24 ± 0.74 |
| $|\Lambda| = 11$ | +**15.42** ± 0.72 | + 3.65 ± 1.12 | + 0.00 ± 0.00 | +13.35 ± 0.74 | +13.55 ± 0.74 | +13.71 ± 0.89 | +11.17 ± 1.05 | +13.41 ± 0.80 | +13.69 ± 0.78 | +10.32 ± 1.08 | +14.76 ± 0.71 |

Table 11: **LFC approaches' results stratified by HP budgets with ENS as HPS criterion.** One column per approach reports the rank compared to the other approaches and the zero-one loss reductions (absolute as percentage points $[\%_p]$ and relative as percentages $[\%]$) compared to the approach mv trained with its default (DEF) HPC. Means and standard errors are computed over all dataset variants for the respective HP budget. The arrows indicate whether a smaller ($\downarrow$) or higher ($\uparrow$) value is better. The **best** and second best values are marked per result type and HP budget.

| Noise Level | Baseline | | Class-dependent | | | | | | | Instance-dependent | | | |
|---|---|---|---|---|---|---|---|---|---|---|---|---|---|
| | mv | ds | cl | trace | conal | union-a | union-b | geo-w | geo-f | madl | crowd-ar | annot-mix | coin |
| | Ranks ($\downarrow$) | | | | | | | | | | | | |
| $|\Lambda| = 31$ | 10.49 ± 0.38 | 9.33 ± 0.56 | 7.90 ± 0.64 | 7.17 ± 0.60 | 7.74 ± 0.48 | 7.17 ± 0.68 | 7.40 ± 0.50 | 5.96 ± 0.60 | **3.63** ± 0.36 | 5.86 ± 0.65 | 9.24 ± 0.45 | 4.99 ± 0.58 | 4.13 ± 0.49 |
| $|\Lambda| = 11$ | 10.10 ± 0.51 | 8.46 ± 0.60 | 8.13 ± 0.66 | 7.23 ± 0.53 | 8.63 ± 0.52 | 7.56 ± 0.65 | 6.97 ± 0.49 | 5.41 ± 0.46 | **3.71** ± 0.44 | 6.51 ± 0.72 | 8.63 ± 0.48 | 5.49 ± 0.70 | 4.17 ± 0.44 |
| | Absolute Zero-one Loss Reductions Compared to mv with DEF ($\Delta_{\text{mv[DEF]}} L_{0/1}$ $[\%_p]$ $\uparrow$) | | | | | | | | | | | | |
| $|\Lambda| = 31$ | + 5.19 ± 0.76 | + 5.61 ± 1.07 | + 7.77 ± 1.21 | + 7.80 ± 0.98 | + 6.97 ± 0.88 | + 7.17 ± 1.06 | + 7.37 ± 0.93 | + 9.51 ± 1.31 | +**10.06** ± 1.06 | + 8.10 ± 1.34 | + 6.57 ± 0.90 | + 9.46 ± 1.00 | + 9.66 ± 1.00 |
| $|\Lambda| = 11$ | + 3.93 ± 0.56 | + 5.04 ± 1.00 | + 6.92 ± 1.21 | + 7.04 ± 0.89 | + 6.11 ± 0.85 | + 6.41 ± 0.94 | + 7.16 ± 0.99 | + 9.20 ± 1.32 | + **9.50** ± 1.08 | + 7.75 ± 1.31 | + 6.09 ± 0.90 | + 8.16 ± 0.98 | + 8.65 ± 0.90 |
| | Relative Zero-one Loss Reductions Compared to mv with DEF ($\Delta_{\text{mv[DEF]}} L_{0/1}$ $[\%]$ $\uparrow$) | | | | | | | | | | | | |
| $|\Lambda| = 31$ | +18.10 ± 2.51 | +19.53 ± 2.89 | +23.11 ± 3.09 | +24.91 ± 2.83 | +23.50 ± 2.77 | +22.36 ± 2.93 | +23.85 ± 2.72 | +28.33 ± 3.09 | +**30.96** ± 2.82 | +26.65 ± 3.44 | +21.93 ± 2.75 | +28.87 ± 2.60 | +30.39 ± 2.92 |
| $|\Lambda| = 11$ | +14.21 ± 1.80 | +18.04 ± 2.61 | +20.81 ± 2.97 | +22.74 ± 2.64 | +20.32 ± 2.73 | +19.70 ± 2.79 | +23.03 ± 2.68 | +27.38 ± 3.07 | +**29.27** ± 2.88 | +23.90 ± 3.23 | +20.95 ± 2.79 | +24.97 ± 2.71 | +27.33 ± 2.66 |

run a targeted study for the dataset variants of label-me and dopanim that includes the classification head architecture in the search. Concretely, instead of using a fixed MLP with $(256, 128)$ neurons in its two hidden layers, we now extend our search to different numbers of layers and neurons sampled according to $\text{uniform}(\{(256), (512), (256, 128), (512, 256)\})$. Because of the more complex HP space $\Omega_\Lambda$, we also increase the number of candidate HPCs to $|\Lambda| = 101$, of which one HPC corresponds to the default one from the main experiments. Table 12 reports results in the same format as Table 4. Nevertheless, both tables' zero-one loss reductions are not directly comparable because the aggregation of Table 4 additionally encompasses the results from variants of the other datasets mgc, reuters, and spc. Moreover, our goal is not to assess the benefit of a potential architecture search, but to investigate whether our main conclusions persist under heterogeneous head architectures. In this context, we observe non-default HPS criteria remain beneficial, and only TRUE surpasses ENS in rank and in zero-one loss reductions.

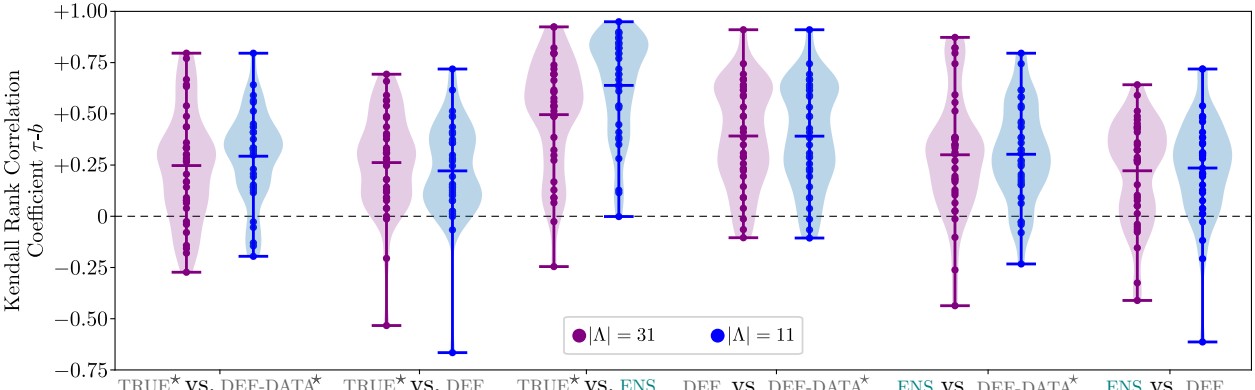

Figure 15: **Rank correlation between HPS criteria stratified by HP budgets.** Each violin plot shows the mean and distribution of pairwise Kendall $\tau$-$b$ coefficients, visualized as dots and obtained from all dataset variants when comparing the ranking of LFC approaches with their HPCs selected via baseline and ensemble-based criteria. The HP budget $|\Lambda|$ is only relevant for the non-default criteria. Higher coefficients indicate stronger correlation. A $\star$ marks criteria that had access to the true validation labels.

Table 12: **HPS criteria's results with architecture search.** One column per criterion reports the rank compared to the other criteria and the zero-one loss reductions (absolute as percentage points [$\%_p$] and relative as percentages [%]) compared to DEF as criterion. Including a simple classification head architecture search, means and standard errors are computed over all combinations of `label-me` and `dopanim` dataset variants and LFC approaches (excluding the approach `mv` that is not compatible with each criterion). The arrows show whether a smaller ($\downarrow$) or higher ($\uparrow$) value is better. The **best** and second best values are marked per result type. A $\star$ marks criteria that had access to the true validation labels.

| Baseline | | | Aggregation-level | | | | Crowd-level | | | Ensemble |
|---|---|---|---|---|---|---|---|---|---|---|
| TRUE$^\star$ | DEF-DATA$^\star$ | DEF | AEU | AEC | ALU | ALC | CXU | CEC | CLC | ENS |
| Ranks ($\downarrow$) | | | | | | | | | | |
| **3.59** | 8.87 | 9.27 | 5.74 | 5.88 | 5.14 | 5.21 | 5.58 | 5.63 | 6.02 | 5.06 |
| $\pm$ 0.23 | $\pm$ 0.21 | $\pm$ 0.17 | $\pm$ 0.20 | $\pm$ 0.19 | $\pm$ 0.19 | $\pm$ 0.22 | $\pm$ 0.19 | $\pm$ 0.17 | $\pm$ 0.21 | $\pm$ 0.15 |
| Absolute Zero-one Loss Reductions Compared to DEF ($\Delta_{\text{DEF}} L_{0/1}$ [$\%_p$] $\uparrow$) | | | | | | | | | | |
| + **3.81** | + 0.30 | + 0.00 | + 2.31 | + 2.18 | + 2.64 | + 2.20 | + 2.67 | + 2.73 | + 1.57 | + 2.76 |
| $\pm$ 0.22 | $\pm$ 0.17 | $\pm$ 0.00 | $\pm$ 0.18 | $\pm$ 0.18 | $\pm$ 0.19 | $\pm$ 0.50 | $\pm$ 0.23 | $\pm$ 0.22 | $\pm$ 0.54 | $\pm$ 0.18 |
| Relative Zero-one Loss Reductions Compared to DEF ($\Delta_{\text{DEF}} L_{0/1}$ [%] $\uparrow$) | | | | | | | | | | |
| +**15.17** | + 0.45 | + 0.00 | +10.46 | + 9.98 | +11.65 | + 8.82 | +10.73 | +11.09 | + 5.65 | +11.82 |
| $\pm$ 0.70 | $\pm$ 0.61 | $\pm$ 0.00 | $\pm$ 0.76 | $\pm$ 0.76 | $\pm$ 0.78 | $\pm$ 2.44 | $\pm$ 0.84 | $\pm$ 0.77 | $\pm$ 2.70 | $\pm$ 0.71 |

**Loss Functions Beyond Zero-one Loss** So far, we have only focused on the zero-one loss (see Eq. (2)) as our target performance measure. However, a trained data classification model $\boldsymbol{f_\theta}$ is often also required to output meaningful probabilities, which can be evaluated using the Brier score (Brier, 1950) as a loss function:

$$L_{\text{BS}}\left(\boldsymbol{y}, \hat{\boldsymbol{y}}\right) \coloneqq (\boldsymbol{y} - \hat{\boldsymbol{y}})^{\text{T}}(\boldsymbol{y} - \hat{\boldsymbol{y}}). \tag{67}$$

At the same time, the data classification model $\boldsymbol{f_\theta}$ ideally achieves a low zero-one loss. In this case, we have a kind of multi-objective optimization problem, whose solution is beyond our scope. Instead, we present a brief analysis demonstrating that the search for the best performing HPS criterion depends on the target loss function(s). Specifically, we compare the non-default criteria using the Brier score as a loss function for selecting the best HPC, whereas the two default criteria remain unchanged. Table 13 shows that conducting HPO with an alternative loss function remains beneficial in the LFC setting. In particular, the proposed criterion ALC using estimates of the crowdworker performance model $h_\psi$ for label aggregation and weighting excels. Overall, the criteria based on aggregation-level risk measures strongly outperform those only relying on crowd-level risk measures. A likely reason is that crowd-level measures overemphasize the ability of the crowdworker classification model $\boldsymbol{g_\phi}$ to assign high probabilities to the workers' noisy labels. Consequently, HPS criteria cannot be transferred naively from one loss function to another. Despite not being the overall best criterion, the ensemble-based approach ENS remains appealing, as its flexible design allows combining

risk measures derived from different loss functions. Exploring such combinations represents a promising direction for future work.

Table 13: **HPS criteria's results with Brier score as loss function.** One column per criterion reports the Brier score reductions (absolute as unitless [−] and relative as percentages [%]) compared to DEF as criterion. Means and standard errors are computed over all combinations of dataset variants and LFC approaches (excluding the approach `mv` that is not compatible with each criterion). The arrows show whether a smaller (↓) or higher (↑) value is better. The **best** and second best values are marked per result type. A ⋆ marks criteria that had access to the true validation labels.

| Baseline | | | Aggregation-level | | | | Crowd-level | | | Ensemble |
|---|---|---|---|---|---|---|---|---|---|---|
| TRUE⋆ | DEF–DATA⋆ | DEF | AEU | AEC | ALU | ALC | CXU | CEC | CLC | ENS |
| Ranks (↓) | | | | | | | | | | |
| **3.93** | 8.16 | 9.13 | 5.45 | 4.98 | 4.52 | 4.46 | 6.58 | 6.73 | 6.81 | 5.25 |
| ± 0.13 | ± 0.15 | ± 0.11 | ± 0.13 | ± 0.11 | ± 0.10 | ± 0.10 | ± 0.14 | ± 0.13 | ± 0.14 | ± 0.11 |
| Absolute Brier Score Reductions Compared to DEF ($\Delta_{\text{DEF}} L_{BS}$ [−] ↑) | | | | | | | | | | |
| +**0.092** | −0.024 | +0.000 | +0.068 | +0.072 | +0.078 | +0.080 | +0.030 | +0.030 | +0.020 | +0.067 |
| ±0.005 | ±0.008 | ±0.000 | ±0.004 | ±0.004 | ±0.004 | ±0.004 | ±0.006 | ±0.006 | ±0.007 | ±0.004 |
| Relative Brier Score Reductions Compared to DEF ($\Delta_{\text{DEF}} L_{BS}$ [%] ↑) | | | | | | | | | | |
| +**18.70** | − 2.59 | + 0.00 | +13.24 | +14.32 | +15.78 | +16.43 | + 7.74 | + 7.65 | + 5.08 | +13.88 |
| ± 0.67 | ± 1.51 | ± 0.00 | ± 0.61 | ± 0.58 | ± 0.59 | ± 0.63 | ± 1.10 | ± 1.10 | ± 1.36 | ± 0.64 |

Table 14: **Zero-one loss results [%] (part I).** The first column lists the LFC approaches and the remaining columns the HPS criteria. Each criterion selects the estimated best HPC per approach, and results are reported as means with standard deviations. The **best-performing** approach per column (excluding gt training with true labels) and the best-performing selection criterion per row (excluding TRUE validating with true labels) are highlighted. The symbol ⋆ marks criteria with access to true validation labels. Some criteria are *not applicable* (N/A) to all approaches.

| Approach | Baseline | | | Aggregation-level | | | | Crowd-level | | | Ensemble |
|---|---|---|---|---|---|---|---|---|---|---|---|
| | TRUE⋆ | DEF-DATA⋆ | DEF | AEU | AEC | ALU | ALC | CXU | CEC | CLC | ENS |
| | | | | | | mgc-worst-1 | | | | | |
| gt | $20.27_{\pm0.83}$ | $20.27_{\pm0.83}$ | $24.60_{\pm1.12}$ | $21.00_{\pm1.94}$ | $21.00_{\pm1.94}$ | N/A | N/A | N/A | N/A | N/A | $21.00_{\pm1.94}$ |
| mv | $81.27_{\pm1.32}$ | $86.93_{\pm1.30}$ | $81.47_{\pm0.96}$ | $79.73_{\pm1.75}$ | $79.73_{\pm1.75}$ | N/A | N/A | N/A | N/A | N/A | $79.73_{\pm1.75}$ |
| ds | $81.27_{\pm1.32}$ | $86.93_{\pm1.30}$ | $81.47_{\pm0.96}$ | $79.73_{\pm1.75}$ | $79.73_{\pm1.75}$ | $79.73_{\pm1.75}$ | $79.73_{\pm1.75}$ | $79.73_{\pm1.75}$ | $79.73_{\pm1.75}$ | $79.73_{\pm1.75}$ | $79.73_{\pm1.75}$ |
| cl | $74.33_{\pm4.29}$ | $85.67_{\pm1.93}$ | $79.73_{\pm1.67}$ | $80.53_{\pm1.54}$ | $80.53_{\pm1.54}$ | $80.53_{\pm1.54}$ | $74.33_{\pm4.29}$ | $80.53_{\pm1.54}$ | $80.53_{\pm1.54}$ | $74.33_{\pm4.29}$ | $74.33_{\pm4.29}$ |
| trace | $\mathbf{70.60}_{\pm2.29}$ | $86.73_{\pm0.83}$ | $81.80_{\pm0.77}$ | $82.27_{\pm0.98}$ | $82.27_{\pm0.98}$ | $82.27_{\pm0.98}$ | $70.60_{\pm2.29}$ | $82.27_{\pm0.98}$ | $82.27_{\pm0.98}$ | $70.60_{\pm2.29}$ | $82.27_{\pm0.98}$ |
| conal | $79.53_{\pm1.71}$ | $85.87_{\pm1.89}$ | $82.07_{\pm1.85}$ | $80.87_{\pm1.19}$ | $80.87_{\pm1.19}$ | $80.87_{\pm1.19}$ | $79.53_{\pm1.71}$ | $80.87_{\pm1.19}$ | $80.87_{\pm1.19}$ | $79.53_{\pm1.71}$ | $80.87_{\pm1.19}$ |
| union-a | $71.07_{\pm3.57}$ | $84.13_{\pm1.39}$ | $79.47_{\pm0.77}$ | $75.93_{\pm3.85}$ | $75.93_{\pm3.85}$ | $75.93_{\pm3.85}$ | $74.20_{\pm2.34}$ | $75.93_{\pm3.85}$ | $75.93_{\pm3.85}$ | $71.07_{\pm3.57}$ | $75.93_{\pm3.85}$ |
| union-b | $78.87_{\pm1.63}$ | $85.73_{\pm1.04}$ | $80.20_{\pm2.13}$ | $78.93_{\pm2.63}$ | $78.93_{\pm2.63}$ | $78.93_{\pm2.63}$ | $85.80_{\pm4.56}$ | $78.93_{\pm2.63}$ | $78.93_{\pm2.63}$ | $87.93_{\pm0.15}$ | $78.93_{\pm2.63}$ |
| geo-w | $75.60_{\pm1.06}$ | $84.87_{\pm1.50}$ | $80.00_{\pm2.04}$ | $74.33_{\pm2.96}$ | $74.33_{\pm2.96}$ | $75.60_{\pm1.06}$ | $74.33_{\pm2.96}$ | $74.33_{\pm2.96}$ | $74.33_{\pm2.96}$ | $74.33_{\pm2.96}$ | $74.33_{\pm2.96}$ |
| geo-f | $71.73_{\pm4.79}$ | $84.60_{\pm1.53}$ | $\mathbf{79.27}_{\pm1.01}$ | $\mathbf{69.80}_{\pm3.18}$ | $\mathbf{69.80}_{\pm3.18}$ | $\mathbf{69.80}_{\pm3.18}$ | $69.80_{\pm3.18}$ | $\mathbf{73.60}_{\pm3.46}$ | $\mathbf{73.60}_{\pm3.46}$ | $69.80_{\pm3.18}$ | $\mathbf{73.60}_{\pm3.46}$ |
| madl | $72.07_{\pm4.01}$ | $87.53_{\pm0.69}$ | $82.27_{\pm1.23}$ | $80.67_{\pm2.36}$ | $80.67_{\pm2.36}$ | $80.67_{\pm2.36}$ | $\mathbf{69.13}_{\pm3.04}$ | $81.27_{\pm1.66}$ | $81.27_{\pm1.66}$ | $\mathbf{69.13}_{\pm3.04}$ | $80.67_{\pm2.36}$ |
| crowd-ar | $80.13_{\pm1.98}$ | $84.67_{\pm1.37}$ | $81.33_{\pm0.85}$ | $78.53_{\pm1.35}$ | $78.53_{\pm1.35}$ | $78.53_{\pm1.35}$ | $89.07_{\pm2.65}$ | $78.53_{\pm1.35}$ | $78.53_{\pm1.35}$ | $89.07_{\pm2.65}$ | $78.53_{\pm1.35}$ |
| annot-mix | $72.40_{\pm3.96}$ | $87.53_{\pm1.91}$ | $80.27_{\pm1.79}$ | $79.87_{\pm2.54}$ | $79.87_{\pm2.54}$ | $79.87_{\pm2.54}$ | $72.40_{\pm3.96}$ | $78.20_{\pm5.50}$ | $78.20_{\pm5.50}$ | $72.40_{\pm3.96}$ | $78.20_{\pm5.50}$ |
| coin | $83.60_{\pm4.95}$ | $\mathbf{83.00}_{\pm4.17}$ | $83.53_{\pm5.68}$ | $80.20_{\pm1.80}$ | $80.20_{\pm1.80}$ | $80.20_{\pm1.80}$ | $79.67_{\pm1.43}$ | $82.80_{\pm6.96}$ | $82.80_{\pm6.96}$ | $82.73_{\pm5.65}$ | $75.40_{\pm2.42}$ |
| | | | | | | mgc-worst-2 | | | | | |
| gt | $19.27_{\pm1.44}$ | $19.27_{\pm1.44}$ | $24.60_{\pm1.12}$ | $19.27_{\pm1.44}$ | $19.27_{\pm1.44}$ | N/A | N/A | N/A | N/A | N/A | $19.27_{\pm1.44}$ |
| mv | $58.53_{\pm2.54}$ | $68.27_{\pm0.86}$ | $58.53_{\pm2.54}$ | $58.60_{\pm1.92}$ | $58.60_{\pm1.92}$ | N/A | N/A | N/A | N/A | N/A | $58.60_{\pm1.92}$ |
| ds | $72.93_{\pm2.13}$ | $79.93_{\pm0.80}$ | $73.47_{\pm2.04}$ | $73.27_{\pm2.25}$ | $73.27_{\pm2.25}$ | $73.27_{\pm2.25}$ | $73.27_{\pm2.25}$ | $73.40_{\pm0.86}$ | $73.40_{\pm0.86}$ | $73.27_{\pm2.25}$ | $73.27_{\pm2.25}$ |
| cl | $52.93_{\pm1.09}$ | $50.73_{\pm2.64}$ | $57.80_{\pm1.12}$ | $60.87_{\pm3.46}$ | $60.87_{\pm3.46}$ | $53.60_{\pm1.62}$ | $57.07_{\pm2.50}$ | $52.93_{\pm1.09}$ | $52.93_{\pm1.09}$ | $53.60_{\pm1.62}$ | $53.60_{\pm1.62}$ |
| trace | $47.93_{\pm1.52}$ | $60.60_{\pm1.55}$ | $59.33_{\pm1.56}$ | $59.47_{\pm2.48}$ | $59.47_{\pm2.48}$ | $47.93_{\pm1.52}$ | $47.93_{\pm1.52}$ | $47.93_{\pm1.52}$ | $59.47_{\pm2.48}$ | $47.93_{\pm1.52}$ | $47.93_{\pm1.52}$ |
| conal | $53.33_{\pm1.03}$ | $55.53_{\pm2.41}$ | $55.80_{\pm0.90}$ | $59.33_{\pm1.75}$ | $54.67_{\pm1.13}$ | $55.73_{\pm1.59}$ | $57.60_{\pm1.77}$ | $54.67_{\pm1.13}$ | $54.67_{\pm1.13}$ | $54.67_{\pm1.13}$ | $54.67_{\pm1.13}$ |
| union-a | $52.07_{\pm1.38}$ | $51.67_{\pm0.97}$ | $\mathbf{54.73}_{\pm1.14}$ | $52.07_{\pm1.38}$ | $52.07_{\pm1.38}$ | $59.20_{\pm3.85}$ | $52.07_{\pm1.38}$ | $48.93_{\pm2.15}$ | $48.93_{\pm2.15}$ | $52.07_{\pm1.38}$ | $51.47_{\pm1.28}$ |
| union-b | $58.80_{\pm2.22}$ | $61.47_{\pm2.29}$ | $58.40_{\pm1.24}$ | $59.87_{\pm2.19}$ | $58.80_{\pm2.22}$ | $58.80_{\pm2.22}$ | $59.53_{\pm2.64}$ | $58.80_{\pm2.22}$ | $56.73_{\pm0.86}$ | $59.53_{\pm2.64}$ | $58.80_{\pm2.22}$ |
| geo-w | $56.27_{\pm1.72}$ | $49.07_{\pm0.72}$ | $57.47_{\pm1.73}$ | $56.87_{\pm0.80}$ | $56.27_{\pm1.95}$ | $56.27_{\pm1.95}$ | $56.27_{\pm1.72}$ | $50.13_{\pm1.71}$ | $56.27_{\pm1.95}$ | $56.27_{\pm1.72}$ | $49.93_{\pm1.42}$ |
| geo-f | $54.40_{\pm2.23}$ | $\mathbf{45.20}_{\pm1.64}$ | $55.87_{\pm1.50}$ | $57.13_{\pm1.39}$ | $57.13_{\pm1.39}$ | $53.00_{\pm1.33}$ | $54.40_{\pm2.23}$ | $54.93_{\pm1.66}$ | $54.93_{\pm1.66}$ | $54.40_{\pm2.23}$ | $48.33_{\pm2.19}$ |
| madl | $47.40_{\pm3.87}$ | $58.20_{\pm4.74}$ | $59.93_{\pm0.55}$ | $\mathbf{47.40}_{\pm3.87}$ | $\mathbf{47.40}_{\pm3.87}$ | $\mathbf{47.40}_{\pm3.87}$ | $\mathbf{47.40}_{\pm3.87}$ | $\mathbf{47.40}_{\pm3.87}$ | $\mathbf{47.40}_{\pm3.87}$ | $\mathbf{47.40}_{\pm3.87}$ | $\mathbf{47.40}_{\pm3.87}$ |
| crowd-ar | $57.00_{\pm1.56}$ | $53.00_{\pm3.01}$ | $56.67_{\pm1.62}$ | $57.47_{\pm1.71}$ | $57.47_{\pm1.71}$ | $54.33_{\pm1.18}$ | $54.33_{\pm1.18}$ | $57.47_{\pm1.71}$ | $57.47_{\pm1.71}$ | $57.47_{\pm1.71}$ | $57.47_{\pm1.71}$ |
| annot-mix | $\mathbf{44.87}_{\pm1.73}$ | $45.60_{\pm1.62}$ | $57.07_{\pm1.19}$ | $47.87_{\pm3.50}$ | $47.87_{\pm3.50}$ | $\mathbf{44.87}_{\pm1.73}$ | $49.00_{\pm0.97}$ | $47.87_{\pm3.50}$ | $47.87_{\pm3.50}$ | $\mathbf{44.87}_{\pm1.73}$ | $47.87_{\pm3.50}$ |
| coin | $45.93_{\pm1.62}$ | $51.27_{\pm6.24}$ | $61.80_{\pm7.12}$ | $57.53_{\pm0.77}$ | $53.53_{\pm1.74}$ | $46.93_{\pm1.19}$ | $53.67_{\pm3.97}$ | $\mathbf{45.93}_{\pm1.62}$ | $\mathbf{45.80}_{\pm2.12}$ | $53.67_{\pm3.97}$ | $\mathbf{45.80}_{\pm2.12}$ |
| | | | | | | mgc-worst-v | | | | | |
| gt | $18.53_{\pm0.73}$ | $18.53_{\pm0.73}$ | $24.60_{\pm1.12}$ | $19.93_{\pm0.60}$ | $19.93_{\pm0.60}$ | N/A | N/A | N/A | N/A | N/A | $19.93_{\pm0.60}$ |
| mv | $53.73_{\pm1.91}$ | $56.07_{\pm1.38}$ | $50.53_{\pm2.18}$ | $50.53_{\pm2.18}$ | $50.53_{\pm0.84}$ | N/A | N/A | N/A | N/A | N/A | $53.73_{\pm1.91}$ |
| ds | $51.87_{\pm0.38}$ | $58.80_{\pm1.98}$ | $52.67_{\pm1.62}$ | $50.93_{\pm1.94}$ | $50.93_{\pm1.94}$ | $50.93_{\pm1.94}$ | $50.93_{\pm1.94}$ | $53.93_{\pm1.53}$ | $53.93_{\pm1.53}$ | $53.93_{\pm1.53}$ | $53.93_{\pm1.21}$ |
| cl | $42.60_{\pm1.71}$ | $47.53_{\pm1.43}$ | $48.47_{\pm1.35}$ | $45.27_{\pm0.55}$ | $45.27_{\pm0.55}$ | $50.13_{\pm2.85}$ | $50.13_{\pm2.85}$ | $46.87_{\pm1.68}$ | $45.27_{\pm0.55}$ | $45.80_{\pm1.26}$ | $45.27_{\pm0.55}$ |
| trace | $40.00_{\pm2.10}$ | $41.53_{\pm1.26}$ | $47.53_{\pm2.05}$ | $48.00_{\pm0.62}$ | $48.00_{\pm0.62}$ | $40.00_{\pm2.10}$ | $40.00_{\pm2.10}$ | $47.60_{\pm2.60}$ | $47.60_{\pm2.60}$ | $40.87_{\pm1.07}$ | $47.60_{\pm2.60}$ |
| conal | $44.00_{\pm0.53}$ | $42.53_{\pm1.76}$ | $46.27_{\pm1.48}$ | $45.67_{\pm1.25}$ | $45.67_{\pm1.25}$ | $43.73_{\pm1.19}$ | $45.47_{\pm2.75}$ | $45.67_{\pm1.25}$ | $45.67_{\pm1.25}$ | $44.00_{\pm0.53}$ | $44.00_{\pm0.53}$ |
| union-a | $41.93_{\pm0.83}$ | $43.67_{\pm2.15}$ | $46.87_{\pm1.92}$ | $43.07_{\pm0.76}$ | $43.07_{\pm0.76}$ | $43.07_{\pm0.76}$ | $41.67_{\pm1.94}$ | $43.07_{\pm0.76}$ | $43.07_{\pm0.76}$ | $42.13_{\pm2.06}$ | $43.07_{\pm0.76}$ |
| union-b | $44.33_{\pm1.11}$ | $49.40_{\pm1.09}$ | $47.80_{\pm0.84}$ | $44.67_{\pm1.90}$ | $46.33_{\pm1.31}$ | $48.13_{\pm1.35}$ | $48.13_{\pm1.35}$ | $43.60_{\pm0.86}$ | $43.53_{\pm1.35}$ | $44.13_{\pm2.19}$ | $48.13_{\pm1.35}$ |
| geo-w | $40.07_{\pm1.99}$ | $42.20_{\pm2.17}$ | $47.47_{\pm0.87}$ | $41.47_{\pm0.93}$ | $41.47_{\pm0.93}$ | $42.93_{\pm1.99}$ | $40.07_{\pm1.99}$ | $39.80_{\pm0.84}$ | $42.13_{\pm0.96}$ | $39.80_{\pm0.84}$ | $42.93_{\pm1.99}$ |
| geo-f | $38.00_{\pm2.78}$ | $38.93_{\pm0.60}$ | $\mathbf{45.33}_{\pm1.78}$ | $41.13_{\pm0.87}$ | $41.13_{\pm0.87}$ | $41.60_{\pm2.41}$ | $41.60_{\pm2.41}$ | $39.80_{\pm1.56}$ | $39.80_{\pm1.56}$ | $39.80_{\pm1.56}$ | $41.13_{\pm0.87}$ |
| madl | $39.20_{\pm3.16}$ | $42.40_{\pm1.85}$ | $47.80_{\pm1.19}$ | $39.20_{\pm3.16}$ | $39.20_{\pm3.16}$ | $39.20_{\pm3.16}$ | $\mathbf{39.20}_{\pm3.16}$ | $39.20_{\pm3.16}$ | $\mathbf{39.20}_{\pm3.16}$ | $39.20_{\pm3.16}$ | $39.20_{\pm3.16}$ |
| crowd-ar | $43.33_{\pm0.97}$ | $44.87_{\pm2.26}$ | $48.07_{\pm0.60}$ | $50.13_{\pm3.27}$ | $50.13_{\pm3.27}$ | $50.13_{\pm3.27}$ | $50.13_{\pm3.27}$ | $50.13_{\pm3.27}$ | $50.13_{\pm3.27}$ | $50.13_{\pm3.27}$ | $50.13_{\pm3.27}$ |
| annot-mix | $\mathbf{37.27}_{\pm1.67}$ | $\mathbf{38.07}_{\pm1.57}$ | $48.00_{\pm1.33}$ | $38.13_{\pm0.80}$ | $38.13_{\pm0.80}$ | $38.13_{\pm0.80}$ | $39.93_{\pm1.09}$ | $\mathbf{39.13}_{\pm1.09}$ | $39.60_{\pm1.19}$ | $38.13_{\pm0.80}$ | $\mathbf{38.13}_{\pm0.80}$ |
| coin | $42.07_{\pm3.02}$ | $40.53_{\pm1.82}$ | $51.27_{\pm5.29}$ | $39.73_{\pm1.09}$ | $39.73_{\pm1.09}$ | $39.73_{\pm1.09}$ | $46.00_{\pm2.44}$ | $39.73_{\pm1.09}$ | $42.40_{\pm1.62}$ | $39.73_{\pm1.09}$ | $39.73_{\pm1.09}$ |
| | | | | | | mgc-rand-1 | | | | | |
| gt | $18.67_{\pm1.16}$ | $18.67_{\pm1.16}$ | $24.60_{\pm1.12}$ | $21.13_{\pm1.98}$ | $21.13_{\pm1.98}$ | N/A | N/A | N/A | N/A | N/A | $21.13_{\pm1.98}$ |
| mv | $39.20_{\pm1.71}$ | $40.13_{\pm2.22}$ | $40.07_{\pm2.10}$ | $40.07_{\pm2.10}$ | $40.07_{\pm2.10}$ | N/A | N/A | N/A | N/A | N/A | $40.07_{\pm2.10}$ |
| ds | $39.20_{\pm1.71}$ | $40.13_{\pm2.22}$ | $40.07_{\pm2.10}$ | $40.07_{\pm2.10}$ | $40.07_{\pm2.10}$ | $40.07_{\pm2.10}$ | $40.07_{\pm2.10}$ | $40.07_{\pm2.10}$ | $40.07_{\pm2.10}$ | $40.07_{\pm2.10}$ | $40.07_{\pm2.10}$ |
| cl | $38.67_{\pm2.30}$ | $49.67_{\pm4.50}$ | $41.60_{\pm1.98}$ | $38.67_{\pm2.30}$ | $38.67_{\pm2.30}$ | $41.00_{\pm4.12}$ | $49.67_{\pm4.50}$ | $41.20_{\pm3.06}$ | $41.20_{\pm3.06}$ | $49.67_{\pm4.50}$ | $41.00_{\pm4.12}$ |
| trace | $41.07_{\pm1.67}$ | $36.73_{\pm1.55}$ | $40.00_{\pm2.44}$ | $39.00_{\pm1.45}$ | $39.00_{\pm1.45}$ | $39.00_{\pm1.45}$ | $41.07_{\pm1.67}$ | $39.00_{\pm1.45}$ | $39.00_{\pm1.45}$ | $41.07_{\pm1.67}$ | $39.00_{\pm1.45}$ |
| conal | $37.53_{\pm0.90}$ | $38.87_{\pm1.80}$ | $40.33_{\pm2.19}$ | $37.07_{\pm2.01}$ | $37.07_{\pm2.01}$ | $37.07_{\pm2.01}$ | $38.00_{\pm1.78}$ | $37.07_{\pm2.01}$ | $37.07_{\pm2.01}$ | $38.00_{\pm1.78}$ | $37.07_{\pm2.01}$ |
| union-a | $33.53_{\pm0.96}$ | $40.80_{\pm4.78}$ | $\mathbf{36.60}_{\pm3.01}$ | $35.53_{\pm3.96}$ | $35.53_{\pm3.96}$ | $\mathbf{35.53}_{\pm3.96}$ | $45.93_{\pm2.13}$ | $35.53_{\pm3.96}$ | $35.53_{\pm3.96}$ | $44.00_{\pm6.58}$ | $35.53_{\pm3.96}$ |
| union-b | $39.40_{\pm1.12}$ | $49.67_{\pm1.79}$ | $40.80_{\pm1.68}$ | $38.80_{\pm1.28}$ | $38.80_{\pm1.28}$ | $38.80_{\pm1.28}$ | $52.47_{\pm1.45}$ | $38.80_{\pm1.28}$ | $38.80_{\pm1.28}$ | $52.47_{\pm1.45}$ | $38.80_{\pm1.28}$ |
| geo-w | $37.40_{\pm0.80}$ | $41.07_{\pm4.81}$ | $40.00_{\pm1.49}$ | $37.40_{\pm0.80}$ | $37.40_{\pm0.80}$ | $37.40_{\pm0.80}$ | $43.80_{\pm2.77}$ | $39.40_{\pm1.09}$ | $39.40_{\pm1.09}$ | $45.60_{\pm4.71}$ | $37.53_{\pm1.76}$ |
| geo-f | $35.33_{\pm0.41}$ | $39.07_{\pm1.91}$ | $37.73_{\pm1.48}$ | $37.13_{\pm1.94}$ | $37.13_{\pm1.94}$ | $37.13_{\pm1.94}$ | $38.60_{\pm1.59}$ | $35.87_{\pm1.26}$ | $35.87_{\pm1.26}$ | $38.60_{\pm1.59}$ | $37.13_{\pm1.94}$ |
| madl | $35.07_{\pm1.48}$ | $\mathbf{36.00}_{\pm2.07}$ | $40.20_{\pm1.54}$ | $36.27_{\pm2.75}$ | $36.27_{\pm2.75}$ | $36.27_{\pm2.75}$ | $\mathbf{35.07}_{\pm1.48}$ | $40.20_{\pm1.54}$ | $40.20_{\pm1.54}$ | $\mathbf{35.07}_{\pm1.48}$ | $36.27_{\pm2.75}$ |
| crowd-ar | $39.07_{\pm2.88}$ | $44.60_{\pm2.13}$ | $39.73_{\pm1.09}$ | $36.87_{\pm2.97}$ | $36.87_{\pm2.97}$ | $36.87_{\pm2.97}$ | $39.07_{\pm2.88}$ | $36.87_{\pm2.97}$ | $36.87_{\pm2.97}$ | $39.07_{\pm2.88}$ | $36.87_{\pm2.97}$ |
| annot-mix | $\mathbf{33.27}_{\pm1.32}$ | $37.13_{\pm3.32}$ | $39.00_{\pm1.79}$ | $\mathbf{33.27}_{\pm1.32}$ | $\mathbf{33.27}_{\pm1.32}$ | $36.53_{\pm0.51}$ | $36.53_{\pm0.51}$ | $\mathbf{33.27}_{\pm1.32}$ | $\mathbf{33.27}_{\pm1.32}$ | $36.53_{\pm0.51}$ | $\mathbf{33.27}_{\pm1.32}$ |
| coin | $35.53_{\pm1.77}$ | $40.93_{\pm3.78}$ | $44.87_{\pm4.31}$ | $35.53_{\pm1.77}$ | $35.53_{\pm1.77}$ | $\mathbf{35.53}_{\pm1.77}$ | $39.27_{\pm2.10}$ | $35.53_{\pm1.77}$ | $35.53_{\pm1.77}$ | $39.27_{\pm2.10}$ | $35.53_{\pm1.77}$ |

Table 14: **Zero-one loss results [%] (part II).** Continued from the previous page.

| Approach | Baseline | | | Aggregation-level | | | | Crowd-level | | | Ensemble |
|---|---|---|---|---|---|---|---|---|---|---|---|
| | TRUE$^\star$ | DEF-DATA$^\star$ | DEF | AEU | AEC | ALU | ALC | CXU | CEC | CLC | ENS |
| **mgc-rand-2** | | | | | | | | | | | |
| gt | $19.07_{\pm0.28}$ | $19.07_{\pm0.28}$ | $24.60_{\pm1.12}$ | $18.93_{\pm0.76}$ | $18.93_{\pm0.76}$ | N/A | N/A | N/A | N/A | N/A | $18.93_{\pm0.76}$ |
| mv | $40.67_{\pm0.78}$ | $43.60_{\pm1.21}$ | $\underline{38.20}_{\pm2.06}$ | $40.67_{\pm0.78}$ | $40.67_{\pm0.78}$ | N/A | N/A | N/A | N/A | N/A | $40.67_{\pm0.78}$ |
| ds | $39.53_{\pm2.09}$ | $44.53_{\pm1.52}$ | $41.07_{\pm1.79}$ | $\underline{38.40}_{\pm0.72}$ | $\underline{38.40}_{\pm0.72}$ | $39.53_{\pm2.09}$ | $40.27_{\pm2.89}$ | $42.13_{\pm0.77}$ | $42.13_{\pm0.77}$ | $42.13_{\pm0.77}$ | $39.53_{\pm2.09}$ |
| cl | $34.80_{\pm0.77}$ | $37.40_{\pm2.46}$ | $39.07_{\pm1.46}$ | $\underline{33.60}_{\pm0.89}$ | $\underline{33.60}_{\pm0.89}$ | $34.80_{\pm0.77}$ | $35.20_{\pm1.12}$ | $34.93_{\pm1.50}$ | $34.93_{\pm1.50}$ | $35.20_{\pm1.12}$ | $34.80_{\pm0.77}$ |
| trace | $33.47_{\pm1.07}$ | $35.00_{\pm1.05}$ | $35.87_{\pm2.17}$ | $\underline{33.47}_{\pm1.07}$ | $\underline{33.47}_{\pm1.07}$ | $33.47_{\pm1.07}$ | $33.47_{\pm1.07}$ | $36.00_{\pm3.26}$ | $36.53_{\pm1.45}$ | $\underline{33.47}_{\pm1.07}$ | $36.53_{\pm1.45}$ |
| conal | $34.07_{\pm1.19}$ | $35.13_{\pm1.73}$ | $37.00_{\pm2.01}$ | $35.00_{\pm1.51}$ | $35.00_{\pm1.51}$ | $35.00_{\pm1.51}$ | $35.60_{\pm0.86}$ | $35.00_{\pm0.91}$ | $\underline{34.07}_{\pm1.19}$ | $35.00_{\pm0.71}$ | $35.60_{\pm0.86}$ |
| union-a | $33.00_{\pm1.05}$ | $33.67_{\pm1.03}$ | $35.40_{\pm2.37}$ | $33.27_{\pm1.26}$ | $33.27_{\pm1.26}$ | $33.27_{\pm1.26}$ | $33.80_{\pm0.77}$ | $\underline{32.00}_{\pm0.47}$ | $\underline{32.00}_{\pm0.47}$ | $33.80_{\pm0.77}$ | $33.80_{\pm0.77}$ |
| union-b | $34.47_{\pm1.48}$ | $40.00_{\pm1.58}$ | $37.93_{\pm1.32}$ | $35.47_{\pm1.46}$ | $35.47_{\pm1.46}$ | $35.47_{\pm1.46}$ | $\underline{34.87}_{\pm1.17}$ | $36.53_{\pm1.30}$ | $36.53_{\pm1.30}$ | $37.00_{\pm1.93}$ | $\underline{34.87}_{\pm1.17}$ |
| geo-w | $33.60_{\pm0.55}$ | $34.27_{\pm0.76}$ | $36.80_{\pm1.50}$ | $32.73_{\pm0.98}$ | $\underline{32.40}_{\pm1.85}$ | $33.13_{\pm0.96}$ | $33.13_{\pm0.96}$ | $\underline{32.40}_{\pm1.85}$ | $\underline{32.40}_{\pm1.85}$ | $34.27_{\pm0.55}$ | $33.13_{\pm0.96}$ |
| geo-f | $34.33_{\pm4.97}$ | $34.87_{\pm1.97}$ | $\mathbf{34.40}_{\pm0.98}$ | $33.53_{\pm1.39}$ | $\underline{33.07}_{\pm1.30}$ | $33.20_{\pm1.28}$ | $33.20_{\pm1.28}$ | $33.40_{\pm1.09}$ | $33.40_{\pm1.09}$ | $\underline{33.07}_{\pm1.30}$ | $33.20_{\pm1.28}$ |
| madl | $34.40_{\pm2.86}$ | $38.60_{\pm1.38}$ | $35.53_{\pm2.29}$ | $\underline{33.53}_{\pm0.90}$ | $\underline{33.53}_{\pm0.90}$ | $\underline{33.53}_{\pm0.90}$ | $34.40_{\pm2.86}$ | $\underline{33.53}_{\pm0.90}$ | $34.87_{\pm1.77}$ | $\underline{33.53}_{\pm0.90}$ | $\underline{33.53}_{\pm0.90}$ |
| crowd-ar | $34.53_{\pm1.09}$ | $36.47_{\pm0.51}$ | $37.40_{\pm3.04}$ | $\underline{35.00}_{\pm0.97}$ | $\underline{35.00}_{\pm0.97}$ | $\underline{35.00}_{\pm0.97}$ | $\underline{35.00}_{\pm0.97}$ | $\underline{35.00}_{\pm0.97}$ | $\underline{35.00}_{\pm0.97}$ | $\underline{35.00}_{\pm0.97}$ | $\underline{35.00}_{\pm0.97}$ |
| annot-mix | $33.07_{\pm1.69}$ | $\mathbf{31.33}_{\pm1.11}$ | $35.67_{\pm2.51}$ | $\mathbf{30.80}_{\pm1.85}$ | $\mathbf{30.80}_{\pm1.85}$ | $31.00_{\pm1.62}$ | $31.00_{\pm1.62}$ | $\mathbf{30.80}_{\pm1.85}$ | $\mathbf{30.80}_{\pm1.85}$ | $\mathbf{30.80}_{\pm1.85}$ | $\mathbf{30.80}_{\pm1.85}$ |
| coin | $\mathbf{32.93}_{\pm2.38}$ | $\underline{31.53}_{\pm1.61}$ | $41.93_{\pm4.82}$ | $33.00_{\pm1.43}$ | $33.00_{\pm1.43}$ | $33.00_{\pm1.43}$ | $33.00_{\pm1.43}$ | $32.93_{\pm2.38}$ | $32.93_{\pm2.38}$ | $31.60_{\pm2.28}$ | $33.00_{\pm1.43}$ |
| **mgc-rand-v** | | | | | | | | | | | |
| gt | $19.40_{\pm0.36}$ | $19.40_{\pm0.36}$ | $24.60_{\pm1.12}$ | $20.00_{\pm0.91}$ | $20.00_{\pm0.91}$ | N/A | N/A | N/A | N/A | N/A | $20.00_{\pm0.91}$ |
| mv | $36.47_{\pm1.50}$ | $\underline{35.40}_{\pm2.35}$ | $36.80_{\pm0.69}$ | $35.67_{\pm3.57}$ | $36.47_{\pm1.52}$ | N/A | N/A | N/A | N/A | N/A | $36.47_{\pm1.52}$ |
| ds | $38.73_{\pm0.55}$ | $\underline{36.87}_{\pm1.26}$ | $37.67_{\pm0.91}$ | $38.73_{\pm0.55}$ | $38.73_{\pm0.55}$ | $38.00_{\pm1.78}$ | $38.00_{\pm1.78}$ | $38.00_{\pm1.78}$ | $38.00_{\pm1.78}$ | $38.00_{\pm1.78}$ | $38.00_{\pm1.78}$ |
| cl | $31.80_{\pm0.61}$ | $39.80_{\pm1.26}$ | $36.00_{\pm3.50}$ | $36.40_{\pm0.60}$ | $36.40_{\pm0.60}$ | $\underline{31.80}_{\pm0.61}$ | $\underline{31.80}_{\pm0.61}$ | $33.07_{\pm1.16}$ | $33.07_{\pm1.16}$ | $33.80_{\pm1.15}$ | $33.07_{\pm1.16}$ |
| trace | $31.07_{\pm1.36}$ | $32.67_{\pm0.62}$ | $35.60_{\pm1.92}$ | $\mathbf{31.53}_{\pm0.69}$ | $\mathbf{31.53}_{\pm0.69}$ | $\mathbf{31.53}_{\pm0.69}$ | $\underline{31.07}_{\pm1.36}$ | $36.27_{\pm2.22}$ | $36.27_{\pm2.22}$ | $31.53_{\pm0.69}$ | $36.27_{\pm2.22}$ |
| conal | $34.80_{\pm2.42}$ | $35.20_{\pm1.73}$ | $\underline{\mathbf{33.67}}_{\pm0.53}$ | $34.33_{\pm2.53}$ | $34.33_{\pm2.53}$ | $34.33_{\pm2.53}$ | $34.40_{\pm1.14}$ | $35.80_{\pm1.07}$ | $34.33_{\pm2.53}$ | $34.80_{\pm2.42}$ | $34.33_{\pm2.53}$ |
| union-a | $\mathbf{29.53}_{\pm1.04}$ | $37.20_{\pm3.35}$ | $35.13_{\pm2.39}$ | $\underline{31.73}_{\pm0.98}$ | $\underline{31.73}_{\pm0.98}$ | $\underline{31.73}_{\pm0.98}$ | $31.73_{\pm0.98}$ | $\underline{\mathbf{31.73}}_{\pm0.98}$ | $\underline{31.73}_{\pm0.98}$ | $31.73_{\pm0.98}$ | $31.73_{\pm0.98}$ |
| union-b | $34.53_{\pm1.32}$ | $39.53_{\pm1.77}$ | $35.40_{\pm1.59}$ | $\underline{33.20}_{\pm1.28}$ | $\underline{33.20}_{\pm1.28}$ | $33.20_{\pm1.28}$ | $38.07_{\pm3.93}$ | $34.00_{\pm2.17}$ | $36.07_{\pm0.83}$ | $34.53_{\pm1.32}$ | $36.07_{\pm0.83}$ |
| geo-w | $32.20_{\pm0.93}$ | $34.73_{\pm2.02}$ | $35.33_{\pm1.75}$ | $\underline{32.20}_{\pm0.93}$ | $32.40_{\pm1.30}$ | $\underline{32.20}_{\pm0.93}$ | $32.67_{\pm1.11}$ | $32.40_{\pm1.30}$ | $32.40_{\pm1.30}$ | $32.40_{\pm1.30}$ | $\underline{32.20}_{\pm0.93}$ |
| geo-f | $31.13_{\pm1.07}$ | $34.07_{\pm1.44}$ | $34.87_{\pm1.52}$ | $33.00_{\pm1.72}$ | $33.00_{\pm1.72}$ | $32.67_{\pm1.55}$ | $31.20_{\pm0.73}$ | $32.67_{\pm1.55}$ | $32.67_{\pm1.55}$ | $\underline{31.13}_{\pm1.07}$ | $32.67_{\pm1.55}$ |
| madl | $32.93_{\pm1.12}$ | $\underline{32.00}_{\pm1.62}$ | $36.27_{\pm0.89}$ | $34.13_{\pm2.15}$ | $35.47_{\pm1.80}$ | $35.67_{\pm2.96}$ | $32.93_{\pm1.12}$ | $36.40_{\pm2.25}$ | $36.40_{\pm2.25}$ | $32.93_{\pm1.12}$ | $34.13_{\pm2.15}$ |
| crowd-ar | $33.60_{\pm1.19}$ | $36.87_{\pm1.98}$ | $35.33_{\pm1.00}$ | $35.87_{\pm1.19}$ | $35.87_{\pm1.19}$ | $35.87_{\pm1.19}$ | $35.87_{\pm1.19}$ | $35.87_{\pm1.19}$ | $35.87_{\pm1.19}$ | $34.13_{\pm1.15}$ | $33.60_{\pm1.19}$ |
| annot-mix | $30.40_{\pm1.50}$ | $31.93_{\pm2.18}$ | $36.00_{\pm1.96}$ | $31.93_{\pm0.93}$ | $31.93_{\pm0.93}$ | $31.93_{\pm0.93}$ | $31.93_{\pm0.93}$ | $31.60_{\pm1.62}$ | $\mathbf{31.73}_{\pm0.86}$ | $\underline{30.60}_{\pm0.15}$ | $31.93_{\pm0.93}$ |
| coin | $33.93_{\pm2.02}$ | $37.60_{\pm4.75}$ | $40.00_{\pm3.46}$ | $33.93_{\pm2.02}$ | $33.93_{\pm2.02}$ | $33.20_{\pm3.00}$ | $33.20_{\pm3.00}$ | $32.00_{\pm1.00}$ | $32.00_{\pm1.00}$ | $32.00_{\pm1.00}$ | $\mathbf{30.87}_{\pm0.77}$ |
| **mgc-full** | | | | | | | | | | | |
| gt | $20.20_{\pm0.96}$ | $20.20_{\pm0.96}$ | $24.60_{\pm1.12}$ | $20.60_{\pm0.28}$ | $20.60_{\pm0.28}$ | N/A | N/A | N/A | N/A | N/A | $20.60_{\pm0.28}$ |
| mv | $34.67_{\pm1.62}$ | $37.73_{\pm1.59}$ | $36.00_{\pm1.33}$ | $\underline{34.67}_{\pm1.62}$ | $\underline{34.67}_{\pm1.62}$ | N/A | N/A | N/A | N/A | N/A | $\underline{34.67}_{\pm1.62}$ |
| ds | $30.40_{\pm1.38}$ | $33.00_{\pm0.82}$ | $\mathbf{33.20}_{\pm1.57}$ | $32.80_{\pm1.35}$ | $31.33_{\pm0.62}$ | $31.33_{\pm0.62}$ | $31.33_{\pm0.62}$ | $31.67_{\pm0.62}$ | $31.67_{\pm0.62}$ | $\underline{31.00}_{\pm0.71}$ | $31.67_{\pm0.62}$ |
| cl | $31.40_{\pm1.04}$ | $\underline{30.47}_{\pm0.90}$ | $37.27_{\pm3.18}$ | $31.40_{\pm1.04}$ | $31.40_{\pm1.04}$ | $31.40_{\pm1.04}$ | $33.20_{\pm1.22}$ | $31.40_{\pm1.04}$ | $31.40_{\pm1.04}$ | $31.40_{\pm1.04}$ | $31.40_{\pm1.04}$ |
| trace | $29.20_{\pm1.69}$ | $35.33_{\pm2.43}$ | $34.07_{\pm1.19}$ | $30.07_{\pm0.64}$ | $30.07_{\pm0.64}$ | $30.07_{\pm0.64}$ | $\underline{29.20}_{\pm1.69}$ | $33.87_{\pm1.57}$ | $33.87_{\pm1.57}$ | $30.07_{\pm0.64}$ | $30.07_{\pm0.64}$ |
| conal | $31.60_{\pm1.34}$ | $\underline{30.47}_{\pm1.46}$ | $33.47_{\pm1.15}$ | $31.87_{\pm1.19}$ | $31.87_{\pm1.19}$ | $32.00_{\pm1.08}$ | $32.67_{\pm1.27}$ | $32.60_{\pm1.44}$ | $32.60_{\pm1.44}$ | $32.60_{\pm1.44}$ | $32.60_{\pm1.44}$ |
| union-a | $31.20_{\pm0.73}$ | $\underline{30.07}_{\pm1.79}$ | $34.47_{\pm2.81}$ | $31.20_{\pm0.73}$ | $31.20_{\pm0.73}$ | $30.53_{\pm1.07}$ | $32.00_{\pm0.82}$ | $31.20_{\pm0.73}$ | $31.20_{\pm0.73}$ | $31.20_{\pm1.04}$ | $31.20_{\pm0.73}$ |
| union-b | $31.07_{\pm0.72}$ | $\underline{30.93}_{\pm1.01}$ | $35.47_{\pm0.90}$ | $31.07_{\pm0.72}$ | $31.07_{\pm0.72}$ | $31.00_{\pm1.05}$ | $32.87_{\pm2.39}$ | $31.07_{\pm0.72}$ | $31.07_{\pm0.72}$ | $31.07_{\pm0.72}$ | $31.07_{\pm0.72}$ |
| geo-w | $30.93_{\pm2.22}$ | $30.40_{\pm1.09}$ | $35.13_{\pm0.96}$ | $30.60_{\pm1.94}$ | $30.60_{\pm1.94}$ | $\underline{30.33}_{\pm0.71}$ | $31.13_{\pm1.92}$ | $\underline{30.33}_{\pm0.71}$ | $30.60_{\pm1.94}$ | $31.27_{\pm1.19}$ | $30.60_{\pm1.94}$ |
| geo-f | $28.67_{\pm1.43}$ | $30.80_{\pm0.77}$ | $34.53_{\pm1.71}$ | $\underline{30.20}_{\pm0.96}$ | $31.13_{\pm0.69}$ | $30.27_{\pm1.38}$ | $30.73_{\pm0.43}$ | $31.13_{\pm0.69}$ | $30.73_{\pm0.43}$ | $30.93_{\pm1.23}$ | $\underline{30.20}_{\pm0.96}$ |
| madl | $29.13_{\pm1.77}$ | $31.00_{\pm1.03}$ | $34.93_{\pm1.82}$ | $31.33_{\pm1.35}$ | $\underline{29.73}_{\pm1.16}$ | $32.27_{\pm2.66}$ | $32.27_{\pm2.66}$ | $\underline{29.73}_{\pm1.16}$ | $\underline{29.73}_{\pm1.16}$ | $\underline{29.73}_{\pm1.16}$ | $\underline{29.73}_{\pm1.16}$ |
| crowd-ar | $31.67_{\pm1.33}$ | $\underline{31.20}_{\pm0.38}$ | $35.00_{\pm1.35}$ | $31.73_{\pm1.32}$ | $31.73_{\pm1.32}$ | $31.73_{\pm1.32}$ | $31.47_{\pm1.74}$ | $31.67_{\pm1.33}$ | $31.67_{\pm1.33}$ | $31.73_{\pm1.32}$ | $31.73_{\pm1.32}$ |
| annot-mix | $\mathbf{27.20}_{\pm2.04}$ | $\mathbf{28.13}_{\pm0.38}$ | $33.93_{\pm2.24}$ | $\mathbf{29.47}_{\pm0.96}$ | $\mathbf{29.47}_{\pm0.96}$ | $\mathbf{29.47}_{\pm0.96}$ | $27.80_{\pm1.43}$ | $26.80_{\pm0.90}$ | $26.80_{\pm0.90}$ | $27.60_{\pm0.72}$ | $\mathbf{26.20}_{\pm1.26}$ |
| coin | $31.60_{\pm2.44}$ | $30.20_{\pm1.89}$ | $40.00_{\pm2.79}$ | $31.80_{\pm1.24}$ | $31.80_{\pm1.24}$ | $30.00_{\pm0.71}$ | $\underline{28.80}_{\pm1.59}$ | $30.00_{\pm0.71}$ | $\underline{28.80}_{\pm1.59}$ | $\underline{28.80}_{\pm1.59}$ | $28.80_{\pm1.59}$ |
| **label-me-worst-1** | | | | | | | | | | | |
| gt | $6.40_{\pm0.27}$ | $6.40_{\pm0.27}$ | $6.31_{\pm0.27}$ | $9.48_{\pm0.85}$ | $9.48_{\pm0.85}$ | N/A | N/A | N/A | N/A | N/A | $9.48_{\pm0.85}$ |
| mv | $30.86_{\pm1.09}$ | $32.76_{\pm0.99}$ | $34.49_{\pm0.44}$ | $\underline{31.67}_{\pm0.70}$ | $\underline{31.67}_{\pm0.70}$ | N/A | N/A | N/A | N/A | N/A | $\underline{31.67}_{\pm0.70}$ |
| ds | $30.86_{\pm1.09}$ | $32.76_{\pm0.99}$ | $34.49_{\pm0.44}$ | $\underline{31.67}_{\pm0.70}$ | $\underline{31.67}_{\pm0.70}$ | $31.67_{\pm0.70}$ | $31.67_{\pm0.70}$ | $31.67_{\pm0.70}$ | $31.67_{\pm0.70}$ | $31.67_{\pm0.70}$ | $\underline{31.67}_{\pm0.70}$ |
| cl | $27.59_{\pm4.75}$ | $33.72_{\pm1.51}$ | $33.27_{\pm0.34}$ | $31.50_{\pm1.41}$ | $31.50_{\pm1.41}$ | $31.50_{\pm1.41}$ | $\underline{27.17}_{\pm0.74}$ | $31.50_{\pm1.41}$ | $31.50_{\pm1.41}$ | $27.59_{\pm4.75}$ | $31.50_{\pm1.41}$ |
| trace | $31.20_{\pm0.59}$ | $32.48_{\pm0.79}$ | $34.38_{\pm0.53}$ | $\underline{31.25}_{\pm0.81}$ | $\underline{31.25}_{\pm0.81}$ | $\underline{31.25}_{\pm0.81}$ | $\underline{31.25}_{\pm0.81}$ | $\underline{31.25}_{\pm0.81}$ | $\underline{31.25}_{\pm0.81}$ | $\underline{31.25}_{\pm0.81}$ | $31.25_{\pm0.76}$ |
| conal | $31.26_{\pm1.50}$ | $31.11_{\pm2.53}$ | $34.33_{\pm0.44}$ | $31.35_{\pm0.73}$ | $31.35_{\pm0.73}$ | $31.35_{\pm0.73}$ | $31.26_{\pm1.50}$ | $31.35_{\pm0.73}$ | $31.35_{\pm0.73}$ | $31.26_{\pm1.50}$ | $\mathbf{30.98}_{\pm0.76}$ |
| union-a | $\mathbf{22.64}_{\pm1.90}$ | $\underline{\mathbf{29.07}}_{\pm1.96}$ | $32.41_{\pm0.75}$ | $\mathbf{29.98}_{\pm1.72}$ | $\mathbf{29.98}_{\pm1.72}$ | $\mathbf{29.98}_{\pm1.72}$ | $29.98_{\pm1.72}$ | $\mathbf{29.41}_{\pm0.87}$ | $\mathbf{29.41}_{\pm0.87}$ | $29.73_{\pm2.20}$ | $\mathbf{29.98}_{\pm1.72}$ |
| union-b | $26.50_{\pm2.23}$ | $32.04_{\pm1.28}$ | $34.34_{\pm0.63}$ | $\underline{31.48}_{\pm1.24}$ | $\underline{31.48}_{\pm1.24}$ | $\underline{31.48}_{\pm1.24}$ | $\underline{31.48}_{\pm1.24}$ | $\underline{31.48}_{\pm1.24}$ | $\underline{31.48}_{\pm1.24}$ | $\underline{31.48}_{\pm1.24}$ | $\underline{31.48}_{\pm1.24}$ |
| geo-w | $28.16_{\pm0.53}$ | $32.17_{\pm1.30}$ | $34.41_{\pm0.55}$ | $31.57_{\pm1.27}$ | $31.57_{\pm1.27}$ | $31.57_{\pm1.27}$ | $\underline{29.02}_{\pm1.19}$ | $31.57_{\pm1.27}$ | $31.57_{\pm1.27}$ | $\underline{29.02}_{\pm1.19}$ | $31.57_{\pm1.27}$ |
| geo-f | $28.48_{\pm1.27}$ | $31.90_{\pm0.66}$ | $34.01_{\pm0.70}$ | $31.40_{\pm1.26}$ | $31.40_{\pm1.26}$ | $31.40_{\pm1.26}$ | $\underline{\mathbf{25.40}}_{\pm2.88}$ | $31.40_{\pm1.26}$ | $31.40_{\pm1.26}$ | $\underline{\mathbf{25.40}}_{\pm2.88}$ | $31.40_{\pm1.26}$ |
| madl | $29.02_{\pm7.43}$ | $32.54_{\pm1.37}$ | $34.78_{\pm0.98}$ | $30.30_{\pm1.62}$ | $30.30_{\pm1.62}$ | $30.30_{\pm1.62}$ | $\underline{29.02}_{\pm7.43}$ | $30.30_{\pm1.62}$ | $30.30_{\pm1.62}$ | $\underline{29.02}_{\pm7.43}$ | $30.30_{\pm1.62}$ |
| crowd-ar | $32.02_{\pm0.24}$ | $\underline{31.03}_{\pm1.63}$ | $34.88_{\pm1.13}$ | $32.37_{\pm0.97}$ | $32.37_{\pm0.97}$ | $32.37_{\pm0.97}$ | $32.37_{\pm0.97}$ | $32.37_{\pm0.97}$ | $32.37_{\pm0.97}$ | $32.37_{\pm0.97}$ | $32.37_{\pm0.97}$ |
| annot-mix | $30.10_{\pm1.22}$ | $30.88_{\pm2.83}$ | $33.18_{\pm0.73}$ | $\underline{30.86}_{\pm0.94}$ | $\underline{30.86}_{\pm0.94}$ | $\underline{30.86}_{\pm0.94}$ | $31.50_{\pm1.38}$ | $\underline{30.86}_{\pm0.94}$ | $\underline{30.86}_{\pm0.94}$ | $\underline{30.86}_{\pm0.94}$ | $30.86_{\pm0.94}$ |
| coin | $31.73_{\pm5.01}$ | $30.15_{\pm2.11}$ | $\mathbf{30.98}_{\pm0.56}$ | $30.25_{\pm0.71}$ | $30.25_{\pm0.71}$ | $30.25_{\pm0.71}$ | $\underline{26.70}_{\pm2.04}$ | $30.25_{\pm0.71}$ | $30.25_{\pm0.71}$ | $\underline{26.70}_{\pm2.04}$ | $30.25_{\pm0.71}$ |

Continued on the next page.

Table 14: **Zero-one loss results [%] (part III).** Continued from the previous page.

| Approach | Baseline | | | Aggregation-level | | | | Crowd-level | | | Ensemble |
|---|---|---|---|---|---|---|---|---|---|---|---|
| | TRUE$^\star$ | DEF-DATA$^\star$ | DEF | AEU | AEC | ALU | ALC | CXU | CEC | CLC | ENS |
| **label-me-worst-2** | | | | | | | | | | | |
| gt | $6.43_{\pm0.44}$ | $6.43_{\pm0.44}$ | $6.31_{\pm0.27}$ | $6.75_{\pm0.14}$ | $6.75_{\pm0.14}$ | N/A | N/A | N/A | N/A | N/A | $6.75_{\pm0.14}$ |
| mv | $18.08_{\pm0.77}$ | $24.97_{\pm0.47}$ | $22.20_{\pm0.95}$ | $17.41_{\pm0.70}$ | $17.41_{\pm0.70}$ | N/A | N/A | N/A | N/A | N/A | $17.41_{\pm0.70}$ |
| ds | $18.00_{\pm0.51}$ | $25.57_{\pm0.50}$ | $22.44_{\pm1.01}$ | $18.13_{\pm1.64}$ | $17.59_{\pm0.66}$ | $17.66_{\pm0.67}$ | $17.66_{\pm0.67}$ | $17.66_{\pm0.67}$ | $17.66_{\pm0.67}$ | $17.66_{\pm0.67}$ | $17.66_{\pm0.67}$ |
| cl | $16.58_{\pm0.89}$ | $17.29_{\pm1.49}$ | $20.82_{\pm0.26}$ | $17.71_{\pm1.17}$ | $17.71_{\pm1.17}$ | $17.71_{\pm1.17}$ | $17.71_{\pm1.17}$ | $17.71_{\pm1.17}$ | $17.71_{\pm1.17}$ | $17.71_{\pm1.17}$ | $17.71_{\pm1.17}$ |
| trace | $16.03_{\pm1.11}$ | $23.42_{\pm0.65}$ | $22.76_{\pm0.45}$ | $17.73_{\pm1.51}$ | $17.73_{\pm1.51}$ | $17.73_{\pm1.51}$ | $17.73_{\pm1.51}$ | $17.73_{\pm1.51}$ | $17.73_{\pm1.51}$ | $17.73_{\pm1.51}$ | $17.73_{\pm1.51}$ |
| conal | $19.14_{\pm0.96}$ | $22.24_{\pm0.70}$ | $22.19_{\pm0.97}$ | $17.07_{\pm0.48}$ | $17.07_{\pm0.48}$ | $19.02_{\pm0.43}$ | $18.86_{\pm1.18}$ | $17.07_{\pm0.48}$ | $17.07_{\pm0.48}$ | $18.13_{\pm1.42}$ | $17.07_{\pm0.48}$ |
| union-a | $\mathbf{14.02}_{\pm1.30}$ | $16.79_{\pm0.60}$ | $20.79_{\pm0.45}$ | $21.09_{\pm0.58}$ | $18.82_{\pm1.45}$ | $\mathbf{15.35}_{\pm1.79}$ | $16.80_{\pm0.72}$ | $17.31_{\pm0.73}$ | $18.03_{\pm0.55}$ | $21.99_{\pm0.95}$ | $21.09_{\pm0.58}$ |
| union-b | $16.55_{\pm1.93}$ | $17.91_{\pm0.43}$ | $21.57_{\pm0.49}$ | $19.21_{\pm0.71}$ | $16.38_{\pm0.57}$ | $16.38_{\pm0.57}$ | $16.38_{\pm0.57}$ | $17.07_{\pm0.88}$ | $17.07_{\pm0.88}$ | $17.07_{\pm0.88}$ | $16.38_{\pm0.57}$ |
| geo-w | $15.74_{\pm1.03}$ | $18.06_{\pm0.54}$ | $21.52_{\pm0.42}$ | $17.32_{\pm0.60}$ | $17.12_{\pm1.05}$ | $15.74_{\pm1.03}$ | $\mathbf{15.74}_{\pm1.03}$ | $17.12_{\pm1.05}$ | $17.12_{\pm1.05}$ | $18.70_{\pm0.81}$ | $17.12_{\pm1.05}$ |
| geo-f | $17.52_{\pm0.86}$ | $18.16_{\pm0.62}$ | $21.46_{\pm0.38}$ | $17.20_{\pm0.50}$ | $16.36_{\pm0.65}$ | $16.21_{\pm0.69}$ | $16.36_{\pm0.65}$ | $\mathbf{15.67}_{\pm0.66}$ | $\mathbf{15.67}_{\pm0.66}$ | $17.98_{\pm0.78}$ | $16.36_{\pm0.65}$ |
| madl | $15.72_{\pm0.94}$ | $20.00_{\pm0.67}$ | $23.21_{\pm0.61}$ | $19.41_{\pm0.74}$ | $19.41_{\pm0.74}$ | $18.00_{\pm0.27}$ | $18.00_{\pm0.27}$ | $19.41_{\pm0.74}$ | $19.41_{\pm0.74}$ | $19.41_{\pm0.74}$ | $19.41_{\pm0.74}$ |
| crowd-ar | $18.06_{\pm1.18}$ | $20.72_{\pm0.68}$ | $21.82_{\pm0.69}$ | $20.03_{\pm0.37}$ | $20.03_{\pm0.37}$ | $18.65_{\pm2.48}$ | $20.03_{\pm0.37}$ | $20.03_{\pm0.37}$ | $20.03_{\pm0.37}$ | $20.03_{\pm0.37}$ | $20.03_{\pm0.37}$ |
| annot-mix | $18.75_{\pm1.47}$ | $20.88_{\pm0.58}$ | $21.72_{\pm1.48}$ | $\mathbf{16.95}_{\pm0.56}$ | $\mathbf{16.95}_{\pm0.56}$ | $20.37_{\pm0.76}$ | $20.37_{\pm0.76}$ | $\mathbf{16.95}_{\pm0.56}$ | $\mathbf{16.95}_{\pm0.56}$ | $\mathbf{16.95}_{\pm0.56}$ | $18.75_{\pm1.47}$ |
| coin | $16.03_{\pm0.61}$ | $\mathbf{16.75}_{\pm0.56}$ | $\mathbf{19.93}_{\pm0.21}$ | $17.59_{\pm1.26}$ | $17.59_{\pm1.26}$ | $18.20_{\pm1.39}$ | $18.20_{\pm1.39}$ | $18.20_{\pm1.39}$ | $16.60_{\pm1.58}$ | $18.20_{\pm1.39}$ | $18.20_{\pm1.39}$ |
| **label-me-worst-v** | | | | | | | | | | | |
| gt | $5.99_{\pm0.33}$ | $5.99_{\pm0.33}$ | $6.31_{\pm0.27}$ | $7.47_{\pm0.25}$ | $7.47_{\pm0.25}$ | N/A | N/A | N/A | N/A | N/A | $7.47_{\pm0.25}$ |
| mv | $19.41_{\pm0.87}$ | $24.44_{\pm0.36}$ | $24.21_{\pm0.43}$ | $19.41_{\pm0.87}$ | $19.41_{\pm0.87}$ | N/A | N/A | N/A | N/A | N/A | $19.41_{\pm0.87}$ |
| ds | $21.04_{\pm0.91}$ | $24.90_{\pm0.57}$ | $24.82_{\pm0.46}$ | $19.36_{\pm0.92}$ | $21.04_{\pm0.91}$ | $19.36_{\pm0.92}$ | $21.04_{\pm0.91}$ | $21.04_{\pm0.91}$ | $21.04_{\pm0.91}$ | $21.04_{\pm0.91}$ | $21.04_{\pm0.91}$ |
| cl | $17.93_{\pm0.85}$ | $22.26_{\pm0.45}$ | $22.37_{\pm0.47}$ | $22.51_{\pm0.63}$ | $22.51_{\pm0.63}$ | $22.07_{\pm0.48}$ | $19.85_{\pm1.73}$ | $18.92_{\pm0.70}$ | $18.92_{\pm0.70}$ | $19.85_{\pm1.73}$ | $22.51_{\pm0.63}$ |
| trace | $19.66_{\pm1.15}$ | $23.15_{\pm0.72}$ | $23.11_{\pm0.86}$ | $20.57_{\pm0.89}$ | $20.57_{\pm0.89}$ | $19.66_{\pm1.10}$ | $19.66_{\pm1.10}$ | $19.66_{\pm1.15}$ | $19.66_{\pm1.10}$ | $19.66_{\pm1.15}$ | $19.66_{\pm1.10}$ |
| conal | $19.19_{\pm0.42}$ | $23.18_{\pm0.41}$ | $23.62_{\pm1.13}$ | $18.18_{\pm0.49}$ | $19.19_{\pm0.42}$ | $18.18_{\pm0.49}$ | $18.08_{\pm1.00}$ | $18.18_{\pm0.49}$ | $19.19_{\pm0.42}$ | $18.18_{\pm0.49}$ | $18.18_{\pm0.49}$ |
| union-a | $17.29_{\pm1.20}$ | $21.23_{\pm0.55}$ | $21.70_{\pm0.47}$ | $18.96_{\pm1.62}$ | $18.96_{\pm1.62}$ | $\mathbf{16.33}_{\pm0.91}$ | $18.23_{\pm0.95}$ | $18.23_{\pm0.95}$ | $18.23_{\pm0.95}$ | $18.23_{\pm0.95}$ | $18.23_{\pm0.95}$ |
| union-b | $\mathbf{15.46}_{\pm1.30}$ | $22.63_{\pm0.47}$ | $22.83_{\pm0.42}$ | $18.92_{\pm0.71}$ | $18.92_{\pm0.71}$ | $19.83_{\pm0.40}$ | $18.59_{\pm0.79}$ | $19.58_{\pm0.39}$ | $19.58_{\pm0.39}$ | $18.92_{\pm0.71}$ | $18.92_{\pm0.71}$ |
| geo-w | $19.55_{\pm0.56}$ | $22.63_{\pm0.57}$ | $22.90_{\pm0.52}$ | $18.84_{\pm0.80}$ | $18.84_{\pm0.80}$ | $21.14_{\pm0.46}$ | $17.56_{\pm1.36}$ | $19.76_{\pm0.46}$ | $19.70_{\pm0.61}$ | $19.70_{\pm0.61}$ | $21.14_{\pm0.46}$ |
| geo-f | $16.60_{\pm0.62}$ | $22.49_{\pm0.68}$ | $22.56_{\pm0.56}$ | $18.01_{\pm0.17}$ | $18.01_{\pm0.17}$ | $19.68_{\pm0.63}$ | $19.68_{\pm0.63}$ | $19.68_{\pm0.63}$ | $19.68_{\pm0.63}$ | $19.68_{\pm0.63}$ | $\mathbf{18.01}_{\pm0.17}$ |
| madl | $19.14_{\pm0.63}$ | $22.76_{\pm0.63}$ | $23.60_{\pm0.70}$ | $19.38_{\pm0.70}$ | $19.38_{\pm0.70}$ | $18.37_{\pm0.66}$ | $18.37_{\pm0.69}$ | $18.37_{\pm0.66}$ | $19.14_{\pm0.63}$ | $18.37_{\pm0.69}$ | $19.38_{\pm0.70}$ |
| crowd-ar | $18.70_{\pm0.31}$ | $23.37_{\pm0.54}$ | $23.25_{\pm0.51}$ | $20.05_{\pm0.64}$ | $20.05_{\pm0.64}$ | $23.25_{\pm0.51}$ | $19.90_{\pm1.00}$ | $18.97_{\pm0.71}$ | $20.05_{\pm0.64}$ | $18.97_{\pm0.71}$ | $20.05_{\pm0.64}$ |
| annot-mix | $18.28_{\pm0.97}$ | $22.41_{\pm0.90}$ | $22.56_{\pm0.60}$ | $19.95_{\pm1.01}$ | $19.95_{\pm1.01}$ | $19.92_{\pm0.67}$ | $20.56_{\pm0.41}$ | $20.56_{\pm0.41}$ | $20.56_{\pm0.41}$ | $20.56_{\pm0.41}$ | $22.05_{\pm0.51}$ |
| coin | $16.90_{\pm3.19}$ | $21.21_{\pm0.44}$ | $20.89_{\pm0.49}$ | $18.13_{\pm1.05}$ | $18.13_{\pm1.05}$ | $18.30_{\pm1.46}$ | $\mathbf{16.90}_{\pm3.19}$ | $18.13_{\pm1.05}$ | $18.13_{\pm1.05}$ | $18.13_{\pm1.05}$ | $18.13_{\pm1.05}$ |
| **label-me-rand-1** | | | | | | | | | | | |
| gt | $6.28_{\pm0.26}$ | $6.28_{\pm0.26}$ | $6.31_{\pm0.27}$ | $7.36_{\pm0.39}$ | $7.36_{\pm0.39}$ | N/A | N/A | N/A | N/A | N/A | $7.36_{\pm0.39}$ |
| mv | $14.76_{\pm0.60}$ | $19.50_{\pm0.57}$ | $18.45_{\pm0.51}$ | $14.49_{\pm0.69}$ | $14.49_{\pm0.69}$ | N/A | N/A | N/A | N/A | N/A | $14.49_{\pm0.69}$ |
| ds | $14.76_{\pm0.60}$ | $19.50_{\pm0.57}$ | $18.45_{\pm0.51}$ | $14.49_{\pm0.69}$ | $14.49_{\pm0.69}$ | $14.49_{\pm0.69}$ | $14.49_{\pm0.69}$ | $14.49_{\pm0.69}$ | $14.49_{\pm0.69}$ | $14.49_{\pm0.69}$ | $14.49_{\pm0.69}$ |
| cl | $15.56_{\pm0.44}$ | $17.26_{\pm1.06}$ | $18.97_{\pm0.47}$ | $15.00_{\pm0.63}$ | $15.00_{\pm0.63}$ | $15.00_{\pm0.63}$ | $15.00_{\pm0.63}$ | $15.00_{\pm0.63}$ | $15.00_{\pm0.63}$ | $15.00_{\pm0.63}$ | $15.00_{\pm0.63}$ |
| trace | $14.34_{\pm1.00}$ | $19.88_{\pm0.66}$ | $18.38_{\pm0.47}$ | $17.52_{\pm0.74}$ | $17.52_{\pm0.74}$ | $17.52_{\pm0.74}$ | $17.52_{\pm0.74}$ | $17.52_{\pm0.74}$ | $17.52_{\pm0.74}$ | $17.52_{\pm0.74}$ | $17.52_{\pm0.74}$ |
| conal | $15.05_{\pm0.68}$ | $18.00_{\pm1.80}$ | $19.24_{\pm0.57}$ | $14.48_{\pm0.38}$ | $14.48_{\pm0.38}$ | $14.48_{\pm0.38}$ | $14.48_{\pm0.38}$ | $14.48_{\pm0.38}$ | $14.48_{\pm0.38}$ | $14.48_{\pm0.38}$ | $14.48_{\pm0.38}$ |
| union-a | $15.07_{\pm0.59}$ | $\mathbf{16.84}_{\pm0.73}$ | $18.65_{\pm0.48}$ | $15.07_{\pm0.59}$ | $15.07_{\pm0.59}$ | $15.07_{\pm0.59}$ | $16.06_{\pm0.70}$ | $15.07_{\pm0.59}$ | $15.07_{\pm0.59}$ | $14.75_{\pm0.60}$ | $15.07_{\pm0.59}$ |
| union-b | $15.37_{\pm0.53}$ | $17.88_{\pm0.42}$ | $19.07_{\pm0.63}$ | $15.39_{\pm0.68}$ | $15.39_{\pm0.68}$ | $15.39_{\pm0.68}$ | $15.39_{\pm0.68}$ | $15.39_{\pm0.68}$ | $15.39_{\pm0.68}$ | $15.39_{\pm0.68}$ | $15.39_{\pm0.68}$ |
| geo-w | $15.74_{\pm1.30}$ | $17.66_{\pm0.91}$ | $19.11_{\pm0.63}$ | $15.40_{\pm0.68}$ | $15.40_{\pm0.68}$ | $15.40_{\pm0.68}$ | $15.39_{\pm0.69}$ | $15.40_{\pm0.68}$ | $15.40_{\pm0.68}$ | $15.76_{\pm0.86}$ | $15.40_{\pm0.68}$ |
| geo-f | $14.46_{\pm0.65}$ | $17.86_{\pm0.39}$ | $19.04_{\pm0.56}$ | $15.30_{\pm0.79}$ | $15.30_{\pm0.79}$ | $15.30_{\pm0.79}$ | $15.54_{\pm1.13}$ | $15.30_{\pm0.79}$ | $15.30_{\pm0.79}$ | $15.54_{\pm1.13}$ | $15.30_{\pm0.79}$ |
| madl | $14.83_{\pm0.53}$ | $20.94_{\pm0.81}$ | $19.01_{\pm0.69}$ | $\mathbf{13.11}_{\pm0.62}$ | $\mathbf{13.11}_{\pm0.62}$ | $\mathbf{13.11}_{\pm0.62}$ | $\mathbf{13.11}_{\pm0.62}$ | $\mathbf{13.11}_{\pm0.62}$ | $\mathbf{13.11}_{\pm0.62}$ | $\mathbf{13.11}_{\pm0.62}$ | $\mathbf{13.11}_{\pm0.62}$ |
| crowd-ar | $15.77_{\pm0.51}$ | $18.10_{\pm0.92}$ | $19.41_{\pm0.22}$ | $16.41_{\pm0.72}$ | $16.41_{\pm0.72}$ | $16.41_{\pm0.72}$ | $16.41_{\pm0.72}$ | $16.41_{\pm0.72}$ | $16.41_{\pm0.72}$ | $16.41_{\pm0.72}$ | $16.41_{\pm0.72}$ |
| annot-mix | $14.46_{\pm0.89}$ | $17.91_{\pm0.29}$ | $18.27_{\pm0.34}$ | $15.76_{\pm1.25}$ | $15.76_{\pm1.25}$ | $15.76_{\pm1.25}$ | $15.76_{\pm1.25}$ | $15.76_{\pm1.25}$ | $15.76_{\pm1.25}$ | $15.76_{\pm1.25}$ | $15.76_{\pm1.25}$ |
| coin | $\mathbf{12.58}_{\pm0.70}$ | $17.46_{\pm1.00}$ | $18.08_{\pm0.28}$ | $13.75_{\pm0.51}$ | $13.75_{\pm0.51}$ | $13.75_{\pm0.51}$ | $\mathbf{12.58}_{\pm1.12}$ | $13.87_{\pm0.25}$ | $13.87_{\pm0.25}$ | $\mathbf{12.58}_{\pm1.12}$ | $13.87_{\pm0.25}$ |
| **label-me-rand-2** | | | | | | | | | | | |
| gt | $6.21_{\pm0.26}$ | $6.21_{\pm0.26}$ | $6.31_{\pm0.27}$ | $6.21_{\pm0.26}$ | $6.21_{\pm0.26}$ | N/A | N/A | N/A | N/A | N/A | $6.21_{\pm0.26}$ |
| mv | $15.42_{\pm0.52}$ | $16.60_{\pm0.68}$ | $19.02_{\pm0.24}$ | $16.16_{\pm0.41}$ | $16.35_{\pm0.41}$ | N/A | N/A | N/A | N/A | N/A | $16.35_{\pm0.41}$ |
| ds | $15.13_{\pm0.63}$ | $15.02_{\pm0.41}$ | $17.24_{\pm0.46}$ | $14.41_{\pm0.35}$ | $15.13_{\pm0.63}$ | $15.02_{\pm0.41}$ | $15.02_{\pm0.41}$ | $15.02_{\pm0.41}$ | $15.02_{\pm0.41}$ | $15.02_{\pm0.41}$ | $15.02_{\pm0.41}$ |
| cl | $13.11_{\pm1.23}$ | $14.60_{\pm0.35}$ | $15.82_{\pm0.41}$ | $15.25_{\pm0.48}$ | $15.13_{\pm0.49}$ | $15.27_{\pm0.50}$ | $14.11_{\pm0.59}$ | $15.27_{\pm0.50}$ | $15.62_{\pm0.79}$ | $13.77_{\pm0.47}$ | $15.27_{\pm0.50}$ |
| trace | $14.53_{\pm0.72}$ | $14.43_{\pm0.72}$ | $17.95_{\pm0.51}$ | $15.15_{\pm0.95}$ | $15.15_{\pm0.95}$ | $15.15_{\pm0.95}$ | $15.15_{\pm0.95}$ | $15.15_{\pm0.95}$ | $15.15_{\pm0.95}$ | $15.15_{\pm0.95}$ | $15.15_{\pm0.95}$ |
| conal | $15.62_{\pm0.71}$ | $14.29_{\pm0.62}$ | $17.12_{\pm1.05}$ | $15.67_{\pm0.68}$ | $15.67_{\pm0.68}$ | $15.67_{\pm0.68}$ | $15.72_{\pm2.79}$ | $13.92_{\pm0.42}$ | $13.92_{\pm0.42}$ | $13.94_{\pm0.78}$ | $14.63_{\pm0.37}$ |
| union-a | $15.44_{\pm4.74}$ | $14.44_{\pm0.78}$ | $15.74_{\pm0.50}$ | $14.78_{\pm0.82}$ | $14.78_{\pm0.82}$ | $15.44_{\pm4.74}$ | $13.72_{\pm0.66}$ | $\mathbf{13.37}_{\pm0.29}$ | $14.16_{\pm0.53}$ | $13.72_{\pm0.66}$ | $14.16_{\pm0.53}$ |
| union-b | $\mathbf{12.95}_{\pm1.11}$ | $14.34_{\pm0.69}$ | $16.60_{\pm0.41}$ | $14.34_{\pm0.62}$ | $14.98_{\pm0.97}$ | $13.16_{\pm0.75}$ | $14.34_{\pm0.62}$ | $17.04_{\pm0.49}$ | $15.79_{\pm0.93}$ | $15.79_{\pm0.93}$ | $15.79_{\pm0.93}$ |
| geo-w | $13.10_{\pm0.49}$ | $14.33_{\pm0.72}$ | $16.57_{\pm0.35}$ | $15.76_{\pm1.04}$ | $15.76_{\pm1.04}$ | $13.10_{\pm0.49}$ | $15.76_{\pm1.04}$ | $14.78_{\pm0.46}$ | $14.78_{\pm0.46}$ | $14.78_{\pm0.46}$ | $15.76_{\pm1.04}$ |
| geo-f | $13.57_{\pm0.33}$ | $14.09_{\pm0.73}$ | $16.36_{\pm0.49}$ | $14.04_{\pm0.78}$ | $14.04_{\pm0.78}$ | $\mathbf{12.21}_{\pm0.32}$ | $14.36_{\pm0.60}$ | $15.61_{\pm0.50}$ | $14.04_{\pm0.78}$ | $13.94_{\pm0.60}$ | $14.36_{\pm0.60}$ |
| madl | $14.28_{\pm0.64}$ | $14.19_{\pm1.06}$ | $18.13_{\pm0.38}$ | $13.55_{\pm0.33}$ | $13.55_{\pm0.33}$ | $14.28_{\pm0.64}$ | $13.55_{\pm0.33}$ | $13.55_{\pm0.33}$ | $13.55_{\pm0.33}$ | $\mathbf{13.55}_{\pm0.33}$ | $13.55_{\pm0.33}$ |
| crowd-ar | $16.25_{\pm0.53}$ | $14.73_{\pm0.38}$ | $16.38_{\pm0.51}$ | $14.26_{\pm0.38}$ | $14.26_{\pm0.38}$ | $16.31_{\pm0.89}$ | $14.26_{\pm0.38}$ | $14.26_{\pm0.38}$ | $14.26_{\pm0.38}$ | $14.26_{\pm0.38}$ | $14.26_{\pm0.38}$ |
| annot-mix | $14.43_{\pm0.46}$ | $13.64_{\pm0.73}$ | $16.52_{\pm0.77}$ | $13.82_{\pm0.54}$ | $13.82_{\pm0.54}$ | $15.29_{\pm0.59}$ | $15.29_{\pm0.59}$ | $18.54_{\pm0.14}$ | $18.54_{\pm0.14}$ | $17.84_{\pm0.86}$ | $13.82_{\pm0.54}$ |
| coin | $13.64_{\pm0.15}$ | $\mathbf{13.01}_{\pm0.68}$ | $\mathbf{14.48}_{\pm0.59}$ | $13.05_{\pm0.57}$ | $13.05_{\pm0.57}$ | $13.82_{\pm0.89}$ | $13.05_{\pm0.57}$ | $15.71_{\pm0.41}$ | $13.05_{\pm0.57}$ | $14.09_{\pm0.56}$ | $\mathbf{13.05}_{\pm0.57}$ |

Continued on the next page.

Table 14: **Zero-one loss results [%] (part IV).** Continued from the previous page.

| Approach | Baseline | | | Aggregation-level | | | | Crowd-level | | | Ensemble |
|---|---|---|---|---|---|---|---|---|---|---|---|
| | TRUE[*] | DEF-DATA[*] | DEF | AEU | AEC | ALU | ALC | CXU | CEC | CLC | ENS |
| *label-me-rand-v* | | | | | | | | | | | |
| gt | $6.35_{\pm 0.39}$ | $6.35_{\pm 0.39}$ | $6.31_{\pm 0.27}$ | $6.35_{\pm 0.39}$ | $6.43_{\pm 0.27}$ | N/A | N/A | N/A | N/A | N/A | $6.43_{\pm 0.27}$ |
| mv | $15.19_{\pm 0.80}$ | $17.96_{\pm 0.86}$ | $19.68_{\pm 0.57}$ | $15.19_{\pm 0.80}$ | $15.19_{\pm 0.80}$ | N/A | N/A | N/A | N/A | N/A | $15.19_{\pm 0.80}$ |
| ds | $14.34_{\pm 0.65}$ | $17.42_{\pm 0.73}$ | $18.60_{\pm 0.61}$ | $15.54_{\pm 0.27}$ | $15.54_{\pm 0.27}$ | $15.54_{\pm 0.27}$ | $15.54_{\pm 0.27}$ | $13.91_{\pm 0.53}$ | $13.91_{\pm 0.53}$ | $17.58_{\pm 0.69}$ | $14.34_{\pm 0.65}$ |
| cl | $14.60_{\pm 0.79}$ | $15.94_{\pm 0.49}$ | $18.18_{\pm 0.71}$ | $19.98_{\pm 0.50}$ | $19.98_{\pm 0.50}$ | $14.88_{\pm 0.55}$ | $15.69_{\pm 0.95}$ | $15.74_{\pm 0.63}$ | $15.39_{\pm 0.95}$ | $15.69_{\pm 0.95}$ | $15.69_{\pm 0.95}$ |
| trace | $13.82_{\pm 0.91}$ | $16.75_{\pm 0.65}$ | $19.31_{\pm 0.79}$ | $16.21_{\pm 1.03}$ | $16.21_{\pm 1.03}$ | $16.67_{\pm 0.82}$ | $16.67_{\pm 0.82}$ | $13.82_{\pm 0.91}$ | $13.82_{\pm 0.91}$ | $13.82_{\pm 0.91}$ | $16.18_{\pm 0.51}$ |
| conal | $14.46_{\pm 0.60}$ | $16.84_{\pm 0.70}$ | $19.23_{\pm 0.95}$ | $16.01_{\pm 0.99}$ | $16.01_{\pm 0.99}$ | $17.71_{\pm 0.59}$ | $14.71_{\pm 0.46}$ | $14.46_{\pm 0.60}$ | $14.46_{\pm 0.60}$ | $14.71_{\pm 0.46}$ | $14.46_{\pm 0.60}$ |
| union-a | $16.50_{\pm 6.75}$ | $15.15_{\pm 0.88}$ | $17.93_{\pm 0.32}$ | $16.50_{\pm 6.75}$ | $16.50_{\pm 6.75}$ | $16.50_{\pm 6.75}$ | $15.69_{\pm 2.48}$ | $15.22_{\pm 0.57}$ | $15.22_{\pm 0.57}$ | $14.95_{\pm 1.35}$ | $18.87_{\pm 0.80}$ |
| union-b | $14.65_{\pm 0.86}$ | $16.38_{\pm 0.82}$ | $18.97_{\pm 0.49}$ | $20.39_{\pm 0.81}$ | $18.62_{\pm 0.58}$ | $15.98_{\pm 0.55}$ | $14.85_{\pm 0.34}$ | $18.62_{\pm 0.58}$ | $18.62_{\pm 0.58}$ | $14.85_{\pm 0.34}$ | $18.62_{\pm 0.58}$ |
| geo-w | $15.76_{\pm 2.80}$ | $16.36_{\pm 0.59}$ | $18.84_{\pm 0.61}$ | $15.20_{\pm 0.57}$ | $15.20_{\pm 0.57}$ | $15.20_{\pm 0.57}$ | $15.20_{\pm 0.57}$ | $20.24_{\pm 0.42}$ | $18.57_{\pm 0.78}$ | $15.20_{\pm 0.57}$ | $15.20_{\pm 0.57}$ |
| geo-f | $14.76_{\pm 0.51}$ | $16.03_{\pm 0.34}$ | $18.77_{\pm 0.64}$ | $\mathbf{13.28}_{\pm 0.45}$ | $\mathbf{13.28}_{\pm 0.45}$ | $17.44_{\pm 0.16}$ | $\mathbf{13.28}_{\pm 0.45}$ | $\mathbf{13.28}_{\pm 0.45}$ | $\mathbf{13.28}_{\pm 0.45}$ | $\mathbf{13.28}_{\pm 0.45}$ | $\mathbf{13.28}_{\pm 0.45}$ |
| madl | $14.75_{\pm 0.98}$ | $17.00_{\pm 0.23}$ | $19.41_{\pm 0.60}$ | $14.75_{\pm 0.98}$ | $14.75_{\pm 0.98}$ | $18.64_{\pm 0.79}$ | $18.64_{\pm 0.79}$ | $14.75_{\pm 0.98}$ | $14.75_{\pm 0.98}$ | $17.49_{\pm 0.43}$ | $14.75_{\pm 0.98}$ |
| crowd-ar | $15.08_{\pm 1.34}$ | $16.33_{\pm 0.65}$ | $19.33_{\pm 0.49}$ | $20.17_{\pm 0.65}$ | $20.17_{\pm 0.65}$ | $15.08_{\pm 1.34}$ | $15.08_{\pm 1.34}$ | $14.65_{\pm 0.70}$ | $14.65_{\pm 0.70}$ | $14.65_{\pm 0.70}$ | $15.08_{\pm 1.34}$ |
| annot-mix | $14.58_{\pm 0.42}$ | $16.08_{\pm 1.10}$ | $18.43_{\pm 0.61}$ | $14.58_{\pm 0.42}$ | $14.58_{\pm 0.42}$ | $14.58_{\pm 0.42}$ | $14.58_{\pm 0.42}$ | $19.55_{\pm 0.85}$ | $19.55_{\pm 0.85}$ | $19.55_{\pm 0.85}$ | $15.52_{\pm 0.68}$ |
| coin | $\mathbf{12.00}_{\pm 0.77}$ | $\mathbf{13.96}_{\pm 0.57}$ | $\mathbf{16.77}_{\pm 0.51}$ | $14.54_{\pm 0.43}$ | $14.54_{\pm 0.43}$ | $\mathbf{14.54}_{\pm 0.43}$ | $\mathbf{14.54}_{\pm 0.43}$ | $14.54_{\pm 0.43}$ | $14.54_{\pm 0.43}$ | $14.54_{\pm 0.43}$ | $14.54_{\pm 0.43}$ |
| *label-me-full* | | | | | | | | | | | |
| gt | $6.01_{\pm 0.25}$ | $6.01_{\pm 0.25}$ | $6.31_{\pm 0.27}$ | $6.60_{\pm 0.38}$ | $6.60_{\pm 0.38}$ | N/A | N/A | N/A | N/A | N/A | $6.60_{\pm 0.38}$ |
| mv | $14.76_{\pm 0.50}$ | $16.89_{\pm 0.71}$ | $18.42_{\pm 0.47}$ | $14.53_{\pm 0.49}$ | $14.38_{\pm 0.91}$ | N/A | N/A | N/A | N/A | N/A | $14.38_{\pm 0.91}$ |
| ds | $13.13_{\pm 0.75}$ | $15.07_{\pm 0.68}$ | $15.96_{\pm 0.26}$ | $14.81_{\pm 0.89}$ | $14.81_{\pm 0.89}$ | $13.13_{\pm 0.75}$ | $14.81_{\pm 0.89}$ | $\mathbf{12.95}_{\pm 0.83}$ | $12.95_{\pm 0.83}$ | $\mathbf{12.95}_{\pm 0.83}$ | $13.13_{\pm 0.75}$ |
| cl | $12.91_{\pm 0.74}$ | $13.77_{\pm 0.78}$ | $15.02_{\pm 0.48}$ | $14.38_{\pm 0.97}$ | $14.38_{\pm 0.97}$ | $14.11_{\pm 1.24}$ | $13.86_{\pm 0.80}$ | $14.73_{\pm 0.31}$ | $14.73_{\pm 0.31}$ | $13.82_{\pm 0.68}$ | $13.82_{\pm 0.68}$ |
| trace | $14.16_{\pm 0.54}$ | $15.17_{\pm 0.65}$ | $16.53_{\pm 0.17}$ | $14.34_{\pm 1.05}$ | $14.34_{\pm 1.05}$ | $14.34_{\pm 1.05}$ | $\mathbf{12.63}_{\pm 0.74}$ | $14.34_{\pm 1.05}$ | $14.34_{\pm 1.05}$ | $14.34_{\pm 1.05}$ | $14.34_{\pm 1.05}$ |
| conal | $13.54_{\pm 0.69}$ | $14.80_{\pm 1.04}$ | $16.80_{\pm 0.57}$ | $13.54_{\pm 0.69}$ | $13.54_{\pm 0.69}$ | $13.54_{\pm 0.69}$ | $16.95_{\pm 2.17}$ | $13.54_{\pm 0.69}$ | $13.54_{\pm 0.69}$ | $16.95_{\pm 2.17}$ | $13.54_{\pm 0.69}$ |
| union-a | $12.73_{\pm 0.49}$ | $13.72_{\pm 0.70}$ | $15.12_{\pm 0.24}$ | $14.16_{\pm 0.84}$ | $14.16_{\pm 0.84}$ | $15.64_{\pm 0.60}$ | $14.06_{\pm 0.55}$ | $14.29_{\pm 0.63}$ | $13.52_{\pm 1.01}$ | $17.69_{\pm 2.35}$ | $14.06_{\pm 0.55}$ |
| union-b | $12.98_{\pm 0.80}$ | $13.97_{\pm 0.83}$ | $16.03_{\pm 0.26}$ | $13.86_{\pm 0.61}$ | $13.86_{\pm 0.61}$ | $13.37_{\pm 0.59}$ | $13.57_{\pm 0.62}$ | $14.58_{\pm 0.16}$ | $14.58_{\pm 0.16}$ | $13.57_{\pm 0.62}$ | $12.95_{\pm 0.57}$ |
| geo-w | $15.19_{\pm 2.91}$ | $13.92_{\pm 0.64}$ | $16.03_{\pm 0.16}$ | $14.51_{\pm 0.83}$ | $14.51_{\pm 0.83}$ | $13.69_{\pm 0.42}$ | $15.69_{\pm 0.63}$ | $14.81_{\pm 0.50}$ | $13.42_{\pm 0.49}$ | $15.69_{\pm 0.63}$ | $14.81_{\pm 0.50}$ |
| geo-f | $26.58_{\pm 35.3}$ | $13.74_{\pm 0.73}$ | $15.82_{\pm 0.21}$ | $\mathbf{12.86}_{\pm 0.48}$ | $\mathbf{12.86}_{\pm 0.48}$ | $26.58_{\pm 35.3}$ | $26.58_{\pm 35.3}$ | $14.76_{\pm 0.61}$ | $12.66_{\pm 0.78}$ | $13.33_{\pm 1.00}$ | $12.66_{\pm 0.78}$ |
| madl | $14.21_{\pm 0.33}$ | $15.07_{\pm 0.58}$ | $16.72_{\pm 0.43}$ | $12.95_{\pm 0.57}$ | $12.95_{\pm 0.57}$ | $\mathbf{12.98}_{\pm 1.53}$ | $12.98_{\pm 1.53}$ | $15.22_{\pm 1.80}$ | $12.95_{\pm 0.57}$ | $15.22_{\pm 1.80}$ | $12.95_{\pm 0.57}$ |
| crowd-ar | $14.66_{\pm 0.58}$ | $14.78_{\pm 0.96}$ | $15.79_{\pm 0.55}$ | $13.77_{\pm 0.51}$ | $13.77_{\pm 0.51}$ | $14.53_{\pm 0.78}$ | $13.77_{\pm 0.51}$ | $13.65_{\pm 0.41}$ | $13.65_{\pm 0.41}$ | $13.77_{\pm 0.51}$ | $13.77_{\pm 0.51}$ |
| annot-mix | $13.64_{\pm 0.38}$ | $14.95_{\pm 0.45}$ | $16.03_{\pm 0.27}$ | $14.90_{\pm 0.73}$ | $14.90_{\pm 0.73}$ | $16.43_{\pm 0.34}$ | $13.64_{\pm 0.38}$ | $16.35_{\pm 0.67}$ | $16.35_{\pm 0.67}$ | $16.35_{\pm 0.67}$ | $14.90_{\pm 0.73}$ |
| coin | $\mathbf{11.06}_{\pm 0.96}$ | $12.95_{\pm 0.51}$ | $13.57_{\pm 0.57}$ | $13.28_{\pm 0.90}$ | $13.28_{\pm 0.90}$ | $13.43_{\pm 0.65}$ | $13.43_{\pm 0.65}$ | $15.62_{\pm 0.51}$ | $\mathbf{12.07}_{\pm 0.54}$ | $15.62_{\pm 0.38}$ | $\mathbf{12.07}_{\pm 0.54}$ |
| *dopanim-worst-1* | | | | | | | | | | | |
| gt | $10.59_{\pm 0.14}$ | $10.59_{\pm 0.14}$ | $10.52_{\pm 0.22}$ | $28.15_{\pm 1.23}$ | $28.15_{\pm 1.23}$ | N/A | N/A | N/A | N/A | N/A | $28.15_{\pm 1.23}$ |
| mv | $66.30_{\pm 1.14}$ | $72.55_{\pm 0.50}$ | $73.28_{\pm 0.58}$ | $68.61_{\pm 0.79}$ | $68.61_{\pm 0.79}$ | N/A | N/A | N/A | N/A | N/A | $68.61_{\pm 0.79}$ |
| ds | $66.30_{\pm 1.14}$ | $72.55_{\pm 0.50}$ | $73.28_{\pm 0.58}$ | $68.61_{\pm 0.79}$ | $68.61_{\pm 0.79}$ | $68.61_{\pm 0.79}$ | $68.61_{\pm 0.79}$ | $68.61_{\pm 0.79}$ | $68.61_{\pm 0.79}$ | $68.61_{\pm 0.79}$ | $68.61_{\pm 0.79}$ |
| cl | $62.67_{\pm 3.58}$ | $69.63_{\pm 2.59}$ | $68.41_{\pm 2.29}$ | $\mathbf{67.77}_{\pm 1.80}$ | $\mathbf{67.77}_{\pm 1.80}$ | $67.77_{\pm 1.80}$ | $62.85_{\pm 2.57}$ | $62.85_{\pm 2.57}$ | $62.85_{\pm 2.57}$ | $62.85_{\pm 2.57}$ | $\mathbf{62.85}_{\pm 2.57}$ |
| trace | $\mathbf{52.79}_{\pm 3.43}$ | $71.63_{\pm 0.25}$ | $73.16_{\pm 0.42}$ | $70.62_{\pm 0.33}$ | $70.62_{\pm 0.33}$ | $70.62_{\pm 0.33}$ | $64.92_{\pm 1.53}$ | $64.92_{\pm 1.53}$ | $64.92_{\pm 1.53}$ | $\mathbf{52.79}_{\pm 3.43}$ | $64.60_{\pm 1.30}$ |
| conal | $68.01_{\pm 0.52}$ | $72.02_{\pm 0.34}$ | $72.55_{\pm 0.55}$ | $70.78_{\pm 0.66}$ | $70.78_{\pm 0.66}$ | $70.78_{\pm 0.66}$ | $70.81_{\pm 0.86}$ | $70.78_{\pm 0.66}$ | $70.78_{\pm 0.66}$ | $70.81_{\pm 0.86}$ | $70.78_{\pm 0.66}$ |
| union-a | $63.49_{\pm 1.07}$ | $67.42_{\pm 2.23}$ | $\mathbf{67.65}_{\pm 0.21}$ | $71.16_{\pm 0.35}$ | $71.16_{\pm 0.35}$ | $\mathbf{67.65}_{\pm 0.21}$ | $67.65_{\pm 0.21}$ | $67.65_{\pm 0.21}$ | $67.65_{\pm 0.21}$ | $67.65_{\pm 0.21}$ | $67.65_{\pm 0.21}$ |
| union-b | $63.01_{\pm 0.17}$ | $66.42_{\pm 0.29}$ | $69.01_{\pm 1.49}$ | $70.56_{\pm 0.60}$ | $70.56_{\pm 0.60}$ | $70.56_{\pm 0.60}$ | $63.01_{\pm 0.17}$ | $66.70_{\pm 0.25}$ | $66.70_{\pm 0.25}$ | $63.01_{\pm 0.17}$ | $66.70_{\pm 0.25}$ |
| geo-w | $65.30_{\pm 2.51}$ | $67.28_{\pm 1.22}$ | $71.55_{\pm 0.82}$ | $71.05_{\pm 0.61}$ | $71.05_{\pm 0.61}$ | $71.05_{\pm 0.61}$ | $66.15_{\pm 0.19}$ | $66.15_{\pm 0.19}$ | $66.15_{\pm 0.19}$ | $66.15_{\pm 0.19}$ | $66.15_{\pm 0.19}$ |
| geo-f | $58.00_{\pm 10.4}$ | $68.90_{\pm 0.20}$ | $70.99_{\pm 0.53}$ | $70.71_{\pm 0.46}$ | $70.71_{\pm 0.46}$ | $70.71_{\pm 0.46}$ | $65.81_{\pm 1.75}$ | $67.62_{\pm 3.87}$ | $67.62_{\pm 3.87}$ | $63.61_{\pm 4.08}$ | $65.81_{\pm 1.75}$ |
| madl | $57.60_{\pm 3.71}$ | $71.98_{\pm 1.18}$ | $73.53_{\pm 1.01}$ | $70.93_{\pm 0.68}$ | $70.93_{\pm 0.68}$ | $70.93_{\pm 0.68}$ | $63.87_{\pm 3.34}$ | $67.17_{\pm 3.40}$ | $67.17_{\pm 3.40}$ | $63.87_{\pm 3.34}$ | $68.88_{\pm 1.63}$ |
| crowd-ar | $70.15_{\pm 2.38}$ | $72.05_{\pm 0.94}$ | $72.01_{\pm 0.45}$ | $72.00_{\pm 0.56}$ | $72.00_{\pm 0.56}$ | $72.00_{\pm 0.56}$ | $70.44_{\pm 0.71}$ | $72.00_{\pm 0.56}$ | $72.00_{\pm 0.56}$ | $70.44_{\pm 0.71}$ | $72.00_{\pm 0.56}$ |
| annot-mix | $59.42_{\pm 4.15}$ | $\mathbf{62.63}_{\pm 1.30}$ | $67.82_{\pm 0.73}$ | $69.75_{\pm 0.94}$ | $69.75_{\pm 0.94}$ | $69.75_{\pm 0.94}$ | $\mathbf{59.42}_{\pm 4.15}$ | $60.35_{\pm 2.26}$ | $60.35_{\pm 2.26}$ | $59.42_{\pm 4.15}$ | $65.74_{\pm 0.87}$ |
| coin | $67.15_{\pm 2.02}$ | $67.04_{\pm 2.15}$ | $68.38_{\pm 1.13}$ | $69.37_{\pm 1.03}$ | $69.37_{\pm 1.03}$ | $69.37_{\pm 1.03}$ | $65.33_{\pm 2.83}$ | $68.06_{\pm 5.02}$ | $68.06_{\pm 5.02}$ | $68.06_{\pm 5.02}$ | $67.72_{\pm 1.11}$ |
| *dopanim-worst-2* | | | | | | | | | | | |
| gt | $11.09_{\pm 0.11}$ | $11.09_{\pm 0.11}$ | $10.52_{\pm 0.22}$ | $12.49_{\pm 0.47}$ | $12.49_{\pm 0.47}$ | N/A | N/A | N/A | N/A | N/A | $12.49_{\pm 0.47}$ |
| mv | $52.77_{\pm 0.38}$ | $54.39_{\pm 0.95}$ | $56.83_{\pm 0.40}$ | $52.42_{\pm 0.40}$ | $52.43_{\pm 0.38}$ | N/A | N/A | N/A | N/A | N/A | $52.42_{\pm 0.40}$ |
| ds | $45.64_{\pm 0.55}$ | $45.34_{\pm 0.33}$ | $48.75_{\pm 0.43}$ | $\mathbf{46.28}_{\pm 0.27}$ | $\mathbf{46.28}_{\pm 0.27}$ | $46.28_{\pm 0.27}$ | $46.28_{\pm 0.27}$ | $46.91_{\pm 0.44}$ | $46.91_{\pm 0.44}$ | $\mathbf{46.91}_{\pm 0.44}$ | $\mathbf{46.91}_{\pm 0.44}$ |
| cl | $50.95_{\pm 0.31}$ | $71.92_{\pm 2.73}$ | $55.63_{\pm 1.68}$ | $55.87_{\pm 2.94}$ | $55.87_{\pm 2.94}$ | $56.59_{\pm 1.07}$ | $52.91_{\pm 2.03}$ | $58.33_{\pm 4.03}$ | $58.33_{\pm 4.03}$ | $58.33_{\pm 4.03}$ | $53.32_{\pm 1.85}$ |
| trace | $48.98_{\pm 2.69}$ | $51.31_{\pm 3.79}$ | $54.61_{\pm 0.45}$ | $52.08_{\pm 0.67}$ | $52.08_{\pm 0.67}$ | $48.34_{\pm 0.45}$ | $\mathbf{42.17}_{\pm 0.60}$ | $\mathbf{42.17}_{\pm 0.60}$ | $\mathbf{39.42}_{\pm 5.16}$ | $67.87_{\pm 1.42}$ | $48.34_{\pm 0.45}$ |
| conal | $52.94_{\pm 1.01}$ | $70.58_{\pm 3.39}$ | $53.94_{\pm 0.78}$ | $52.99_{\pm 0.22}$ | $52.99_{\pm 0.22}$ | $52.99_{\pm 0.22}$ | $53.51_{\pm 1.43}$ | $52.99_{\pm 0.22}$ | $52.99_{\pm 0.22}$ | $53.51_{\pm 1.43}$ | $52.95_{\pm 0.19}$ |
| union-a | $52.35_{\pm 2.34}$ | $73.77_{\pm 4.48}$ | $51.89_{\pm 4.25}$ | $53.53_{\pm 1.89}$ | $53.53_{\pm 1.89}$ | $51.89_{\pm 4.25}$ | $51.89_{\pm 4.25}$ | $51.89_{\pm 4.25}$ | $51.89_{\pm 4.25}$ | $51.89_{\pm 4.25}$ | $51.89_{\pm 4.25}$ |
| union-b | $51.40_{\pm 3.11}$ | $72.28_{\pm 2.00}$ | $50.55_{\pm 0.30}$ | $54.16_{\pm 2.28}$ | $52.82_{\pm 0.81}$ | $51.40_{\pm 3.11}$ | $51.40_{\pm 3.11}$ | $52.95_{\pm 4.03}$ | $51.40_{\pm 3.11}$ | $51.40_{\pm 3.11}$ | $51.40_{\pm 3.11}$ |
| geo-w | $48.05_{\pm 0.41}$ | $70.58_{\pm 2.80}$ | $50.46_{\pm 0.72}$ | $53.02_{\pm 0.19}$ | $53.02_{\pm 0.19}$ | $52.18_{\pm 2.20}$ | $52.18_{\pm 2.20}$ | $48.45_{\pm 2.06}$ | $49.43_{\pm 1.33}$ | $49.43_{\pm 1.33}$ | $49.43_{\pm 1.33}$ |
| geo-f | $52.04_{\pm 2.09}$ | $61.45_{\pm 7.15}$ | $50.36_{\pm 0.18}$ | $51.99_{\pm 0.33}$ | $51.86_{\pm 0.21}$ | $49.77_{\pm 1.85}$ | $51.57_{\pm 1.87}$ | $51.57_{\pm 1.87}$ | $51.57_{\pm 1.87}$ | $51.57_{\pm 1.87}$ | $49.77_{\pm 1.85}$ |
| madl | $46.38_{\pm 2.18}$ | $72.72_{\pm 2.59}$ | $52.96_{\pm 3.88}$ | $49.65_{\pm 1.31}$ | $49.65_{\pm 1.31}$ | $49.65_{\pm 1.31}$ | $48.53_{\pm 2.09}$ | $49.65_{\pm 1.31}$ | $49.65_{\pm 1.31}$ | $48.53_{\pm 2.09}$ | $49.65_{\pm 1.31}$ |
| crowd-ar | $55.13_{\pm 1.44}$ | $64.58_{\pm 4.46}$ | $54.12_{\pm 0.42}$ | $54.27_{\pm 1.73}$ | $54.27_{\pm 1.73}$ | $54.27_{\pm 1.73}$ | $53.93_{\pm 0.71}$ | $54.27_{\pm 1.73}$ | $54.27_{\pm 1.73}$ | $53.93_{\pm 0.71}$ | $53.93_{\pm 0.71}$ |
| annot-mix | $\mathbf{44.16}_{\pm 3.15}$ | $50.74_{\pm 1.79}$ | $\mathbf{47.75}_{\pm 0.72}$ | $49.98_{\pm 1.29}$ | $49.98_{\pm 1.29}$ | $47.60_{\pm 1.14}$ | $43.95_{\pm 3.35}$ | $47.35_{\pm 0.91}$ | $47.60_{\pm 1.14}$ | $47.35_{\pm 0.91}$ | $47.35_{\pm 0.91}$ |
| coin | $45.57_{\pm 4.94}$ | $58.23_{\pm 5.19}$ | $50.17_{\pm 0.24}$ | $51.91_{\pm 0.24}$ | $50.09_{\pm 0.80}$ | $50.09_{\pm 0.80}$ | $51.35_{\pm 3.61}$ | $45.57_{\pm 4.94}$ | $50.09_{\pm 0.80}$ | $50.09_{\pm 0.80}$ | $50.09_{\pm 0.80}$ |

Continued on the next page.

Table 14: **Zero-one loss results [%] (part V).** Continued from the previous page.

| Approach | Baseline | | | Aggregation-level | | | | Crowd-level | | | Ensemble |
|---|---|---|---|---|---|---|---|---|---|---|---|
| | TRUE[*] | DEF-DATA[*] | DEF | AEU | AEC | ALU | ALC | CXU | CEC | CLC | ENS |
| **dopanim-worst-v** | | | | | | | | | | | |
| gt | $10.74_{\pm0.20}$ | $10.74_{\pm0.20}$ | $10.52_{\pm0.22}$ | $11.03_{\pm0.18}$ | $11.31_{\pm0.14}$ | N/A | N/A | N/A | N/A | N/A | $11.03_{\pm0.18}$ |
| mv | $34.12_{\pm0.43}$ | $36.47_{\pm0.49}$ | $41.50_{\pm0.69}$ | $34.12_{\pm0.43}$ | $34.09_{\pm0.73}$ | N/A | N/A | N/A | N/A | N/A | $34.09_{\pm0.73}$ |
| ds | $29.73_{\pm0.55}$ | $30.58_{\pm0.46}$ | $35.22_{\pm0.58}$ | $29.05_{\pm0.38}$ | $29.05_{\pm0.38}$ | $29.05_{\pm0.38}$ | $29.05_{\pm0.38}$ | $29.05_{\pm0.38}$ | $29.05_{\pm0.38}$ | $29.05_{\pm0.38}$ | $29.05_{\pm0.38}$ |
| cl | $35.03_{\pm3.68}$ | $43.26_{\pm5.96}$ | $38.46_{\pm1.71}$ | $35.03_{\pm3.68}$ | $35.03_{\pm3.68}$ | $37.22_{\pm1.40}$ | $35.03_{\pm3.68}$ | $44.13_{\pm2.40}$ | $44.13_{\pm2.40}$ | $35.03_{\pm3.68}$ | $35.03_{\pm3.68}$ |
| trace | $21.16_{\pm0.38}$ | $29.82_{\pm0.15}$ | $34.89_{\pm0.18}$ | $28.17_{\pm0.20}$ | $28.17_{\pm0.20}$ | $24.91_{\pm0.50}$ | $53.56_{\pm8.60}$ | $21.16_{\pm0.38}$ | $21.16_{\pm0.38}$ | $53.56_{\pm8.60}$ | $21.16_{\pm0.38}$ |
| conal | $30.75_{\pm1.47}$ | $32.52_{\pm0.42}$ | $33.93_{\pm0.27}$ | $32.40_{\pm0.32}$ | $32.40_{\pm0.32}$ | $32.40_{\pm0.32}$ | $32.08_{\pm1.92}$ | $32.34_{\pm0.36}$ | $30.86_{\pm0.40}$ | $32.08_{\pm1.92}$ | $33.17_{\pm0.22}$ |
| union-a | $33.65_{\pm2.42}$ | $43.92_{\pm5.72}$ | $37.73_{\pm0.68}$ | $36.84_{\pm4.34}$ | $36.84_{\pm4.34}$ | $37.73_{\pm0.68}$ | $37.73_{\pm0.68}$ | $37.73_{\pm0.68}$ | $37.73_{\pm0.68}$ | $37.73_{\pm0.68}$ | $37.73_{\pm0.68}$ |
| union-b | $31.69_{\pm0.78}$ | $31.85_{\pm1.45}$ | $33.35_{\pm1.08}$ | $31.69_{\pm0.78}$ | $31.69_{\pm0.78}$ | $32.62_{\pm3.13}$ | $31.85_{\pm0.55}$ | $36.76_{\pm1.72}$ | $32.62_{\pm3.13}$ | $31.85_{\pm0.55}$ | $31.69_{\pm0.78}$ |
| geo-w | $27.41_{\pm0.26}$ | $29.84_{\pm0.55}$ | $32.76_{\pm0.67}$ | $27.41_{\pm0.26}$ | $27.41_{\pm0.26}$ | $27.41_{\pm0.26}$ | $30.08_{\pm0.95}$ | $30.78_{\pm3.17}$ | $30.78_{\pm3.17}$ | $30.08_{\pm0.95}$ | $27.41_{\pm0.26}$ |
| geo-f | $21.91_{\pm0.38}$ | $26.32_{\pm0.73}$ | $28.99_{\pm0.44}$ | $25.44_{\pm0.40}$ | $29.45_{\pm0.52}$ | $23.16_{\pm1.32}$ | $25.84_{\pm0.77}$ | $24.40_{\pm2.51}$ | $23.16_{\pm1.32}$ | $25.84_{\pm0.77}$ | $23.16_{\pm1.32}$ |
| madl | $20.74_{\pm0.35}$ | $29.60_{\pm2.76}$ | $31.85_{\pm0.87}$ | $27.58_{\pm0.77}$ | $29.90_{\pm1.18}$ | $20.74_{\pm0.35}$ | $20.74_{\pm0.35}$ | $20.74_{\pm0.35}$ | $20.74_{\pm0.35}$ | $20.74_{\pm0.35}$ | $20.74_{\pm0.35}$ |
| crowd-ar | $31.59_{\pm0.57}$ | $31.89_{\pm0.41}$ | $34.07_{\pm0.97}$ | $32.66_{\pm2.17}$ | $32.66_{\pm2.17}$ | $32.66_{\pm2.17}$ | $31.96_{\pm1.40}$ | $30.98_{\pm0.58}$ | $30.98_{\pm0.58}$ | $31.96_{\pm1.40}$ | $31.55_{\pm1.37}$ |
| annot-mix | $21.61_{\pm0.51}$ | $24.09_{\pm0.62}$ | $26.29_{\pm0.47}$ | $28.09_{\pm1.26}$ | $28.09_{\pm1.26}$ | $22.32_{\pm0.38}$ | $22.32_{\pm0.38}$ | $22.32_{\pm0.38}$ | $23.45_{\pm0.24}$ | $23.82_{\pm0.58}$ | $23.45_{\pm0.24}$ |
| coin | $21.26_{\pm3.81}$ | $28.21_{\pm1.31}$ | $23.20_{\pm0.33}$ | $29.51_{\pm0.40}$ | $29.51_{\pm0.40}$ | $20.11_{\pm0.22}$ | $26.14_{\pm0.77}$ | $20.11_{\pm0.22}$ | $20.11_{\pm0.22}$ | $25.79_{\pm0.92}$ | $20.11_{\pm0.22}$ |
| **dopanim-rand-1** | | | | | | | | | | | |
| gt | $10.97_{\pm0.39}$ | $10.97_{\pm0.39}$ | $10.52_{\pm0.22}$ | $11.28_{\pm0.26}$ | $11.28_{\pm0.26}$ | N/A | N/A | N/A | N/A | N/A | $11.28_{\pm0.26}$ |
| mv | $20.79_{\pm0.62}$ | $21.50_{\pm0.60}$ | $27.66_{\pm0.31}$ | $20.56_{\pm0.30}$ | $20.56_{\pm0.30}$ | N/A | N/A | N/A | N/A | N/A | $20.56_{\pm0.30}$ |
| ds | $20.79_{\pm0.62}$ | $21.50_{\pm0.60}$ | $27.66_{\pm0.31}$ | $20.56_{\pm0.30}$ | $20.56_{\pm0.30}$ | $20.56_{\pm0.30}$ | $20.56_{\pm0.30}$ | $20.56_{\pm0.30}$ | $20.56_{\pm0.30}$ | $20.56_{\pm0.30}$ | $20.56_{\pm0.30}$ |
| cl | $23.04_{\pm2.10}$ | $60.00_{\pm5.17}$ | $26.92_{\pm2.65}$ | $23.04_{\pm2.10}$ | $23.04_{\pm2.10}$ | $23.04_{\pm2.10}$ | $29.24_{\pm2.73}$ | $23.04_{\pm2.10}$ | $23.04_{\pm2.10}$ | $29.24_{\pm2.73}$ | $26.49_{\pm4.36}$ |
| trace | $22.33_{\pm3.73}$ | $23.53_{\pm2.66}$ | $27.48_{\pm0.64}$ | $17.34_{\pm0.44}$ | $17.34_{\pm0.44}$ | $20.08_{\pm0.54}$ | $27.47_{\pm0.60}$ | $17.34_{\pm0.44}$ | $17.34_{\pm0.44}$ | $46.79_{\pm2.83}$ | $17.34_{\pm0.44}$ |
| conal | $19.61_{\pm1.00}$ | $48.42_{\pm3.66}$ | $23.14_{\pm0.60}$ | $19.61_{\pm1.00}$ | $19.61_{\pm1.00}$ | $19.61_{\pm1.00}$ | $19.61_{\pm1.00}$ | $19.61_{\pm1.00}$ | $19.61_{\pm1.00}$ | $19.61_{\pm1.00}$ | $19.61_{\pm1.00}$ |
| union-a | $20.36_{\pm2.11}$ | $68.81_{\pm3.45}$ | $25.32_{\pm3.71}$ | $20.36_{\pm2.11}$ | $20.36_{\pm2.11}$ | $25.32_{\pm3.71}$ | $25.32_{\pm3.71}$ | $25.32_{\pm3.71}$ | $25.32_{\pm3.71}$ | $25.32_{\pm3.71}$ | $22.43_{\pm1.77}$ |
| union-b | $20.04_{\pm0.34}$ | $45.95_{\pm5.03}$ | $22.01_{\pm0.44}$ | $20.18_{\pm1.55}$ | $20.18_{\pm1.55}$ | $20.18_{\pm1.55}$ | $21.74_{\pm2.25}$ | $20.32_{\pm2.01}$ | $20.32_{\pm2.01}$ | $21.74_{\pm2.25}$ | $20.32_{\pm2.01}$ |
| geo-w | $18.88_{\pm0.57}$ | $45.03_{\pm7.25}$ | $22.50_{\pm0.30}$ | $18.88_{\pm0.57}$ | $18.88_{\pm0.57}$ | $18.88_{\pm0.57}$ | $19.68_{\pm2.50}$ | $19.20_{\pm0.35}$ | $19.20_{\pm0.35}$ | $19.68_{\pm2.50}$ | $19.20_{\pm0.35}$ |
| geo-f | $16.59_{\pm0.56}$ | $23.57_{\pm3.20}$ | $21.95_{\pm0.37}$ | $16.59_{\pm0.56}$ | $16.59_{\pm0.56}$ | $16.59_{\pm0.56}$ | $16.59_{\pm0.56}$ | $16.45_{\pm0.21}$ | $16.45_{\pm0.21}$ | $16.59_{\pm0.56}$ | $16.59_{\pm0.56}$ |
| madl | $16.78_{\pm0.98}$ | $24.20_{\pm0.94}$ | $27.79_{\pm0.96}$ | $16.78_{\pm0.98}$ | $16.78_{\pm0.98}$ | $16.78_{\pm0.98}$ | $16.78_{\pm0.98}$ | $16.78_{\pm0.98}$ | $16.78_{\pm0.98}$ | $16.78_{\pm0.98}$ | $16.78_{\pm0.98}$ |
| crowd-ar | $19.95_{\pm0.43}$ | $61.13_{\pm6.61}$ | $22.18_{\pm0.24}$ | $19.95_{\pm0.43}$ | $19.95_{\pm0.43}$ | $19.95_{\pm0.43}$ | $18.89_{\pm0.52}$ | $19.95_{\pm0.43}$ | $19.95_{\pm0.43}$ | $18.89_{\pm0.52}$ | $18.89_{\pm0.52}$ |
| annot-mix | $17.79_{\pm0.32}$ | $20.52_{\pm4.22}$ | $21.40_{\pm0.37}$ | $17.79_{\pm0.32}$ | $17.79_{\pm0.32}$ | $17.79_{\pm0.32}$ | $18.39_{\pm0.22}$ | $17.79_{\pm0.32}$ | $17.79_{\pm0.32}$ | $18.81_{\pm0.40}$ | $17.79_{\pm0.32}$ |
| coin | $17.09_{\pm2.69}$ | $21.04_{\pm2.92}$ | $19.16_{\pm0.61}$ | $17.78_{\pm0.29}$ | $17.78_{\pm0.29}$ | $17.78_{\pm0.29}$ | $17.09_{\pm2.69}$ | $17.09_{\pm2.69}$ | $17.09_{\pm2.69}$ | $17.09_{\pm2.69}$ | $17.09_{\pm2.69}$ |
| **dopanim-rand-2** | | | | | | | | | | | |
| gt | $10.85_{\pm0.15}$ | $10.85_{\pm0.15}$ | $10.52_{\pm0.22}$ | $10.75_{\pm0.07}$ | $10.91_{\pm0.39}$ | N/A | N/A | N/A | N/A | N/A | $10.91_{\pm0.39}$ |
| mv | $20.76_{\pm0.30}$ | $23.31_{\pm0.57}$ | $28.22_{\pm0.41}$ | $20.76_{\pm0.30}$ | $20.76_{\pm0.30}$ | N/A | N/A | N/A | N/A | N/A | $20.76_{\pm0.30}$ |
| ds | $19.78_{\pm0.43}$ | $21.36_{\pm0.40}$ | $24.43_{\pm0.36}$ | $20.13_{\pm0.30}$ | $20.13_{\pm0.30}$ | $20.38_{\pm1.19}$ | $19.78_{\pm0.43}$ | $20.02_{\pm0.60}$ | $20.02_{\pm0.60}$ | $20.13_{\pm0.30}$ | $20.13_{\pm0.30}$ |
| cl | $23.72_{\pm3.45}$ | $33.33_{\pm4.67}$ | $24.26_{\pm3.93}$ | $23.72_{\pm3.45}$ | $23.72_{\pm3.45}$ | $23.72_{\pm3.45}$ | $23.43_{\pm3.46}$ | $31.02_{\pm2.75}$ | $31.02_{\pm2.75}$ | $23.80_{\pm1.76}$ | $23.72_{\pm3.45}$ |
| trace | $16.10_{\pm0.39}$ | $16.98_{\pm0.33}$ | $23.59_{\pm0.26}$ | $19.19_{\pm0.74}$ | $19.19_{\pm0.74}$ | $16.77_{\pm0.40}$ | $16.10_{\pm0.39}$ | $16.77_{\pm0.40}$ | $19.19_{\pm0.74}$ | $16.77_{\pm0.40}$ | $16.77_{\pm0.40}$ |
| conal | $18.59_{\pm0.30}$ | $18.90_{\pm0.39}$ | $20.42_{\pm0.17}$ | $18.77_{\pm0.41}$ | $18.48_{\pm0.25}$ | $18.48_{\pm0.25}$ | $19.79_{\pm0.23}$ | $18.59_{\pm0.30}$ | $18.59_{\pm0.30}$ | $19.79_{\pm0.23}$ | $18.59_{\pm0.30}$ |
| union-a | $21.30_{\pm4.32}$ | $34.20_{\pm4.06}$ | $23.75_{\pm3.17}$ | $21.30_{\pm4.32}$ | $21.30_{\pm4.32}$ | $23.75_{\pm3.17}$ | $23.75_{\pm3.17}$ | $23.75_{\pm3.17}$ | $23.75_{\pm3.17}$ | $23.75_{\pm3.17}$ | $20.20_{\pm1.99}$ |
| union-b | $18.80_{\pm0.36}$ | $24.79_{\pm2.82}$ | $19.17_{\pm0.24}$ | $19.48_{\pm0.63}$ | $19.48_{\pm0.63}$ | $20.03_{\pm2.55}$ | $18.80_{\pm0.36}$ | $20.03_{\pm2.55}$ | $20.13_{\pm2.28}$ | $19.48_{\pm0.63}$ | $19.48_{\pm0.63}$ |
| geo-w | $18.02_{\pm0.22}$ | $19.70_{\pm2.84}$ | $19.61_{\pm0.05}$ | $18.02_{\pm0.22}$ | $18.02_{\pm0.22}$ | $18.02_{\pm0.22}$ | $18.81_{\pm0.45}$ | $18.02_{\pm0.22}$ | $18.02_{\pm0.22}$ | $18.02_{\pm0.22}$ | $18.02_{\pm0.22}$ |
| geo-f | $15.29_{\pm0.25}$ | $15.93_{\pm0.32}$ | $19.07_{\pm0.45}$ | $15.29_{\pm0.25}$ | $15.29_{\pm0.25}$ | $15.29_{\pm0.25}$ | $17.42_{\pm0.14}$ | $15.29_{\pm0.25}$ | $17.49_{\pm0.33}$ | $17.49_{\pm0.33}$ | $15.29_{\pm0.25}$ |
| madl | $17.30_{\pm0.58}$ | $16.37_{\pm0.77}$ | $22.18_{\pm0.97}$ | $17.30_{\pm0.58}$ | $16.72_{\pm0.41}$ | $16.64_{\pm0.47}$ | $16.64_{\pm0.47}$ | $16.64_{\pm0.47}$ | $16.72_{\pm0.41}$ | $16.72_{\pm0.41}$ | $16.72_{\pm0.41}$ |
| crowd-ar | $18.74_{\pm0.48}$ | $21.10_{\pm1.81}$ | $19.18_{\pm0.52}$ | $18.74_{\pm0.48}$ | $18.74_{\pm0.48}$ | $18.74_{\pm0.48}$ | $18.74_{\pm0.48}$ | $18.74_{\pm0.48}$ | $18.74_{\pm0.48}$ | $18.74_{\pm0.48}$ | $18.74_{\pm0.55}$ |
| annot-mix | $17.18_{\pm0.37}$ | $16.43_{\pm0.49}$ | $18.12_{\pm0.33}$ | $17.18_{\pm0.37}$ | $17.18_{\pm0.37}$ | $17.18_{\pm0.37}$ | $16.92_{\pm0.52}$ | $16.92_{\pm0.52}$ | $16.92_{\pm0.52}$ | $16.92_{\pm0.52}$ | $16.92_{\pm0.52}$ |
| coin | $15.57_{\pm0.34}$ | $14.75_{\pm0.31}$ | $17.15_{\pm0.41}$ | $17.23_{\pm0.16}$ | $17.23_{\pm0.16}$ | $16.18_{\pm0.44}$ | $17.02_{\pm0.36}$ | $16.22_{\pm0.19}$ | $17.23_{\pm0.16}$ | $17.15_{\pm0.41}$ | $17.23_{\pm0.16}$ |
| **dopanim-rand-v** | | | | | | | | | | | |
| gt | $10.43_{\pm0.15}$ | $10.43_{\pm0.15}$ | $10.52_{\pm0.22}$ | $11.23_{\pm0.15}$ | $11.23_{\pm0.15}$ | N/A | N/A | N/A | N/A | N/A | $11.23_{\pm0.15}$ |
| mv | $18.51_{\pm0.51}$ | $21.26_{\pm0.26}$ | $23.83_{\pm0.55}$ | $18.55_{\pm0.41}$ | $18.55_{\pm0.41}$ | N/A | N/A | N/A | N/A | N/A | $18.55_{\pm0.41}$ |
| ds | $17.59_{\pm0.42}$ | $19.34_{\pm0.52}$ | $21.08_{\pm0.36}$ | $17.59_{\pm0.42}$ | $17.59_{\pm0.42}$ | $17.59_{\pm0.42}$ | $17.59_{\pm0.42}$ | $17.59_{\pm0.42}$ | $17.59_{\pm0.42}$ | $17.59_{\pm0.42}$ | $17.59_{\pm0.42}$ |
| cl | $20.54_{\pm3.71}$ | $32.44_{\pm5.76}$ | $22.84_{\pm4.37}$ | $20.54_{\pm3.71}$ | $20.54_{\pm3.71}$ | $20.54_{\pm3.71}$ | $20.54_{\pm3.71}$ | $31.40_{\pm7.11}$ | $31.40_{\pm7.11}$ | $20.54_{\pm3.71}$ | $20.54_{\pm3.71}$ |
| trace | $14.62_{\pm0.17}$ | $17.75_{\pm0.28}$ | $20.36_{\pm0.26}$ | $17.83_{\pm0.21}$ | $17.83_{\pm0.21}$ | $15.87_{\pm0.11}$ | $14.62_{\pm0.17}$ | $16.89_{\pm0.27}$ | $16.89_{\pm0.27}$ | $16.89_{\pm0.27}$ | $16.89_{\pm0.27}$ |
| conal | $17.49_{\pm0.68}$ | $17.15_{\pm0.42}$ | $18.52_{\pm0.20}$ | $17.26_{\pm0.16}$ | $17.26_{\pm0.16}$ | $17.51_{\pm0.20}$ | $18.63_{\pm1.85}$ | $17.51_{\pm0.20}$ | $17.51_{\pm0.20}$ | $18.63_{\pm1.85}$ | $17.26_{\pm0.16}$ |
| union-a | $18.86_{\pm2.34}$ | $30.60_{\pm6.42}$ | $22.97_{\pm4.49}$ | $19.05_{\pm2.97}$ | $19.05_{\pm2.97}$ | $22.97_{\pm4.49}$ | $22.97_{\pm4.49}$ | $22.97_{\pm4.49}$ | $22.97_{\pm4.49}$ | $22.97_{\pm4.49}$ | $22.97_{\pm4.49}$ |
| union-b | $17.73_{\pm0.21}$ | $18.54_{\pm2.82}$ | $16.94_{\pm0.27}$ | $17.73_{\pm0.21}$ | $17.73_{\pm0.21}$ | $19.01_{\pm2.60}$ | $17.75_{\pm0.50}$ | $17.84_{\pm0.32}$ | $16.57_{\pm0.43}$ | $17.75_{\pm0.50}$ | $16.57_{\pm0.43}$ |
| geo-w | $17.14_{\pm0.38}$ | $18.70_{\pm2.64}$ | $17.44_{\pm0.30}$ | $17.14_{\pm0.38}$ | $17.14_{\pm0.38}$ | $18.81_{\pm2.89}$ | $17.23_{\pm0.16}$ | $17.97_{\pm0.48}$ | $17.97_{\pm0.48}$ | $17.14_{\pm0.38}$ | $17.14_{\pm0.38}$ |
| geo-f | $14.62_{\pm0.32}$ | $15.35_{\pm0.52}$ | $16.89_{\pm0.27}$ | $14.90_{\pm0.12}$ | $14.90_{\pm0.12}$ | $14.51_{\pm0.12}$ | $16.15_{\pm0.18}$ | $14.90_{\pm0.12}$ | $15.73_{\pm0.21}$ | $15.73_{\pm0.21}$ | $15.73_{\pm0.21}$ |
| madl | $14.24_{\pm0.33}$ | $15.64_{\pm0.52}$ | $20.12_{\pm1.24}$ | $14.88_{\pm0.65}$ | $14.88_{\pm0.65}$ | $14.88_{\pm0.65}$ | $14.88_{\pm0.65}$ | $14.73_{\pm0.57}$ | $14.73_{\pm0.57}$ | $15.11_{\pm0.47}$ | $14.73_{\pm0.57}$ |
| crowd-ar | $16.79_{\pm0.38}$ | $18.28_{\pm2.30}$ | $17.58_{\pm0.27}$ | $16.79_{\pm0.38}$ | $16.79_{\pm0.38}$ | $16.79_{\pm0.38}$ | $16.79_{\pm0.38}$ | $16.79_{\pm0.38}$ | $16.79_{\pm0.38}$ | $16.79_{\pm0.38}$ | $16.79_{\pm0.38}$ |
| annot-mix | $14.82_{\pm0.47}$ | $15.65_{\pm0.60}$ | $16.57_{\pm0.36}$ | $14.82_{\pm0.47}$ | $15.46_{\pm0.26}$ | $14.82_{\pm0.47}$ | $14.82_{\pm0.47}$ | $15.46_{\pm0.26}$ | $15.46_{\pm0.26}$ | $15.91_{\pm0.38}$ | $15.46_{\pm0.26}$ |
| coin | $14.27_{\pm0.13}$ | $16.10_{\pm0.75}$ | $15.63_{\pm0.32}$ | $16.32_{\pm0.31}$ | $16.32_{\pm0.31}$ | $14.36_{\pm0.28}$ | $16.17_{\pm0.14}$ | $15.86_{\pm0.49}$ | $16.32_{\pm0.31}$ | $16.17_{\pm0.14}$ | $16.32_{\pm0.31}$ |

Table 14: **Zero-one loss results [%] (part VI).** Continued from the previous page.

| Approach | Baseline | | | Aggregation-level | | | | Crowd-level | | | Ensemble |
|---|---|---|---|---|---|---|---|---|---|---|---|
| | TRUE[*] | DEF-DATA[*] | DEF | AEU | AEC | ALU | ALC | CXU | CEC | CLC | ENS |
| **dopanim-full** | | | | | | | | | | | |
| gt | $11.02_{\pm0.28}$ | $11.02_{\pm0.28}$ | $10.52_{\pm0.22}$ | $10.57_{\pm0.26}$ | $10.57_{\pm0.26}$ | N/A | N/A | N/A | N/A | N/A | $10.57_{\pm0.26}$ |
| mv | $17.32_{\pm0.58}$ | $17.82_{\pm0.28}$ | $20.59_{\pm0.17}$ | $17.32_{\pm0.58}$ | $17.32_{\pm0.58}$ | N/A | N/A | N/A | N/A | N/A | $17.32_{\pm0.58}$ |
| ds | $16.82_{\pm0.51}$ | $17.16_{\pm0.27}$ | $19.07_{\pm0.17}$ | $17.45_{\pm0.29}$ | $17.45_{\pm0.29}$ | $16.82_{\pm0.51}$ | $16.82_{\pm0.51}$ | $17.45_{\pm0.29}$ | $17.33_{\pm0.31}$ | $17.33_{\pm0.31}$ | $17.45_{\pm0.29}$ |
| cl | $18.20_{\pm0.52}$ | $42.66_{\pm1.87}$ | $22.36_{\pm4.39}$ | $18.20_{\pm0.52}$ | $18.20_{\pm0.52}$ | $22.09_{\pm5.16}$ | $19.63_{\pm2.30}$ | $33.90_{\pm3.31}$ | $25.66_{\pm4.06}$ | $19.63_{\pm2.30}$ | $18.20_{\pm0.52}$ |
| trace | $13.80_{\pm0.20}$ | $15.19_{\pm0.15}$ | $18.55_{\pm0.40}$ | $\mathbf{14.12}_{\pm0.27}$ | $15.89_{\pm0.17}$ | $14.12_{\pm0.27}$ | $14.12_{\pm0.27}$ | $16.85_{\pm0.25}$ | $17.20_{\pm0.76}$ | $17.20_{\pm0.76}$ | $15.89_{\pm0.17}$ |
| conal | $16.62_{\pm0.14}$ | $19.24_{\pm2.28}$ | $17.31_{\pm0.12}$ | $16.76_{\pm0.28}$ | $16.76_{\pm0.28}$ | $16.76_{\pm0.28}$ | $19.39_{\pm2.07}$ | $16.58_{\pm0.13}$ | $16.58_{\pm0.13}$ | $17.09_{\pm0.40}$ | $16.46_{\pm0.18}$ |
| union-a | $20.12_{\pm3.93}$ | $34.56_{\pm4.62}$ | $23.14_{\pm3.22}$ | $20.12_{\pm3.93}$ | $20.12_{\pm3.93}$ | $23.14_{\pm3.22}$ | $23.14_{\pm3.22}$ | $23.14_{\pm3.22}$ | $23.14_{\pm3.22}$ | $23.14_{\pm3.22}$ | $20.12_{\pm3.93}$ |
| union-b | $17.82_{\pm1.94}$ | $20.57_{\pm4.03}$ | $16.26_{\pm0.38}$ | $17.82_{\pm1.94}$ | $17.82_{\pm1.94}$ | $18.88_{\pm2.70}$ | $18.03_{\pm1.58}$ | $17.73_{\pm2.75}$ | $17.82_{\pm1.94}$ | $18.03_{\pm1.58}$ | $16.28_{\pm0.33}$ |
| geo-w | $16.06_{\pm0.29}$ | $19.94_{\pm2.55}$ | $16.50_{\pm0.25}$ | $16.06_{\pm0.29}$ | $16.06_{\pm0.29}$ | $16.54_{\pm0.18}$ | $17.48_{\pm0.53}$ | $16.59_{\pm0.15}$ | $16.06_{\pm0.29}$ | $16.06_{\pm0.29}$ | $16.06_{\pm0.29}$ |
| geo-f | $\mathbf{13.71}_{\pm0.49}$ | $14.90_{\pm0.21}$ | $16.25_{\pm0.30}$ | $15.80_{\pm0.19}$ | $15.80_{\pm0.19}$ | $\mathbf{13.71}_{\pm0.49}$ | $16.77_{\pm0.20}$ | $14.93_{\pm0.42}$ | $15.52_{\pm0.31}$ | $15.52_{\pm0.31}$ | $14.93_{\pm0.42}$ |
| madl | $14.15_{\pm0.28}$ | $14.99_{\pm0.50}$ | $17.86_{\pm1.98}$ | $16.44_{\pm0.12}$ | $16.44_{\pm0.12}$ | $15.26_{\pm1.17}$ | $15.26_{\pm1.17}$ | $14.91_{\pm0.92}$ | $17.95_{\pm1.37}$ | $\mathbf{14.22}_{\pm0.28}$ | $14.91_{\pm0.92}$ |
| crowd-ar | $16.27_{\pm0.19}$ | $21.47_{\pm1.74}$ | $16.63_{\pm0.13}$ | $16.27_{\pm0.19}$ | $16.27_{\pm0.19}$ | $16.27_{\pm0.19}$ | $16.27_{\pm0.19}$ | $16.27_{\pm0.19}$ | $16.27_{\pm0.19}$ | $16.27_{\pm0.19}$ | $16.27_{\pm0.19}$ |
| annot-mix | $14.38_{\pm0.28}$ | $\mathbf{14.77}_{\pm0.35}$ | $15.96_{\pm0.13}$ | $15.35_{\pm0.10}$ | $15.34_{\pm0.44}$ | $15.54_{\pm0.33}$ | $\mathbf{14.38}_{\pm0.28}$ | $15.76_{\pm0.22}$ | $15.40_{\pm0.54}$ | $15.40_{\pm0.54}$ | $15.54_{\pm0.33}$ |
| coin | $14.12_{\pm0.12}$ | $21.41_{\pm9.11}$ | $\mathbf{14.72}_{\pm0.31}$ | $14.90_{\pm0.19}$ | $14.90_{\pm0.19}$ | $15.12_{\pm0.31}$ | $15.87_{\pm0.28}$ | $15.12_{\pm0.31}$ | $\mathbf{14.90}_{\pm0.19}$ | $14.90_{\pm0.19}$ | $\mathbf{14.90}_{\pm0.19}$ |
| **reuters-worst-1** | | | | | | | | | | | |
| gt | $3.98_{\pm0.17}$ | $3.98_{\pm0.17}$ | $4.14_{\pm0.07}$ | $7.25_{\pm0.52}$ | $7.25_{\pm0.52}$ | N/A | N/A | N/A | N/A | N/A | $7.25_{\pm0.52}$ |
| mv | $48.80_{\pm2.40}$ | $59.46_{\pm1.73}$ | $58.90_{\pm1.18}$ | $49.41_{\pm3.00}$ | $49.41_{\pm3.00}$ | N/A | N/A | N/A | N/A | N/A | $49.41_{\pm3.00}$ |
| ds | $48.80_{\pm2.40}$ | $59.46_{\pm1.73}$ | $58.90_{\pm1.18}$ | $49.41_{\pm3.00}$ | $49.41_{\pm3.00}$ | $49.41_{\pm3.00}$ | $49.41_{\pm3.00}$ | $49.41_{\pm3.00}$ | $49.41_{\pm3.00}$ | $49.41_{\pm3.00}$ | $49.41_{\pm3.00}$ |
| cl | $\mathbf{32.66}_{\pm2.17}$ | $58.10_{\pm1.63}$ | $57.20_{\pm1.04}$ | $\mathbf{32.66}_{\pm2.17}$ | $\mathbf{32.66}_{\pm2.17}$ | $\mathbf{32.66}_{\pm2.17}$ | $34.92_{\pm9.83}$ | $\mathbf{32.66}_{\pm2.17}$ | $\mathbf{32.66}_{\pm2.17}$ | $34.92_{\pm9.83}$ | $\mathbf{32.66}_{\pm2.17}$ |
| trace | $48.77_{\pm2.32}$ | $58.69_{\pm0.77}$ | $58.50_{\pm1.25}$ | $51.75_{\pm5.58}$ | $51.75_{\pm5.58}$ | $51.75_{\pm5.58}$ | $51.75_{\pm5.58}$ | $52.53_{\pm0.61}$ | $52.53_{\pm0.61}$ | $52.53_{\pm0.61}$ | $51.75_{\pm5.58}$ |
| conal | $47.89_{\pm5.64}$ | $60.64_{\pm1.19}$ | $59.65_{\pm1.61}$ | $51.03_{\pm1.54}$ | $51.03_{\pm1.54}$ | $51.03_{\pm1.54}$ | $51.03_{\pm1.54}$ | $51.03_{\pm1.54}$ | $51.03_{\pm1.54}$ | $51.03_{\pm1.54}$ | $51.03_{\pm1.54}$ |
| union-a | $47.38_{\pm4.54}$ | $58.64_{\pm0.46}$ | $59.13_{\pm1.64}$ | $43.80_{\pm5.54}$ | $43.80_{\pm5.54}$ | $43.80_{\pm5.54}$ | $43.80_{\pm5.54}$ | $51.62_{\pm4.91}$ | $51.62_{\pm4.91}$ | $51.62_{\pm4.91}$ | $43.80_{\pm5.54}$ |
| union-b | $48.51_{\pm3.05}$ | $59.30_{\pm1.90}$ | $57.88_{\pm1.32}$ | $48.30_{\pm7.86}$ | $48.30_{\pm7.86}$ | $48.30_{\pm7.86}$ | $37.18_{\pm3.44}$ | $37.18_{\pm3.44}$ | $37.18_{\pm3.44}$ | $37.18_{\pm3.44}$ | $37.18_{\pm3.44}$ |
| geo-w | $48.47_{\pm2.96}$ | $\mathbf{57.97}_{\pm0.94}$ | $57.77_{\pm1.30}$ | $51.26_{\pm4.89}$ | $51.26_{\pm4.89}$ | $51.26_{\pm4.89}$ | $36.11_{\pm1.45}$ | $36.11_{\pm1.45}$ | $36.11_{\pm1.45}$ | $36.11_{\pm1.45}$ | $36.11_{\pm1.45}$ |
| geo-f | $44.55_{\pm3.29}$ | $58.07_{\pm0.89}$ | $58.69_{\pm1.39}$ | $42.89_{\pm4.13}$ | $42.89_{\pm4.13}$ | $42.89_{\pm4.13}$ | $42.63_{\pm4.60}$ | $42.63_{\pm4.60}$ | $42.63_{\pm4.60}$ | $42.63_{\pm4.60}$ | $37.04_{\pm1.97}$ |
| madl | $51.41_{\pm0.78}$ | $59.77_{\pm1.62}$ | $59.32_{\pm1.10}$ | $51.41_{\pm0.78}$ | $51.41_{\pm0.78}$ | $51.41_{\pm0.78}$ | $51.41_{\pm0.78}$ | $41.45_{\pm0.27}$ | $41.45_{\pm0.27}$ | $41.45_{\pm0.27}$ | $41.45_{\pm0.27}$ |
| crowd-ar | $44.60_{\pm1.26}$ | $58.86_{\pm1.36}$ | $58.62_{\pm0.65}$ | $44.60_{\pm1.26}$ | $44.60_{\pm1.26}$ | $44.60_{\pm1.26}$ | $44.60_{\pm1.26}$ | $44.60_{\pm1.26}$ | $44.60_{\pm1.26}$ | $44.60_{\pm1.26}$ | $44.60_{\pm1.26}$ |
| annot-mix | $47.11_{\pm4.60}$ | $61.35_{\pm1.59}$ | $62.35_{\pm0.61}$ | $48.46_{\pm2.38}$ | $48.46_{\pm2.38}$ | $48.46_{\pm2.38}$ | $44.36_{\pm5.20}$ | $44.36_{\pm5.20}$ | $44.36_{\pm5.20}$ | $44.36_{\pm5.20}$ | $44.36_{\pm5.20}$ |
| coin | $48.06_{\pm1.59}$ | $59.24_{\pm0.54}$ | $62.95_{\pm2.82}$ | $48.06_{\pm1.59}$ | $48.06_{\pm1.59}$ | $48.06_{\pm1.59}$ | $53.01_{\pm7.20}$ | $72.12_{\pm5.67}$ | $72.12_{\pm5.67}$ | $53.01_{\pm7.20}$ | $48.06_{\pm1.59}$ |
| **reuters-worst-2** | | | | | | | | | | | |
| gt | $3.79_{\pm0.14}$ | $3.79_{\pm0.14}$ | $4.14_{\pm0.07}$ | $4.14_{\pm0.07}$ | $4.14_{\pm0.07}$ | N/A | N/A | N/A | N/A | N/A | $4.14_{\pm0.07}$ |
| mv | $26.22_{\pm1.65}$ | $40.89_{\pm1.25}$ | $43.12_{\pm0.79}$ | $26.22_{\pm1.65}$ | $26.22_{\pm1.65}$ | N/A | N/A | N/A | N/A | N/A | $26.22_{\pm1.65}$ |
| ds | $23.34_{\pm0.58}$ | $33.63_{\pm0.49}$ | $34.38_{\pm0.72}$ | $23.34_{\pm0.58}$ | $23.34_{\pm0.58}$ | $23.34_{\pm0.58}$ | $23.34_{\pm0.58}$ | $23.12_{\pm1.07}$ | $23.12_{\pm1.07}$ | $23.12_{\pm1.07}$ | $23.34_{\pm0.58}$ |
| cl | $19.32_{\pm1.52}$ | $27.11_{\pm1.17}$ | $31.03_{\pm0.69}$ | $20.20_{\pm2.44}$ | $20.20_{\pm2.44}$ | $19.32_{\pm1.52}$ | $19.32_{\pm1.52}$ | $19.32_{\pm1.52}$ | $19.32_{\pm1.52}$ | $19.32_{\pm1.52}$ | $19.32_{\pm1.52}$ |
| trace | $20.58_{\pm1.58}$ | $36.63_{\pm1.01}$ | $35.76_{\pm0.83}$ | $20.58_{\pm1.58}$ | $20.58_{\pm1.58}$ | $20.58_{\pm1.58}$ | $26.25_{\pm1.32}$ | $26.25_{\pm1.32}$ | $26.25_{\pm1.32}$ | $26.25_{\pm1.32}$ | $20.58_{\pm1.58}$ |
| conal | $22.11_{\pm1.20}$ | $32.99_{\pm0.71}$ | $35.70_{\pm0.53}$ | $22.11_{\pm1.20}$ | $22.11_{\pm1.20}$ | $22.11_{\pm1.20}$ | $27.75_{\pm0.96}$ | $22.11_{\pm1.20}$ | $22.11_{\pm1.20}$ | $27.75_{\pm0.96}$ | $22.11_{\pm1.20}$ |
| union-a | $19.33_{\pm1.97}$ | $34.01_{\pm2.53}$ | $41.08_{\pm0.56}$ | $\mathbf{19.33}_{\pm1.97}$ | $\mathbf{19.33}_{\pm1.97}$ | $39.77_{\pm20.3}$ | $25.25_{\pm3.59}$ | $26.07_{\pm1.40}$ | $26.07_{\pm1.40}$ | $39.77_{\pm20.3}$ | $39.77_{\pm20.3}$ |
| union-b | $\mathbf{18.17}_{\pm1.89}$ | $32.73_{\pm1.25}$ | $33.81_{\pm1.06}$ | $21.12_{\pm2.04}$ | $21.12_{\pm2.04}$ | $\mathbf{18.17}_{\pm1.89}$ | $\mathbf{18.17}_{\pm1.89}$ | $\mathbf{18.17}_{\pm1.89}$ | $\mathbf{18.17}_{\pm1.89}$ | $\mathbf{18.17}_{\pm1.89}$ | $\mathbf{18.17}_{\pm1.89}$ |
| geo-w | $20.86_{\pm2.02}$ | $30.19_{\pm1.60}$ | $33.34_{\pm0.80}$ | $20.86_{\pm2.02}$ | $20.86_{\pm2.02}$ | $\mathbf{17.23}_{\pm3.02}$ | $18.49_{\pm1.39}$ | $18.49_{\pm1.39}$ | $18.49_{\pm1.39}$ | $18.49_{\pm1.39}$ | $\mathbf{17.23}_{\pm3.02}$ |
| geo-f | $20.53_{\pm1.69}$ | $\mathbf{27.00}_{\pm1.32}$ | $33.61_{\pm0.61}$ | $20.53_{\pm1.69}$ | $20.53_{\pm1.69}$ | $20.53_{\pm1.69}$ | $40.60_{\pm1.76}$ | $40.60_{\pm1.76}$ | $40.60_{\pm1.76}$ | $40.60_{\pm1.76}$ | $22.59_{\pm1.08}$ |
| madl | $23.38_{\pm0.61}$ | $35.00_{\pm5.88}$ | $37.64_{\pm3.49}$ | $23.38_{\pm0.61}$ | $23.38_{\pm0.61}$ | $24.60_{\pm3.76}$ | $19.57_{\pm5.84}$ | $19.57_{\pm5.84}$ | $19.57_{\pm5.84}$ | $19.57_{\pm5.84}$ | $19.57_{\pm5.84}$ |
| crowd-ar | $22.36_{\pm1.62}$ | $32.49_{\pm0.64}$ | $33.08_{\pm0.65}$ | $22.36_{\pm1.62}$ | $22.36_{\pm1.62}$ | $22.36_{\pm1.62}$ | $22.36_{\pm1.62}$ | $22.36_{\pm1.62}$ | $22.36_{\pm1.62}$ | $22.36_{\pm1.62}$ | $22.36_{\pm1.62}$ |
| annot-mix | $21.67_{\pm1.28}$ | $35.45_{\pm1.79}$ | $43.81_{\pm2.11}$ | $26.73_{\pm1.54}$ | $26.73_{\pm1.54}$ | $20.91_{\pm0.81}$ | $20.91_{\pm0.81}$ | $20.91_{\pm0.81}$ | $20.91_{\pm0.81}$ | $20.91_{\pm0.81}$ | $26.73_{\pm1.54}$ |
| coin | $23.79_{\pm1.16}$ | $31.78_{\pm2.40}$ | $36.86_{\pm1.47}$ | $23.79_{\pm1.16}$ | $23.79_{\pm1.16}$ | $23.79_{\pm1.16}$ | $29.43_{\pm1.76}$ | $69.49_{\pm2.90}$ | $69.49_{\pm2.90}$ | $69.49_{\pm2.90}$ | $26.07_{\pm1.32}$ |
| **reuters-worst-v** | | | | | | | | | | | |
| gt | $3.88_{\pm0.09}$ | $3.88_{\pm0.09}$ | $4.14_{\pm0.07}$ | $4.13_{\pm0.11}$ | $3.75_{\pm0.14}$ | N/A | N/A | N/A | N/A | N/A | $4.13_{\pm0.11}$ |
| mv | $20.55_{\pm0.97}$ | $38.84_{\pm0.41}$ | $40.13_{\pm0.62}$ | $20.55_{\pm0.97}$ | $20.55_{\pm0.97}$ | N/A | N/A | N/A | N/A | N/A | $20.55_{\pm0.97}$ |
| ds | $18.92_{\pm0.96}$ | $30.49_{\pm1.15}$ | $32.04_{\pm1.14}$ | $18.92_{\pm0.96}$ | $18.92_{\pm0.96}$ | $18.92_{\pm0.96}$ | $18.92_{\pm0.96}$ | $23.31_{\pm1.09}$ | $24.64_{\pm0.40}$ | $24.64_{\pm0.40}$ | $18.92_{\pm0.96}$ |
| cl | $21.11_{\pm2.09}$ | $\mathbf{17.88}_{\pm1.39}$ | $29.06_{\pm0.71}$ | $20.40_{\pm1.04}$ | $20.40_{\pm1.04}$ | $20.40_{\pm1.04}$ | $21.11_{\pm2.09}$ | $21.11_{\pm2.09}$ | $21.11_{\pm2.09}$ | $21.11_{\pm2.09}$ | $20.40_{\pm1.04}$ |
| trace | $18.48_{\pm0.51}$ | $31.18_{\pm0.47}$ | $31.48_{\pm1.24}$ | $18.48_{\pm0.51}$ | $18.48_{\pm0.51}$ | $18.48_{\pm0.51}$ | $24.66_{\pm2.03}$ | $24.66_{\pm2.03}$ | $24.66_{\pm2.03}$ | $24.66_{\pm2.03}$ | $18.48_{\pm0.51}$ |
| conal | $17.93_{\pm2.19}$ | $29.04_{\pm1.22}$ | $32.16_{\pm1.02}$ | $18.52_{\pm0.78}$ | $17.93_{\pm2.19}$ | $18.52_{\pm0.78}$ | $17.93_{\pm2.19}$ | $18.52_{\pm0.78}$ | $17.93_{\pm2.19}$ | $17.93_{\pm2.19}$ | $17.93_{\pm2.19}$ |
| union-a | $40.03_{\pm23.7}$ | $37.20_{\pm4.43}$ | $36.41_{\pm0.79}$ | $17.96_{\pm1.41}$ | $17.96_{\pm1.41}$ | $40.03_{\pm23.7}$ | $40.03_{\pm23.7}$ | $16.93_{\pm2.53}$ | $\mathbf{16.93}_{\pm2.53}$ | $\mathbf{16.93}_{\pm2.53}$ | $16.93_{\pm2.53}$ |
| union-b | $17.99_{\pm1.86}$ | $22.21_{\pm0.55}$ | $30.00_{\pm0.20}$ | $17.99_{\pm1.86}$ | $17.99_{\pm1.86}$ | $21.96_{\pm2.79}$ | $24.43_{\pm2.02}$ | $21.96_{\pm2.79}$ | $21.96_{\pm2.79}$ | $24.43_{\pm2.02}$ | $21.96_{\pm2.79}$ |
| geo-w | $18.20_{\pm1.84}$ | $\mathbf{17.25}_{\pm0.44}$ | $30.25_{\pm0.61}$ | $18.20_{\pm1.84}$ | $18.20_{\pm1.84}$ | $19.85_{\pm1.26}$ | $19.85_{\pm1.26}$ | $19.85_{\pm1.26}$ | $19.85_{\pm1.26}$ | $25.73_{\pm2.57}$ | $19.85_{\pm1.26}$ |
| geo-f | $\mathbf{14.76}_{\pm0.60}$ | $18.81_{\pm1.11}$ | $30.05_{\pm0.67}$ | $18.05_{\pm1.79}$ | $\mathbf{14.76}_{\pm0.60}$ | $15.62_{\pm1.64}$ | $34.91_{\pm17.0}$ | $34.91_{\pm17.0}$ | $34.91_{\pm17.0}$ | $27.70_{\pm2.90}$ | $15.62_{\pm1.64}$ |
| madl | $17.48_{\pm0.82}$ | $29.76_{\pm8.49}$ | $32.66_{\pm2.65}$ | $17.48_{\pm0.82}$ | $17.48_{\pm0.82}$ | $17.06_{\pm0.83}$ | $\mathbf{17.06}_{\pm0.83}$ | $17.06_{\pm0.83}$ | $17.06_{\pm0.83}$ | $17.06_{\pm0.83}$ | $17.06_{\pm0.83}$ |
| crowd-ar | $18.57_{\pm1.55}$ | $23.49_{\pm1.07}$ | $31.45_{\pm1.38}$ | $18.57_{\pm1.55}$ | $18.57_{\pm1.55}$ | $18.57_{\pm1.55}$ | $18.57_{\pm1.55}$ | $\mathbf{16.02}_{\pm1.01}$ | $18.57_{\pm1.55}$ | $18.57_{\pm1.55}$ | $18.57_{\pm1.55}$ |
| annot-mix | $18.62_{\pm0.76}$ | $34.92_{\pm4.11}$ | $37.74_{\pm3.37}$ | $16.91_{\pm1.13}$ | $16.91_{\pm1.13}$ | $16.91_{\pm1.13}$ | $20.07_{\pm0.85}$ | $20.07_{\pm0.85}$ | $20.07_{\pm0.85}$ | $20.07_{\pm0.85}$ | $16.91_{\pm1.13}$ |
| coin | $16.86_{\pm0.93}$ | $27.14_{\pm1.86}$ | $35.68_{\pm0.83}$ | $\mathbf{16.86}_{\pm0.93}$ | $16.86_{\pm0.93}$ | $19.13_{\pm3.81}$ | $19.13_{\pm3.81}$ | $19.13_{\pm3.81}$ | $19.13_{\pm3.81}$ | $19.13_{\pm3.81}$ | $19.13_{\pm3.81}$ |

Continued on the next page.

Table 14: **Zero-one loss results [%] (part VII).** Continued from the previous page.

| Approach | Baseline | | | Aggregation-level | | | | Crowd-level | | | Ensemble |
|---|---|---|---|---|---|---|---|---|---|---|---|
| | TRUE[*] | DEF-DATA[*] | DEF | AEU | AEC | ALU | ALC | CXU | CEC | CLC | ENS |
| | | | | | | reuters-rand-1 | | | | | |
| gt | $3.94_{\pm0.15}$ | $3.94_{\pm0.15}$ | $4.14_{\pm0.07}$ | $4.05_{\pm0.21}$ | $4.05_{\pm0.21}$ | N/A | N/A | N/A | N/A | N/A | $4.05_{\pm0.21}$ |
| mv | $13.29_{\pm0.67}$ | $25.86_{\pm0.66}$ | $26.55_{\pm0.58}$ | $\underline{15.16}_{\pm0.66}$ | $\underline{15.16}_{\pm0.66}$ | N/A | N/A | N/A | N/A | N/A | $\underline{15.16}_{\pm0.66}$ |
| ds | $13.29_{\pm0.67}$ | $25.86_{\pm0.66}$ | $26.55_{\pm0.58}$ | $\underline{15.16}_{\pm0.66}$ | $\underline{15.16}_{\pm0.66}$ | $\underline{15.16}_{\pm0.66}$ | $\underline{15.16}_{\pm0.66}$ | $\underline{15.16}_{\pm0.66}$ | $\underline{15.16}_{\pm0.66}$ | $\underline{15.16}_{\pm0.66}$ | $\underline{15.16}_{\pm0.66}$ |
| cl | $14.64_{\pm0.84}$ | $22.47_{\pm3.56}$ | $24.72_{\pm0.78}$ | $\underline{14.28}_{\pm1.16}$ | $\underline{14.28}_{\pm1.16}$ | $\underline{14.28}_{\pm1.16}$ | $21.37_{\pm3.56}$ | $14.64_{\pm0.84}$ | $14.64_{\pm0.84}$ | $21.37_{\pm3.56}$ | $14.64_{\pm0.84}$ |
| trace | $13.86_{\pm0.47}$ | $\mathbf{15.17}_{\pm0.88}$ | $26.64_{\pm0.74}$ | $\underline{14.91}_{\pm0.41}$ | $\underline{14.91}_{\pm0.41}$ | $\underline{14.91}_{\pm0.41}$ | $18.23_{\pm0.97}$ | $\underline{14.91}_{\pm0.41}$ | $\underline{14.91}_{\pm0.41}$ | $18.23_{\pm0.97}$ | $\underline{14.91}_{\pm0.41}$ |
| conal | $13.85_{\pm1.13}$ | $22.22_{\pm4.04}$ | $25.33_{\pm0.39}$ | $\underline{13.85}_{\pm1.13}$ | $\underline{13.85}_{\pm1.13}$ | $\underline{13.85}_{\pm1.13}$ | $\underline{13.85}_{\pm1.13}$ | $\underline{13.85}_{\pm1.13}$ | $\underline{13.85}_{\pm1.13}$ | $\underline{13.85}_{\pm1.13}$ | $\underline{13.85}_{\pm1.13}$ |
| union-a | $14.79_{\pm1.28}$ | $34.62_{\pm14.2}$ | $26.58_{\pm0.71}$ | $14.79_{\pm1.28}$ | $14.79_{\pm1.28}$ | $\underline{12.99}_{\pm0.88}$ | $\mathbf{12.99}_{\pm0.88}$ | $\underline{12.99}_{\pm0.88}$ | $\underline{12.99}_{\pm0.88}$ | $\mathbf{12.99}_{\pm0.88}$ | $\underline{12.99}_{\pm0.88}$ |
| union-b | $14.34_{\pm0.92}$ | $22.25_{\pm1.81}$ | $24.75_{\pm0.98}$ | $\underline{13.88}_{\pm0.68}$ | $\underline{13.88}_{\pm0.68}$ | $\underline{13.88}_{\pm0.68}$ | $20.66_{\pm3.02}$ | $\underline{13.88}_{\pm0.68}$ | $\underline{13.88}_{\pm0.68}$ | $20.66_{\pm3.02}$ | $\underline{13.88}_{\pm0.68}$ |
| geo-w | $14.44_{\pm0.94}$ | $20.68_{\pm1.09}$ | $25.00_{\pm0.72}$ | $\underline{13.92}_{\pm0.61}$ | $\underline{13.92}_{\pm0.61}$ | $\underline{13.92}_{\pm0.61}$ | $19.53_{\pm2.90}$ | $15.24_{\pm1.83}$ | $15.24_{\pm1.83}$ | $19.53_{\pm2.90}$ | $15.24_{\pm1.83}$ |
| geo-f | $12.69_{\pm0.44}$ | $18.31_{\pm2.70}$ | $\mathbf{24.39}_{\pm0.63}$ | $\underline{12.69}_{\pm0.44}$ | $\underline{12.69}_{\pm0.44}$ | $\underline{12.69}_{\pm0.44}$ | $23.23_{\pm1.79}$ | $\underline{12.69}_{\pm0.44}$ | $\underline{12.69}_{\pm0.44}$ | $23.23_{\pm1.79}$ | $\underline{12.69}_{\pm0.44}$ |
| madl | $14.97_{\pm1.34}$ | $23.88_{\pm12.7}$ | $27.26_{\pm1.41}$ | $18.36_{\pm0.42}$ | $18.36_{\pm0.42}$ | $18.36_{\pm0.42}$ | $21.78_{\pm2.97}$ | $\underline{14.97}_{\pm1.34}$ | $\underline{14.97}_{\pm1.34}$ | $21.78_{\pm2.97}$ | $14.98_{\pm1.19}$ |
| crowd-ar | $14.90_{\pm0.93}$ | $21.62_{\pm1.10}$ | $24.87_{\pm1.00}$ | $\underline{14.46}_{\pm0.61}$ | $\underline{14.46}_{\pm0.61}$ | $\underline{14.46}_{\pm0.61}$ | $\underline{14.46}_{\pm0.61}$ | $\underline{14.46}_{\pm0.61}$ | $\underline{14.46}_{\pm0.61}$ | $\underline{14.46}_{\pm0.61}$ | $\underline{14.46}_{\pm0.61}$ |
| annot-mix | $13.22_{\pm0.49}$ | $15.41_{\pm1.89}$ | $29.56_{\pm0.51}$ | $\underline{13.22}_{\pm0.49}$ | $\underline{13.22}_{\pm0.49}$ | $\underline{13.22}_{\pm0.49}$ | $17.33_{\pm0.90}$ | $13.58_{\pm0.80}$ | $13.58_{\pm0.80}$ | $17.33_{\pm0.90}$ | $\underline{13.22}_{\pm0.49}$ |
| coin | $\mathbf{11.01}_{\pm0.57}$ | $53.89_{\pm9.36}$ | $28.43_{\pm0.84}$ | $\mathbf{11.01}_{\pm0.57}$ | $\mathbf{11.01}_{\pm0.57}$ | $\mathbf{11.01}_{\pm0.57}$ | $37.92_{\pm5.72}$ | $\mathbf{11.01}_{\pm0.57}$ | $\mathbf{11.01}_{\pm0.57}$ | $37.92_{\pm5.72}$ | $\mathbf{11.01}_{\pm0.57}$ |
| | | | | | | reuters-rand-2 | | | | | |
| gt | $3.84_{\pm0.09}$ | $3.84_{\pm0.09}$ | $4.14_{\pm0.07}$ | $4.14_{\pm0.07}$ | $4.14_{\pm0.07}$ | N/A | N/A | N/A | N/A | N/A | $4.14_{\pm0.07}$ |
| mv | $13.05_{\pm0.76}$ | $27.22_{\pm0.67}$ | $29.22_{\pm0.98}$ | $\underline{13.05}_{\pm0.76}$ | $\underline{13.05}_{\pm0.76}$ | N/A | N/A | N/A | N/A | N/A | $\underline{13.05}_{\pm0.76}$ |
| ds | $12.87_{\pm0.95}$ | $19.77_{\pm0.65}$ | $20.53_{\pm0.60}$ | $\underline{12.87}_{\pm0.95}$ | $\underline{12.87}_{\pm0.95}$ | $14.99_{\pm0.37}$ | $14.99_{\pm0.37}$ | $14.76_{\pm0.64}$ | $14.76_{\pm0.64}$ | $14.76_{\pm0.64}$ | $14.99_{\pm0.37}$ |
| cl | $11.45_{\pm0.77}$ | $13.17_{\pm0.33}$ | $\mathbf{15.31}_{\pm0.31}$ | $11.45_{\pm0.77}$ | $11.45_{\pm0.77}$ | $\underline{11.41}_{\pm1.77}$ | $15.04_{\pm1.08}$ | $\underline{11.41}_{\pm1.77}$ | $11.45_{\pm0.77}$ | $15.04_{\pm1.08}$ | $\underline{11.41}_{\pm1.77}$ |
| trace | $11.22_{\pm0.58}$ | $20.91_{\pm1.10}$ | $18.92_{\pm0.68}$ | $\underline{11.22}_{\pm0.58}$ | $\underline{11.22}_{\pm0.58}$ | $\underline{11.22}_{\pm0.58}$ | $11.55_{\pm0.73}$ | $11.55_{\pm0.73}$ | $\underline{11.22}_{\pm0.58}$ | $11.55_{\pm0.73}$ | $\underline{11.22}_{\pm0.58}$ |
| conal | $11.51_{\pm0.57}$ | $15.58_{\pm0.85}$ | $18.44_{\pm0.94}$ | $\underline{11.36}_{\pm0.59}$ | $\underline{11.36}_{\pm0.59}$ | $12.19_{\pm0.38}$ | $12.19_{\pm0.38}$ | $\underline{11.36}_{\pm0.59}$ | $\underline{11.36}_{\pm0.59}$ | $11.51_{\pm0.57}$ | $\underline{11.36}_{\pm0.59}$ |
| union-a | $11.45_{\pm0.48}$ | $27.37_{\pm2.46}$ | $28.28_{\pm1.02}$ | $11.45_{\pm0.48}$ | $11.45_{\pm0.48}$ | $\underline{10.67}_{\pm1.05}$ | $\underline{10.67}_{\pm1.05}$ | $\underline{10.67}_{\pm1.05}$ | $\underline{10.67}_{\pm1.05}$ | $\underline{10.67}_{\pm1.05}$ | $\underline{10.67}_{\pm1.05}$ |
| union-b | $11.53_{\pm0.57}$ | $15.46_{\pm0.17}$ | $17.36_{\pm0.64}$ | $\underline{11.06}_{\pm0.73}$ | $\underline{11.06}_{\pm0.73}$ | $\underline{11.06}_{\pm0.73}$ | $11.85_{\pm1.39}$ | $11.85_{\pm1.39}$ | $\underline{11.06}_{\pm0.73}$ | $11.85_{\pm1.39}$ | $11.85_{\pm1.39}$ |
| geo-w | $11.53_{\pm0.46}$ | $14.63_{\pm1.36}$ | $17.50_{\pm0.59}$ | $\underline{11.06}_{\pm0.83}$ | $\underline{11.06}_{\pm0.83}$ | $11.53_{\pm0.46}$ | $18.55_{\pm2.09}$ | $11.98_{\pm0.48}$ | $11.98_{\pm0.48}$ | $18.55_{\pm2.09}$ | $11.53_{\pm0.46}$ |
| geo-f | $11.15_{\pm0.44}$ | $\mathbf{12.98}_{\pm0.49}$ | $16.96_{\pm0.26}$ | $11.15_{\pm0.44}$ | $11.15_{\pm0.44}$ | $\underline{10.72}_{\pm0.55}$ | $15.95_{\pm2.60}$ | $14.40_{\pm1.72}$ | $\underline{10.72}_{\pm0.55}$ | $15.95_{\pm2.60}$ | $\underline{10.72}_{\pm0.55}$ |
| madl | $11.58_{\pm0.72}$ | $20.23_{\pm6.91}$ | $21.27_{\pm2.06}$ | $12.36_{\pm1.05}$ | $11.58_{\pm0.72}$ | $\underline{9.55}_{\pm1.64}$ | $\underline{9.55}_{\pm1.64}$ | $\underline{9.55}_{\pm1.64}$ | $\underline{9.55}_{\pm1.64}$ | $\underline{9.55}_{\pm1.64}$ | $\underline{9.55}_{\pm1.64}$ |
| crowd-ar | $11.87_{\pm0.80}$ | $14.64_{\pm0.79}$ | $17.14_{\pm0.47}$ | $\underline{11.00}_{\pm0.34}$ | $\underline{11.00}_{\pm0.34}$ | $\underline{11.00}_{\pm0.34}$ | $21.66_{\pm0.88}$ | $\underline{11.00}_{\pm0.34}$ | $\underline{11.00}_{\pm0.34}$ | $\underline{11.00}_{\pm0.34}$ | $\underline{11.00}_{\pm0.34}$ |
| annot-mix | $12.02_{\pm1.07}$ | $19.94_{\pm0.49}$ | $29.09_{\pm1.31}$ | $12.02_{\pm1.07}$ | $12.02_{\pm1.07}$ | $\underline{11.74}_{\pm0.61}$ | $\underline{11.74}_{\pm0.61}$ | $14.08_{\pm0.62}$ | $14.08_{\pm0.62}$ | $14.08_{\pm0.62}$ | $\underline{11.74}_{\pm0.61}$ |
| coin | $\mathbf{9.25}_{\pm0.44}$ | $19.82_{\pm0.77}$ | $21.23_{\pm1.46}$ | $\mathbf{9.53}_{\pm0.62}$ | $\mathbf{9.53}_{\pm0.62}$ | $\mathbf{9.25}_{\pm0.44}$ | $\mathbf{9.25}_{\pm0.44}$ | $\mathbf{9.25}_{\pm0.44}$ | $\mathbf{9.25}_{\pm0.44}$ | $\mathbf{9.25}_{\pm0.44}$ | $\mathbf{9.25}_{\pm0.44}$ |
| | | | | | | reuters-rand-v | | | | | |
| gt | $3.86_{\pm0.14}$ | $3.86_{\pm0.14}$ | $4.14_{\pm0.07}$ | $4.14_{\pm0.07}$ | $4.14_{\pm0.07}$ | N/A | N/A | N/A | N/A | N/A | $4.14_{\pm0.07}$ |
| mv | $14.11_{\pm0.47}$ | $24.86_{\pm0.75}$ | $26.32_{\pm0.80}$ | $\underline{14.11}_{\pm0.47}$ | $\underline{14.11}_{\pm0.47}$ | N/A | N/A | N/A | N/A | N/A | $\underline{14.11}_{\pm0.47}$ |
| ds | $13.63_{\pm0.59}$ | $22.15_{\pm0.98}$ | $22.66_{\pm0.61}$ | $\underline{13.63}_{\pm0.59}$ | $\underline{13.63}_{\pm0.59}$ | $17.25_{\pm1.13}$ | $17.25_{\pm1.13}$ | $17.25_{\pm1.13}$ | $17.25_{\pm1.13}$ | $17.25_{\pm1.13}$ | $17.25_{\pm1.13}$ |
| cl | $13.09_{\pm0.42}$ | $15.80_{\pm2.67}$ | $\mathbf{18.30}_{\pm0.73}$ | $\underline{13.09}_{\pm0.42}$ | $\underline{13.09}_{\pm0.42}$ | $15.22_{\pm1.79}$ | $18.84_{\pm3.45}$ | $14.53_{\pm1.43}$ | $15.22_{\pm1.79}$ | $18.84_{\pm3.45}$ | $15.22_{\pm1.79}$ |
| trace | $13.75_{\pm0.56}$ | $21.16_{\pm1.22}$ | $19.91_{\pm1.09}$ | $16.17_{\pm0.64}$ | $16.17_{\pm0.64}$ | $\underline{13.75}_{\pm0.56}$ | $16.34_{\pm0.93}$ | $\underline{13.75}_{\pm0.56}$ | $\underline{13.75}_{\pm0.56}$ | $16.34_{\pm0.93}$ | $\underline{13.75}_{\pm0.56}$ |
| conal | $13.72_{\pm1.02}$ | $16.28_{\pm0.98}$ | $20.40_{\pm0.50}$ | $\underline{13.72}_{\pm1.02}$ | $\underline{13.72}_{\pm1.02}$ | $\underline{13.72}_{\pm1.02}$ | $\underline{13.72}_{\pm1.02}$ | $\underline{13.72}_{\pm1.02}$ | $\underline{13.72}_{\pm1.02}$ | $\underline{13.72}_{\pm1.02}$ | $\underline{13.72}_{\pm1.02}$ |
| union-a | $15.41_{\pm1.42}$ | $20.84_{\pm2.62}$ | $24.98_{\pm1.02}$ | $15.41_{\pm1.42}$ | $15.41_{\pm1.42}$ | $12.67_{\pm1.02}$ | $\underline{11.99}_{\pm1.64}$ | $13.60_{\pm1.09}$ | $13.60_{\pm1.09}$ | $13.60_{\pm1.09}$ | $13.60_{\pm1.09}$ |
| union-b | $14.94_{\pm0.98}$ | $16.09_{\pm1.98}$ | $19.50_{\pm0.53}$ | $\underline{14.94}_{\pm0.98}$ | $\underline{14.94}_{\pm0.98}$ | $15.12_{\pm2.83}$ | $15.12_{\pm2.83}$ | $15.12_{\pm2.83}$ | $\underline{14.94}_{\pm0.98}$ | $19.61_{\pm1.40}$ | $15.12_{\pm2.83}$ |
| geo-w | $10.86_{\pm0.49}$ | $\mathbf{14.20}_{\pm0.66}$ | $18.90_{\pm0.45}$ | $14.85_{\pm1.04}$ | $14.85_{\pm1.04}$ | $\underline{10.86}_{\pm0.49}$ | $\underline{10.86}_{\pm0.49}$ | $12.39_{\pm1.01}$ | $12.57_{\pm1.86}$ | $12.57_{\pm1.86}$ | $12.57_{\pm1.86}$ |
| geo-f | $12.03_{\pm0.53}$ | $19.19_{\pm2.04}$ | $18.74_{\pm0.67}$ | $12.03_{\pm0.53}$ | $\mathbf{12.03}_{\pm0.53}$ | $\underline{10.78}_{\pm1.14}$ | $14.37_{\pm2.76}$ | $\underline{10.78}_{\pm1.14}$ | $\underline{10.78}_{\pm1.14}$ | $\underline{10.78}_{\pm1.14}$ | $\underline{10.78}_{\pm1.14}$ |
| madl | $13.01_{\pm0.48}$ | $20.61_{\pm10.2}$ | $22.02_{\pm2.45}$ | $14.99_{\pm0.97}$ | $14.99_{\pm0.97}$ | $13.46_{\pm2.34}$ | $13.46_{\pm2.34}$ | $13.46_{\pm2.34}$ | $\underline{13.01}_{\pm0.48}$ | $13.46_{\pm2.34}$ | $13.46_{\pm2.34}$ |
| crowd-ar | $12.34_{\pm0.70}$ | $15.43_{\pm0.56}$ | $19.88_{\pm0.57}$ | $\underline{14.16}_{\pm0.58}$ | $\underline{14.16}_{\pm0.58}$ | $\underline{14.16}_{\pm0.58}$ | $21.99_{\pm1.34}$ | $\underline{14.16}_{\pm0.58}$ | $\underline{14.16}_{\pm0.58}$ | $21.99_{\pm1.34}$ | $\underline{14.16}_{\pm0.58}$ |
| annot-mix | $13.59_{\pm1.05}$ | $21.78_{\pm5.18}$ | $27.51_{\pm0.75}$ | $\underline{15.26}_{\pm1.77}$ | $15.42_{\pm0.68}$ | $\underline{15.26}_{\pm1.77}$ | $\underline{15.26}_{\pm1.77}$ | $15.37_{\pm1.38}$ | $15.42_{\pm0.68}$ | $15.37_{\pm1.38}$ | $\underline{15.26}_{\pm1.77}$ |
| coin | $\mathbf{10.17}_{\pm1.32}$ | $34.36_{\pm3.17}$ | $24.04_{\pm1.06}$ | $\underline{10.17}_{\pm1.32}$ | $12.47_{\pm0.34}$ | $\underline{10.17}_{\pm1.32}$ | $\underline{10.17}_{\pm1.32}$ | $\underline{10.17}_{\pm1.32}$ | $\underline{10.17}_{\pm1.32}$ | $\underline{10.17}_{\pm1.32}$ | $\underline{10.17}_{\pm1.32}$ |
| | | | | | | reuters-full | | | | | |
| gt | $3.80_{\pm0.15}$ | $3.80_{\pm0.15}$ | $4.14_{\pm0.07}$ | $4.20_{\pm0.25}$ | $3.80_{\pm0.15}$ | N/A | N/A | N/A | N/A | N/A | $4.16_{\pm0.29}$ |
| mv | $16.71_{\pm0.52}$ | $22.84_{\pm0.41}$ | $24.32_{\pm0.23}$ | $\underline{13.96}_{\pm0.98}$ | $\underline{13.96}_{\pm0.98}$ | N/A | N/A | N/A | N/A | N/A | $\underline{13.96}_{\pm0.98}$ |
| ds | $11.64_{\pm0.33}$ | $17.78_{\pm0.42}$ | $19.88_{\pm0.92}$ | $\underline{11.64}_{\pm0.33}$ | $\underline{11.64}_{\pm0.33}$ | $\underline{11.64}_{\pm0.33}$ | $\underline{11.64}_{\pm0.33}$ | $\underline{11.64}_{\pm0.33}$ | $16.91_{\pm1.05}$ | $17.68_{\pm0.58}$ | $\underline{11.64}_{\pm0.33}$ |
| cl | $10.52_{\pm0.79}$ | $12.20_{\pm1.51}$ | $\mathbf{14.90}_{\pm0.39}$ | $\underline{10.52}_{\pm0.79}$ | $\underline{10.52}_{\pm0.79}$ | $12.54_{\pm2.09}$ | $16.93_{\pm0.33}$ | $12.54_{\pm2.09}$ | $\underline{10.52}_{\pm0.79}$ | $16.93_{\pm0.33}$ | $12.54_{\pm2.09}$ |
| trace | $11.50_{\pm0.64}$ | $16.80_{\pm0.77}$ | $16.61_{\pm0.61}$ | $\underline{11.50}_{\pm0.64}$ | $\underline{11.50}_{\pm0.64}$ | $\underline{11.50}_{\pm0.64}$ | $18.44_{\pm3.40}$ | $18.44_{\pm3.40}$ | $\underline{11.50}_{\pm0.64}$ | $18.44_{\pm3.40}$ | $\underline{11.50}_{\pm0.64}$ |
| conal | $11.53_{\pm0.74}$ | $16.32_{\pm0.82}$ | $17.07_{\pm1.02}$ | $11.46_{\pm0.76}$ | $\underline{11.26}_{\pm0.60}$ | $\underline{11.26}_{\pm0.60}$ | $11.53_{\pm0.74}$ | $11.53_{\pm0.74}$ | $11.53_{\pm0.74}$ | $11.53_{\pm0.74}$ | $11.53_{\pm0.74}$ |
| union-a | $15.53_{\pm13.4}$ | $20.36_{\pm3.28}$ | $23.36_{\pm0.37}$ | $11.08_{\pm0.64}$ | $15.53_{\pm13.4}$ | $15.53_{\pm13.4}$ | $15.53_{\pm13.4}$ | $\underline{10.95}_{\pm0.72}$ | $\underline{10.95}_{\pm0.72}$ | $\underline{10.95}_{\pm0.72}$ | $\underline{10.95}_{\pm0.72}$ |
| union-b | $12.13_{\pm0.56}$ | $13.34_{\pm0.42}$ | $15.79_{\pm0.25}$ | $12.13_{\pm0.56}$ | $12.13_{\pm0.56}$ | $11.61_{\pm0.50}$ | $17.15_{\pm1.94}$ | $14.90_{\pm0.72}$ | $\underline{10.81}_{\pm0.88}$ | $17.15_{\pm1.94}$ | $13.79_{\pm0.59}$ |
| geo-w | $12.11_{\pm0.40}$ | $11.42_{\pm0.43}$ | $15.10_{\pm0.28}$ | $12.11_{\pm0.40}$ | $12.11_{\pm0.40}$ | $10.47_{\pm0.56}$ | $\underline{10.24}_{\pm1.93}$ | $\underline{10.24}_{\pm1.93}$ | $12.01_{\pm0.68}$ | $\underline{10.24}_{\pm1.93}$ | $10.29_{\pm0.27}$ |
| geo-f | $10.22_{\pm0.31}$ | $\mathbf{9.53}_{\pm0.89}$ | $14.99_{\pm0.52}$ | $10.46_{\pm0.61}$ | $10.46_{\pm0.61}$ | $10.35_{\pm0.83}$ | $12.96_{\pm2.48}$ | $12.96_{\pm2.48}$ | $10.35_{\pm0.83}$ | $12.96_{\pm2.48}$ | $10.35_{\pm0.83}$ |
| madl | $\mathbf{9.45}_{\pm1.40}$ | $14.38_{\pm1.50}$ | $20.16_{\pm3.18}$ | $13.12_{\pm0.70}$ | $11.47_{\pm0.49}$ | $\mathbf{9.45}_{\pm1.40}$ | $\mathbf{9.45}_{\pm1.40}$ | $\mathbf{9.45}_{\pm1.40}$ | $\mathbf{9.45}_{\pm1.40}$ | $\mathbf{9.45}_{\pm1.40}$ | $\mathbf{9.45}_{\pm1.40}$ |
| crowd-ar | $11.78_{\pm0.30}$ | $14.55_{\pm0.37}$ | $16.11_{\pm0.31}$ | $\underline{11.78}_{\pm0.30}$ | $\underline{11.78}_{\pm0.30}$ | $\underline{11.78}_{\pm0.30}$ | $\underline{11.78}_{\pm0.30}$ | $\underline{11.78}_{\pm0.30}$ | $\underline{11.78}_{\pm0.30}$ | $22.59_{\pm1.06}$ | $\underline{11.78}_{\pm0.30}$ |
| annot-mix | $10.33_{\pm1.13}$ | $17.37_{\pm1.83}$ | $27.09_{\pm0.76}$ | $11.95_{\pm0.55}$ | $11.95_{\pm0.55}$ | $\underline{10.33}_{\pm1.13}$ | $\underline{10.33}_{\pm1.13}$ | $\underline{10.33}_{\pm1.13}$ | $\underline{10.33}_{\pm1.13}$ | $\underline{10.33}_{\pm1.13}$ | $\underline{10.33}_{\pm1.13}$ |
| coin | $10.11_{\pm1.01}$ | $28.18_{\pm3.04}$ | $20.50_{\pm0.83}$ | $\underline{10.11}_{\pm1.01}$ | $\underline{10.11}_{\pm1.01}$ | $\underline{10.11}_{\pm1.01}$ | $10.77_{\pm0.93}$ | $27.27_{\pm4.41}$ | $13.44_{\pm1.01}$ | $27.27_{\pm4.41}$ | $\underline{10.11}_{\pm1.01}$ |

Table 14: **Zero-one loss results [%] (part VIII).** Continued from the previous page.

| Approach | Baseline | | | Aggregation-level | | | | Crowd-level | | | Ensemble |
|---|---|---|---|---|---|---|---|---|---|---|---|
| | TRUE[*] | DEF-DATA[*] | DEF | AEU | AEC | ALU | ALC | CXU | CEC | CLC | ENS |
| **spc-worst-1** | | | | | | | | | | | |
| gt | $15.47_{\pm0.33}$ | $15.47_{\pm0.33}$ | $17.27_{\pm0.31}$ | $16.17_{\pm0.33}$ | $16.17_{\pm0.33}$ | N/A | N/A | N/A | N/A | N/A | $16.17_{\pm0.33}$ |
| mv | $53.87_{\pm0.43}$ | $\mathbf{51.47}_{\pm0.85}$ | $51.51_{\pm0.97}$ | $53.44_{\pm2.57}$ | $53.44_{\pm2.57}$ | N/A | N/A | N/A | N/A | N/A | $53.44_{\pm2.57}$ |
| ds | $53.87_{\pm0.43}$ | $\mathbf{51.47}_{\pm0.85}$ | $51.51_{\pm0.97}$ | $53.44_{\pm2.57}$ | $53.44_{\pm2.57}$ | $53.44_{\pm2.57}$ | $53.44_{\pm2.57}$ | $53.44_{\pm2.57}$ | $53.44_{\pm2.57}$ | $53.44_{\pm2.57}$ | $53.44_{\pm2.57}$ |
| cl | $32.29_{\pm27.6}$ | $51.72_{\pm1.08}$ | $52.56_{\pm0.77}$ | $32.29_{\pm27.6}$ | $32.29_{\pm27.6}$ | $32.29_{\pm27.6}$ | $32.29_{\pm27.6}$ | $32.29_{\pm27.6}$ | $32.29_{\pm27.6}$ | $32.29_{\pm27.6}$ | $32.29_{\pm27.6}$ |
| trace | $52.37_{\pm4.65}$ | $51.73_{\pm1.11}$ | $51.46_{\pm1.04}$ | $50.76_{\pm3.69}$ | $50.76_{\pm3.69}$ | $52.37_{\pm4.65}$ | $52.37_{\pm4.65}$ | $52.37_{\pm4.65}$ | $52.37_{\pm4.65}$ | $52.37_{\pm4.65}$ | $52.37_{\pm4.65}$ |
| conal | $53.18_{\pm1.43}$ | $52.03_{\pm1.22}$ | $51.82_{\pm0.75}$ | $52.03_{\pm1.26}$ | $52.03_{\pm1.26}$ | $52.03_{\pm1.26}$ | $52.03_{\pm1.26}$ | $52.03_{\pm1.26}$ | $52.03_{\pm1.26}$ | $52.03_{\pm1.26}$ | $52.03_{\pm1.26}$ |
| union-a | $39.57_{\pm26.2}$ | $52.34_{\pm0.60}$ | $52.69_{\pm0.51}$ | $81.50_{\pm0.56}$ | $81.50_{\pm0.56}$ | $39.57_{\pm26.2}$ | $39.57_{\pm26.2}$ | $39.57_{\pm26.2}$ | $39.57_{\pm26.2}$ | $39.57_{\pm26.2}$ | $39.57_{\pm26.2}$ |
| union-b | $49.98_{\pm0.00}$ | $51.67_{\pm1.18}$ | $51.96_{\pm0.94}$ | $49.98_{\pm0.00}$ | $49.98_{\pm0.00}$ | $49.98_{\pm0.00}$ | $49.98_{\pm0.00}$ | $49.98_{\pm0.00}$ | $49.98_{\pm0.00}$ | $49.98_{\pm0.00}$ | $49.98_{\pm0.00}$ |
| geo-w | $\mathbf{18.21}_{\pm0.44}$ | $51.99_{\pm1.09}$ | $52.23_{\pm0.79}$ | $\mathbf{18.00}_{\pm0.23}$ | $\mathbf{18.00}_{\pm0.23}$ | $\mathbf{18.00}_{\pm0.23}$ | $\mathbf{18.00}_{\pm0.23}$ | $\mathbf{18.00}_{\pm0.23}$ | $\mathbf{18.00}_{\pm0.23}$ | $\mathbf{18.00}_{\pm0.23}$ | $\mathbf{18.00}_{\pm0.23}$ |
| geo-f | $42.60_{\pm34.9}$ | $51.78_{\pm0.96}$ | $51.86_{\pm1.37}$ | $31.33_{\pm29.1}$ | $31.33_{\pm29.1}$ | $31.33_{\pm29.1}$ | $31.33_{\pm29.1}$ | $31.33_{\pm29.1}$ | $31.33_{\pm29.1}$ | $31.33_{\pm29.1}$ | $31.33_{\pm29.1}$ |
| madl | $62.25_{\pm27.0}$ | $52.03_{\pm0.92}$ | $\mathbf{47.86}_{\pm5.05}$ | $50.69_{\pm2.59}$ | $50.69_{\pm2.59}$ | $69.72_{\pm27.7}$ | $56.28_{\pm24.2}$ | $69.72_{\pm27.7}$ | $69.72_{\pm27.7}$ | $69.61_{\pm25.9}$ | $69.61_{\pm25.9}$ |
| crowd-ar | $52.92_{\pm2.63}$ | $52.25_{\pm0.77}$ | $51.21_{\pm0.47}$ | $52.33_{\pm0.98}$ | $52.33_{\pm0.98}$ | $50.10_{\pm0.29}$ | $50.10_{\pm0.29}$ | $52.33_{\pm0.98}$ | $52.33_{\pm0.98}$ | $52.33_{\pm0.98}$ | $50.10_{\pm0.29}$ |
| annot-mix | $44.33_{\pm12.6}$ | $53.09_{\pm2.93}$ | $50.13_{\pm1.27}$ | $42.61_{\pm26.1}$ | $42.61_{\pm26.1}$ | $31.50_{\pm16.9}$ | $31.50_{\pm16.9}$ | $40.36_{\pm16.6}$ | $40.36_{\pm16.6}$ | $31.50_{\pm16.9}$ | $42.61_{\pm26.1}$ |
| coin | $43.31_{\pm35.3}$ | $51.85_{\pm1.27}$ | $52.28_{\pm0.86}$ | $31.77_{\pm20.5}$ | $31.77_{\pm20.5}$ | $31.77_{\pm20.5}$ | $31.77_{\pm20.5}$ | $43.31_{\pm35.3}$ | $43.31_{\pm35.3}$ | $43.31_{\pm35.3}$ | $31.77_{\pm20.5}$ |
| **spc-worst-2** | | | | | | | | | | | |
| gt | $15.67_{\pm0.29}$ | $15.67_{\pm0.29}$ | $17.27_{\pm0.31}$ | $16.03_{\pm0.19}$ | $16.03_{\pm0.19}$ | N/A | N/A | N/A | N/A | N/A | $16.03_{\pm0.19}$ |
| mv | $28.93_{\pm0.64}$ | $30.45_{\pm1.00}$ | $38.55_{\pm0.33}$ | $28.93_{\pm0.64}$ | $28.93_{\pm0.64}$ | N/A | N/A | N/A | N/A | N/A | $28.93_{\pm0.64}$ |
| ds | $20.85_{\pm0.88}$ | $\mathbf{20.85}_{\pm0.88}$ | $28.28_{\pm0.47}$ | $19.63_{\pm0.59}$ | $19.63_{\pm0.59}$ | $19.63_{\pm0.59}$ | $19.63_{\pm0.59}$ | $19.63_{\pm0.59}$ | $19.63_{\pm0.59}$ | $19.63_{\pm0.59}$ | $19.63_{\pm0.59}$ |
| cl | $25.84_{\pm1.12}$ | $25.84_{\pm1.12}$ | $31.54_{\pm0.83}$ | $25.28_{\pm0.72}$ | $25.28_{\pm0.72}$ | $16.21_{\pm0.52}$ | $16.21_{\pm0.52}$ | $17.15_{\pm0.76}$ | $17.15_{\pm0.76}$ | $17.15_{\pm0.76}$ | $16.21_{\pm0.52}$ |
| trace | $25.75_{\pm0.76}$ | $28.72_{\pm1.39}$ | $35.13_{\pm1.28}$ | $25.75_{\pm0.76}$ | $28.95_{\pm1.11}$ | $19.36_{\pm0.33}$ | $19.36_{\pm0.33}$ | $19.49_{\pm1.06}$ | $19.49_{\pm1.06}$ | $19.49_{\pm1.06}$ | $19.49_{\pm1.06}$ |
| conal | $25.61_{\pm1.36}$ | $29.52_{\pm0.52}$ | $35.80_{\pm1.08}$ | $23.85_{\pm0.87}$ | $23.85_{\pm0.87}$ | $23.85_{\pm0.87}$ | $23.85_{\pm0.87}$ | $23.85_{\pm0.87}$ | $23.85_{\pm0.87}$ | $23.85_{\pm0.87}$ | $23.85_{\pm0.87}$ |
| union-a | $\mathbf{16.73}_{\pm1.15}$ | $22.49_{\pm0.34}$ | $30.14_{\pm0.71}$ | $16.73_{\pm1.15}$ | $16.73_{\pm1.15}$ | $16.73_{\pm1.15}$ | $16.73_{\pm1.15}$ | $16.73_{\pm1.15}$ | $16.73_{\pm1.15}$ | $16.73_{\pm1.15}$ | $16.73_{\pm1.15}$ |
| union-b | $23.25_{\pm1.13}$ | $26.03_{\pm1.28}$ | $34.04_{\pm0.53}$ | $20.95_{\pm4.76}$ | $20.95_{\pm4.76}$ | $20.95_{\pm4.76}$ | $20.95_{\pm4.76}$ | $20.95_{\pm4.76}$ | $20.95_{\pm4.76}$ | $20.95_{\pm4.76}$ | $20.95_{\pm4.76}$ |
| geo-w | $22.82_{\pm1.24}$ | $25.47_{\pm1.17}$ | $32.03_{\pm0.76}$ | $26.89_{\pm0.92}$ | $\mathbf{16.20}_{\pm0.78}$ | $\mathbf{16.20}_{\pm0.78}$ | $\mathbf{16.20}_{\pm0.78}$ | $\mathbf{16.20}_{\pm0.78}$ | $\mathbf{16.20}_{\pm0.78}$ | $\mathbf{16.20}_{\pm0.78}$ | $\mathbf{16.20}_{\pm0.78}$ |
| geo-f | $22.79_{\pm1.10}$ | $25.26_{\pm1.15}$ | $31.47_{\pm0.92}$ | $17.44_{\pm0.54}$ | $17.44_{\pm0.54}$ | $16.55_{\pm0.97}$ | $16.55_{\pm0.97}$ | $16.55_{\pm0.97}$ | $16.55_{\pm0.97}$ | $17.44_{\pm0.54}$ | $17.44_{\pm0.54}$ |
| madl | $21.78_{\pm1.76}$ | $28.74_{\pm6.19}$ | $28.61_{\pm12.2}$ | $16.20_{\pm0.23}$ | $16.20_{\pm0.23}$ | $16.98_{\pm0.51}$ | $17.16_{\pm0.74}$ | $18.10_{\pm0.47}$ | $18.10_{\pm0.47}$ | $18.10_{\pm0.47}$ | $18.10_{\pm0.47}$ |
| crowd-ar | $24.85_{\pm1.27}$ | $29.46_{\pm1.06}$ | $35.72_{\pm0.78}$ | $24.85_{\pm1.27}$ | $24.85_{\pm1.27}$ | $25.76_{\pm0.94}$ | $25.20_{\pm3.64}$ | $24.85_{\pm1.27}$ | $24.85_{\pm1.27}$ | $25.20_{\pm3.64}$ | $25.20_{\pm3.64}$ |
| annot-mix | $17.39_{\pm0.81}$ | $26.30_{\pm2.04}$ | $\mathbf{25.48}_{\pm0.86}$ | $16.94_{\pm0.40}$ | $16.94_{\pm0.40}$ | $18.46_{\pm0.26}$ | $16.70_{\pm0.68}$ | $16.94_{\pm0.40}$ | $16.94_{\pm0.40}$ | $16.70_{\pm0.68}$ | $16.94_{\pm0.40}$ |
| coin | $24.65_{\pm1.41}$ | $23.68_{\pm1.29}$ | $31.00_{\pm0.50}$ | $22.20_{\pm1.01}$ | $22.20_{\pm1.01}$ | $16.35_{\pm0.35}$ | $16.35_{\pm0.35}$ | $16.35_{\pm0.35}$ | $16.35_{\pm0.35}$ | $17.28_{\pm0.34}$ | $16.35_{\pm0.35}$ |
| **spc-worst-v** | | | | | | | | | | | |
| gt | $15.85_{\pm0.43}$ | $15.85_{\pm0.43}$ | $17.27_{\pm0.31}$ | $15.16_{\pm0.09}$ | $15.16_{\pm0.09}$ | N/A | N/A | N/A | N/A | N/A | $15.16_{\pm0.09}$ |
| mv | $20.44_{\pm0.70}$ | $22.86_{\pm0.28}$ | $27.91_{\pm0.72}$ | $18.50_{\pm0.53}$ | $18.50_{\pm0.53}$ | N/A | N/A | N/A | N/A | N/A | $18.50_{\pm0.53}$ |
| ds | $18.64_{\pm0.49}$ | $18.64_{\pm0.49}$ | $22.72_{\pm0.63}$ | $16.98_{\pm0.31}$ | $16.98_{\pm0.31}$ | $16.98_{\pm0.31}$ | $16.98_{\pm0.31}$ | $16.98_{\pm0.31}$ | $16.98_{\pm0.31}$ | $16.98_{\pm0.31}$ | $16.98_{\pm0.31}$ |
| cl | $18.09_{\pm0.28}$ | $18.09_{\pm0.28}$ | $21.50_{\pm0.86}$ | $16.54_{\pm0.36}$ | $16.54_{\pm0.36}$ | $16.13_{\pm0.51}$ | $16.13_{\pm0.51}$ | $16.13_{\pm0.51}$ | $16.54_{\pm0.36}$ | $16.22_{\pm0.26}$ | $16.13_{\pm0.51}$ |
| trace | $16.50_{\pm0.40}$ | $20.20_{\pm0.72}$ | $26.02_{\pm0.62}$ | $16.50_{\pm0.40}$ | $16.50_{\pm0.40}$ | $16.86_{\pm0.25}$ | $17.44_{\pm0.38}$ | $16.50_{\pm0.40}$ | $16.50_{\pm0.40}$ | $16.50_{\pm0.40}$ | $16.50_{\pm0.40}$ |
| conal | $17.36_{\pm0.60}$ | $20.30_{\pm0.74}$ | $25.43_{\pm0.54}$ | $16.86_{\pm0.57}$ | $16.86_{\pm0.57}$ | $16.86_{\pm0.57}$ | $16.86_{\pm0.57}$ | $17.36_{\pm0.60}$ | $17.36_{\pm0.60}$ | $17.36_{\pm0.60}$ | $16.86_{\pm0.57}$ |
| union-a | $16.27_{\pm0.47}$ | $18.06_{\pm0.25}$ | $20.43_{\pm0.61}$ | $16.13_{\pm0.49}$ | $\mathbf{16.13}_{\pm0.49}$ | $25.12_{\pm14.0}$ | $16.13_{\pm0.49}$ | $25.12_{\pm14.0}$ | $25.12_{\pm14.0}$ | $25.12_{\pm14.0}$ | $25.12_{\pm14.0}$ |
| union-b | $17.81_{\pm0.33}$ | $18.31_{\pm0.42}$ | $22.50_{\pm0.42}$ | $17.96_{\pm0.51}$ | $17.96_{\pm0.51}$ | $16.20_{\pm0.71}$ | $16.20_{\pm0.71}$ | $16.03_{\pm0.53}$ | $17.24_{\pm0.33}$ | $17.24_{\pm0.33}$ | $16.03_{\pm0.53}$ |
| geo-w | $17.72_{\pm0.26}$ | $18.01_{\pm0.18}$ | $21.33_{\pm0.55}$ | $17.72_{\pm0.26}$ | $17.72_{\pm0.26}$ | $\mathbf{15.74}_{\pm0.08}$ | $17.53_{\pm1.00}$ | $\mathbf{15.74}_{\pm0.08}$ | $17.78_{\pm0.38}$ | $17.78_{\pm0.38}$ | $\mathbf{15.74}_{\pm0.08}$ |
| geo-f | $17.69_{\pm0.31}$ | $18.03_{\pm0.28}$ | $21.22_{\pm0.37}$ | $17.69_{\pm0.31}$ | $17.69_{\pm0.31}$ | $15.78_{\pm0.60}$ | $\mathbf{15.78}_{\pm0.60}$ | $16.12_{\pm0.24}$ | $17.69_{\pm0.31}$ | $17.74_{\pm0.47}$ | $16.23_{\pm0.35}$ |
| madl | $18.10_{\pm1.46}$ | $18.59_{\pm1.49}$ | $\mathbf{18.15}_{\pm0.73}$ | $16.10_{\pm0.43}$ | $16.16_{\pm0.33}$ | $16.16_{\pm0.33}$ | $17.83_{\pm1.14}$ | $16.05_{\pm0.40}$ | $16.16_{\pm0.33}$ | $16.87_{\pm0.69}$ | $16.16_{\pm0.33}$ |
| crowd-ar | $17.82_{\pm0.85}$ | $20.38_{\pm1.53}$ | $25.34_{\pm0.27}$ | $17.56_{\pm0.47}$ | $17.56_{\pm0.47}$ | $17.56_{\pm0.47}$ | $19.56_{\pm1.46}$ | $17.56_{\pm0.47}$ | $17.56_{\pm0.47}$ | $17.56_{\pm0.47}$ | $17.56_{\pm0.47}$ |
| annot-mix | $\mathbf{15.96}_{\pm0.39}$ | $19.61_{\pm1.26}$ | $19.31_{\pm0.44}$ | $\mathbf{15.96}_{\pm0.39}$ | $16.87_{\pm0.72}$ | $15.96_{\pm0.39}$ | $16.26_{\pm0.33}$ | $15.96_{\pm0.39}$ | $16.87_{\pm0.72}$ | $\mathbf{15.96}_{\pm0.39}$ | $15.96_{\pm0.39}$ |
| coin | $16.83_{\pm0.69}$ | $\mathbf{17.58}_{\pm0.31}$ | $20.93_{\pm0.27}$ | $16.71_{\pm0.35}$ | $16.71_{\pm0.35}$ | $16.68_{\pm0.58}$ | $16.29_{\pm0.52}$ | $17.39_{\pm1.02}$ | $16.71_{\pm0.35}$ | $16.71_{\pm0.35}$ | $16.71_{\pm0.35}$ |
| **spc-rand-1** | | | | | | | | | | | |
| gt | $15.18_{\pm0.24}$ | $15.18_{\pm0.24}$ | $17.27_{\pm0.31}$ | $15.18_{\pm0.24}$ | $15.18_{\pm0.24}$ | N/A | N/A | N/A | N/A | N/A | $15.18_{\pm0.24}$ |
| mv | $16.21_{\pm0.33}$ | $16.21_{\pm0.33}$ | $22.89_{\pm0.41}$ | $16.07_{\pm0.61}$ | $16.07_{\pm0.61}$ | N/A | N/A | N/A | N/A | N/A | $16.07_{\pm0.61}$ |
| ds | $16.21_{\pm0.33}$ | $16.21_{\pm0.33}$ | $22.89_{\pm0.41}$ | $16.07_{\pm0.61}$ | $16.07_{\pm0.61}$ | $16.07_{\pm0.61}$ | $16.07_{\pm0.61}$ | $16.07_{\pm0.61}$ | $16.07_{\pm0.61}$ | $16.07_{\pm0.61}$ | $16.07_{\pm0.61}$ |
| cl | $\mathbf{15.58}_{\pm0.29}$ | $15.58_{\pm0.29}$ | $21.60_{\pm0.60}$ | $\mathbf{15.58}_{\pm0.29}$ | $\mathbf{15.58}_{\pm0.29}$ | $15.58_{\pm0.29}$ | $15.58_{\pm0.29}$ | $15.58_{\pm0.29}$ | $15.58_{\pm0.29}$ | $15.58_{\pm0.29}$ | $15.58_{\pm0.29}$ |
| trace | $18.88_{\pm0.53}$ | $16.21_{\pm0.44}$ | $23.12_{\pm0.49}$ | $16.15_{\pm0.67}$ | $16.15_{\pm0.67}$ | $16.15_{\pm0.67}$ | $16.15_{\pm0.67}$ | $16.15_{\pm0.67}$ | $16.15_{\pm0.67}$ | $16.15_{\pm0.67}$ | $16.15_{\pm0.67}$ |
| conal | $16.05_{\pm0.40}$ | $16.14_{\pm0.42}$ | $22.35_{\pm0.31}$ | $16.05_{\pm0.40}$ | $16.05_{\pm0.40}$ | $16.05_{\pm0.40}$ | $16.05_{\pm0.40}$ | $16.05_{\pm0.40}$ | $16.05_{\pm0.40}$ | $16.05_{\pm0.40}$ | $16.05_{\pm0.40}$ |
| union-a | $17.36_{\pm0.43}$ | $15.64_{\pm0.23}$ | $21.56_{\pm0.73}$ | $17.36_{\pm0.43}$ | $17.36_{\pm0.43}$ | $17.36_{\pm0.43}$ | $17.36_{\pm0.43}$ | $17.36_{\pm0.43}$ | $17.36_{\pm0.43}$ | $17.36_{\pm0.43}$ | $17.36_{\pm0.43}$ |
| union-b | $17.90_{\pm0.58}$ | $15.59_{\pm0.34}$ | $21.92_{\pm0.49}$ | $17.42_{\pm0.43}$ | $17.42_{\pm0.43}$ | $15.42_{\pm0.34}$ | $15.42_{\pm0.34}$ | $15.42_{\pm0.34}$ | $15.42_{\pm0.34}$ | $16.70_{\pm0.52}$ | $15.42_{\pm0.34}$ |
| geo-w | $17.97_{\pm0.65}$ | $15.57_{\pm0.28}$ | $21.84_{\pm0.82}$ | $17.40_{\pm0.44}$ | $17.40_{\pm0.44}$ | $17.40_{\pm0.44}$ | $17.40_{\pm0.44}$ | $\mathbf{15.40}_{\pm0.34}$ | $\mathbf{15.40}_{\pm0.34}$ | $\mathbf{15.40}_{\pm0.34}$ | $\mathbf{15.40}_{\pm0.34}$ |
| geo-f | $17.94_{\pm0.59}$ | $15.57_{\pm0.27}$ | $21.72_{\pm0.91}$ | $17.36_{\pm0.45}$ | $17.36_{\pm0.45}$ | $17.36_{\pm0.45}$ | $17.03_{\pm0.65}$ | $16.28_{\pm0.33}$ | $16.28_{\pm0.33}$ | $17.03_{\pm0.65}$ | $17.36_{\pm0.45}$ |
| madl | $16.67_{\pm0.59}$ | $16.49_{\pm0.73}$ | $\mathbf{19.01}_{\pm1.07}$ | $16.79_{\pm0.87}$ | $16.79_{\pm0.87}$ | $16.79_{\pm0.87}$ | $16.67_{\pm0.59}$ | $16.79_{\pm0.87}$ | $16.79_{\pm0.87}$ | $38.75_{\pm15.4}$ | $16.79_{\pm0.87}$ |
| crowd-ar | $16.32_{\pm0.52}$ | $16.20_{\pm0.33}$ | $22.92_{\pm0.57}$ | $16.21_{\pm0.36}$ | $16.21_{\pm0.36}$ | $16.32_{\pm0.52}$ | $16.32_{\pm0.52}$ | $16.21_{\pm0.36}$ | $16.21_{\pm0.36}$ | $16.32_{\pm0.52}$ | $16.32_{\pm0.52}$ |
| annot-mix | $16.37_{\pm0.81}$ | $16.16_{\pm0.39}$ | $21.43_{\pm0.48}$ | $16.63_{\pm0.91}$ | $16.63_{\pm0.91}$ | $16.63_{\pm0.91}$ | $16.63_{\pm0.91}$ | $16.36_{\pm0.61}$ | $16.36_{\pm0.61}$ | $16.36_{\pm0.61}$ | $16.63_{\pm0.91}$ |
| coin | $15.66_{\pm0.23}$ | $15.55_{\pm0.29}$ | $21.02_{\pm0.31}$ | $15.66_{\pm0.23}$ | $15.66_{\pm0.23}$ | $15.66_{\pm0.23}$ | $16.45_{\pm0.33}$ | $15.66_{\pm0.23}$ | $15.66_{\pm0.23}$ | $16.45_{\pm0.33}$ | $15.66_{\pm0.23}$ |

Continued on the next page.

Table 14: **Zero-one loss results [%] (part IX).** Continued from the previous page.

| Approach | Baseline | | | Aggregation-level | | | | Crowd-level | | | Ensemble |
|---|---|---|---|---|---|---|---|---|---|---|---|
| | TRUE[*] | DEF-DATA[*] | DEF | AEU | AEC | ALU | ALC | CXU | CEC | CLC | ENS |
| **spc-rand-2** | | | | | | | | | | | |
| gt | $15.19_{\pm0.22}$ | $15.19_{\pm0.22}$ | $17.27_{\pm0.31}$ | $15.93_{\pm0.21}$ | $15.19_{\pm0.22}$ | N/A | N/A | N/A | N/A | N/A | $15.93_{\pm0.21}$ |
| mv | $19.53_{\pm0.68}$ | $\underline{16.58}_{\pm0.58}$ | $22.82_{\pm0.40}$ | $\underline{16.58}_{\pm0.58}$ | $\underline{16.58}_{\pm0.58}$ | N/A | N/A | N/A | N/A | N/A | $\underline{16.58}_{\pm0.58}$ |
| ds | $15.87_{\pm0.16}$ | $15.87_{\pm0.16}$ | $20.28_{\pm0.50}$ | $\underline{15.78}_{\pm0.26}$ | $15.87_{\pm0.16}$ | $15.87_{\pm0.16}$ | $15.87_{\pm0.16}$ | $15.87_{\pm0.16}$ | $15.87_{\pm0.16}$ | $16.27_{\pm0.29}$ | $15.87_{\pm0.16}$ |
| cl | $17.02_{\pm0.66}$ | $\underline{15.69}_{\pm0.36}$ | $18.99_{\pm0.42}$ | $15.97_{\pm0.17}$ | $\underline{15.69}_{\pm0.36}$ | $15.75_{\pm0.14}$ | $15.75_{\pm0.14}$ | $\underline{15.69}_{\pm0.36}$ | $\underline{15.69}_{\pm0.36}$ | $16.46_{\pm0.34}$ | $\underline{15.69}_{\pm0.36}$ |
| trace | $\mathbf{15.21}_{\pm0.29}$ | $\underline{15.79}_{\pm0.23}$ | $20.26_{\pm0.72}$ | $16.22_{\pm0.45}$ | $16.22_{\pm0.45}$ | $16.05_{\pm0.39}$ | $16.05_{\pm0.39}$ | $16.22_{\pm0.45}$ | $16.22_{\pm0.45}$ | $16.22_{\pm0.45}$ | $16.22_{\pm0.45}$ |
| conal | $16.41_{\pm0.17}$ | $15.46_{\pm0.56}$ | $19.83_{\pm0.54}$ | $\mathbf{\underline{15.20}}_{\pm0.70}$ | $\mathbf{\underline{15.20}}_{\pm0.70}$ | $\mathbf{\underline{15.20}}_{\pm0.70}$ | $16.21_{\pm0.38}$ | $\mathbf{\underline{15.20}}_{\pm0.70}$ | $\mathbf{\underline{15.20}}_{\pm0.70}$ | $16.21_{\pm0.38}$ | $\mathbf{\underline{15.20}}_{\pm0.70}$ |
| union-a | $16.34_{\pm0.25}$ | $\underline{15.35}_{\pm0.09}$ | $18.70_{\pm0.18}$ | $16.34_{\pm0.25}$ | $16.34_{\pm0.25}$ | $16.34_{\pm0.25}$ | $18.00_{\pm0.50}$ | $16.65_{\pm0.12}$ | $16.65_{\pm0.12}$ | $18.00_{\pm0.50}$ | $16.34_{\pm0.25}$ |
| union-b | $16.39_{\pm0.35}$ | $\underline{15.65}_{\pm0.29}$ | $19.06_{\pm0.50}$ | $16.39_{\pm0.35}$ | $15.67_{\pm0.20}$ | $15.67_{\pm0.20}$ | $15.67_{\pm0.20}$ | $15.67_{\pm0.20}$ | $15.67_{\pm0.20}$ | $16.39_{\pm0.35}$ | $15.67_{\pm0.20}$ |
| geo-w | $15.65_{\pm0.28}$ | $15.60_{\pm0.32}$ | $18.63_{\pm0.56}$ | $\underline{15.46}_{\pm0.28}$ | $15.65_{\pm0.28}$ | $15.65_{\pm0.28}$ | $15.65_{\pm0.28}$ | $15.65_{\pm0.28}$ | $15.65_{\pm0.28}$ | $16.58_{\pm0.54}$ | $15.65_{\pm0.28}$ |
| geo-f | $15.62_{\pm0.27}$ | $\underline{15.61}_{\pm0.38}$ | $18.93_{\pm0.31}$ | $15.62_{\pm0.27}$ | $15.62_{\pm0.27}$ | $15.62_{\pm0.27}$ | $\mathbf{15.62}_{\pm0.27}$ | $15.62_{\pm0.27}$ | $15.62_{\pm0.27}$ | $\mathbf{15.62}_{\pm0.27}$ | $15.62_{\pm0.27}$ |
| madl | $17.87_{\pm2.96}$ | $\underline{15.85}_{\pm0.51}$ | $\mathbf{17.90}_{\pm0.62}$ | $17.43_{\pm0.52}$ | $17.43_{\pm0.52}$ | $17.52_{\pm0.67}$ | $17.52_{\pm0.67}$ | $16.59_{\pm0.71}$ | $16.59_{\pm0.71}$ | $16.59_{\pm0.71}$ | $16.59_{\pm0.71}$ |
| crowd-ar | $16.98_{\pm0.52}$ | $\underline{15.49}_{\pm0.59}$ | $19.81_{\pm0.33}$ | $16.62_{\pm0.51}$ | $16.06_{\pm0.22}$ | $16.48_{\pm0.71}$ | $16.48_{\pm0.71}$ | $16.48_{\pm0.71}$ | $16.48_{\pm0.71}$ | $16.48_{\pm0.71}$ | $16.48_{\pm0.71}$ |
| annot-mix | $16.10_{\pm0.54}$ | $\mathbf{15.21}_{\pm0.37}$ | $18.79_{\pm0.28}$ | $15.99_{\pm0.26}$ | $15.99_{\pm0.26}$ | $16.05_{\pm0.67}$ | $15.99_{\pm0.26}$ | $15.99_{\pm0.26}$ | $15.99_{\pm0.26}$ | $15.99_{\pm0.26}$ | $15.99_{\pm0.26}$ |
| coin | $15.77_{\pm0.30}$ | $\underline{15.63}_{\pm0.42}$ | $19.15_{\pm0.51}$ | $15.77_{\pm0.30}$ | $15.77_{\pm0.30}$ | $15.77_{\pm0.30}$ | $15.77_{\pm0.30}$ | $15.77_{\pm0.30}$ | $15.77_{\pm0.30}$ | $16.33_{\pm0.30}$ | $15.77_{\pm0.30}$ |
| **spc-rand-v** | | | | | | | | | | | |
| gt | $15.76_{\pm0.29}$ | $15.76_{\pm0.29}$ | $17.27_{\pm0.31}$ | $15.28_{\pm0.28}$ | $15.28_{\pm0.28}$ | N/A | N/A | N/A | N/A | N/A | $15.28_{\pm0.28}$ |
| mv | $16.69_{\pm0.73}$ | $\underline{16.32}_{\pm0.61}$ | $18.89_{\pm0.17}$ | $16.69_{\pm0.73}$ | $\underline{16.32}_{\pm0.61}$ | N/A | N/A | N/A | N/A | N/A | $16.69_{\pm0.73}$ |
| ds | $15.72_{\pm0.26}$ | $15.71_{\pm0.40}$ | $18.29_{\pm0.34}$ | $15.72_{\pm0.26}$ | $15.72_{\pm0.26}$ | $15.44_{\pm0.14}$ | $15.72_{\pm0.26}$ | $\mathbf{15.11}_{\pm0.26}$ | $\mathbf{\underline{14.89}}_{\pm0.43}$ | $\mathbf{\underline{14.89}}_{\pm0.43}$ | $15.44_{\pm0.14}$ |
| cl | $15.72_{\pm0.51}$ | $15.91_{\pm0.56}$ | $16.37_{\pm0.49}$ | $15.72_{\pm0.51}$ | $15.69_{\pm0.60}$ | $15.61_{\pm0.63}$ | $15.61_{\pm0.63}$ | $15.61_{\pm0.63}$ | $\underline{15.40}_{\pm0.31}$ | $15.54_{\pm0.31}$ | $15.69_{\pm0.60}$ |
| trace | $\mathbf{15.22}_{\pm0.41}$ | $16.31_{\pm0.81}$ | $18.34_{\pm0.38}$ | $\mathbf{\underline{15.22}}_{\pm0.41}$ | $\mathbf{\underline{15.22}}_{\pm0.41}$ | $\underline{15.22}_{\pm0.41}$ | $\mathbf{15.22}_{\pm0.41}$ | $\underline{15.22}_{\pm0.41}$ | $15.99_{\pm0.25}$ | $\underline{15.22}_{\pm0.41}$ | $\mathbf{15.22}_{\pm0.41}$ |
| conal | $16.47_{\pm0.41}$ | $\mathbf{15.63}_{\pm0.38}$ | $17.45_{\pm0.55}$ | $16.04_{\pm0.29}$ | $16.04_{\pm0.29}$ | $15.68_{\pm0.45}$ | $\underline{15.42}_{\pm0.43}$ | $16.04_{\pm0.29}$ | $16.04_{\pm0.29}$ | $16.04_{\pm0.29}$ | $16.04_{\pm0.29}$ |
| union-a | $15.57_{\pm0.32}$ | $15.83_{\pm0.08}$ | $16.43_{\pm0.52}$ | $\underline{15.57}_{\pm0.32}$ | $\underline{15.57}_{\pm0.32}$ | $\underline{15.57}_{\pm0.32}$ | $\underline{15.57}_{\pm0.32}$ | $\underline{15.57}_{\pm0.32}$ | $\underline{15.57}_{\pm0.32}$ | $\underline{15.57}_{\pm0.32}$ | $\underline{15.57}_{\pm0.32}$ |
| union-b | $15.30_{\pm0.41}$ | $15.90_{\pm0.59}$ | $16.48_{\pm0.65}$ | $\underline{15.30}_{\pm0.41}$ | $\underline{15.30}_{\pm0.41}$ | $15.58_{\pm0.38}$ | $\underline{15.30}_{\pm0.41}$ | $\underline{15.30}_{\pm0.41}$ | $\underline{15.30}_{\pm0.41}$ | $\underline{15.30}_{\pm0.41}$ | $\underline{15.30}_{\pm0.41}$ |
| geo-w | $15.78_{\pm0.53}$ | $15.80_{\pm0.59}$ | $16.41_{\pm0.41}$ | $15.61_{\pm0.37}$ | $15.61_{\pm0.37}$ | $\underline{15.32}_{\pm0.37}$ | $\underline{15.32}_{\pm0.37}$ | $\underline{15.32}_{\pm0.37}$ | $15.78_{\pm0.53}$ | $15.78_{\pm0.53}$ | $15.61_{\pm0.37}$ |
| geo-f | $15.76_{\pm0.57}$ | $15.80_{\pm0.59}$ | $\mathbf{16.14}_{\pm0.50}$ | $\underline{15.63}_{\pm0.24}$ | $\underline{15.63}_{\pm0.24}$ | $15.81_{\pm0.38}$ | $15.81_{\pm0.38}$ | $15.81_{\pm0.38}$ | $15.76_{\pm0.57}$ | $16.03_{\pm0.57}$ | $\underline{15.63}_{\pm0.24}$ |
| madl | $15.88_{\pm0.54}$ | $16.41_{\pm1.33}$ | $16.24_{\pm0.46}$ | $15.59_{\pm0.63}$ | $15.59_{\pm0.63}$ | $\mathbf{\underline{15.19}}_{\pm0.17}$ | $\mathbf{\underline{15.19}}_{\pm0.17}$ | $15.37_{\pm0.50}$ | $15.59_{\pm0.63}$ | $15.59_{\pm0.63}$ | $15.59_{\pm0.63}$ |
| crowd-ar | $16.74_{\pm0.41}$ | $\underline{15.94}_{\pm0.48}$ | $17.70_{\pm0.33}$ | $16.29_{\pm0.90}$ | $16.29_{\pm0.90}$ | $16.29_{\pm0.90}$ | $16.29_{\pm0.90}$ | $16.29_{\pm0.90}$ | $16.29_{\pm0.90}$ | $16.29_{\pm0.90}$ | $16.29_{\pm0.90}$ |
| annot-mix | $15.71_{\pm0.40}$ | $16.15_{\pm0.30}$ | $16.58_{\pm0.77}$ | $15.66_{\pm0.37}$ | $15.66_{\pm0.37}$ | $15.66_{\pm0.37}$ | $15.66_{\pm0.37}$ | $\underline{15.14}_{\pm0.23}$ | $15.52_{\pm0.23}$ | $15.52_{\pm0.23}$ | $\mathbf{15.14}_{\pm0.23}$ |
| coin | $15.67_{\pm0.30}$ | $15.80_{\pm0.48}$ | $16.15_{\pm0.25}$ | $15.67_{\pm0.30}$ | $15.67_{\pm0.30}$ | $\underline{15.54}_{\pm0.26}$ | $15.67_{\pm0.29}$ | $15.72_{\pm0.39}$ | $15.55_{\pm0.30}$ | $15.67_{\pm0.29}$ | $15.67_{\pm0.30}$ |
| **spc-full** | | | | | | | | | | | |
| gt | $15.24_{\pm0.18}$ | $15.24_{\pm0.18}$ | $17.27_{\pm0.31}$ | $15.09_{\pm0.20}$ | $15.09_{\pm0.20}$ | N/A | N/A | N/A | N/A | N/A | $15.09_{\pm0.20}$ |
| mv | $15.64_{\pm0.28}$ | $\underline{15.10}_{\pm0.48}$ | $17.93_{\pm0.51}$ | $15.24_{\pm0.30}$ | $15.24_{\pm0.30}$ | N/A | N/A | N/A | N/A | N/A | $15.24_{\pm0.30}$ |
| ds | $15.23_{\pm0.10}$ | $\underline{15.14}_{\pm0.33}$ | $16.80_{\pm0.50}$ | $15.23_{\pm0.10}$ | $15.23_{\pm0.10}$ | $15.23_{\pm0.10}$ | $15.23_{\pm0.10}$ | $15.23_{\pm0.10}$ | $15.26_{\pm0.29}$ | $15.26_{\pm0.29}$ | $15.23_{\pm0.10}$ |
| cl | $14.89_{\pm0.17}$ | $\underline{14.87}_{\pm0.43}$ | $15.24_{\pm0.36}$ | $14.89_{\pm0.17}$ | $14.89_{\pm0.17}$ | $15.33_{\pm0.31}$ | $15.33_{\pm0.36}$ | $15.33_{\pm0.31}$ | $\underline{14.87}_{\pm0.31}$ | $15.23_{\pm0.19}$ | $\underline{14.87}_{\pm0.31}$ |
| trace | $16.56_{\pm0.58}$ | $\underline{14.66}_{\pm0.11}$ | $16.74_{\pm0.38}$ | $\mathbf{14.73}_{\pm0.33}$ | $\mathbf{14.73}_{\pm0.33}$ | $\mathbf{14.73}_{\pm0.33}$ | $\mathbf{14.73}_{\pm0.33}$ | $\mathbf{14.73}_{\pm0.33}$ | $14.94_{\pm0.38}$ | $14.94_{\pm0.38}$ | $14.94_{\pm0.38}$ |
| conal | $15.60_{\pm0.43}$ | $\underline{14.79}_{\pm0.29}$ | $16.81_{\pm0.24}$ | $15.60_{\pm0.43}$ | $15.60_{\pm0.43}$ | $14.85_{\pm0.39}$ | $15.60_{\pm0.43}$ | $15.60_{\pm0.43}$ | $15.60_{\pm0.43}$ | $14.85_{\pm0.39}$ | $14.85_{\pm0.39}$ |
| union-a | $15.70_{\pm0.32}$ | $\underline{14.79}_{\pm0.37}$ | $15.30_{\pm0.23}$ | $15.63_{\pm0.27}$ | $15.63_{\pm0.27}$ | $15.63_{\pm0.27}$ | $14.93_{\pm0.29}$ | $15.33_{\pm0.47}$ | $15.33_{\pm0.47}$ | $15.63_{\pm0.27}$ | $15.63_{\pm0.27}$ |
| union-b | $15.33_{\pm0.52}$ | $14.82_{\pm0.47}$ | $\mathbf{15.11}_{\pm0.33}$ | $15.28_{\pm0.42}$ | $15.28_{\pm0.42}$ | $\underline{14.74}_{\pm0.13}$ | $15.28_{\pm0.42}$ | $15.28_{\pm0.42}$ | $15.28_{\pm0.42}$ | $15.17_{\pm0.33}$ | $15.28_{\pm0.42}$ |
| geo-w | $15.30_{\pm0.56}$ | $\underline{14.77}_{\pm0.49}$ | $15.34_{\pm0.38}$ | $15.30_{\pm0.56}$ | $15.30_{\pm0.56}$ | $15.11_{\pm0.47}$ | $14.97_{\pm0.35}$ | $15.21_{\pm0.41}$ | $14.97_{\pm0.35}$ | $15.15_{\pm0.35}$ | $14.97_{\pm0.35}$ |
| geo-f | $15.32_{\pm0.64}$ | $\underline{14.86}_{\pm0.40}$ | $15.26_{\pm0.21}$ | $\underline{14.86}_{\pm0.49}$ | $\underline{14.86}_{\pm0.49}$ | $\underline{14.86}_{\pm0.49}$ | $\underline{14.86}_{\pm0.49}$ | $15.15_{\pm0.75}$ | $\mathbf{\underline{14.86}}_{\pm0.49}$ | $15.43_{\pm0.24}$ | $\underline{14.86}_{\pm0.49}$ |
| madl | $15.53_{\pm0.79}$ | $15.09_{\pm0.52}$ | $15.71_{\pm0.60}$ | $15.53_{\pm0.79}$ | $15.53_{\pm0.79}$ | $15.07_{\pm0.31}$ | $15.53_{\pm0.79}$ | $15.08_{\pm0.37}$ | $15.20_{\pm0.60}$ | $\mathbf{\underline{14.57}}_{\pm0.34}$ | $15.53_{\pm0.79}$ |
| crowd-ar | $16.60_{\pm3.12}$ | $\underline{14.68}_{\pm0.31}$ | $16.28_{\pm0.39}$ | $15.06_{\pm0.53}$ | $15.06_{\pm0.53}$ | $15.06_{\pm0.53}$ | $15.30_{\pm0.40}$ | $16.20_{\pm0.40}$ | $15.41_{\pm0.43}$ | $15.30_{\pm0.40}$ | $16.20_{\pm0.40}$ |
| annot-mix | $\mathbf{14.73}_{\pm0.25}$ | $\mathbf{\underline{14.25}}_{\pm0.35}$ | $15.83_{\pm0.20}$ | $\mathbf{14.73}_{\pm0.25}$ | $\mathbf{14.73}_{\pm0.25}$ | $\mathbf{14.73}_{\pm0.25}$ | $\mathbf{14.73}_{\pm0.25}$ | $\mathbf{14.73}_{\pm0.25}$ | $15.66_{\pm0.69}$ | $15.66_{\pm0.69}$ | $\mathbf{14.73}_{\pm0.25}$ |
| coin | $14.99_{\pm0.30}$ | $\underline{14.75}_{\pm0.42}$ | $15.35_{\pm0.39}$ | $14.99_{\pm0.30}$ | $14.99_{\pm0.30}$ | $14.99_{\pm0.30}$ | $14.99_{\pm0.30}$ | $15.25_{\pm0.73}$ | $15.12_{\pm0.31}$ | $15.12_{\pm0.31}$ | $14.99_{\pm0.30}$ |

