# OpenReview forum: "crowd-hpo: Realistic Hyperparameter Optimization and Benchmarking for Learning from Crowds with Noisy Labels"
_TMLR — Accepted by TMLR_

### Review · Reviewer_q474 · 2025-08-14

**Summary Of Contributions:**

Hyper-parameter optimization is a central challenge in label noise learning due to the lack of clean validation data subsets. However, this issue has been overlooked by the research community in noisy label learning and hence, not been properly studied. The paper proposes *crowd-hpo* as a framework to evaluate different label noise learning methods, while selecting hyper-parameters following certain criteria on noisy validation sets. Empirical demonstrations have shown that selecting hyper-parameters based on these criteria improves the performnace of many state-of-the-art methods in label noise learning.

**Audience:**

Yes

**Audience Explanation:**

The most valuable contribution of the paper is to study the hyper-parameter optimization in label noise learning. In standard machine learning, such a hyper-parameter optimization relies on a validation set consisting of samples with their corresponding ground truth labels. However, the setting in noisy label learning often assumes the absence of such a validation set. Instead, only noisy label validation set is available. This poses a central challenge in label noise learning as most current methods often fine-tune their hyper-parameters directly on testing sets, resulting in bias reported results. Hence, the motivation of this study is worthy and significant.

**Claims And Evidence:**

No

**Claims Explanation:**

Despite the significance in the introduction and related work about hyper-parameter optimization, the methodology presented in section 3 is too vague and unclear in terms of addressing the challenge. In addition, as the paper presents several formulation, the notations are abused, causing more confusion than clarification, and hence, significantly reduces the readability of the paper. After going through the whole section 3, I cannot understand the main point presented there personally. The main idea presented in Figure 4 is unclear, especially for each component.
- It is hard to understand the meaning of Eqs. (11) and (13). To what I can understand, Eq. (11) calculate the risk on the validation set given some label aggregation and weighting mechanism. However, it is unclear what exactly those aggregation and weighting mechanisms are. Is this only an illustration that the authors want to use to demonstrate the risk calculated on the validation set? If this is the case, the introduction of these two things cause confusion.
- Eq. (13) is to present the risk of the model $g$ on the validation set. It is unclear why this risk is taken into account because according to Eq. (4), $g$ is used to model the prediction of each annotator, while the target in noisy label learning is to have the highest performance when predicting ground truth labels. Do the authors mean to enforce consistency to match with the annotations made by humans? This case is only helpful if $g$ and $f$ are independent, but it has not been specified in the paper.
- The presentation about *Crowdworker Performance* is unclear. Eqs. (15) and (16) are fine, but the justification for Eq. (17) is vague. Could the authors elaborate more why predicting $\Pr(y | x)$ will result in *double-dipping*. To what I understand, it happens when we use the same samples for training and validation. However, here, the authors already slit the dataset into two disjoint subsets.
    - The proof for Proposition 1 should be revised significantly. In its current form, it is unclear why one can go from $l\_{max}$ and $l\_{nk}$ to $\Pr(y = e\_{c} | x)$. It would be helpful if the authors could provide detailed explanation of each step.
    - In Eq. (18), it is unclear why there is a division by $C - 1$. Due to that division, it is no long a Bernoulli distribution.
    - Similar to the above two points with Eq. (19).
- Nevertheless, it is unclear about the estimation of posterior of the ground truth label given the annotations in (15) and (19).
- For *Label Weights*, it is unclear why there are only two ways to weight samples: uniform and confidence. Could the authors elaborate further more about this?
- For the *Hyper-parameter Selection Criteria*, could the authors elaborate further on how to select hyper-parameter based on such criteria? Does it reply on gradient based methods or the the classic Borda count as mentioned in the paper?

Due to these concerns, I have not proceeded further but stopped right before section 4. I believe that it should be clear in the methodology before moving to the empirical demonstration.

**Requested Changes:**

It seems that the paper contains too much of information without giving a big picture to explain what it is going to do. I would like to request the authors to elaborate, add discussion and explanation for each step in section 3, so that reader can understand why we need to go through such formulations and why such formulations make sense.

---
### **Minors**
- The abbreviation "HP" is often used in the paper, but it has never been defined.
- Figure 2 could be improved further to increase the readability of the paper. In its current form, it is too thin but long. In addition, it is helpful if the model parameter, such as $\theta$, is included into the graphical model to describe the data generation process more clearly.
- In section 3.1 - paragraph Data Generation Process, the terminology "multiset" is used without being properly defined. A set is defined to contain multiple elements, but it is unclear why in such a context, it is called "multiset".
- The paper considers the case of missing labels at the early part of section 3, but later on, discard that case. Hence, I suggest to specify that explicitly at the beginning, saying that only the case with annotations would be considered. That reduces the confusion because of the graphical model presented in Figure 2. If missing labels are considered, then the random variable $z$ is observed for some samples, while being hidden for others.
- The definition of 0-1 loss in Eq. (2) is inaccurate. As $\mathbf{e}\_{c}$ and $\mathbf{y}$ are defined as one-hot vectors, their dot product $\operatorname*{argmax}\_{\mathbf{e}\_{c}} \mathbf{e}\_{c}^{\top} \mathbf{y}$ is eventually $\mathbf{y}$. Why would one go through a complicated definition without writing that: $L\_{0-1}(\mathbf{y}, \hat{\mathbf{y}}) = 1 - \mathbf{y}^{\top} \hat{\mathbf{y}}$
- Although the authors mention the abuse of notation in footnote 2 on page 5, using the same notation $\theta$ for the functions f, g and h causes confusion. Please update the manuscript with a clearer notation.

---

> ### Author Response · Authors · 2025-08-23
> **Rebuttal: Part 1**
>
> **Summary:** Following your suggestions, we revised Section 3 by updating the wording and adding more detailed explanations. Changes are $\color{blue}blue$.
>
> Foremost, many thanks for your detailed and helpful criticism. Moreover, sorry for any confusion caused by our writing in Section 3. We hope to resolve each issue by briefly explaining and summarizing the associated changes marked in $\color{blue}blue$ in the revised paper. Finally, we look forward to receiving your feedback regarding the rest of the paper once the methodology explanation is straightforward.
>
> > After going through the whole section 3, I cannot understand the main point presented there personally. The main idea presented in Figure 4 is unclear, especially for each component.
>
> **Explanation:**
> - Because we only have crowd-labeled validation data, the empirical risk for validation instances with true labels cannot be computed. We therefore define several HPS criteria, each built on a distinct empirical risk measure that reflects specific design choices and assumptions about the crowd. We do not know which of these design choices and assumptions actually perform best for a concrete combination of an LFC approach and a real-world dataset.
> - Figure 4 organizes these criteria as a tree. Along the path within a tree, we specify the risk template, crowdworker performance model, and set the label weighting scheme. Each leaf is a concrete HPS criterion that evaluates an LFC approach on a held-out validation set and selects the HPC with the smallest estimated risk according to Eq. (8). For example, AEU instantiates the aggregation-level template, assumes Equal crowdworker performances, and uses Uniform label weights.
>
> **Revision:** We revised Figure 4 and inserted a comparable explanation when introducing Figure 4 in Section 3.4 for the first time.
>
> > It is hard to understand the meaning of Eqs. (11) and (13). To what I can understand, Eq. (11) calculate the risk on the validation set given some label aggregation and weighting mechanism. However, it is unclear what exactly those aggregation and weighting mechanisms are. Is this only an illustration that the authors want to use to demonstrate the risk calculated on the validation set? If this is the case, the introduction of these two things cause confusion.
>
> **Explanation:**
> - We introduce the HPS criteria in a top-down manner, starting with the empirical-risk template. In this view, Eqs. (11) and (13) are just templates: Eq. (11) computes an aggregation-level risk that uses a single proxy label per instance, whereas Eq. (13) computes a crowd-level risk that uses the set of crowd labels per instance. Which template is instantiated and how is determined by the design choices along the path to a criterion’s leaf.
> - For example, AEU is an aggregation-level criterion and therefore instantiates Eq. (11). It first fixes the aggregation function $\boldsymbol{\overline{z}}$ via Eq. (22), which requires the posterior class label probabilities $\widehat{\Pr}(\boldsymbol{y}_n = \boldsymbol{e}_c \mid \boldsymbol{x}_n, \mathcal{Z}_n)$  from Eq. (19). These posteriors depend on crowdworker-performance estimates, of which two implementations are given in Eqs. (20) and (21). Because AEU assumes equal performances, it uses Eq. (20) inside Eq. (19), yielding the aggregated proxy label in Eq. (22). AEU also assumes uniform label weighting, such that the label weighting function $w$ is defined through Eq. (23). Substituting these concrete definitions of $\boldsymbol{\overline{z}}$ and $w$ into the template of Eq. (11) gives the AEU empirical risk. Given an LFC approach, we compute this risk for each HPC in $\Lambda$ and pick the HPC $\boldsymbol{\hat{\lambda}}$ with the minimal risk estimate, as in Eq. (8).
>
> **Revision:** We inserted the first part of the explanation when talking about the empirical risk estimation template in Section 3.4, while the second part is an example added (in reduced form) to the HPS criteria paragraph in Section 3.4.

---

> ### Author Response · Authors · 2025-08-23
> **Rebuttal: Part 2**
>
> > Eq. (13) is to present the risk of the model $\boldsymbol{g}$ on the validation set. It is unclear why this risk is taken into account because according to Eq. (4), $\boldsymbol{g}$ is used to model the prediction of each annotator, while the target in noisy label learning is to have the highest performance when predicting ground truth labels. Do the authors mean to enforce consistency to match with the annotations made by humans? This case is only helpful if $\boldsymbol{g}$ and $\boldsymbol{f}$ are independent, but it has not been specified in the paper.
>
> **Explanation:**
> - Your observation regarding the target mismatch between the data classification model $\boldsymbol{f}$ and the crowdworker classification $\boldsymbol{g}$ is correct. The introduction of the crowd-level risk is motivated by the loss function of many LFC approaches. In fact, these approaches do not directly optimize the loss between the outputs of the data classification model $\boldsymbol{f}$ and estimated ground truth labels. Instead, the outputs of the crowdworker classification model $\boldsymbol{g}$ are optimized to match the crowdworkers’ noisy class labels. To ensure that the data classification model $\boldsymbol{f}$ still learns to predict ground truth class labels, the crowdworker classifier is typically designed as a concatenation of the instance classification model with crowdworker-specific heads.
> - The most common implementation is to introduce a confusion matrix for each crowdworker, which is then combined with the probabilities estimated by the data classification model $\boldsymbol{f}$ to obtain a distribution of the potential crowdworker’s class label assignment (see Eq. (32)). To keep $\boldsymbol{f}$ oriented toward the ground-truth signal rather than overfitting crowdworkers’ label noise, LFC methods approaches impose inductive biases on confusion matrices. For example, the confusion matrices are typically initialized at (or near) identity and regularized so that deviations from identity must be justified by data.
> - While Eq. (13) seems to be a rather unintuitive surrogate for evaluating the data classification model, it has the benefit that we do not need to estimate true class labels via an aggregation function for the cost of not evaluating on the target we actually care about.
>
> **Revision:** We added a small statement in Section 3.4 to show the downfalls of both risk templates, therefore motivating the usage of both.
>
> > The presentation about Crowdworker Performance is unclear. Eqs. (15) and (16) are fine, but the justification for Eq. (17) is vague. Could the authors elaborate more why predicting $\Pr(\boldsymbol{y} | \boldsymbol{x})$ will result in double-dipping. To what I understand, it happens when we use the same samples for training and validation. However, here, the authors already slit the dataset into two disjoint subsets.
>
> **Explanation**:
> - We use the term *double-dipping* in a generalized sense to describe mixing up model training and validation. As we cannot observe the ground truth values $\boldsymbol{y}_n$, we can only form an estimate from the noisy crowd labels $\mathcal{Z}_n$ by aggregation. To do this aggregation, we resort to a maximum a posteriori (MAP) estimate. If we were to now use the same probability model for making this MAP estimate, which we used for training, we would, to put it pointedly, evaluate a model on targets biased by itself and therefore create a form of *double-dipping*. This also applies despite having separate training and validation instances. To overcome this problem, we need to model the prior $\widehat{\Pr}\left(\boldsymbol{y}_n=\boldsymbol{e}_c \mid \boldsymbol{x}_n\right)$ and likelihood for our MAP estimate carefully. As shown in Proposition 1, a sufficiently biased model results in an MAP estimate, which equals the prior estimate, i.e., the model's prediction. For this reason, we choose the typical uniform prior to make it uninformative.
>
> **Revision:** We made it clearer in Section 3.4 that an optimistic risk estimate can arise despite having strictly separate training and validation instances. If the term “double-dipping” is confusing in this context, we can also remove it.
>
> > The proof for Proposition 1 should be revised significantly. In its current form, it is unclear why one can go from $l_{\mathrm{max}}$ and $l_{nk}$ to $\Pr(\boldsymbol{y} = \boldsymbol{e}_c | \boldsymbol{x})$. It would be helpful if the authors could provide detailed explanation of each step.
>
> **Revision:** Thanks for reviewing in such detail and pointing out that this proof, in its current form, is unclear. Therefore, we revised the proof significantly. First, to make the proof clearer, we split up the previous chain of inequalities into separate arguments. Second, every step now follows from a basic argument. We further added a textual justification for each step, which does not include algebraic manipulations.

---

> > ### Author Response · Authors · 2025-08-23
> > **Rebuttal: Part 3**
> >
> > > In Eq. (18), it is unclear why there is a division by $C-1$. Due to that division, it is no long a Bernoulli distribution. Similar to the above two points with Eq. (19).
> >
> > **Explanation:**
> > - We resort to a Bernoulli model (not distribution) in the sense that we use our model to estimate the probability of whether a crowdworker is correct in their prediction for instance $\boldsymbol{x}_n$ or not, i.e., we predict a probability distribution over $\boldsymbol{z}\_{nm}^T \boldsymbol{y}\_n$, which takes values in $\\{0, 1\\}$.
> > - However, for producing our MAP estimate, we need to model the distribution over the outputs of $\boldsymbol{z}\_{nm}$ given that $\boldsymbol{y}\_n = \boldsymbol{e}\_c$. Therefore, we extend our predicted probability distribution over the outputs $\boldsymbol{z}\_{nm}^T \boldsymbol{y}\_n$ to a distribution over the outputs of $\boldsymbol{z}\_{nm}$ given that $\boldsymbol{y}\_n = \boldsymbol{e}\_c$ by defining $\widehat{\Pr}(\boldsymbol{z}\_{nm} = \boldsymbol{e}\_k \mid \boldsymbol{x}\_n, \boldsymbol{y}\_n = \boldsymbol{e}\_c)$ as in Eq. (19). As there are $C-1$ classes in $ \Omega_Y\setminus\\{\boldsymbol{e}\_k\\}$ for which $\boldsymbol{e}\_k^\mathrm{T}\boldsymbol{e}\_c = 0$, we get a well-defined probability distribution.
> >
> > **Revision:** To prevent this confusion, we deviate from using the term Bernoulli model.
> >
> > > Nevertheless, it is unclear about the estimation of the posterior of the ground truth label given the annotations in (15) and (19).
> >
> > **Explanation:**  We need to carefully model the posterior by defining prior and likelihood to prevent double-dipping, but at the same time, use the learned knowledge about the individual crowdworkers' performances. We do this by using an uninformative prior and a restricted likelihood, which is defined via the crowdworkers’ performance estimate (scalar).
> >
> > **Revision:** We hope the additional explanations for the individual modeling steps now make the process of coming from Eq. (15) to Eq. (19) more clear.
> >
> > > For Label Weights, it is unclear why there are only two ways to weight samples: uniform and confidence. Could the authors elaborate further more about this?
> >
> > **Explanation:** We chose uniform and confidence label weighting because they are simple baselines and avoid introducing additional design choices into HPS. However, you are absolutely right that there could be other choices, e.g., filtering instances or scaling the confidences.
> >
> > **Revision:** In Section 3.4, we introduce Figure 4 as a non-exhaustive tree of design choices that can be made to define surrogate empirical risk measures. Moreover, we explicitly state the above motivation for the two label weightings in the corresponding paragraph of Section 3.4.
> >
> > > For the Hyper-parameter Selection Criteria, could the authors elaborate further on how to select hyper-parameter based on such criteria? Does it reply on gradient based methods or the the classic Borda count as mentioned in the paper?
> >
> > **Explanation:**
> > - Each HPS criterion is defined in Eq. (8) as a rule for selecting the HPC with the minimum risk. Accordingly, HPS criteria differ only in their definition of the risk measure $R\_{L,\mathcal{S}, \mathcal{I}}$.
> > - Such a risk measure assigns the HPC $\boldsymbol{\lambda}$ for an LFC approach $\boldsymbol{A}\_{\boldsymbol{\lambda}}$ a single scalar $R\_{L,\mathcal{S}, \mathcal{I}}(\boldsymbol{A}_{\boldsymbol{\lambda}}).$
> > - To perform an actual selection, we need to measure the risk for each candidate HPC in $\Lambda$.
> > - Once we have these scalar values, an HPS criterion outputs the HPC with the minimum risk measurement.
> > - The definition of the risk measure for DEF is just a workaround to formalize that the outputted HPC equals the default HPC $\boldsymbol{\lambda}\_{DEF}$.
> > - In contrast, the criterion TRUE and all criteria in Figure 4 involve actual validation via $K$-fold cross-validation. They differ on how the respective risk measures are defined. Almost all of these measures return a kind of average zero-one loss.
> > - The only exception is the criterion ENS, which returns the rank sum (see Eq. (29)) of an HPC determined by ranking this HPC using different risk measures. For example, let us assume we have $J=3$ risk measures and $|\Lambda| = 5$ candidate HPCs. Then, we evaluate the $3$ risk measures for each of these $5$ HPCs. Subsequently, we can rank the $5$ candiate HPCs for each of these $3$ risk measures (see Eq. (28)). Finally, the criterion ENS outputs the HPC with the minimum rank sum (plug Eq. (29) into Eq. (8)).
> >
> > **Revision:** Within Section 3.3 + 3.4 and in Figure 4, we emphasized Eq. (8) as the starting point for defining and evaluating an HPS criterion. Further, we made it clearer that the risk measure in Eq. (8) is just a placeholder whose concrete definition depends on the respective criterion.
> >
> > > The abbreviation HP is often used in the paper, but it has never been defined.
> >
> > **Revision:** We have now introduced HP (hyperparameter) on its first occurrence

---

> ### Author Response · Authors · 2025-08-23
> **Rebuttal: Part 4**
>
> > Figure 2 could be improved further to increase the readability of the paper. In its current form, it is too thin but long. In addition, it is helpful if the model parameter, such as $\boldsymbol{\theta}$, is included into the graphical model to describe the data generation process more clearly.
>
> **Explanation**:
> - In principle, adding $\boldsymbol{\theta}$ to the graphical model in Figure 2 is possible by linking it to $\boldsymbol{y}\_n$ and $\boldsymbol{z}\_{nm}$.
> - However, we would like to keep the data generation and modeling process separate from each other because some LFC approaches make simplified assumptions about this data generation process (see Appendix A).
>
> **Revision:** We reduced the width of Figure 2 and increased its height.
>
> > In section 3.1 - paragraph Data Generation Process, the terminology "multiset" is used without being properly defined. A set is defined to contain multiple elements, but it is unclear why in such a context, it is called "multiset".
>
> **Explanation**: The term multiset means a set that can contain multiple elements of the same kind, e.g., the multiset $\\{a, a, b\\}$ is different from $\\{a, b\\}$. This is relevant to properly defining the set of target values.
>
> **Revision:** We added a footnote giving a brief explanation of the term multiset. If you still consider the usage of the term multiset to be confusing, we would consider simply using the term set instead, though we consider it to be a slight misuse of the word.
>
> > The paper considers the case of missing labels at the early part of section 3, but later on, discard that case. Hence, I suggest to specify that explicitly at the beginning, saying that only the case with annotations would be considered. That reduces the confusion because of the graphical model presented in Figure 2. If missing labels are considered, then the random variable $\boldsymbol{z}$ is observed for some samples, while being hidden for others.
>
> **Explanation**:
> - Thanks for pointing this out, because it seems that we have not sufficiently made clear that none of the evaluated LFC approaches and the HPS criteria assume that all crowdworkers must label each instance.
> - The $\boldsymbol{z}\_{nm}$ must not be observed for each pair $(n, m) \in [N] \times [M]$. Our evaluation explicitly covers the extreme scenario where only a single noisy class label is observed per instance.
> - However, we do not consider the scenario where some instances have no assigned label at all. Moreover, the considered LFC approaches learn only from the observed noisy class labels.
>
> **Revision:**
> - We revised the plate notation in the probabilistic model of Figure 2 to explicitly indicate that not all noisy class labels are observed.
> - We adjusted the wording at the beginning of Section 3.3 when introducing $\mathcal{D}$.
>
> > The definition of 0-1 loss in Eq. (2) is inaccurate. As $\boldsymbol{e}\_c$ and $\boldsymbol{y}$ are defined as one-hot vectors, their dot product $\mathrm{arg\,max}_{\boldsymbol{e}\_c} \boldsymbol{e}\_c^T \boldsymbol{y}$ is eventually $\boldsymbol{y}$. Why would one go through a complicated definition without writing that: $L\_{0-1} = 1 - \boldsymbol{y}^T \boldsymbol{\hat{y}}$
>
> **Explanation:** We do not think that the definition of the 0-1 loss is inaccurate. Note that $\boldsymbol{y}$  and $\hat{\boldsymbol{y}}$  are not necessarily one-hot vectors but can be probability vectors (in particular, the prediction $\hat{\boldsymbol{y}}$ ).
>
> **Revision:** If this explanation is insufficient, we could use an indicator function.
>
> > Although the authors mention the abuse of notation in footnote 2 on page 5, using the same notation $\boldsymbol{\theta}$ for the functions $f$, $g$ and $h$ causes confusion. Please update the manuscript with a clearer notation.
>
> **Explanation:**
> - You are right about the slight abuse of notation because $\boldsymbol{\theta}$ represents the parameters for the data classification model in Eq. (1), whereas in all other cases, $\boldsymbol{\theta}$ encompasses all trainable parameters of the LFC model.
> - The LFC model allows us to make different predictions, i.e., an estimate for the true label probabilities at a specific instance, an estimate for the label probabilities at a specific instance for a specific crowdworker, and an estimate for the probability of a crowdworker to provide the correct label for a specific instance, of which all depend on $\boldsymbol{\theta}$
> - We prefer this notation because the functions $\boldsymbol{f}$, $\boldsymbol{g}$, and $h$ typically share a subset of parameters, and other notation would require more variables.
>
> **Revision**: We made our definition of $\boldsymbol{\theta}$ more explicit and moved it from a footnote to the main text. However, we still require some abuse of notation to keep the number of variables minimal, i.e., do not use an extra variable just for Eq. (1).

---

> > ### Comment · Reviewer_q474 · 2025-09-27
> > **Incomprehensible paper**
> >
> > Thank the authors for explaining the concerns raised initially in details. Despite the explanation provided by the authors, the revision is not updated accordingly, making the methodology of the paper incompreehnsible. There are certain points are mathematically incorrect.
> > - The disconnection between the methodology relating to risk minimization and the paragraph "Crowdworker Performance". As explained by the authors, it follows a top-down principle. However, in the current writing, it is difficult to understand the purpose of "Crowdworker Performance".
> > - There are a number of concerns in the "Crowdworker Performance":
> >     - The information leakage explained via Proposition 1 is vague. It is unclear why doing that would lead to information leakage.
> >     - The proof of Proposition 1 is, again, quite problematic. For example, Equality (45) seems to be the Bayes' rule. However, only the numerator of the Bayes' rule is included, while the denominator term $\Pr(\mathcal{Z} | x)$ is missing.
> >     - In the proof of Proposition 2, it is unclear where the assumption (58) is from.
> >     - Again, Eq. (18) is problematic because it is not satisfied the properties of a probability when having $C - 1$. In addition, the formulation is in probability notation, and it is unclear why such a probability is "defined", not derived based on probability theory. I understand that the authors want to replace the "cumbersome" transition matrix by a noise rate, which is simpler. However, the formulation in Eq. (18) is confusing.
> >     - The naming "equal" and "learned" in Eqs. (20) and (21) could be done better. For the case "equal", I think that $p$ should be denoted as $p_{m}$ represents the accuracy for the crowdworker indexed by $m$. The current naming just means that the authors set a special case of a special case, in which $p_{m} = \mathrm{const.} \forall m$.
> > - For the final part in the methodology (e.g., Eqs. (27) - (29)), it is unclear how such risks are "surrogate", in which the optimal hyperparameters obtained from them will give a consistent classifier. If this is not proved, it diminishes the main contribution of the paper.
> > - Again, I disagree with the usage of notation $\theta$ for all of the models. That causes confusion, especially when those models are mentioned in the same line.

---

> > > ### Author Response · Authors · 2025-10-01
> > > **Second Round Response: Part 1**
> > >
> > > Dear Reviewer,
> > >
> > > Thanks for acknowledging our previous explanation, and many thanks for your new feedback regarding our revision and for continuing the discussion with us. Again, we address each of your concerns by providing an explanation and a summary of the associated changes marked in $\color{blue}blue$ in the revised paper.
> > >
> > > > In the proof of Proposition 2, it is unclear where the assumption (58) is from.
> > >
> > > **Explanation:**
> > >
> > > - Eq. (58) is not an assumption in a classic sense but denotes a sufficient condition where even under uniform priors, sufficiently skewed confusion probabilities (e.g., when the HPC  tends to overfitting) can dominate the posterior and, thus, the MAP estimate (the aggregated label), potentially overriding the observed labels.
> > > - As an example, let us assume $\\boldsymbol{z}\_{n1}=\\boldsymbol{z}\_{n2}=\\boldsymbol{e}\_2$ as observed class labels with $C=3$, $M=2$, and uniform class priors.
> > > - Define the estimated confusion probabilities (only the needed ones) via:
> > >
> > > $$
> > > \qquad\widehat{\Pr}(z_{nm}=\boldsymbol e_2\mid \boldsymbol x_n, \boldsymbol y_n=\boldsymbol e_c,m)=
> > > \begin{cases}
> > > 0.90 & (c=1,\ m \in \\{1, 2\\}),\newline
> > > 0.40 & (c=2,\ m \in \\{1, 2\\}),\newline
> > > 0.20 & (c=3,\ m \in \\{1, 2\\}).
> > > \end{cases}
> > > $$
> > >
> > > - Then, the unnormalized posteriors are:
> > >
> > > $$
> > > \qquad S_c = \prod_{m=1}^2 \widehat{\Pr}(z_{nm}=\boldsymbol e_2\mid \boldsymbol x_n, \boldsymbol y_n=\boldsymbol e_c,m)
> > > \quad\Rightarrow\quad
> > > S_1=0.90^2=0.81,\\; S_2=0.40^2=0.16,\\; S_3=0.20^2=0.04.
> > > $$
> > >
> > > - As a result, the MAP estimate is $\boldsymbol e_1$ even though both crowdworkers labeled $\boldsymbol e_2$.
> > > - Moreover, if validation compares the classifier’s predictions to the MAP computed under the same HPC $\boldsymbol{\lambda}$, the target itself depends on $\boldsymbol{\lambda}$, HPO maximizes agreement with self-generated labels rather than ground truth, systematically favoring configurations that encode the skewed confusion probabilities.
> > >
> > > **Revision:** We refer to the condition in Eq. (58) when explaining the issue of the bias (see Appendix B.2).
> > >
> > > > Again, Eq. (18) is problematic because it is not satisfied the properties of a probability when having $C-1$. In addition, the formulation is in probability notation, and it is unclear why such a probability is "defined", not derived based on probability theory. I understand that the authors want to replace the "cumbersome" transition matrix by a noise rate, which is simpler. However, the formulation in Eq. (18) is confusing.
> > >
> > > **Explanation:**
> > >
> > > - Let us start with a concrete example for Eq. (18), where we set: $\widehat{\Pr}(\boldsymbol z^{\top}_{nm}\boldsymbol y_n=1\mid\boldsymbol x_n,m)=0.7$ with $C=4$ and $\boldsymbol{y}_n=\boldsymbol{e}_3$. Then, we get:
> > >
> > > $$
> > > \qquad \widehat{\Pr}(z_{nm}=\boldsymbol e_k\mid \boldsymbol x_n, \boldsymbol y_n=\boldsymbol e_3,m)=
> > > \\begin{cases}
> > > 0.70 & k=3,\newline
> > > \frac{1-0.7}{4-1}=0.1 & k \in \\{1, 2, 4\\}.\newline
> > > \\end{cases}
> > > $$
> > >
> > > - To show that Eq. (18) is also a valid probability distribution in the more general setting, we prove that it satisfies:
> > >
> > > $$
> > > \qquad\text{non-negativity: } 0\le\widehat{\Pr}(\boldsymbol z_{nm}^{\top}\boldsymbol y_n=1\mid\boldsymbol x_n,m)\le1,\ C\ge2\ \Rightarrow\ 0\le\frac{\widehat{\Pr}(\boldsymbol z_{nm}^{\top}\boldsymbol y_n=0\mid\boldsymbol x_n,m)}{C-1}\le1,\ \boldsymbol e_k^{\top}\boldsymbol e_c\in\\{0,1\\}\ \Rightarrow \widehat{\Pr}(\boldsymbol z_{nm}=\boldsymbol e_k\mid\boldsymbol x_n,\boldsymbol y_n=\boldsymbol e_c,m)\in[0,1],
> > > $$
> > >
> > > $$
> > > \qquad\text{normalization: }\sum_{k=1}^C \widehat{\Pr}(\boldsymbol z_{nm}=\boldsymbol e_k\mid\boldsymbol x_n,\boldsymbol y_n=\boldsymbol e_c,m)=\widehat{\Pr}(\boldsymbol z_{nm}^{\top}\boldsymbol y_n=1\mid\boldsymbol x_n,m)+(C-1)\frac{\widehat{\Pr}(\boldsymbol z_{nm}^{\top}\boldsymbol y_n=0\mid\boldsymbol x_n,m)}{C-1}=1.
> > > $$
> > >
> > > - The probability is defined because it is a design choice not to allow confusion probabilities (or matrices) without restrictions.
> > > - This design choice is supported by Proposition 2 (see also our previous explanation regarding Eq. (58)) because otherwise sufficiently skewed confusion probabilities can override the observed labels.
> > >
> > > > Again, I disagree with the usage of notation for all of the models. That causes confusion, especially when those models are mentioned in the same line.
> > >
> > > **Explanation:** Although we think that there is also a benefit to a more lightweight notation without separate parameter symbols, we can also understand that this can lead to confusion.
> > >
> > > **Revision:** We have now reworked our notation by introducing distinct symbols (see Section 3.2) for the parameters of all three models, while noting that the models may share parameters (e.g., if they use the same backbone).

---

> > > ### Author Response · Authors · 2025-10-01
> > > **Second Round Response: Part 2**
> > >
> > > > The naming "equal" and "learned" in Eqs. (20) and (21) could be done better. For the case "equal", I think that $p$ should be denoted as $p_m$ represents the accuracy for the crowdworker indexed by $m$. The current naming just means that the authors set a special case of a special case, in which $p_m = \mathrm{const}. \forall m$.
> > >
> > > **Explanation:**
> > >
> > > - The name “equal” indicates that the correctness probability of a label is equal across all crowdworkers, classes, and instances.
> > > - This most simplistic case is intended by us because it does not involve any learning and, thus, serves as a baseline.
> > > - If we write $p_m$ instead of $p$, we would need a method to learn the crowdworker-specific accuracies (performances).
> > > - However, for this purpose, we already have the alternative case of learned crowdworker accuracies.
> > > - For example, if we have an LFC approach with a crowdworker performance model estimating a crowdworker’s performance independent of the instance input, we get $\forall \boldsymbol{x}: h_{\boldsymbol{\psi}}(\boldsymbol{x}, m) = p_m$.
> > >
> > > **Revision:** In the paragraph “Crowdworker Performance” in Section 3.4, we indicated that modeling instance-dependent crowdworker performances is optional and depends on the implementation of the crowdworker performance model $h_{\boldsymbol{\psi}}$.
> > >
> > > > The proof of Proposition 1 is, again, quite problematic. For example, Equality (45) seems to be the Bayes' rule. However, only the numerator of the Bayes' rule is included, while the denominator term $\Pr(\mathcal{Z} \mid \boldsymbol{x})$.
> > >
> > > **Explanation:** Thanks for this pointer to this symbol error because Eq. (45) must reflect Eqs. (15), (16), where we correctly used $\propto$ instead of $=$.
> > >
> > > **Revision:** In the proof of Propositions 1 and 2, we now use the normalization $\Pr(\mathcal{Z}\_n \mid \boldsymbol{x}\_n)$ instead of using any $\propto$.
> > >
> > > > The information leakage explained via Proposition 1 is vague. It is unclear why doing that would lead to information leakage.
> > >
> > > **Explanation:**
> > >
> > > - We agree that our terminology was confusing/imprecise.
> > > - With strictly disjoint training and validation sets, the issue we highlight is not information leakage in the narrow sense but rather, it is a circular evaluation bias: using $\boldsymbol{f}_{\boldsymbol{\theta}}$ to set the instance-specific prior $\widehat{\Pr}\left(\boldsymbol{y}_n \mid \boldsymbol{x}_n\right)$ biases the posterior toward the model’s own predictions, and by Proposition 1 this can dominate the MAP regardless of the observed crowd labels.
> > > - Our statement "… which can be interpreted as a form of double-dipping …" (see paragraph on “Crowdworker Performance” in Section 3.4) was meant to refer to this circular logic, not actual data reuse across splits.
> > >
> > > **Revision:** We replaced the corresponding sentence (see paragraph on "Crowdworker Performance" in Section 3.4).
> > >
> > > > For the final part in the methodology (e.g., Eqs. (27) - (29)), it is unclear how such risks are "surrogate", in which the optimal hyperparameters obtained from them will give a consistent classifier. If this is not proved, it diminishes the main contribution of the paper.
> > >
> > > **Explanation:**
> > >
> > > - We make no claim of theoretical consistency for classifiers whose HPs were determined via our HPS criteria with only access to crowd-labeled validation data.
> > > - We merely used “surrogate” to refer to HPS criteria with only access to noisy validation labels as proxies of the HPS criterion with access to the true validation labels. We changed that terminology (see revision below).
> > > - Because HPS criterion TRUE depends on unobserved validation labels, it is not identifiable from noisy validation data in general. With differing LFC assumptions, possible model misspecification, and heterogeneous crowdworkers, distinct data-generating processes can be observationally equivalent yet induce different TRUE rankings over HPCs. Therefore, any universal selection guarantee requires explicit assumptions (e.g., identifiable noise structure, bounded error rates).
> > > - We do not investigate such assumptions here and do not claim universal guarantees.
> > > - Accordingly, our abstract and contribution box emphasize a comprehensive benchmark on real crowd-labeled data that empirically evaluates the proposed HPS criteria.
> > > - We sum ranks in Eq. (29) because the empirical risk measures are individually noisy, which leads to the HPC criterion ENS attaining the best mean rank across datasets and LFC approaches according to Table 4 (excluding the criterion TRUE).
> > > - Finally, our conclusion (see Section 5) indicates that this work establishes a baseline (starting point) for HPO with noisy crowd-labeled validation data.
> > > - We recommend that new LFC approaches ship with a matching HPS criterion for noisy crowd-labeled validation.
> > >
> > > **Revision:**
> > >
> > > - To avoid confusion with classical “surrogate losses”, we replaced “surrogate” with “proxy” and added a footnote with our intent (see first paragraph of Section 3.4).

---

> ### Author Response · Authors · 2025-10-01
> **Second Round Response: Part 3**
>
> > The disconnection between the methodology relating to risk minimization and the paragraph "Crowdworker Performance". As explained by the authors, it follows a top-down principle. However, in the current writing, it is difficult to understand the purpose of "Crowdworker Performance".
>
> **Explanation:**
>
> - Once you pick a risk template for validation, you still need targets or weights for those validation items. Those come from how you model crowdworker performance.
> - Your modeling of crowdworker performances tells you how to combine the individual labels (aggregation) and how much trust to place in each crowdworker (weighting).
> - Since this decision affects the definition of the label aggregation and label weighting function, there is a clear connection between crowdworker performance and the empirical risk measure, and thus the actual implementation of an HPS criterion.
>
> **Revision:** We adjusted the beginning of the paragraph on “Crowdworker Performance” in Section 3.4.
>
> > Despite the explanation provided by the authors, the revision is not updated accordingly, making the methodology of the paper incompreehnsible.
>
> **Explanation:** In this new round of smaller revisions, we have further refined and clarified the methodological presentation for readability according to your comments. We are ready to provide additional details on any specific points if needed.
>
> **Question:** We are a bit confused by the formulation “the revision is not updated accordingly”. Do you mean our made revisions in the first round did not meet your expectations, or could you not see our changes (e.g., due to a technical issue)?

---

### Review · Reviewer_7Mew · 2025-08-17

**Summary Of Contributions:**

This paper addresses the critical problem of hyperparameter optimization (HPO) in the Learning from Crowds (LFC) literature, where methods are often evaluated unfairly using hyperparameters tuned on clean labels or unrealistically with fixed default settings. The central contribution is `crowd-hpo`, a novel framework for selecting hyperparameters using only the available noisy, crowd-labeled validation data. This removes the need for either suboptimal defaults or an idealized clean validation set. The authors systematically propose several selection criteria based on different risk estimation strategies and worker performance assumptions, culminating in a robust ensemble method (ENS).

A major strength is its comprehensive and large-scale experimental study. It provides compelling evidence for the framework's effectiveness while also yielding critical insights, such as how the choice of selection criterion can drastically alter the final performance and ranking of LFC methods across different datasets.

**Audience:**

Yes

**Audience Explanation:**

Its systematic review of literature from top ML venues (Table 1) reveals that prior studies are often less experimentally thorough and commonly rely on flawed protocols, such as using unrealistic clean validation sets or datasets with simulated, non-crowdsourced labels. By demonstrating that these common practices can alter method rankings and lead to unfair conclusions, the paper provides an indispensable guide. Its methodology is essential for any researcher aiming to develop or benchmark LFC algorithms with real-world relevance and scientific rigor.

**Broader Impact Concerns:**

The paper includes a "Broader Impact Statement" section. The authors have acknowledged the primary societal and ethical implications of their work, including the practical limitations of the framework and the crucial importance of fair labor conditions for crowdworkers.

The provided statement is sufficient and well-addressed. I have not identified any further broader impact concerns.

**Claims And Evidence:**

Yes

**Claims Explanation:**

The paper is structured as a comprehensive investigative study, and its conclusions are directly grounded in the results of its extensive and well-designed experiments.

- **The claim that existing HPO practices are suboptimal and lead to unfair comparisons** is clearly supported. Figure 1 provides a stark visual demonstration of how performance and rankings change between default (`DEF`) and optimized (`TRUE`) hyperparameters. Table 4 further quantifies this by showing that both `DEF` and `DEF-DATA` baselines consistently rank the lowest and offer minimal performance gains, validating the paper's central motivation.

-  **The claim that the proposed `crowd-hpo` framework effectively addresses hyperparameter selection using only noisy data** is convincingly demonstrated. Table 4 shows that all proposed criteria achieve significant zero-one loss reductions compared to the `DEF` baseline. Crucially, the proposed ensemble criterion (ENS) stands out as the top-performing realistic method, significantly narrowing the performance gap to the TRUE upper-bound baseline.

-  **The claim that the choice of selection criterion is itself critical and can alter the ranking of LFC methods** is also well-supported. The low Kendall's rank correlation coefficients presented in Figure 7 provide direct, quantitative evidence that rankings produced by naive baselines (`DEF`, `DEF-DATA`) differ significantly from those produced by more robust criteria (`ENS`, `TRUE`). Figure 6 further illustrates this by showing how the relative performance gains between different LFC approaches shift depending on the HPS criterion used and the datasets.

**Requested Changes:**

The authors highlight the considerable performance dispersion of HPS criteria across datasets, as in the end of Section 4.2. Since Table 2 shows that these datasets also feature a wide range of noise levels, the paper would be even more insightful with an analysis stratified by noise. For instance, presenting the Absolute Zero-one Loss Reductions or Rank Correlations for low-noise versus high-noise variant groups could offer a clearer understanding of how noise severity impacts the effectiveness of different selection criteria. This analysis would likely further strengthen the necessity of the robust ensemble (ENS) strategy.

---

> ### Author Response · Authors · 2025-08-23
> **Rebuttal**
>
> **Summary:** We added an analysis stratified by label noise to Appendix 4.2. Changes are $\color{olive}olive$.
>
> Foremost, many thanks for your detailed and helpful criticism to strengthen our paper. Moreover, thanks for acknowledging the importance of our contributions to benchmarking future LFC approaches with real-world relevance. In the following, we provide a brief explanation and a summary of the revision, which we mark in $\color{olive}olive$ in the revised paper.
>
> > The authors highlight the considerable performance dispersion of HPS criteria across datasets, as in the end of Section 4.2. Since Table 2 shows that these datasets also feature a wide range of noise levels, the paper would be even more insightful with an analysis stratified by noise. For instance, presenting the Absolute Zero-one Loss Reductions or Rank Correlations for low-noise versus high-noise variant groups could offer a clearer understanding of how noise severity impacts the effectiveness of different selection criteria. This analysis would likely further strengthen the necessity of the robust ensemble (ENS) strategy.
>
> **Explanation:**
> - Thanks for this valuable suggestion. We implemented it by dividing the 35 dataset variants into a low- and high-noise group and by analyzing the HPS criteria and LFC approaches for each group separately.
> - As expected, the non-default criteria have a greater benefit for the high-noise group, yielding larger absolute and relative loss reductions. Still, non-default criteria also provide noticeable gains in the low-noise setting.
> - In both groups, the upper baseline TRUE (with access to true validation labels) performs best, followed by ENS (with only access to crowd-labeled validation data).
> - Concerning the LFC approaches, two key takeaways are (1) that the performance gains of one-stage approaches over two-stage approaches are larger in the high-noise setting and (2) that the ranking between the two settings can be quite different.
> - A comparison of the rank correlation coefficients between the two groups shows differently distributed values. In particular, there is a higher spread of the Kendall $\tau$-$b$ coefficients for the high-noise group when comparing ranking for the default (DEF, DEF-DATA) HPCs with the rankings after HPO (via TRUE or ENS), although their means remain similar.
>
> **Revision:** We added a paragraph with tabular and graphical results from this noise-stratified analysis to Appendix C.2 and referred to it at the start of Section 4.2.

---

### Review · Reviewer_gS1P · 2025-08-17

**Summary Of Contributions:**

The paper attempts to solve the common issue in learning from crowds, namely how to tune models when the only available validation labels are noisy crowd annotations. They (1) propose a hyperparameter-selection (HPS) criteria that estimate validation risk directly from crowd labels and recommends a simple rank-ensemble over these criteria to make selection more robust; and (2) build a benchmark, including 13 LFC approaches across 5 real datasets with 7 noise variants for realistic, fair comparison.

Key strengths: This work provides empirical evidence that the proposed criteria (especially the ensemble) reliably beat defaults in zero-one loss. The benchmarking setting is reasonable with the same candidate set, the same CV, and multiple seeds. It is a large, reproducible benchmark.

Key weaknesses: these experimental results emphasize zero-one loss, a notable limitation, but the authors have already discussed it in the manuscript. As a benchmark, the sensitivity analysis in this benchmark is also insufficient; for example, they fix a Sobol, $|\Lambda|=51$ candidate set.

**Audience:**

Yes

**Audience Explanation:**

This paper can attract the readers who care about weak supervision, crowdsourcing, practical HPO/AutoML, and reproducible benchmarking.

**Broader Impact Concerns:**

I have no additional ethical concerns beyond what the authors already discuss.

**Claims And Evidence:**

Yes

**Claims Explanation:**

Mostly yes for the paper’s central claims, the evidence is careful, repeated, and clearly presented.

**Requested Changes:**

1. This paper attempts to compress baseline methods used $C \times C$ confusion matrices to scalar throws, this disadvantages some methods whose core strength is modeling per-class worker biases (e.g., DS and GLAD style methods).

2. This paper freezes strong pretrained backbones (e.g., DINOv2 and MPNet) and put a small MLP head on top for fair comparison. The author can report an end-to-end fine-tuning setting with light fine-tuning or at least vary the head size to show conclusions are stable.

3. The statistical significance should be reported, add paired tests.

---

> ### Author Response · Authors · 2025-08-23
> **Rebuttal: Part 1**
>
> **Summary:** We added analyses regarding an alternative loss function and the size of the HPC candidate set $\Lambda$, reported statistical test results, and clarified potential misunderstandings. Changes are  $\color{red}red$ in the uploaded revision.
>
> Foremost, many thanks for your detailed and helpful criticism. Moreover, thanks for acknowledging our paper’s central claims, including our large, reproducible benchmark, to be of interest to several groups of readers. In the following, we provide explanations and summaries of the changes associated with your criticism, which we mark in $\color{red}red$ in the revised paper.
>
> > Key weaknesses: these experimental results emphasize zero-one loss, a notable limitation, but the authors have already discussed it in the manuscript.
>
> **Explanation:**
> - We agree that this is a notable limitation, which we only briefly noted. Therefore, we have performed a brief analysis of the Brier score that shows that the target loss function has a non-negligible impact on the choice of the HPS criteria.
> - Our key takeaway that the proposed (non-default) HPS criteria improve performance even in an LFC setting with only crowd-labeled validation data is confirmed.
> - However, the best criterion for optimizing the Brier score is not ENS but ALC using estimates of the crowdworker performance model $\boldsymbol{h}_{\boldsymbol{\theta}}$ for label aggregation and weighting.
> - In our opinion, the ensemble-based approach ENS remains appealing, as its flexible design even allows combining risk measures derived from different loss functions. Exploring such combinations represents a promising direction for future work.
>
> **Revision:** We have added this analysis to Appendix C.2 and a summary of it to the limitations paragraph in Section 4.3.
>
> > As a benchmark, the sensitivity analysis in this benchmark is also insufficient; for example, they fix a Sobol, $|\Lambda|=51$ candidate set.
>
> **Explanation:**
> - Thanks for pointing out this weakness, which we hopefully resolved by adding a corresponding analysis to study the effect of changing the size of the candidate set $\Lambda$.
> - Regarding the comparison of the HPS criteria, we observe that the loss reductions compared to using default  HPCs decrease with a decreasing number $|\Lambda|$ of candidate HPCs. This is likely because the selection is more restricted. The results also confirm that ENS is the best-performing HPS criterion with only access to crowd-labeled validation data across all tested budgets.
> - Regarding the comparison of the LFC approaches, we observe that the performances of the LFC approaches also decrease with a decreasing number $|\Lambda|$ of candidate HPCs, while the one-stage approaches still take the lead over the two-stage approaches.
>
> **Revision:** We added a paragraph with tabular and graphical results from this analysis stratified by $|\Lambda|$ to Appendix C.2 and referred to it at the start of Section 4.2.
>
> > This paper attempts to compress baseline methods used $C \times C$ confusion matrices to scalar throws, this disadvantages some methods whose core strength is modeling per-class worker biases (e.g., DS and GLAD style methods).
>
> **Explanation:**
> - We think that there could be a misunderstanding since we do not make any changes to the training procedures of the individual LFC approaches. Accordingly, approaches that estimate full confusion matrices per crowdworker or even per instance-crowdworker pair were allowed to do so. Only during validation, we use the scalar version from Eq. (21) for the reasons provided in the derivation of Section 3.4 and Proposition 2. Moreover, for estimating this scalar, we use the full inference capacity of each approach, which we detail in Appendix A.
> - Despite this simplification during validation, Figure 5 demonstrates that all LFC approaches profit from the HPO via the non-default HPS criteria. The results from Table 5 additionally confirm that the approaches $\texttt{geo-reg-f}$ and $\texttt{coin}$ modeling per-class crowdworker biases ($C \times C$ confusion matrices) perform best.
> - Finally, we totally agree with you that defining additional HPC criteria using full confusion matrices could be a future research direction. For now, this is out of the scope of this paper because it needs solutions to avoid harmful class-specific biases during validation (see Proposition 2).
>
> **Revision:**
> - In the paragraph on crowdworker performance in Section 3.4, we made explicit that we use the scalar version only during validation.
> - In Section 5, we noted the potential for using a different aggregation method for validation, which can include per-class crowdworker biases.

---

> ### Author Response · Authors · 2025-08-23
> **Rebuttal: Part 2**
>
> > The statistical significance should be reported, add paired tests.
>
> **Explanation:**
> - Thanks for this suggestion to obtain a more rigorous evaluation. We implemented it by adopting the following test procedure.
>     - A Friedman omnibus test checks whether there are any significant differences at all.
>     - If the omnibus is significant, we perform all pairwise Wilcoxon signed-rank tests and control the family-wise error with Holm’s correction.
> - We performed this test procedure for the comparison of the HPS criteria as part of $RQ_1$ and for the comparison of the LFC approaches as part of $RQ_2$.
> - For $RQ_1$, TRUE and ENS significantly outperform the other criteria
> - For $RQ_2$, $\texttt{geo-reg-f}$ and $\texttt{coin}$ significantly outperform most competing approaches in pairwise comparisons.
>
> **Revision:** We added the description of the test procedure to Section 4.2 and the number of significant wins, significant losses, and non-significant comparisons (ties) to Table 4 for the HPS criteria and Table 5 for the LFC approaches. The full results of the pairwise comparisons are given in Appendix C.2.
>
> > This paper freezes strong pretrained backbones (e.g., DINOv2 and MPNet) and put a small MLP head on top for fair comparison. The author can report an end-to-end fine-tuning setting with light fine-tuning or at least vary the head size to show conclusions are stable.
>
> **Explanation:**
> We are planning the fine-tuning experiments and aim to provide (at least preliminary) results within the next week. Given the large number of dataset variants, LFC approaches, and cross-validation folds, careful planning is required to ensure feasibility. In particular, we are examining which fine-tuning strategy is most suitable (e.g., full fine-tuning with block-wise learning rate decay or selectively unfreezing certain blocks). As a fallback, testing only a subset of these combinations or, as you suggested, varying the head size provides a straightforward way to demonstrate the stability of our conclusions.
>
> **Revision:** We will add an analysis to Appendix C.2 once experiments are completed.

---

> ### Author Response · Authors · 2025-09-01
> **Rebuttal: Part 3**
>
> Update on:
>
> > This paper freezes strong pretrained backbones (e.g., DINOv2 and MPNet) and put a small MLP head on top for fair comparison. The author can report an end-to-end fine-tuning setting with light fine-tuning or at least vary the head size to show conclusions are stable.
>
> **Explanation:**
> - While our initial response discussed adding light end-to-end fine-tuning, the revision instead implements a targeted head-architecture search. In our view, this is an appropriate robustness test because it
>    - addresses your request to vary at least the classification head design,
>    - preserves a single training/evaluation protocol for comparability, and
>    - is budget-feasible.
> - In contrast, end-to-end fine-tuning is computationally (GPU-wise) expensive across our many dataset variants, approaches, and folds, and requires heterogeneous training protocols (e.g., block-wise/layer-wise learning-rate decay, staged unfreezing, warmup) exceeding our paper's scope.
> - Concretely, for the $\\texttt{label-me}$ and $\\texttt{dopanim}$ dataset variants, we expand the search from a fixed MLP with $(256, 128)$ neurons in its two hidden layers to different numbers of layers and neurons sampled according to $\\texttt{uniform}\\left(\\{(256), (512), (256, 128), (512, 256)\\}\\right)$.
> - Because of the more complex HP space $\Omega_\Lambda$, we also increase the number of candidate HPCs to $|\Lambda|=101$.
> - The associated results keep our paper's main conclusion unchanged/stable: non-default HPS criteria with only access to crowd-labeled validation data remain beneficial.
> - Moreover, only TRUE with access to true validation labels outranks ENS and achieves larger zero-one-loss reductions.
>
> **Revision:**
> - We added these new experiments with their analysis to Appendix C.2 and referenced them in Section 4.2.
> - We noted the limitation of not evaluating the HPO in combination with fine-tuning of pre-trained backbones in Section 4.3.

---

### Author Response · Authors · 2025-08-23
**Revision Summary**

Many thanks again to all reviewers for their detailed and constructive feedback. We use this comment to summarize the revisions we made to the uploaded paper based on your suggestions. Changes are highlighted according to the reviewer who proposed them.

**Revisions**:
- $\color{blue}\text{Reviewer q474}$: Added more detailed explanations in Section 3 and in the proof of Propositions 1 and 2.
- $\color{red}\text{Reviewer gS1P}$: Reported aggregated statistical test results of the HPC criteria and LFC approaches in Section 4.2, with detailed pairwise comparisons moved to Appendix C.2.
- $\color{olive}\text{Reviewer 7Mew}$: Added an analysis stratified by noise level in Appendix C.2 and referred to it in Section 4.2.
- $\color{red}\text{Reviewer gS1P}$: Added an analysis stratified by the number of candidate HPCs in Appendix C.2 and referred to it in Section 4.2.
- $\color{red}\text{Reviewer gS1P}$: Added an analysis using the Brier score as an alternative loss function in Appendix C.2 and referred to it in Section 4.3.
- $\color{red}\text{Reviewer gS1P}$: Added a case study on HPO with an architecture search for the classification head of the DINOv2 backbone in Appendix C.2 and referred to it in Section 4.3.
- $\color{blue}\text{Reviewer q474}$: Revised the mathematical notation, in particular the notation of model parameters.

**Conclusion**:
In our view, these revisions strengthen both the contributions and the readability of our paper, while preserving its main takeaways: the proposed HPS criteria enhance the performance of LFC approaches and enable more realistic benchmarking of them when only crowd-labeled validation data are available.

---

### Decision · Action_Editor_cNzM · 2025-10-22

**Recommendation:** Accept with minor revision

**Additional Comments:**

I appreciate the authors' responsiveness to reviews and their multiple updates. Given the reviews and discussions, the following point may improve the paper further in the camera ready version.

The paper does not study the relationship between the proxy HPS criteria and the underlying classification risk, e.g., unbiasedness or consistency. I understand that such analysis may require assumptions on the data generation or the workers' labeling process (as the rebuttal explained), but in its absence it remains unclear what a practitioner working with crowdsourced training/validation labels should do in practice. The experiments show that the most suitable criterion varies by dataset and setup, so it would help to provide more guidance on how to choose a criterion, when a future researcher/engineer is working a project with a new dataset without ground-truth evaluation data. The rebuttal provided a good discussion that clarifies the contributions of the paper starting from "We make no claim of theoretical consistency...". Adding a lot of this discussion also into footnote 4 would be very helpful for the reader.

I recommend a light edit so this point is clearer before the camera-ready submission.

**Audience:**

Yes

**Audience Explanation:**

There are many researchers studying learning from crowds.

**Claims And Evidence:**

Yes

**Claims Explanation:**

This paper introduces crowd‑hpo, a framework for hyperparameter selection (HPS) when clean validation data are unavailable and only crowd‑labeled validation data exist. It defines several proxy risk criteria (at the aggregation level and the crowd level) and an ensemble (ens). Across different settings and approaches, the paper shows that crowd‑only HPO substantially improves over using defaults. "ens" is the only crowd‑only criterion with no significant loss relative to a clean‑label oracle. Evidence and discussions are provided for the two research questions stated in the paper.

---

> ### Author Response · Authors · 2025-11-20
> **Camera-ready Submission and Minor Revisions**
>
> Dear Action Editor,
>
> Thank you for acknowledging the contributions of our paper and for suggesting minor revisions to further strengthen the work. We apologize that our response took a few weeks. We wanted to carefully implement and verify all requested changes before submitting our suggestion for the camera-ready version.
>
> The minor revisions are as follows:
> - **Section 4.3**: We have expanded the previous list of recommendations into a more systematic guideline, including a simple flowchart, to support researchers and practitioners in optimizing the hyperparameters (HPs) of a learning-from-crowds (LFC) approach for a (new) crowd-labeled target dataset.
> - **Section 5:** We now provide a dedicated section on limitations. It includes a paragraph reflecting the helpful discussion with the reviewers regarding the missing theoretical results on unbiasedness and (selection) consistency of our proxy hyperparameter selection (HPS) criteria, whose underlying empirical risk estimates rely solely on crowd-labeled validation data.
> - **Footnote 4:** We extended this footnote to reference the new discussion of our empirical viewpoint in Section 4.3 and its limitations in Section 5.
> - **Minor edits:** We corrected several typographical errors and refined the wording in several places for clarity.
> - **Deanonymization:** We have added the author names, the actual link to our GitHub codebase, and the link to this OpenReview discussion.
>
> We hope that these revisions address all outstanding concerns. Please let us know if any further changes are required.
>
> Best regards,
>
> Authors